# Generating Creative Chess Puzzles

**Xidong Feng**[◇] **Vivek Veeriah**[◇] **Marcus Chiam**[◇] **Michael Dennis**[◇] **Federico Barbero**[‡*]
**Johan Obando-Ceron**[§*] **Jiaxin Shi**[◇] **Satinder Singh**[◇] **Shaobo Hou**[◇]
**Nenad Tomašev**[◇] **Tom Zahavy**[◇] [†]
[◇]Google DeepMind [‡]University of Oxford [§]Mila, University of Montreal
{xidong,tomzahavy}@google.com

## Abstract

While Generative AI rapidly advances in various domains, generating truly creative, aesthetic, and counter-intuitive outputs remains a challenge. This paper presents an approach to tackle these difficulties in the domain of chess puzzles. We start by benchmarking Generative AI architectures, and then introduce an RL framework with novel rewards based on chess engine search statistics to overcome some of those shortcomings. The rewards are designed to enhance a puzzle's uniqueness, counter-intuitiveness, diversity, and realism. Our RL approach dramatically increases counter-intuitive puzzle generation by 10x, from 0.22% (supervised) to 2.5%, surpassing existing dataset rates (2.1%) and the best Lichess-trained model (0.4%). Our puzzles meet novelty and diversity benchmarks, retain aesthetic themes, and are rated by human experts as more creative, enjoyable, and counter-intuitive than composed book puzzles, even approaching classic compositions. Our final outcome is a curated booklet (Appendix M) of these novel AI-generated puzzles, which is acknowledged for creativity by three world-renowned experts.

The idea of computers achieving human-level intelligence, first proposed by the Turing test [90], has been a subject of ongoing debate since the dawn of computing [64]. From outperforming humans in complex games [72, 59, 8, 58] to demonstrating remarkable abilities in fields like medicine [79], law, business [44], mathematics [28], and coding [49, 24] AI's capabilities are rapidly expanding. Initial signs even suggest that LLMs pass a basic version of the Turing test [42], prompting the question of what cognitive functions will remain uniquely human.

Creativity is often viewed as the "final frontier" for AI [15]. Despite recent Generative AI demonstrating notable abilities across various domains, they continue to fall short in creative problem-solving, abstract reasoning, and compositional complexity [40]. For example, while LLM-generated poetry can seem indistinguishable from human creations [68], expert evaluations indicate a deficiency in complex structural elements [19]. More generally, LLMs continue to struggle with creative writing [83], and tend to generate practical but less original business ideas than humans [7].

This paper investigates the application of Gen AI techniques for creating aesthetic and creative chess puzzles [2, 48]. Chess puzzles have a history in education, online entertainment, and computational creativity research [39, 37], yet, the generation of creative chess puzzles is difficult due to a number of reasons. Chess puzzles have non-trivial pattern that only reveals itself when inspecting the solution. Moreover, a standardized definition of a chess puzzle is absent, making it challenging to objectively measure its creativity or aesthetic qualities. The limited and diverse nature of available chess puzzle data, which varies in puzzle types, objectives, piece utilization further complicates the process.

Our work begins by formalizing the definition: what makes a board position creative and interesting that can engage chess players? We concentrate on well-established aspects of creativity in chess [48, 2]: counter-intuitiveness, aesthetics and novelty. Counter-intuitive solutions are those that, at

[*]Research conducted during an internship at Google DeepMind
[†]Ryan Pachauri and Thomas Tumiel contributed but were inadvertently omitted from the author list.

39th Conference on Neural Information Processing Systems (NeurIPS 2025).

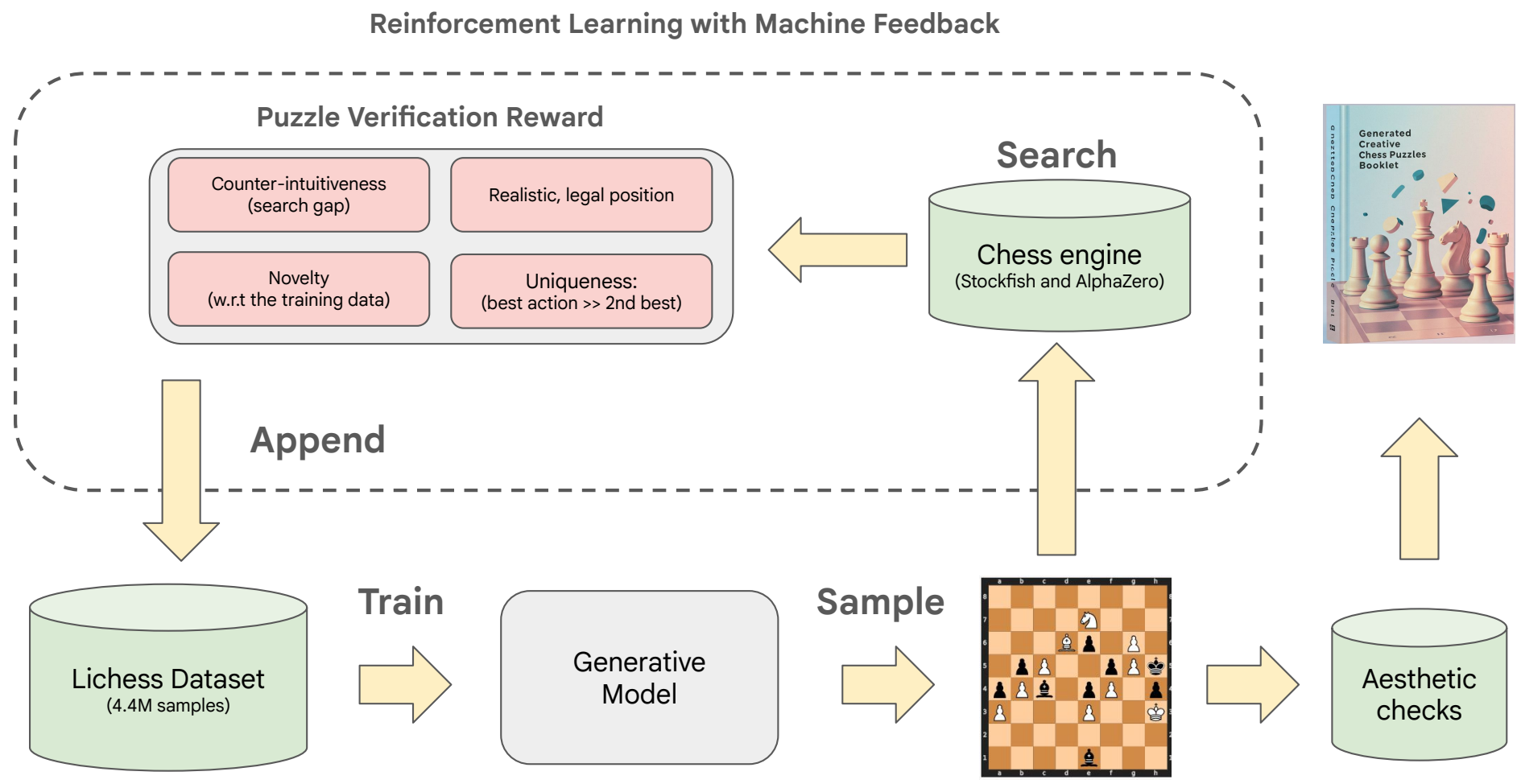

Figure 1: Our approach begins by training a generative model on the Lichess dataset (Section 3.2), followed by RL training (Section 2.1). Each position generated by the RL model is verified for legality, uniqueness, novelty, and counter-intuitiveness using chess engine search statistics (Section 1). The positions are filtered for aesthetics (Appendix I) and selected based on reward for our booklet.

first glance, seem are terrible or out-of-mind, but are, in fact, brilliantly effective. Aesthetics refers to the visually or intellectually beauty or elegance of a chess position and its solution. Novelty is straightforward: players are often drawn to positions or solutions that are uncommon or haven't been seen frequently in normal chess game. In Section 1, we detail the logic of these metrics and present how we can quantify them with computation. These proposed quantifications are not merely descriptive; they form the basis of the reward functions and evaluation metrics used throughout our study. To help readers better appreciate what makes a puzzle creative, Fig. 2 offers a detailed analysis on booklet examples. Next, we prioritize chess puzzles with a single, unambiguous solution: multiple 'correct' paths dilute the satisfaction of finding the sharpest, most brilliant line. This criterion also allows us to leverage the large open-source Lichess Puzzler [52] dataset. As shown in Fig. 6 about the volume of games played online (2021-2025), only 2.1% of annually one million puzzles meet our counter-intuitiveness criterion – a comparatively small number for AI applications.

Given these chess puzzle metrics, we initiated our work by evaluating several Generative AI architectures, including auto-regressive transformers [91], latent diffusion models [70], masked discrete diffusion models [77], and MaskGIT [11] – all trained on the Lichess Puzzler dataset. We benchmarked these models by generating one million positions from each and assessing them based on legality, uniqueness, counter-intuitiveness, and novelty (results detailed in Tables 2 and 3).

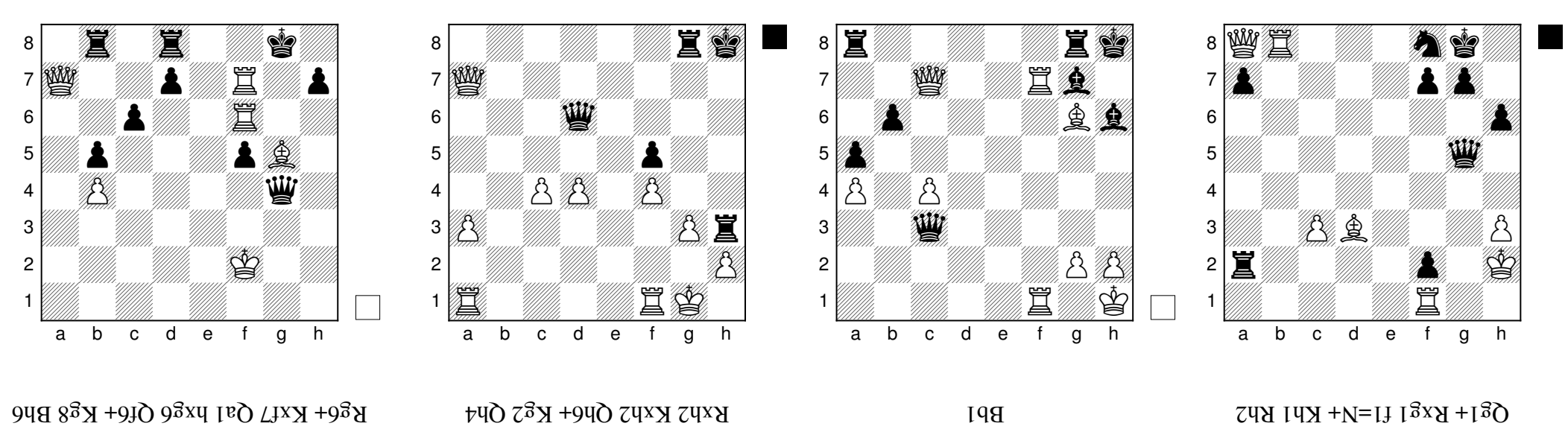

Figure 2: Booklet examples of creative chess puzzle, generated with our methods, with a unique, counter intuitive and aesthetic solution (written upside down). *1 (left):* White undefends both rooks with Rg6+. After one of the hanging rooks is captured, White plays the slow Qa1 move, sacrificing the second rook. The remaining White queen and bishop coordinate very well after Qf6+, with an unstoppable attack. The double rook sacrifice is counter intuitive and aesthetic for a human: both rooks are initially very active and it is surprising that the remaining queen and bishop are sufficient to win the game. The winning move in this position is both counter-intuitive and the only one for White. Even Stockfish needs a moment to identify it (see analysis on lichess), but then confidently confirms it's the sole path to victory. For a description of positions *2–4*, please see Appendix I.

To tackle the limited availability of chess puzzle data, we further trained the transformer model with RL. This reward function included a Lichess-style uniqueness check and a counter-intuitiveness measure designed to identify positions solvable by strong engines but not weak ones. This latter component, tuned with human feedback (Section 3.1), was based on a linear combination of search statistics from StockFish [86] and AlphaZero, with the 'critical depth' (the depth StockFish first finds the final solution) being the dominant term. In addition, we found that RL with pure puzzle reward would quickly succumb to 'entropy collapse' by repeatedly generating one single high-reward puzzle. Therefore, we developed a diversity filtering mechanism (Section 2.1) that promotes exploration and diversity. This was proved essential for preventing reward hacking and largely increased training stability, enabling the continuous generation of novel and diverse puzzles (Section 3.4).

Our findings demonstrate that RL markedly enhances the generation of counter-intuitive puzzles. The probability increased significantly from 0.22% (transformer trained on Lichess) to 2.5%, exceeding both the best Lichess-trained model's rate (0.4%) and the baseline in the Lichess training data (2.1%), while satisfying novelty and diversity criteria. While aesthetic features were not directly verified by the reward and thus not optimised for during the RL training, we observed that aesthetic themes were well-preserved in the RL samples (Section 3.3). Crucially, expert evaluation confirmed the success of our approach, demonstrating that some of our generated chess puzzles are fun, creative and counter intuitive as puzzle compositions from the chess literature. Our final result is a booklet of curated chess puzzles in Appendix M; a few examples can be seen in Fig. 2. The booklet is further reviewed by three world-renowned experts and garners the acknowledgement for their creativity.

## 1 Quantifying the standard of creative chess puzzles

Originating in the ninth century, chess puzzles predate modern chess rules and present chess-related problems solvable through knowledge of the game. This work concentrates on legal chess positions with a unique, optimal, and ideally aesthetic solution sequence, excluding puzzles with multiple solutions, non-standard elements, special stipulations, retrogrades, missing information, shortest mates, or mathematical aspects. Mate-in-x puzzles are considered only if they have a single correct solution. For an overview of various chess puzzle types, see [17, 87, 31, 85].

To quantify the standard of creative puzzles, we focus on puzzles with a unique solution. Following this, we examine creative aspects of puzzles such as novelty, counter-intuitiveness, and aesthetics.

**Uniqueness:** To confirm that a chess position represents a puzzle with a single unique solution, we utilize the StockFish engine, following the Lichess Puzzler methodology. This involves recursively examining the principal variation to compare the winning chances $w(a \mid s)$ of the best move $a^*$ against the second best move (see Appendix C for more details). A move is considered unique if the difference in these winning chances meets or exceeds an empirically chosen threshold ($\tau_{uni} = 0.5$):

$$\mathbf{r}_{uni} = \max_{a \in A(s)} w(a \mid s) - \max_{a \in A(s) \setminus \{a^*\}} w(a \mid s) \geq \tau_{uni}. \tag{1}$$

**Creativity:** This is a paradoxical concept, often linked to rule-breaking and unconventional thinking. It can be challenging to define precisely. For instance, Wikipedia [96] defines creativity as "the production of novel, useful products" [61], while also acknowledging the existence of significant variations among different definitions. Similarly in chess, professional players often share beautiful combinations from their games, and some games are considered aesthetically pleasing. However, articulating the specific reasons why these games or combinations are considered beautiful remains challenging, even for experts [48]. Nevertheless, a few aspects of chess positions had been associated with creativity in the chess literature [66, 2, 48]. Its important to note that while these aspects are not necessary or sufficient conditions for creativity, they are often associated with it.

**Novelty:** Creativity is the generation or expression of new ideas perceived as valuable or interesting. A key aspect of this definition is the subjective nature of "newness," which depends on the context—an idea might be novel to an AI agent, specific individuals, or universally. While genuine invention occurs, most chess creativity involves applying existing concepts in innovative ways. For instance, grandmasters engage in creative work during opening preparation by seeking lines that are challenging, suit their playing style, or exploit potential inaccuracies in computer evaluations. This form of finding contextually effective new ideas aligns closely with the concept of "novelty" in AI terminology.

To quantify the novelty of generated chess positions we propose several metrics. First, we measure syntactic similarity using the Levenshtein distance [47] calculated over the FEN string representing the board (piece configuration) and the Principal Variation ([12], PV) derived from StockFish . For the board distance, the Levenshtein distance roughly corresponds to the number of pieces that have

to be changed to get from one position to the other. For any two board positions b and b', the board distance is defined as $dist^{board}(b, b') = \text{Levenshtein}(\text{FEN}(b), \text{FEN}(b'))$ and the PV distance is defined as $dist^{PV}(b, b') = \text{Levenshtein}(\text{PV}(b), \text{PV}(b'))/\max\left(length\left(PV(b)\right), length\left(PV(b')\right)\right)$

To further capture the distance in the semantic space, we also leverage the generative model's uncertainty – sequence-level entropy as a metric for diversity. High entropy suggests the model finds the current state less predictable, potentially indicating a more novel, complex, or unusual semantic context compared to low-entropy states representing common patterns. Take an auto-regressive transformer as an example, for a model $\pi_\theta$ and a board sequence string $s_T$, the sequence-level entropy is calculated as the average token-level entropy: $H^{\text{seq}}(s_T) = \frac{1}{T}\sum_{i=0}^{T-1} H(s_{i+1}|s_{0:i})$, where $s_i$ indicates the i-th token in the sequence and $s_{0:i}$ indicates the sequence of tokens.

**Counter intuitiveness.** In chess, intuition often obscures advantageous yet seemingly unsound moves like sacrifices or tempo loss [66]. Players may overlook certain options or not even consider them [48]. As formal logic alone can also be misleading [66], counter-intuitive chess puzzles are considered creative because they demand players to think unconventionally, and question their intuition [2].

To quantify a chess move's counter-intuitiveness, we compare evaluation scores from a chess engine using different search depths. A "shallow" search approximates an intuitive assessment, while a "deep" search serves as a proxy for a more accurate evaluation. This concept is extended through multiple searches with varied search budgets (t). The resulting data allows us to measure counter-intuitiveness by: the evaluation difference (Eq. (2)), the area under the curve (AUC) of this difference over search time (Eq. (3)), or the critical search point where the evaluation becomes stable (Eq. (4)).

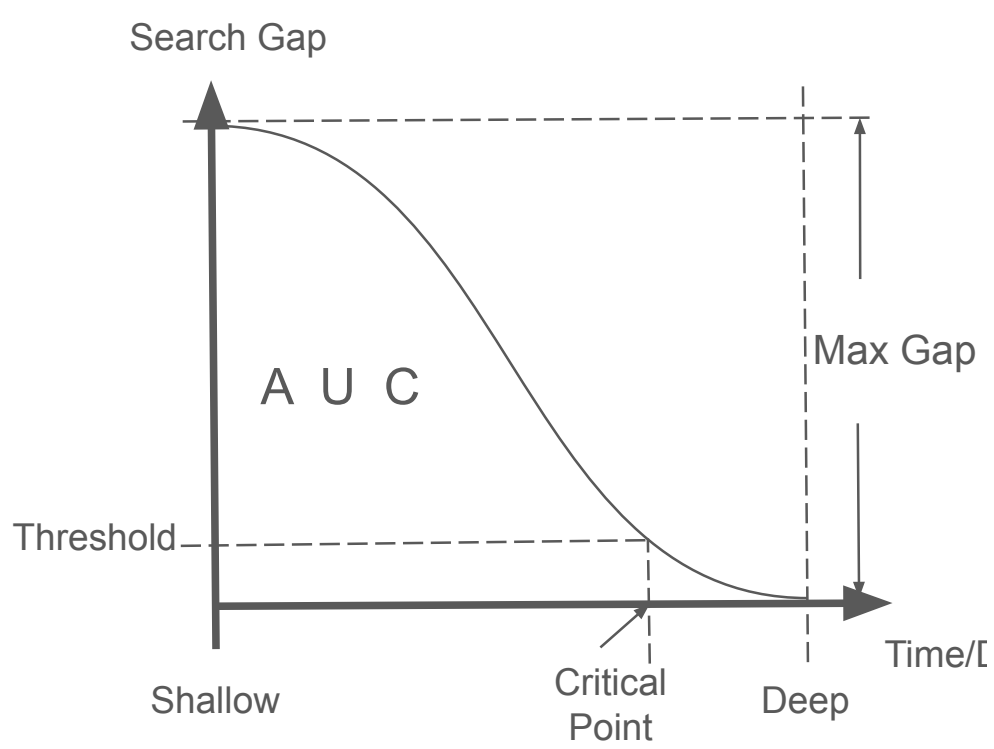

$$\mathbf{v}_{gap} = |\mathbf{V_T} - \mathbf{V_0}|, \tag{2}$$

$$\mathbf{v}_{auc} = \int_{t=0}^{t=T} |\mathbf{V}_T - \mathbf{V_t}|\, dt, \tag{3}$$

$$\mathbf{v}_{cp} = \min t, \text{subject to } |\mathbf{V}_T - \mathbf{V_t}| \leq \tau, \tag{4}$$

$$\mathbf{r}_{cnt} = \sum_i \mathbf{w}_i \mathbf{v}_i, \qquad \mathbf{I}_{cnt} = \mathbf{r}_{cnt} > \tau_{cnt}. \tag{5}$$

Figure 3: Counter-intuitiveness calculation visualization and equations.

More broadly, we can represent a counter-intuitiveness score $\mathbf{r}_{cnt}$ as a linear combination of various search statistics $\mathbf{v}_i$ (Eq. (5)). We will discuss candidates for $\mathbf{v}_i$ in Appendix F and a method for determining the weights $\mathbf{w}_i$ by tuning them on a curated Golden Set of counter-intuitive positions collected from diverse sources in Section 3.1. $\mathbf{I}_{cnt}$ will serve as a counter-intuitiveness indicator.

**Aesthetics.** Chess beauty, like creativity, is an elusive concept, but it is often associated with specific patterns, structures, or themes within the game. The notion of introducing order into a concept like creativity, which is by definition opposed to order, seems paradoxical, but there is logic in this madness. Creativity is not such a mysterious non-consequent and unfathomable process as it appears to many people [2]. This is because many of these structures are designed to defy some human heuristics for what constitutes a good move (like not losing material or tempo). Other structures may be simply appealing to human players. Previous research has explored classifying chess positions by aesthetic themes. Notably, Iqbal et al. [38] identified 17 such elements and showed that a corresponding score function aligned positively with expert aesthetic judgments. Lichess also defines themes for their puzzles. Our work combines elements from both these sources, along with themes drawn from chess literature [2, 48]. Using this consolidated set, we developed detectors to automatically classify puzzles across diverse aesthetic themes, as detailed in Appendix I.

## 2 Methods

**Training discrete generative models.** A chessboard has 64 squares that are either empty or occupied by one of six chess pieces. We represent it using the Forsyth-Edwards Notation ([13], FEN) as a

discrete sequence and adopt a chess-specialized tokenization [71] with a vocabulary of 31 tokens. Using this representation, we can use established discrete generative models to generate chess puzzles. The models were trained exclusively on the Lichess Puzzler [52] dataset (without any pre-training), excluding any chess compositions or other datasets. We experimented with different candidates including auto-regressive transformer [91], MaskGIT [11], latent diffusion model [70, LDMs] and masked diffusion model [77], more details can be found in Appendix D.

### 2.1 Reinforcement Learning from puzzle feedback

RL [84] enables us to further train our generative model to generate better chess puzzles by verifying the puzzle metrics (Section 1). For implementation simplicity, we focus on the auto-regressive transformer and utilize a critic-free variant of Proximal Policy Optimization [73, PPO]. This version estimates the value/advantage function using Monte-Carlo samples [50, 1, 75, 43, 33].

**Puzzle generation as an RL problem.** Unlike scenarios with per-step rewards (e.g., navigating a maze or playing an Atari game step-by-step), the relevant quality metrics for a chess puzzle can only be comprehensively evaluated after the entire puzzle has been generated. This feedback shows resemblance to the paradigm of Reinforcement Learning from Human Feedback [14, 62] and reasoning models that are trained with outcome-based reward [30, 41]. In both scenarios, the model generates a complete output (a puzzle or a text response), and a reward signal, derived from evaluating this entire output (e.g., based on puzzle quality checks or human preference ratings/answer correctness). Thus, puzzle generation with an autoregressive transformer aligns with a token-level Markov Decision Process (MDP) with outcome reward. In this view, the state $s_t = (x_1, ..., x_t)$ is the sequence generated up to step t, and taking an action $a_t$ corresponds to selecting the next token $x_{t+1}$ from the vocabulary V. The transition is deterministic: $s_{t+1} = s_t \oplus x_{t+1}$ (concatenation). The transformer itself acts as the policy $\pi(a_t|s_t) = P_{transformer}(x_{t+1}|s_t;\theta)$, providing the probability for the next token. Rewards are typically sparse, with zero intermediate reward and a final outcome reward $R_{outcome}$ assigned to the complete sequence $s_T$, reflecting puzzle's quality or validity.

We define our outcome reward $R_{outcome}(s_T)$ with the **uniqueness** and **counter intuitiveness** shown in Eq. (1) and Eq. (5). Specifically, a position is considered a qualified puzzle and receives a reward of $+1$ only if it satisfies uniqueness standard (Eq. (1)) and its counter-intuitiveness score ($\mathbf{r}_{cnt}$ in Eq. (5)) surpasses a threshold $\tau_{cnt} = 0.1$. Unqualified but legal position receives a reward of $0$. Positions identified as illegal are penalized with a reward of $-2$, such that the final reward is

$$R_{\text{outcome}}(s_T) = \begin{cases} 1, & \text{If } \mathbf{r}_{uni} \geq \tau_{uni} \wedge \mathbf{r}_{cnt} \geq \tau_{cnt} \\ 0, & \text{if} \quad \text{Legal} \quad \text{else} \quad -2. \end{cases} \tag{6}$$

**Realism.** Ensuring the realism of generated chess puzzles is crucial. Unlike puzzles derived from actual games, composed or computationally generated puzzles can sometimes feature unrealistic positions. While the precise impact of realism on puzzle appeal or educational value is debatable, it's often considered desirable. Previous work [37] addressed this via post-generation modifications and will be discussed in Appendix A. In contrast, we employ several RL-based techniques, drawing inspiration from RLHF practices [62] to maintain realism and prevent deviation from the the lichess dataset with three mechanisms: (a) KL Divergence constraint: We incorporate a token-level Kullback-Leibler (KL) divergence penalty $r_t = -KL(\pi_\theta(a_t|s_t)|\pi_{pretrained}(a_t|s_t))$ into the reward function. This discourages the RL policy $\pi_\theta$ from straying far from the policy trained on the lichess data $\pi_{lichess}$, thus preserving similarity to the training data distribution. (b) Training data mixture: We collect 100k high-quality examples filtered from the lichess dataset using the $R_{outcome}$ check in Eq. (6). These puzzles are used to seed the replay buffer discussed in the next sub-section. (c) Piece regularization. We find that the model can easily learn to artificially inflate reward and entropy by adding pieces (e.g., two white queens or three black knights). Thus, we zero out the board's reward if the count of any piece exceeds its count in the beginning of the game.

**Creating novel puzzles with diversity-filtering RL.** The RL setup detailed above, can in principle, be satisfied by an agent that produces a single high reward puzzle with very little or no diversity. As we will soon see in Appendix H.1, this indeed happens frequently when maximizing reward, even with explicit entropy regularization. To address this, we leverage novelty filters. First, we check that a generated position is novel by measuring its board and pv distance against all other qualified positions within the same batch (intra-batch) and preceding batches (inter-batch). A position $b$ passes the novelty test if $I_{\text{board}} = \text{dist}^{board}(b) \geq \tau_{board}$ and $I_{\text{PV}} = \text{dist}^{PV}(b) \geq \tau_{PV}$. We maintain a replay buffer to store preceding qualified and novel samples and subsample a small set of it for each inter-batch checking. In each RL step, we sample 16 positions from this replay buffer to enrich the qualified

samples in the training batch. Subsequently, we calculate all samples' sequence-level entropy in the training batch (including those from replay buffer samples) and mark a position as novel if the board's sequence-level entropy $H^{seq}(S_T)$ surpasses the predefined entropy threshold: $I_{\text{ent}} = H(b) \geq \tau_{ent}$. Note that this filter is different from entropy regularization because it only penalizes low-entropy generations, thereby avoiding the direct incentivization of high entropy generation. Qualified positions ($R_{\text{outcome}(s_T)} = 1$) that pass these diversity filters are considered novel, receive a reward of 1 and are added to the replay buffer. All other positions, regardless of their score, get a reward of 0 or -2 (if illegal). Thus the final reward with diversity filtering is defined as

$$R^{div}_{\text{outcome}}(s_T) = 1 \quad \text{If} \quad R_{\text{outcome}}(s_T) = 1 \wedge I_{\text{board}} \wedge I_{\text{PV}} \wedge I_{\text{ent}} \quad \text{else} \quad R_{\text{outcome}}(s_T) \tag{7}$$

## 3 Experiments

We begin with a study of the counter intuitiveness reward component in Section 3.1, which is then used, in addition to novelty and uniqueness, to compare and select the best model in a supervised-training experiment in Section 3.2. Once we identified good generative models, we post-train them using RL to generate counter intuitivene puzzles. We ablate and analyze the RL objectives in Section 3.4. We then analyze the creativity of our generations by studying their aesthetics in Section 3.3. We conclude with a human expert study comparing our generations with baselines in Section 3.5. Lastly, we present our best cherry picked and auto-selected generations in a booklet that can be found in the supplementary. We refer the reader to Appendix D for implementation details.

### 3.1 Tuning the counter-intuitiveness reward $\mathbf{r}_{cnt}$

We begin our experiments with a study of the counter intuitiveness component. We extract counter intuitiveness features from the search statistic of chess engines and then compute the reward as a weighted combination of these features. Explicitly, let $\mathbf{v}_i(s)$ be a vector of search statistics compute for a chess position $s$ then, the counter intuitiveness reward $r_i(s) = \mathbf{w}_i^T \cdot \mathbf{v}_i(s)$, where the weight vector $\mathbf{w}_i$ contain values in the $[0, 1]$ interval. The subscript $i$ represents a reward candidates.

**The Golden Set .** The weights $\mathbf{w}_i$ are tuned on a small, manually curated set of 39 positive and 45 negative TRAIN examples. The resulting weights were subsequently evaluated on a similar, independent TEST set containing 21 positive and 20 negative examples. These examples are chosen to capture human intuitions about the counter-intuitiveness aspect of chess puzzles that human players find appealing or interesting, and are selected from a variety of sources including books [66, 2, 48], the Lichess Puzzler dataset, and preliminary samples generated from our AR model after filtering for uniqueness. See Table 8 and Table 9 in the appendix for the examples. Given a particular $\mathbf{w}_i$ and positive and negative examples $\{s_i\}_{i=1}^{n}$, we first sort the examples using the counter-intuitiveness reward to produce a sorted dataset $D'$. We then compute the Average Precision (AP) metric [97] as:

$$\text{AP}_i = \frac{1}{npos} \sum_{k=1}^{n} \text{Precision}(k) * I_{\text{pos}}(k), \qquad \text{Precision}(k) = \frac{1}{k} \sum_{j=1}^{k} pos(j), \tag{8}$$

where $npos$ is the total number of positive examples, $I_{\text{pos}}(k)$ is an indicator function equaling 1 if the example at rank $k$ in the sorted set $D'$ is a positive one, and Precision($k$) is the precision at cutoff $k$, i.e. the proportion of positive examples at rank $k$. Intuitively, the better the counter-intuitiveness score is at ranking the positive examples ahead of the negative ones, the higher the AP metric should be. In practice, we tune the weights $\mathbf{w}_i$ to maximise the AP metric using a form of random search on a discretized grid with step size of $0.1$, i.e. $0.0, 0.1, 0.2, ..., 1.0$.

| | TRAIN | TRAIN+TEST | TEST |
|---|---|---|---|
| Uniqueness | 0.4768 | 0.4326 | 0.3788 |
| (1) Search Features | 0.6619 | 0.6222 | 0.5487 |
| (2) + SF through time | 0.7280 | 0.7312 | 0.7401 |
| (3) + AZ through time | 0.7475 | 0.7403 | 0.7360 |

Table 1: The average precision (Eq. (8)) of counter-intuitiveness score functions.

Table 1 presents a comparison of the average precision achieved by various candidate counter-intuitiveness scoring functions on our curated Golden Set . Each scoring function was optimized using grid search to maximise average precision on the TRAIN subset. Each candidate score function

also leverage more StockFish and AlphaZero features than the ones before it. We selected the best performing configuration on the TEST set, (2) Search Features + SF through time, as our counter-intuitiveness. This configuration yielded sparse weights, with only two features having non-zero values: StockFish critical point (Eq. (4)) with a weight of $0.8$ and normalized negative capture material (the negative value of captured material, weighted according to standard valuation ([95], divided by 9) with a weight of $0.1$. A complete list of weights, other tuned configurations, details on the uniqueness baseline, correlation analysis with the lichess rating and popularity scores and the average precision of specific search features can be found in Appendix F.

### 3.2 Generative models

**Models.** For autoregressive transformer, MaskGIT and masked diffusion model, we leverage the same tokenization and 200M parameter transformer from [71]. The model is a decoder-only transformer with causal masking, post-normalization and SwiGLU [76]. For the latent diffusion model experiment, we investigate the impact of different chessboard representations on the generative model's performance. Since latent diffusion models typically operate on 2D image data, we explore representing the chessboard in a 2D format. We first train a VAE model on the same puzzle dataset. For the further diffusion model training, we employ the U-net transformer architecture from [67] with the frozen VAE encoder. To ensure comparability, the U-Net Transformer's hyperparameters are configured to yield approximately 220M parameters. More details can be found in Appendix D. For each model the checkpoint with the lowest test loss is selected for the final evaluation. We compare our models with two **baselines**: the lichess dataset and a standard game dataset (2024-11) [51].

| | Lichess-Games | Lichess-Puzzles | Transformer | MaskGIT | Latent Diffusion | Masked Diffusion |
|---|---|---|---|---|---|---|
| Legal | **100.00** | **100.00** | 99.07 | 97.61 | 97.14 | 99.72 |
| Unique | 4.32 | **95.25** | 23.44 | 27.76 | 22.31 | 30.89 |
| Counter intuitive | 0.71 | **2.25** | 0.93 | 1.44 | 0.85 | 1.11 |
| Puzzle | 0.03 | **2.14** | 0.22 | 0.40 | 0.19 | 0.34 |

Table 2: **Generative models benchmark.** Metrics are reported as percentages (%), for 1M generated positions (generative models) and 1M sub-samples (Lichess datasets). Puzzle refers to positions that meet both the uniqueness and counter-intuitiveness criteria.

**Results.** Table 2 and Table 3 summarize our benchmark results for puzzle quality and novelty scores, respectively. As expected, the lichess dataset, which served as the training data, exhibits the best performance across all metrics, acting as a practical upper bound. All four generative models significantly outperform the standard game dataset baseline. Additionally, the high board/PV distance w.r.t the lichess dataset, as well as the self metric, suggest that they are capable of generating novel and diverse positions. Among the generative approaches, Masked Diffusion and the Transformer achieve high legality ratios, while MaskGIT and Masked Diffusion excel in uniqueness and counter-intuitiveness. The Transformer and MaskGIT demonstrate strong performance on the various diversity metrics. Latent Diffusion, however, yields relatively moderate results across most indicators.

Notably, it is challenging to declare a single superior model, as none dominates across all scores, perhaps because the metrics present trade-offs. For instance, inspecting Fig. 8 in the appendix, reveals that the AR transformer achieves higher puzzle scores and less novelty as it is trained for longer at the cost of overfitting to the training data. Our checkpoint selection strategy, based on minimizing test loss on a held-out set, might also influence these relative performances. We observed that the AR model's test loss begins to increase after approximately 15k training steps, leading to earlier checkpoint selection. In contrast, MaskGIT and the diffusion models did not show a similar increase in test loss even after 1 million training steps. Consequently, this selection strategy might curtail the AR model's training before it reaches its peak puzzle score potential.

### 3.3 Creativity

**Novelty.** Inspecting Table 3 indicates that the datasets sampled from the generative models exhibit both novelty (when compared to the Lichess data) and internal diversity (when compared to other samples from the same model). For instance, the top model, MaskGIT, achieves a self-comparison board distance of 14.57, surpassing the diversity baseline of the Lichess puzzles (11.914). Its novelty score relative to Lichess is 11.44, which is close to the Lichess baseline (11.914), suggesting its outputs are distinct from the training data. The board distance roughly equates to the number of piece changes separating a generated position from its nearest neighbour. Values around 10 thus suggest relatively high novelty. To explore this qualitatively, the booklet presents each generated puzzle next to its three closest matches from Lichess, allowing readers to visually assess its distinctiveness.

| | Lichess-Game | Lichess-Puzzle | Transformer | MaskGIT | Latent Diffusion | Masked Diffusion |
|---|---|---|---|---|---|---|
| Board distance (Lichess) | 8.984 | N/A | 9.013 | 11.444 | 8.928 | 8.528 |
| Board distance (Self) | 10.715 | 11.914 | 11.486 | **14.570** | 11.393 | 10.859 |
| PV distance (Lichess) | **0.995** | N/A | 0.747 | 0.855 | 0.758 | 0.637 |
| PV distance (Self) | **1.118** | 0.991 | 0.949 | 1.084 | 0.947 | 0.837 |

Table 3: **Novelty benchmark.** For each position, we compute the distance to the Lichess dataset and the distance to other samples from the same model ('Self'). The final metrics are averaged over all unique positions. We use 100k samples per method for Lichess-distance and 40k (the number of unique positions in the Lichess-Game's 1M positions) samples for self-distance.

**Emergence of Aesthetics.** Fig. 4 illustrates how the positions in each data set are distributed across various aesthetic themes (detailed in Appendix I), whereas Fig. 18 and Fig. 19 in the appendix show the distribution across positions with a unique solution, and across positions with a unique counter intuitive solution respectively. Puzzles from the Lichess dataset are shown in blue, those from our AR generative model in orange, and those from the RL agent in green. Aesthetic themes were identified only after the puzzles had been created, so they did not inform or guide the training process.

Analysis of the figures reveals diverse aesthetic themes across all three datasets. Their appearance in the Lichess dataset indicates that aesthetically pleasing positions naturally occur in human games, without deliberate creation. However, theme prevalence varies, with certain mate patterns like Boden's Mate and Arabian Mate being less frequent in Lichess data, potentially due to the dataset's filtering for unique positions where multiple winning moves can exist in mate scenarios. The figures also show that the AR model largely mirrors the original dataset's distribution of aesthetic themes; some themes are even more common in AR samples due to sampling variations. Notably, the RL distribution aligns with AR and Lichess, despite not being explicitly trained for aesthetics.

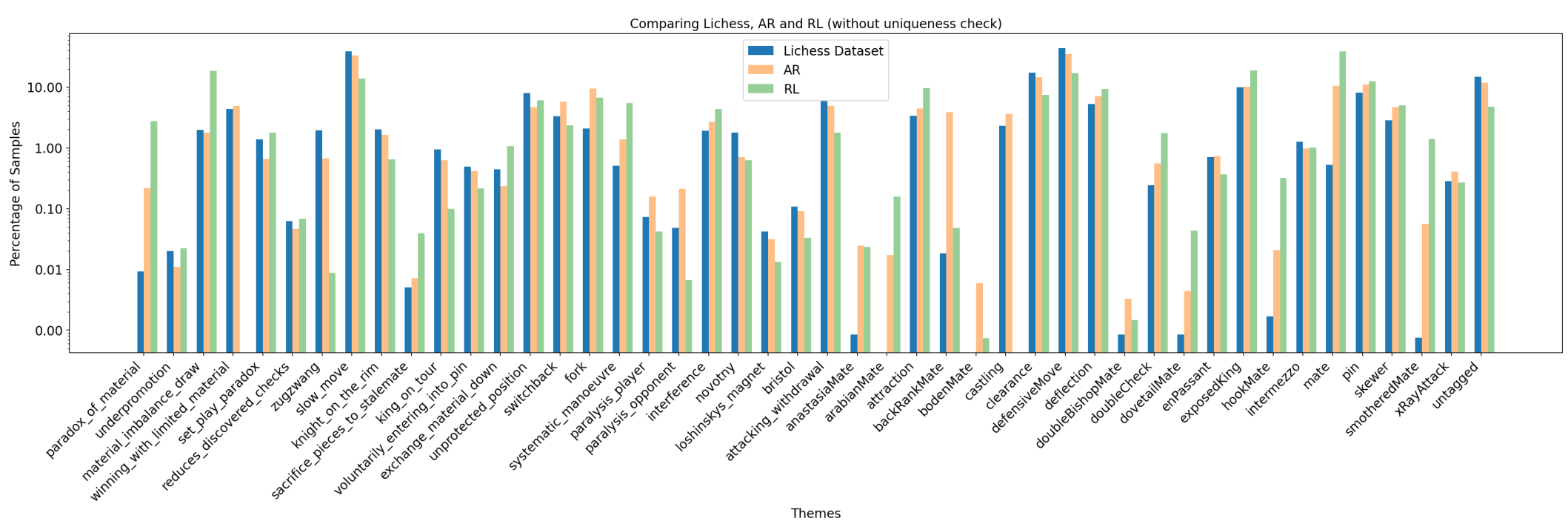

Figure 4: Distribution of aesthetic themes in chess positions across different datasets.

### 3.4 Reinforcement Learning

For simplicity, we conduct the RL training phase with the auto-regressive transformer. Our first results performed Reward maximization with the reward in Eq. (6) and an entropy regularization, starting from the lichess-trained supervised model. However, as the results in Appendix H.1 suggest, this setup suffered from entropy collapse and non realistic positions. Appendix H.2 and Appendix H.3 explain how we solved these issues respectively. Further ablation studies can be found in Appendix H.4.

We conducted our RL runs and compared the performance of the RL model with that of the generative model baselines that were trained only on lichess. Fig. 5.(a)-(c) present the results for two RL variants: one trained solely with RL updates and another incorporating auxiliary high-quality Lichess puzzle data during training. Fig. 5.(a) demonstrates that RL significantly enhances the model's performance, achieving an average puzzle score that surpasses even the strongest baseline, the Lichess puzzle dataset itself. Interestingly, despite the uniqueness score being a component of the reward function (via the qualification threshold), the uniqueness ratio remains relatively stagnant around 20% in Fig. 5.(c). Instead, the primary gains stem from a substantial increase in the counter-intuitiveness score. Consequently, the overall rate of qualified puzzles (passing both uniqueness and counter-intuitiveness thresholds) generated by our RL model becomes comparable or slightly superior to the Lichess puzzle baseline (Fig. 5.(b)). It is crucial to note that this parity is achieved even though the Lichess puzzle baseline has been explicitly filtered for uniqueness in advance while RL does not.

Fig. 5.(d)-(h) also illustrates the evolution of diversity metrics throughout the training process. Across all measured aspects (e.g., board distance, PV distance, entropy), the generated puzzles demonstrate

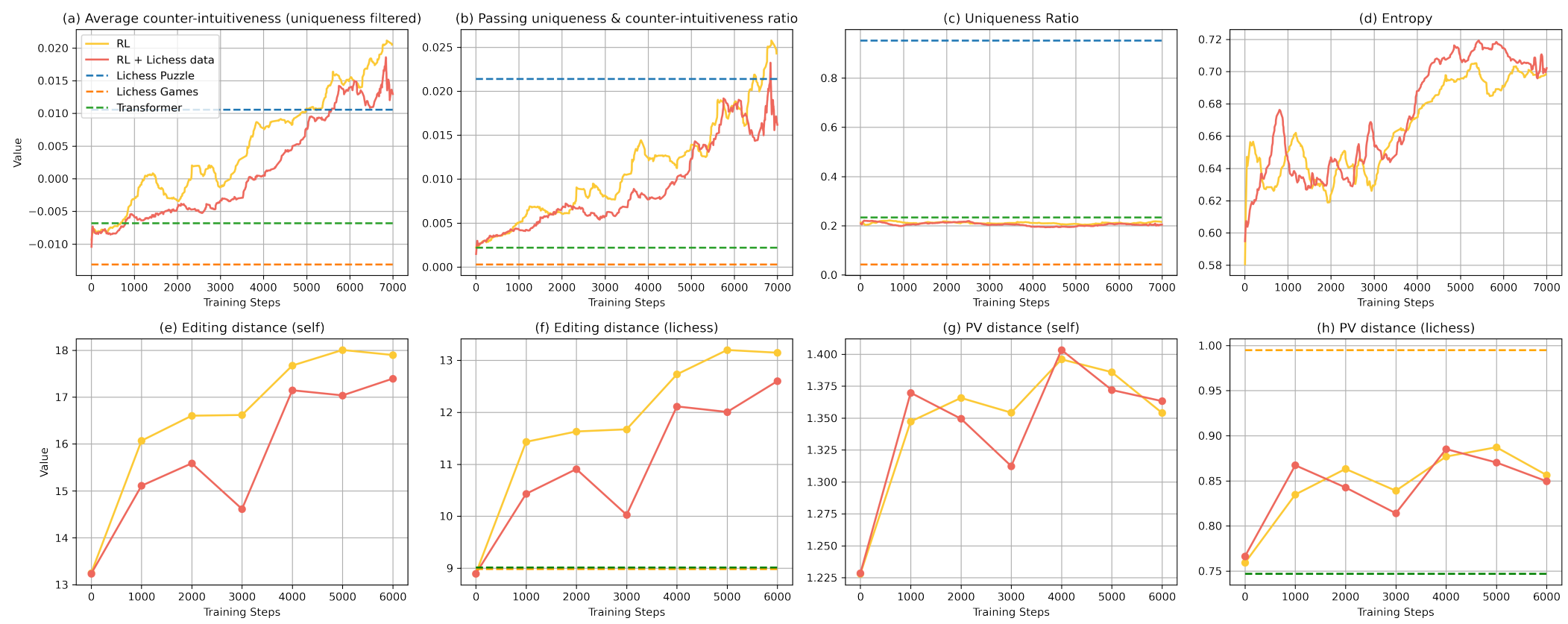

Figure 5: RL run surpasses the baselines in puzzle metrics and keeps improving diversity through the training. We smooth the curve in (a)-(d) for better visualization.

a consistent upward trend as training progresses. This observation strongly validates the efficacy of our proposed diversity filtering mechanisms. The final RL model not only produces a higher rate of qualified puzzles but also generates results that are significantly more diverse than the initial lichess-trained supervised model or models trained with less sophisticated RL setups.

### 3.5 Human study and booklet

We further conducted human study with 8 chess experts (Lichess Elo 2000-2400) to evaluate the quality of the generated puzzles and compare them with creative puzzles recognized in the chess literature (Full results are in supplementary materials). The study included puzzles from four different sources. (1) Lichess Puzzler (2) RL (auto-selected) (3) Booklet (4) Chess books (human selected) [2, 48, 66]. 10 puzzles were selected from each source and were presented to the raters in random order. The selected puzzles were required to satisfy our puzzle check, i.e., they had to pass the counter-intuitiveness and uniqueness checks. We also verified that the book puzzles satisfy the theme function. The booklet puzzles were collected from the outputs of various methods. We then filtered the puzzles based on aesthetic themes and sorted the positions in each theme by the counter intuitiveness reward. Experienced players (FIDE rated 2200-2300) reviewed the top 50 puzzles from this process and selected a small number as potential candidates for the booklet, more details can be found in the supplementary material. For each puzzle we include the three closest puzzles from the Lichess training dataset, identified using edit distance, and reference book puzzles for each theme.

To reduce bias, the Lichess and RL puzzles were selected from a sub sampled set of exactly 1 million puzzles. Furthermore, to maintain an unbiased estimation across 7 aesthetic themes were selected for the study (sacrifice, underpromotion, attacking withdrawal, novotny, interference, unprotected position and knight on the rim is dim). The puzzles for each source were selected with the same aesthetic theme distribution as the other methods. The themes were selected such that all the methods could satisfy these counts. For example, the Lichess data did not include all the themes from the books, and some of the book examples were not unique or did not satisfy the theme when analyzed with StockFish . Given the aesthetic theme counts, the puzzles from each method were ranked according to the counter intuitiveness-reward, and the top samples were selected for the study. The puzzles were randomly shuffled to further reduce bias. They were rated from 0 (poor) to 3 (perfect) across five different metrics: realism, difficulty, creativity, fun, and counter-intuitiveness.

| Puzzle Source | Realism | Difficulty | Creative | Fun | Counter Intuitiveness |
|---|---|---|---|---|---|
| Book puzzles | **2.74 $\pm$ 0.45** | **2.6 $\pm$ 0.33** | 2.39 $\pm$ 0.39 | 2.22 $\pm$ 0.3 | 2.05 $\pm$ 0.2 |
| Lichess | 2.53 $\pm$ 0.41 | 2.41 $\pm$ 0.34 | 1.7 $\pm$ 0.2 | 1.49 $\pm$ 0.17 | 1.7 $\pm$ 0.18 |
| RL (auto-selected) | 2.28 $\pm$ 0.38 | 2.59 $\pm$ 0.24 | 2.1 $\pm$ 0.37 | 1.96 $\pm$ 0.19 | 1.98 $\pm$ 0.19 |
| Booklet | 2.59 $\pm$ 0.33 | 2.39 $\pm$ 0.15 | **2.48 $\pm$ 0.25** | **2.56 $\pm$ 0.24** | **2.12 $\pm$ 0.2** |

Table 4: Avg ratings from 8 chess experts from our rating task.

**Results.** The study's findings, detailed in Table 4 with average scores and standard deviations across five metrics, reveal distinct puzzle performance. In terms of Realism and Difficulty, Book puzzles ranked highest, followed by Booklet puzzles, then RL (auto-selected) puzzles, and finally Lichess

Puzzler puzzles. For Creativity, Fun, and Counter-intuitiveness, Booklet puzzles led, with Book puzzles next, followed by RL (auto-selected) puzzles, and Lichess Puzzler puzzles again ranking last. Notably, these results are significant because they demonstrate that AI-generated puzzles surpassed the data they were trained on and even human-composed puzzles in certain aspects.

To set an even higher standard of evaluation, we invited three world-renowned experts to provide an in-depth review of our booklet: International Master for chess compositions Amatzia Avni, Grandmaster Jonathan Levitt, and Grandmaster Matthew Sadler. The experts' assessments were exceptionally positive, commending the puzzles for their creativity, novelty, and aesthetic merit. They regarded the collection as a pioneering advancement in the human-AI partnership for chess composition. The complete, detailed review is provided here.

## 4 Discussion and limitation

This paper presents a significant step towards generating creative chess puzzles using GenAI and RL techniques. The research successfully demonstrates that an RL framework, equipped with novel reward functions promoting uniqueness, counter-intuitiveness, diversity, and realism, can substantially enhance the generation rate of such puzzles. This is particularly compelling because existing methods, like those used by Lichess, depend on a finite stream of human games and involve intensive re-analysis of positions primarily for uniqueness. We found that among Lichess's unique puzzles, only about 2.1% align with our definition of counter-intuitive, yielding a limited set of approximately 20,000 new creative puzzles per year. Our AI-driven approach surpasses this, achieving a 2.5% rate for unique, counter-intuitive puzzles. Furthermore, the efficiency of generating candidates with our comparatively small generative models (e.g., 200M parameters ) contrasts favorably with the engine analysis Lichess performs per puzzle. This underscores AI-powered generation as a scalable and promising path to continuously deliver a rich and varied stream of creative chess puzzles.

A critical observation from our experiments is that achieving a high reward does not automatically equate to high human-perceived creativity. The models exhibited tendencies towards "reward hacking," such as entropy collapse (generating the same puzzle repeatedly), producing unrealistic samples (e.g., boards with excessive pieces) or exploiting chess engine weaknesses (making the position artificially more complicated for engines, for example, by allowing the defender to deliver meaningless checks). While diversity filtering and realism constraints helped mitigate these issues, there remain a gap in our understanding of creative chess positions due to the subjective, multifaceted nature of human creativity. The iterative refinement of theme-detecting functions also highlighted the difficulty in encoding nuanced human intuition about what makes a theme's presence "significant" or "creative". This emphasizes the necessity of human expert evaluation as the arbiter of creativity. The human evaluation further corroborates these findings, with experts rating the booklet puzzles as more creative, fun and counter-intuitive than the book compositions, while the auto-selected puzzles were rated lower than the book compositions but higher than the typical Lichess puzzles, and approaching the quality of classic compositions. The creation of a chess booklet featuring curated puzzles and the subsequent human expert study were invaluable components of this research. This feedback loop is essential for validating the quality of generated puzzles and refining the generation process.

For future work, we believe key components of our methodology are broadly generalizable and could be applied beyond the domain of chess in computational creativity. For instance, the counter-intuitiveness reward that contrasts shallow and deep search evaluations, is not specific to chess. This principle of rewarding 'surprise' or non-obvious solutions could be readily transferred to other domains that rely on search or iterative reasoning, such as the game of Go, automated theorem proving, or even prompting 'deeper thinking' in large language models. Furthermore, the 'entropy collapse' we observed (detailed in Appendix H) is a fundamental challenge when using RL for novelty-driven tasks. Our diversity-filtering framework, which actively maintains novelty and prevents distributional collapse, offers a generalizable blueprint for training generative models to produce a rich and varied stream of creative outputs in other complex fields.

## Acknowledgements

We would like to thank Demis Hassabis for discussion and feedback on our work. We thank Aleksei Ostapenko, Arpit Hamirwasia, Daniel Körnlein, John Reid, Joon Lee, Kola Adeyemi, Matteo Tortora, Preet Sardhara, Sachin Ravichandran, Sagar Jha, and Temirlan Ulugbek uulu for their participation in evaluating our chess puzzles through various rating tasks, and particularly for their engagement in our Human study (Section 3.5). We also thank them for their valuable discussions. The feedback from these tasks was instrumental in directing our work. We would also like to thank Lisa Schut for feedback on the booklet, David Abel for reviewing the paper, Simon Osindero, Arnaud Doucet and Clare Lyle for joining discussions and providing feedback, and Gabriela Fernandez-Cuervo for engaging with chess experts.

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

# A Related work

While much computational chess research has focused on enhancing engine strength, exemplified by engines like StockFish , AlphaZero [78], and Leela, a line of work has identified puzzles that remain challenging even for top engines [22, 65, 16, 80]. This has spurred the development of specialized solvers like Crystal, specialized methods [29] and creative problem-solving approaches [98]. In parallel, large-scale puzzle generation for platforms like Chess.com and Lichess typically extracts tactical sequences from games, prioritizing criteria such as solution uniqueness (often a single best move per turn, barring final mating moves ) and achieving a clearly winning state, usually defined by material gain or forced mate (chess.com blog). Assessing puzzle difficulty computationally involves heuristic models analyzing simulated human search tree features [81] and machine learning models trained on platform data [99, 57]. Furthermore, tools like the novelty grinder identify moves that are strategically or tactically sound but statistically rare in human play. Despite these advancements in analyzing solvability, difficulty, aesthetics, and novelty, the generation of puzzles that are specifically creative or counter-intuitive remains an underexplored area.

A particularly relevant line of prior research is the work by Iqbal et al. on the Chesthetica program [36, 38, 39, 37], which focused on computationally modeling and optimizing for human appreciation of beauty in chess. Chesthetica employed a detailed aesthetic score built upon 17 terms, combining core aesthetic principles (like sacrifice, paradox, economy) with specific tactical themes (such as pins and zugzwang) derived from chess literature. This score was validated against expert human judgments, comparing composed studies to game sequences. Significantly, this research also found that the aesthetic score did not positively correlate with computational complexity, measured via engine solving time, indicating that difficulty and beauty are distinct dimensions [35]. Differing from the objective of generating broader unique positions as in Lichess Puzzler and the present work, Chesthetica's generation capability was primarily aimed at creating mate-in-x problems, utilizing realism heuristics. Its proposed refinement method involved applying seven sequential steps intended to preserve the core solution: 1) removing pieces, 2) substituting white pieces with weaker ones, 3) substituting black pieces with stronger ones, 4) moving the white king out of check, 5) seeking a shorter mate, 6) relocating the white king closer to other pieces, and 7) preventing major black pieces from being immediately capturable. It is noteworthy that the generation process itself was not guided by optimizing the aesthetic score; rather, the aesthetic evaluation functioned primarily as a filter applied after generation to select suitable outputs. This focus on generating realistic mate scenarios often resulted in compositions resembling endgames with relatively few pieces, as visually evidenced on the Chesthetica youtube channel.

The automated generation of puzzles for games more broadly has also been studied extensively in the field of procedural content generation (PCG)[89]. The Procedural content generation via machine learning (PCGML) [82, 88] community studies how to bring machine learning to bear on this problem, and many of these puzzles have been generated using RL-generative policies in (PCGRL) [45, 55, 69, 20]. There is a hope that similar algorithms could be used as a source of problems to form an increasingly complex curriculum to train generally capable agents [46, 21, 23], exhibiting an open-ended [34] stream of increasingly complex and novel problems which replicates the open-ended complexity of evolution [4, 5]. Given its origin, many approaches to achieving open-endedness use evolutionary algorithms [74, 93, 94, 63], especially building on the power of modern LLMs [92, 27, 56, 100, 25, 53, 26]. The integration of evolutionary methods is a promising avenue to extend our approach.

While this research primarily utilized Stockfish and AlphaZero for analysis and reward formulation, future work could delve deeper into improving measuring counter-intuitivness. For example, by incorporating insights from other chess engines like Leels or Maia [54], which is designed to mimic human play more closely. This could lead to puzzles that are not only counter-intuitive for strong engines but also align better with human notions of surprise or beauty.

## B The Lichess dataset

The lichess puzzles were generated from 300,000,000 analysed games from the Lichess database. Interesting positions were re-analyzed with Stockfish 12/13/14/15 NNUE at 40 meganodes. The resulting puzzles were then automatically tagged. The number of games played at the platform, and the number of puzzles through time can be viewed in Fig. 6.

A user-driven popularity metric for each puzzle, ranging from 100 (best) to -100 (worst), is calculated based on weighted upvotes and downvotes. The weighting considers factors like successful solves and the rating difference between the solver and the puzzle. Lichess Puzzler also assigns an Elo rating to each puzzle, treating each solving attempt as a rated game.

We train all our models purely on the public Lichess Puzzler dataset with cutoff date on 2024-12-01, with 4.4M data points. We split train and test set by 99% and 1%, resulting in 4.36M train samples and 44k test samples correspondingly. Note that the "FEN" column in the raw dataset is the position before the opponent makes their move, so we apply the first move in solution to get the puzzle board.

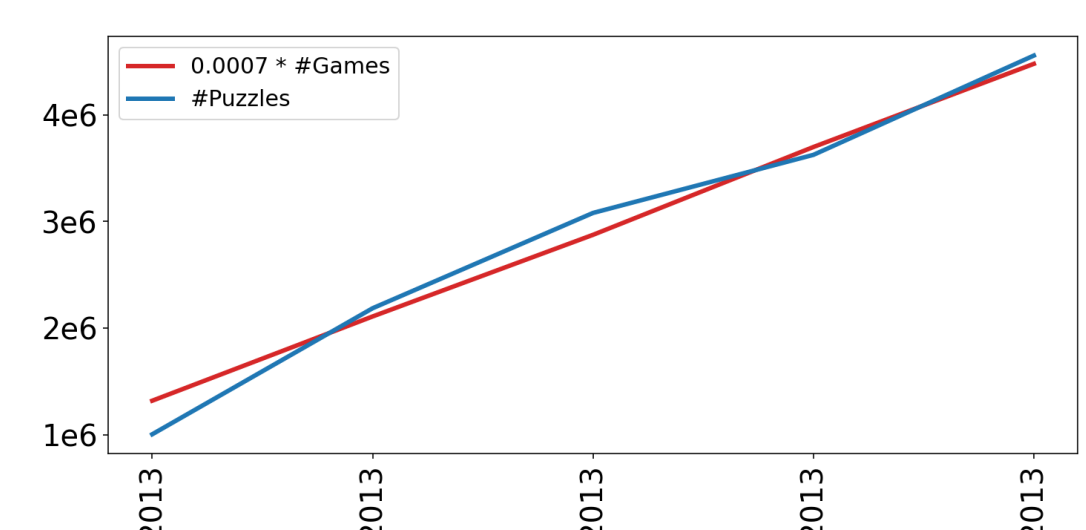

Figure 6: The amount of games played in lichess between 2021-2025 (red) and the amount of puzzles extract from them (blue). The amount of games is divided by 100 so its in the same scale as the puzzles.

## C Uniqueness

To verify that a chess position has only one solution we use StockFish and analyze its search results. We begin by confirming that the position has a single solution, using the approach from the Lichess Puzzler project, which utilizes StockFish analysis. The first step is to see if there's a checkmate sequence in the primary line of play within 15 moves.

- If **yes**: check that the puzzle player's main line-of-play is the only one that leads to checkmate. This check is then recursively applied by playing the top move from the main variation and then recomputing the variations from the resulting position. If any alternative mating sequence appears before checkmate, then it is not puzzle.
- If **not**: check that the puzzle player's main variation's chance of winning $w(a|s)$ is substantially greater than the second best variation, i.e. it is unique using Eq. (1) below. This check is then recursively applied by playing the top move from the main variation and recomputing the variation from the resulting position. If the criterion holds for at least the first move from the initial position, the it is considered a puzzle.

**Note** that in both cases, the puzzle criterion is only applied to the puzzle player's turn. For the opponent, the top move from the main variation is always played. The chance of winning for non-checkmate variation is compute using the equation below based on Stockfish's centi-pawn score.

Figure 7 evaluates our implementation of the Lichess puzzle uniqueness check against 1% of the Lichess puzzles dataset (approx 44K examples) and a set of 30K non-puzzle examples, for different values of uniqueness thresholds. The value of $0.5$ is used for experiments in this paper, where Lichess Puzzler uses $0.7$. There are other minor implementation differences between the Lichess Puzzler version and ours. For example, Lichess Puzzler ignored positions that don't win by enough. These modifications were ignored in our version for simplicity.

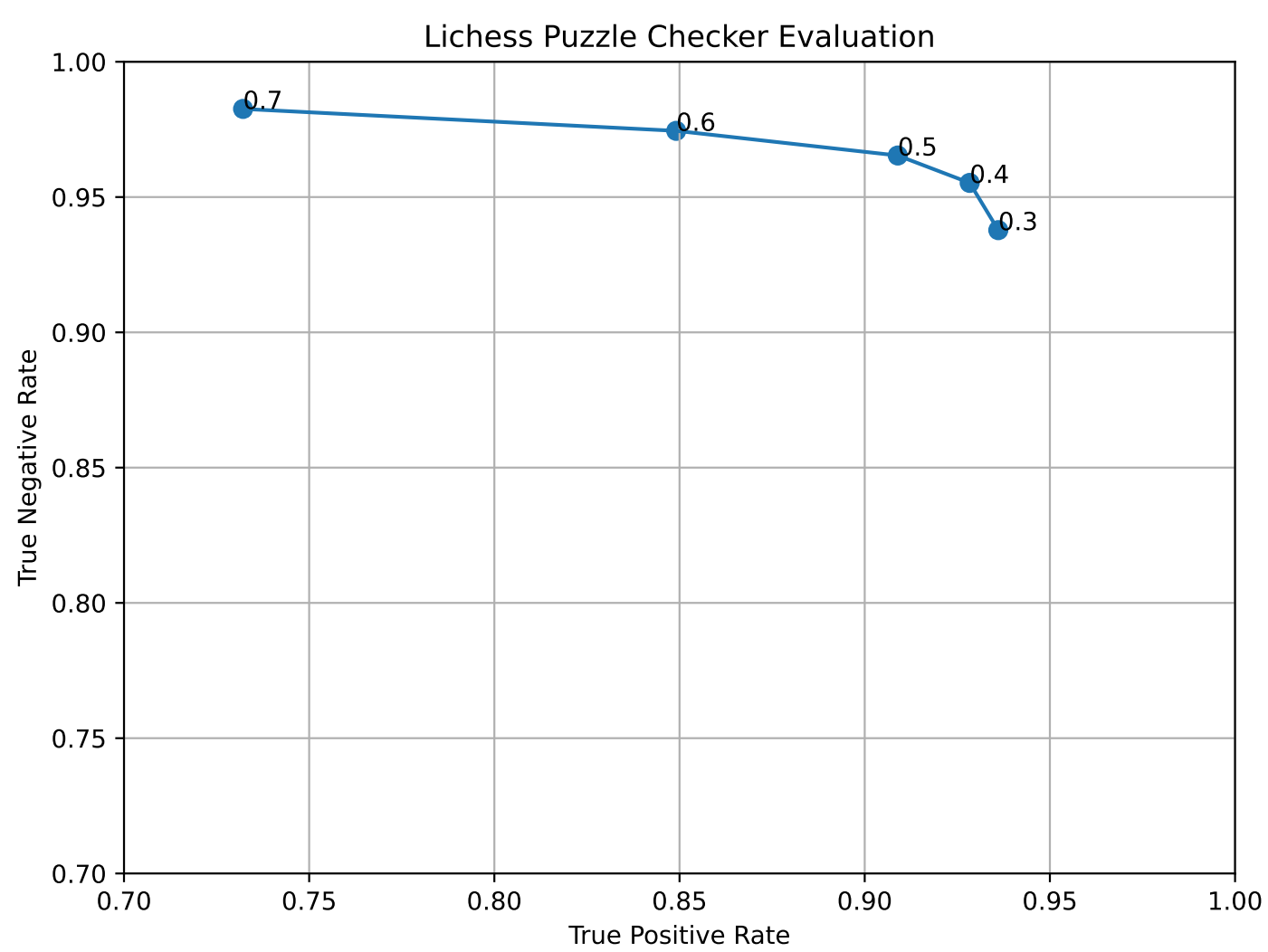

Figure 7: Evaluation of Lichess Puzzle Checker (Uniqueness) for different threshold values

# D Implementation Details

## D.1 Generative model implementation details

We detail our model and training implementation details.

**Autoregressive Transformer** We adopt the same architecture from [71], a decoder-only transformer with causal masking, post-normalization and SwiGLU [76]. We use 8 heads, 16 layers, and an embedding dimension of 1024, resulting in 200M parameters in total. We adopt the AdamW optimizer, with 1e-4 learing rate and 1e-4 weight decay coefficient. We conduct the training with 1024 batch size and 100k steps. Finally, we pick the checkpoint by measuring the test loss on our 44k test set.

**Latent Diffusion Model** For the latent diffusion model experiments, we investigated the impact of different chessboard representations on the generative model's performance. Since latent diffusion models typically operate on 2D image data, we explored representing the chessboard in a 2D format. Specifically, we padded the original sequence representation (length 77) to a length of 80 and reshaped it into an 8x10 grid. The first 8x8 section of this grid represents the chessboard squares, while the remaining 2x8 section encodes metadata such as the current turn, castling rights, and move number.

Latent diffusion models require an encoder to map the input into a continuous latent space. Therefore, we trained a separate Variational Autoencoder (VAE) to serve as this encoder for our 2D chessboard representation. For VAE training, the 8x10 grid input was first converted into an 8x10x31 one-hot tensor (corresponding to a vocabulary size of 31). The VAE was trained using a standard objective comprising a reconstruction likelihood loss and a KL divergence penalty. Following common practices in latent diffusion [70], we set the KL divergence coefficient to a small value (1e-3). The VAE training proceeded for 50,000 steps.

Upon completion of VAE training, its parameters were frozen. We then trained the diffusion model itself, which operates in the latent space generated by the VAE encoder. We employed a U-Net Transformer architecture, specifically adapting the design from [67]. The network features a downsampling path and an upsampling path. The downsampling path consists of three Cross-Attention 2D Downsizing blocks (each combining cross-attention and ResNet convolutional blocks) followed by one standard 2D Downsizing block (ResNet convolutional blocks only). The upsampling path mirrors this structure in reverse: one standard 2D Upsampling block followed by three Cross-Attention 2D Upsampling blocks. The channel dimensions progress through 160, 320, 640, and 640 across the stages. All attention mechanisms utilize 8 heads. This configuration resulted in a U-Net Transformer with approximately 220M parameters, making it comparable in scale to other generative model.

For training the diffusion model, we used the Adam optimizer with a learning rate of 2e-4. We used a batch size of 512. The training followed the Denoising Diffusion Probabilistic Models (DDPM) [32] framework. The diffusion model was trained for 300,000 steps.

**Masked Diffusion Model** We follow MD4 [77] to train a masked diffusion model by maximizing a variational lower bound of the log likelihood of data. The loss function is

$$-\log p_\theta(x) \leq \mathcal{L}_\theta(x) = \int_0^1 \frac{\alpha'_t}{1-\alpha_t} \mathbb{E}_{q(x^t|x^0)} \Big[ \sum_{i: x_i^t = [mask]} (x_i^0)^\top \log \mu_\theta(x^t)_i \Big] \, dt, \tag{9}$$

We reuse the transformer architecture from the AR model to parameterize the denoising network $\mu_\theta$, except that we remove the causal attention mask to use bidirectional attention. We randomly draw $t \sim \mathcal{U}(0, 1)$ to estimate the integral. Then we corrupt the clean data point (a sequence of length 77) by independently masking each element with probability $\alpha_t$, this corresponds to drawing a sample from $q(x_t|x_0)$ to estimate the inner expectation. Finally, we compute the desnoising loss via cross entropy at the masked positions and weight them by $\frac{\alpha'_t}{1-\alpha_t}$. We use a linear schedule $\alpha_t = 1 - t$ for our experiments.

**MaskGIT** Our MaskGIT model is the same as the training of our auto-regressive model, we choose the masking rate per batch by sampling from a uniform distribution $r \sim \mathcal{U}(0, 0.5)$, we then mask each token IID with probability $r$. During inference, the MaskGIT model works by iterative demasking. We find that the inference optimizations originally used in MaskGIT [11] for the image domain, prioritizing decoding order by a confidence in the generation, results in a bias for generating empty chess boards. Intuitively, if the most likely token at any board position is the empty square, so the

first several generated squares are likely empty. Conditional on this, it is much more likely much of the rest of the board will be empty. We address this by selecting which token to keep randomly, keeping one token at a time until the entire board has been generated.

### D.2 Generative model evaluation details

For the PV distance evaluation, we truncate the full principle variation sequence length to 6. We use 1M samples for the puzzle score calculation in Table 2. For Table 3, we use 100k samples per method for Lichess-distance and 40k samples for self-distance. We adopt 40k for self-distance calculation because the Lichess-Games dataset only have around 4% uniqueness ratio, which corresponds to 40k samples from 1M generated positions. We need to align the number of samples because we are calculating minimal distance rather than the average one.

### D.3 RL implementation details

For the diversity filtering, we set $\tau_{\text{board}}$ as 6 and $\tau_{\text{PV}}$ as 1. We find that the truncation length 6 used in generative model evaluation is too strict for selecting samples in RL, thus we reduce it to 1, leading to a check over board's first-move. For the final RL run, we lower the training learning rate to 3e-5 and we set the replay buffer diversity filtering subsample size as 2k.

## E Evolutionary Search

Complementary to generative models trained on pre-existing data, evolutionary methods provide us with the opportunity to go beyond the training data distribution and set up an optimization process that would explore the set of possible board configurations in relation to a given counter-intuitiveness objective, and possibly with additional imposed stylistic constraints. Our initial implementation of evolutionary search in the context of this task has been rather straightforward, in that we've not explored many of the perhaps more advanced or custom ideas that have been proposed in the literature over the years, having instead focused on the more 'basic' setup, where we have prioritized finding the right way of incorporating the score and the constraints to produce the desired outcome.

In our ES approach, we would run a flexible number of distinct ES 'workers', each running its own local ES process. These processes would not communicate between each other, rather producing independent outputs. Each worker would start by sampling a random non-puzzle chess position (or alternatively, a set of positions) from the provided pre-existing dataset. Different workers would initiate their local ES processes from different sampled seed positions, to facilitate diversity across the whole worker population. In principle, one could also alternatively start from a randomly generated position instead, though starting from sampled pre-existing position enables us to potentially stay closer to the original distribution – under a certain set of parameter settings.

Each ES process running on its own separate ES worker would run for a fixed pre-specified number of iterations, this being a parameter of the process. Each process would also have its own separate buffer. In each ES iteration, candidate solutions from the previous generation would be sampled from the buffer, proportionally to their counter-intuitiveness, uniqueness, and conformity to additional constraints. To trade-off between exploration and exploitation, this sampling would involve a temperature parameter, modulating its stochasticity. The temperature parameter would be annealed between the start and the end of each ES process, so as to sample more stochastically initially, and more directly proportional to the score towards the end of the ES run. The number of sampled solution candidates is a tunable parameter, as is the temperature. For each sampled solution from the previous generation, the ES process would generate and propose one or more mutations, new solutions obtained via a number of random edits on the board. These edits would involve random piece removals and additions, as well as potentially playing a randomly selected number of random moves from the sampled position. Given that it is possible to have mutations result in illegal board positions, this process would be repeated if necessary for each mutation to ensure it conforms to the basic criteria of legality, so as to always get the requested number of candidates for the next ES iteration. The exact number of edits for these mutations is another controllable parameter. Once the new candidate solutions have been generated through mutations from the sampled set, we would compute their own fitness in terms of counter-intuitiveness, uniqueness, and constraint satisfaction. We would then use that composite score to sort the old and the new solutions jointly, and then discard the lowest performing candidates, keeping the ES buffer at a fixed size throughout the process. At the end of the final iteration, the ES process would write the best generated solutions, reset all the counters and parameters, re-sample the starting position, and start from scratch.

As is clear from the description above, even in this basic setup there are many controllable parameters. Rather than being a weakness, this is actually a form of strength, given that different parameter configurations work in different ways, leading to different types of resulting puzzle positions. For example, running broad but shallow search, with a large buffer, a small number of iterations, and a low mutation rate, results in largely on-distribution positions that tend to turn the original sampled non-puzzle chess boards into chess puzzles that still feel largely similar to the original. On the other hand, running the ES process deeper with a higher mutation rate results in positions that are far more OOD and unusual. As this is a spectrum, the exact configuration represents a creative choice.

While chess experts generally favor realistic puzzle positions, the pursuit of true novelty—such as inventing entirely new categories of chess puzzles—necessitates venturing beyond currently known and realistic configurations. Evolutionary Search (ES) emerges as a promising complementary technique in this regard. As detailed in the paper, ES can directly optimize for specific objectives and constraints, enabling exploration of puzzle space regions that models trained solely on existing data might not reach. Future research could fruitfully investigate hybrid models, for instance, by using Generative AI to initialize ES populations or applying Reinforcement Learning to refine puzzles initially identified through ES.

## F Search Features

The following features were used to compute the counter-intuitiveness score. Features that are mentioned to have penalty were added with negative weights.

**Stockfish** computes a win rate for each move in a given position, Win$(a)$. Using this win rate we computed:

- **top move gap**: the win rate gap between the top move and the second top move, this is similar to uniqueness.
- **top move miseval gap**: the win rate gap given the top move between deep-search and shallow search.

**AlphaZero** [78] has a prior network that computes move probabilities $\text{Pr}_{prior}(a)$. Once AlphaZero completes its search, its action probabilities $\text{Pr}_{post-search}(a)$ are computed as the number of visits MCTS has assigned to action $a$ divides by the total number of visits. Using these statistics we computed:

- **score policy**: Probability gap of top action (identified by deep-search) between pre and post search. I.e., Let $a^* = \arg\max \text{Pr}_{post-search}(a)$, then score policy $= \text{Pr}_{post-search}(a^*) - \text{Pr}_{prior}(a^*)$.
- **score prior drop**: How much the action probability of the action (with highest prior probability) drops in post search. I.e., Let $a^* = \arg\max \text{Pr}_{prior}(a)$, then score policy $= \text{Pr}_{post-search}(a^*) - \text{Pr}_{prior}(a^*)$.
- **prior entropy**: (penalty) The entropy of the action probabilities in the prior.

**Misc**

- **capture material**: (penalty) the material captured by the top move, weighted according to standard valuation (Wikipedia), divided by 9
- **promote material**: (penalty) the material promoted by the top move, weighted according to standard valuation (Wikipedia), divided by 9
- **giving check**: (penalty) 1 if the best move is giving check.
- **mate in one**: (penalty) 1 if the best move is giving checkmate in one move.
- **check**: (penalty) 1 if the player to play is under check.

The tuned weights of the (1) Search Features config in Table 1 using the features listed above were

$$\begin{aligned} \textit{check} &= 1.0 \\ \textit{giving check} &= 0.4 \\ \textit{score policy} &= 0.1 \\ \textit{capture material} &= 1.0 \end{aligned}$$

For the Stockfish Through-Time features, we computed the critical point and area under the curve from Section 1 (Eq. (3) and Eq. (4)).

- (time, nodes, depth) AUC: It is calculated as the area under a curve where: the x-axis represents the search resources used, including time, nodes and depth, and the y axis refers to the absolute difference between the engine's current evaluated win rate for the final solution move and the final win rate evaluation obtained after the longest possible search duration.
- (time, nodes, depth) critical point (cp) move: The time, number of nodes searched, or search depth at which the Stockfish engine first identifies the move that is eventually determined to be the final solution move after the longest search duration.
- (time, nodes, depth) critical point (cp) value: The time, number of nodes searched, or search depth at which the Stockfish engine's evaluation (expressed as a win rate) for the final solution move first reaches a value that is within a specified threshold of the final win rate evaluation obtained after the longest search duration.

| | TRAIN | TRAIN+TEST | TEST |
|---|---|---|---|
| score policy auc (az) | 0.47 | 0.47 | 0.47 |
| score policy cp (az) | 0.46 | 0.45 | 0.44 |
| move cp (az) | 0.46 | 0.50 | 0.61 |
| top move miseval gap (sf) | 0.41 | 0.38 | 0.32 |
| score policy (sf) | 0.44 | 0.41 | 0.34 |
| depth auc (sf) | 0.46 | 0.42 | 0.32 |
| depth cp move (sf) | 0.73 | 0.74 | 0.74 |
| depth cp value (sf) | 0.53 | 0.45 | 0.34 |
| time auc (sf) | 0.49 | 0.46 | 0.38 |
| time cp move (sf) | 0.64 | 0.63 | 0.60 |
| time cp value (sf) | 0.51 | 0.44 | 0.33 |
| nodes auc (sf) | 0.50 | 0.46 | 0.38 |
| nodes cp move (sf) | 0.65 | 0.64 | 0.66 |
| nodes cp value (sf) | 0.51 | 0.43 | 0.33 |

Table 5: The average precision (Eq. (8)) of specific search features.

Using these weights, the weights of (2) search Features + SF through time that we reported in Section 1 were computed.

The weights of (3) Search Features + SF through time + AZ through time were

$$\begin{aligned} \textit{az move cp} &= 0.2 \\ \textit{az score policy entropy mean} &= 0.3 \\ \textit{depth cp move} &= 1.0 \\ \textit{score prior drop} &= 0.1 \\ \textit{capture material} &= 0.3 \end{aligned}$$

Finally, Table 5 present the average precision for standalone search features detailed above. Analyzing these results for StockFish reveals clear trends. Firstly, features derived from the search process over time consistently outperformed static positional features. Secondly, when comparing methods for summarizing through-time data, the 'critical point (move)' approach yielded better results than the 'value' method, and both critical point strategies were superior to the AUC method. Thirdly, considering the search dimension (depth, nodes, time), depth-based metrics generally outperformed node-based and time-based metrics across most through-time measures. This superiority of depth-based features explains why they received high weights in the tuned configurations.

To demonstrate that simply having a unique solution is not enough to fully represent the counter-intuitiveness of chess puzzles, we also present uniqueness as a baseline. This metric is calculated as an indicator according to Eq. (1). To break ties ties, we performed 100 random shuffles of the samples and computed the average score. Finally, Table 6 in the supplementary presents a correlation analysis between various counter-intuitiveness scores and the lichess rating and popularity scores (described in Appendix B) conducted on $1\%$ of the Lichess dataset (approximately 36,000 examples). The results indicate a positive correlation between counter-intuitiveness and rating, but no correlation with popularity.

| | Rating | | Popularity | |
|---|---|---|---|---|
| | Pearson | Spearman | Pearson | Spearman |
| Uniqueness | -0.044057 | -0.044483 | 0.021340 | 0.023032 |
| (1) Search Features | 0.374290 | 0.406505 | 0.020066 | -0.023285 |
| (2) + SF through time | 0.343795 | 0.291719 | 0.016591 | 0.013052 |
| (3) + AZ through time | 0.333493 | 0.222792 | 0.021130 | 0.014161 |

Table 6: Correlation analysis between candidate score functions and rating/popularity in Lichess

# G Additional results

We report additional results on AR training curves in Fig. 8.

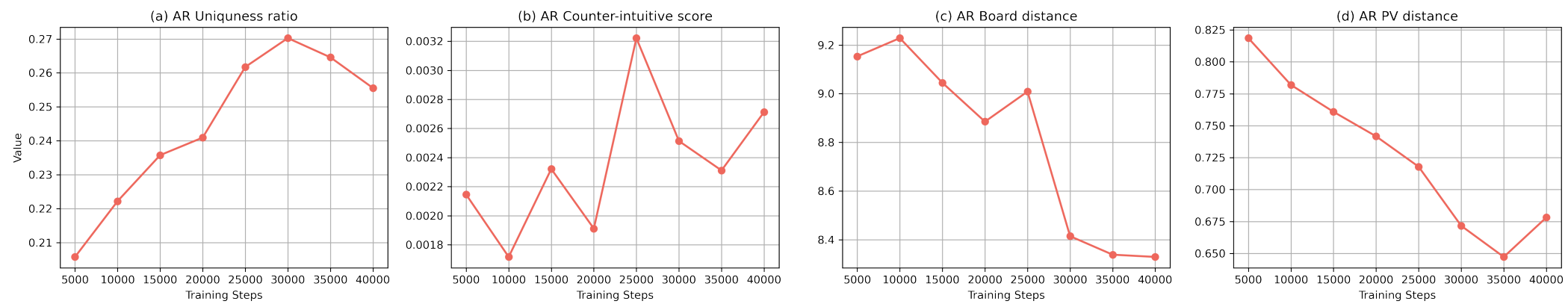

Figure 8: AR training curves. More AR training steps bring in higher uniqueness and counter-intuitiveness score, with the price of lower board distance and PV distance.

# H Additional RL results

## H.1 Vanilla RL fails with entropy-collapse or out-of-distribution samples

We initiate our experiments with vanilla Reinforcement Learning (RL) training, incorporating an entropy bonus and starting from the lichess-trained supervised model. Fig. 9 illustrates the progression of reward and model generation entropy during training under different entropy bonus coefficients.

With relatively low entropy bonus coefficients (e.g., coef=1e-2 or 2e-2), the training exhibits distinct failure stages. Initially, after several hundred steps, the reward sharply increases towards 1.0, suggesting that nearly all generated positions are classified as qualified puzzles. However, this coincides with the generation entropy plummeting towards zero—a clear indication of reward hacking. The model learns to maximize reward simply by collapsing its output distribution onto a single, easily generated qualified puzzle (an example is shown in Fig. 9, 3rd board). In the later training phase, this severe entropy collapse leads to complete training failure, where the model's output degenerates further into predominantly illegal positions (reward = -2, see example in Fig. 9, 1st board).

Addressing this entropy collapse issue proves non-trivial. Simply increasing the entropy bonus (e.g., coef=4e-2) successfully mitigates the drastic drop in entropy. However, this approach yields negligible reward improvement (the reward never exceeds 0) and introduces severe out-of-distribution (OOD) issues. As illustrated by the 2nd board of Fig. 9, the model adopts undesirable generation strategies, such as placing an excessive number of pieces or creating densely packed piece clusters. While these tactics likely succeed in maximizing the entropy score component of the reward, they produce positions that lack the strategic interest and natural game flow expected by human chess players.

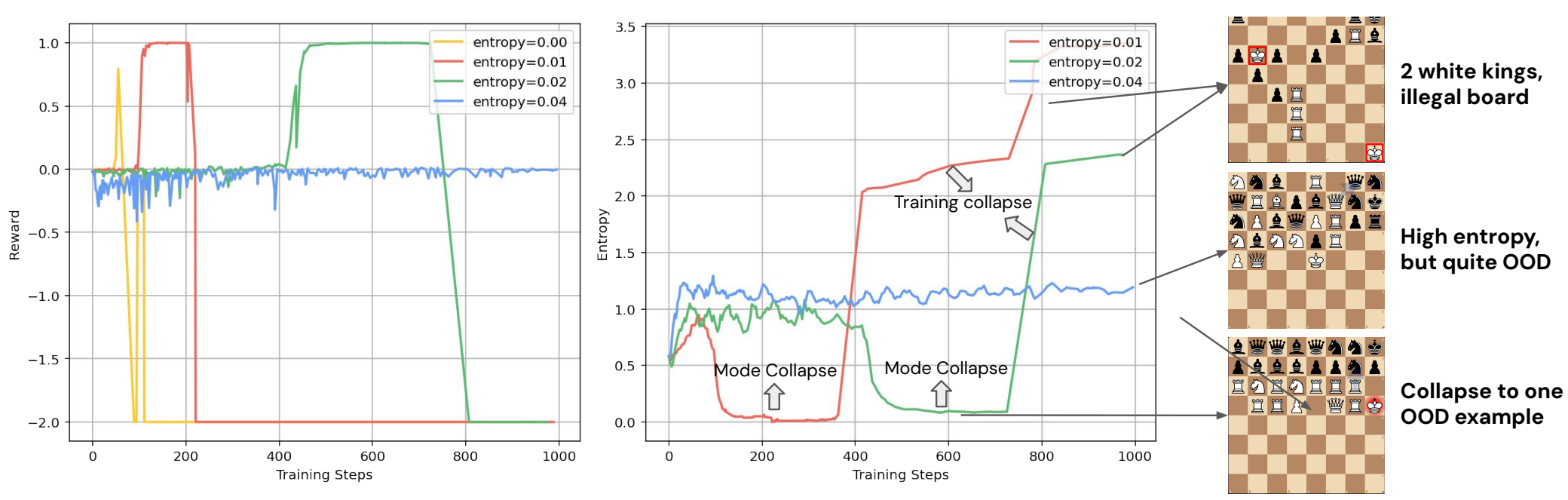

Figure 9: Vanilla RL with entropy loss fails with entropy-collapse or out-of-distribution samples.

## H.2 RL with diversity filtering stabilizes training

We introduced a few filtering mechanisms in Section 2.1, including board distance, PV distance, entropy-based filtering and intra/inter-batch checking. To analyze their effects, we investigate the impact of progressively applying them.

Our first experiment explores only adding entropy-based filtering on the vanilla RL setting. Fig. 10 indicates this method partially mitigates entropy collapse, preventing the rapid decrease towards zero observed previously, although it does not fully stabilize the training process.

Fig. 11 shows stricter constraints, which combines the entropy-based filtering mechanism with the inter-batch diversity filtering based on board distance. With an appropriate entropy filtering threshold $\tau_{ent}$, this configuration yields the first observed stable training trajectory for both reward and entropy. However, manual inspection of generated examples identifies a new diversity hacking issue. As exemplified in Fig. 11, the model learns to generate positions with high superficial diversity (e.g., large board distance) but identical core tactics and solutions (principal variations). This indicates the model circumvents the filter by manipulating puzzle-irrelevant elements while preserving the rewarded puzzle core.

To address this more sophisticated reward/diversity hacking, we employ all filtering mechanisms, detailed in Fig. 12. It illustrates how different components and their configurations contribute to

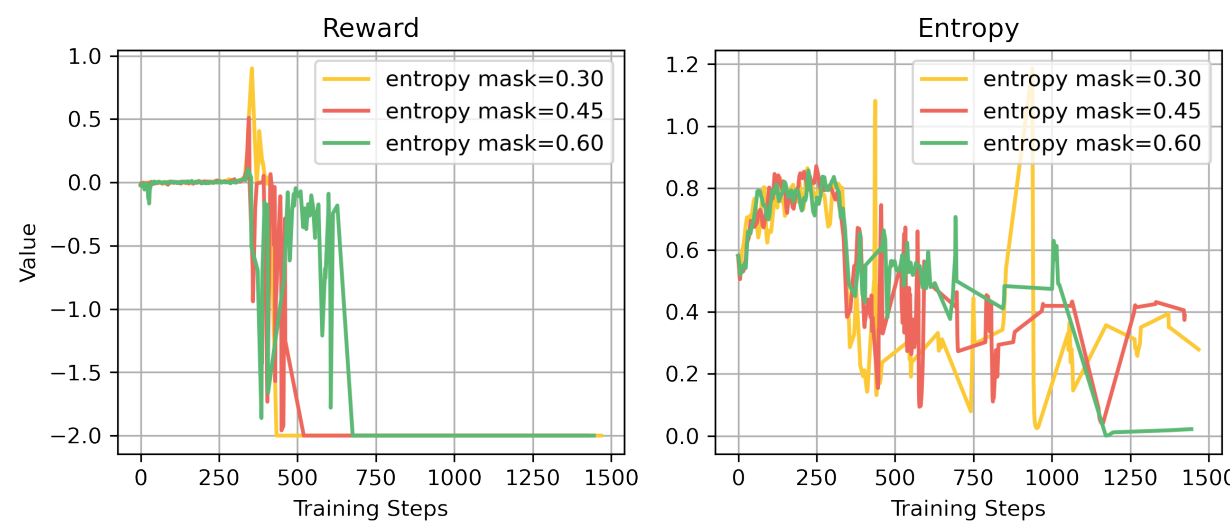

Figure 10: RL with pure entropy reward mask filtering.

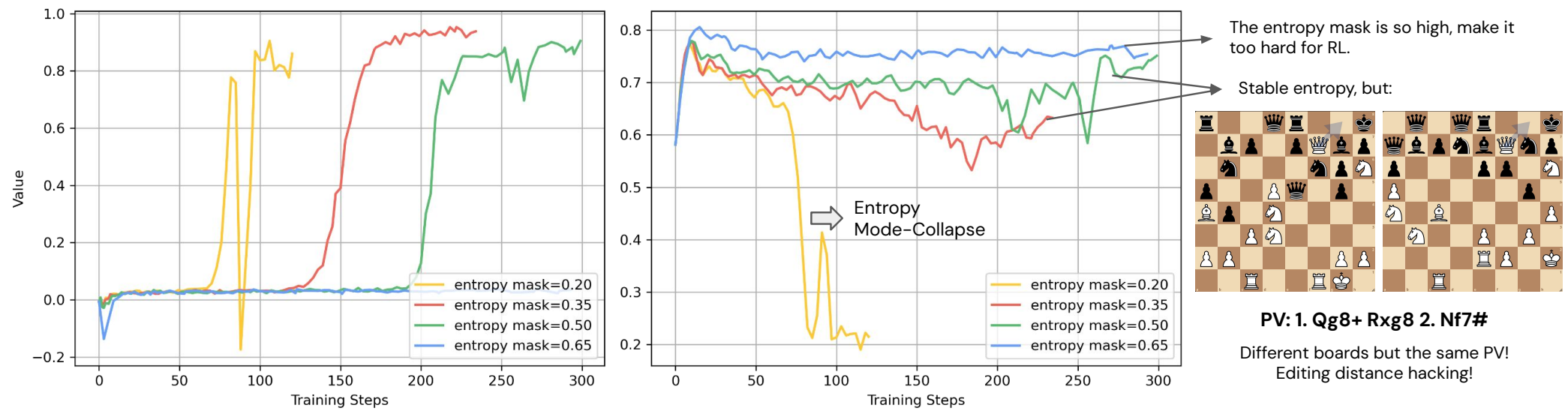

Figure 11: Reward entropy mask + inter-batch board-distance filtering can stabilize training. However, further RL hacks the board-distance filtering.

training performance. With optimal hyperparameter settings (e.g., entropy-filtering threshold = 0.6, utilizing all Board/PV/entropy filters), we achieve stable training and maintain a high sample pass ratio (samples are both qualified and diverse) over 5,000 training steps. Consequently, we adopt this comprehensive filtering strategy as our optimal setup for ensuring generation diversity.

### H.3 Make puzzles more human game like

Beyond entropy collapse, the second major challenge identified in vanilla RL is Out-of-Distribution (OOD) generation. To address this, we aim to maintain the RL model's alignment with the distribution learned by the initial pretrained model, primarily by minimizing their KL divergence.

Experimental results in Fig. 13.a suggest that even without employing more complex filtering mechanisms like entropy filtering, maintaining proximity to the pretrained model via a KL penalty offers some mitigation against entropy collapse. When combined with entropy filtering and other filtering techniques (Fig. 13.b-d), the KL penalty proves crucial for alleviating OOD phenomena. Specifically, when a higher entropy bonus is applied (e.g., entropy-coef=0.03), a low KL penalty coefficient (kl-coef=0.01) allows the generation entropy to surge above 1.0, consistent with the entropy levels observed during OOD generation in Fig. 9. However, increasing the KL penalty (kl-coef=0.03) effectively counteracts this surge, leading to lower KL divergence and entropy, thereby reducing the occurrence of OOD generations.

We also explore another technique common in RLHF: incorporating auxiliary supervised data to help anchor the model and reduce KL divergence. We curate a high-quality subset of approximately 100k puzzles by filtering the original Lichess dataset based on our defined uniqueness and counter-intuitiveness metrics. These puzzles are then used to seed the replay buffer, ensuring that samples drawn during RL training include these high-quality, human-like examples. Fig. 13.d demonstrates that this strategy successfully assists the model in further reducing its KL divergence from the pretrained model.

Finally, we address a specific reward hacking behavior related to OOD generation observed in Fig. 14.a-c. The model learns to artificially inflate reward (Fig. 14.a) and entropy Fig. 14.b) by adding an excessive number of pieces to the board (e.g., two white queens or three black knights). Fig. 14.c reveals that without intervention, the proportion of generated puzzles containing more pieces than a standard chess set can exceed 30% and tends to increase during training. While these positions might be technically legal, they are often unrealistic, rarely occur in standard gameplay, and are generally

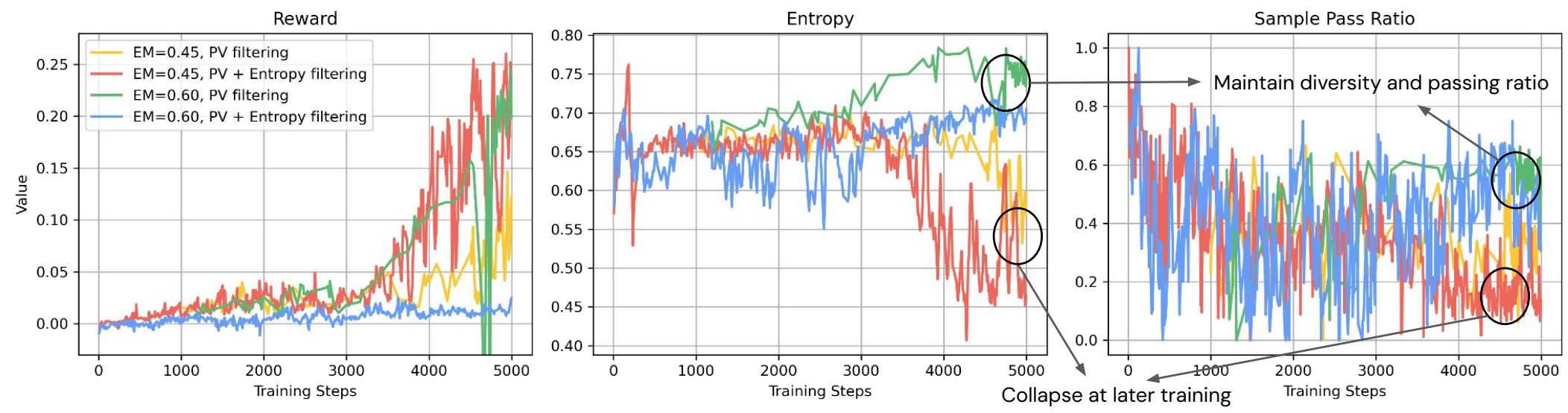

Figure 12: Further add pv distance and entropy filtering for replay buffer's new sample selection. We conduct hyperparameter search on entropy mask + Board/PV/Entropy-based replay buffer filtering.

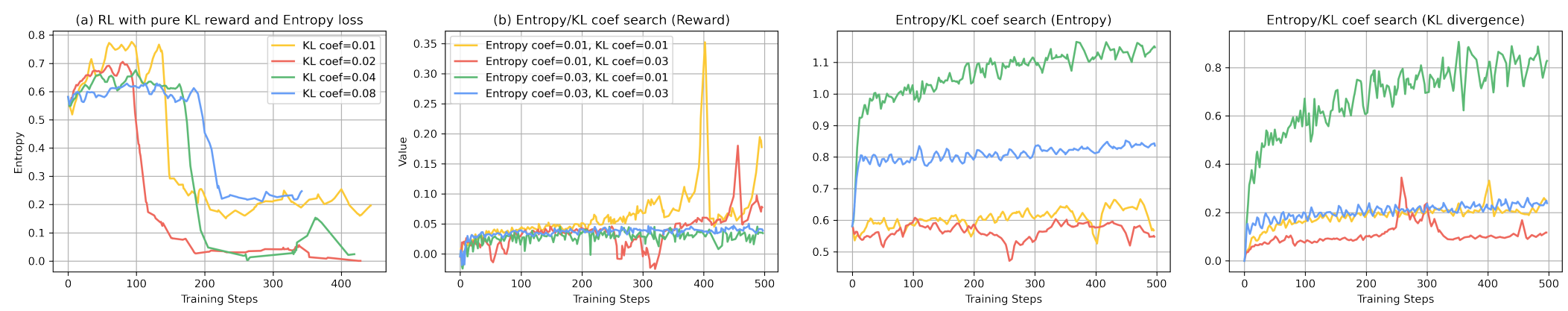

Figure 13: KL coef hyper-parameter search. It can effectively control the OOD phenomenon and make puzzle more human like.

less appreciated by human players. To discourage this, we introduce a piece count penalty: the reward for any generated position exceeding the standard piece count is zeroed out. The effectiveness of this approach is evident in Fig. 14.c, where the penalty successfully reduces the prevalence of such positions to approximately 5%.

### H.4 More ablation studies

We conduct several additional ablation studies and analyses to further understand the training dynamics and justify our methodological choices:

**PPO Update Epochs**: We investigate the effect of the number of PPO update epochs per data batch. Fig. 15.a presents results for different epoch numbers (epoch=1, 2, 4). While epoch=2 yields slightly better performance than epoch=1, increasing to epoch=4 leads to highly unstable reward progression. Unlike traditional RL applications prioritizing sample efficiency, our primary focus is on training stability and mitigating entropy collapse through effective diversity management. Therefore, to ensure maximum stability, we forgo the common practice of multiple updates per batch (epoch > 1) and consistently use epoch=1 in the analyses presented above and in our final configuration.

**Necessity of Continuous Training**: We examine the rationale for continuous model training alongside our diversity filtering mechanisms. Fig. 15.b provides a crucial comparison: when applying the same filtering mechanisms but keeping the model parameters frozen, the rate of discovering novel samples (eligible for the replay buffer) progressively diminishes over sampling steps, eventually approaching zero. This demonstrates that continuous model updates are essential for the sustained generation of novel, qualified puzzles under our filtering regime. This finding reframes the training objective: instead of converging to a single, static optimal policy, our approach functions as a **dynamic search process**. The model continually adapts based on the regions of the puzzle space it has already explored, enabling ongoing exploration. This contrasts sharply with standard RL aiming for convergence to a stationary policy and shares conceptual similarities with exploration strategies driven by intrinsic reward in RL [10, 6, 9].

**Late-Stage Training Oscillations**: Despite the significant stabilization achieved through comprehensive diversity filtering, Fig. 15.d reveals oscillatory behavior emerging in the very late stages of extensive training (e.g, more than 5k steps) in our experiments. We observe periodic fluctuations where reward and entropy move in opposition, suggesting the model alternates between prioritizing reward maximization and satisfying diversity/entropy constraints. This behavior is reminiscent of dynamics observed in adversarial training [18, 3] or constrained optimization [60] when operating near a Pareto frontier, where small parameter adjustments (e.g., a few gradient steps) can cause

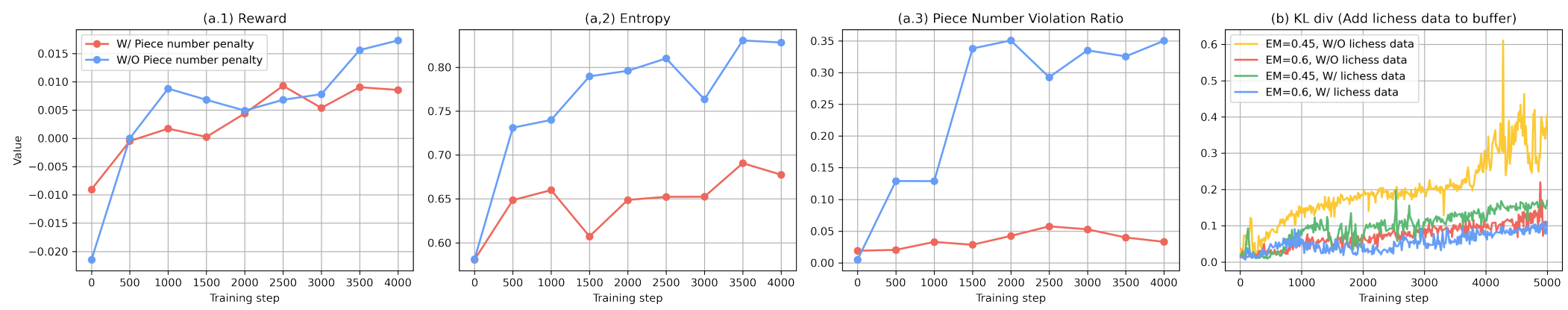

Figure 14: (a) shows the effects on adding piece count penalty. Model can easily learn to violate piece count standard (nearly 35% are violating it from a.3) to get high reward (a.1) and entropy (a.2). (b) Adding lichess samples into replay buffer can lower KL divergence.

significant shifts between conflicting objectives. This indicates our model is effectively optimized to the boundary defined by the reward and the imposed constraints. Potential avenues for mitigating these late-stage oscillations include: (1) employing softer constraint enforcement mechanisms, such as adaptive Lagrange multipliers, instead of the hard entropy mask threshold, and (2) exploring optimization algorithms specifically designed to achieve last-iterate convergence, e.g. [60].

**Necessity of Supervised training ("Zero-RL")**: To underscore the necessity of the initial supervised pretraining phase, we attempt to train the model using RL directly from a randomly initialized Transformer. Fig. 16 illustrates the results of this "Zero-RL" approach under various entropy bonus settings. The outcome unequivocally demonstrates the infeasibility of this strategy: the randomly initialized model consistently fails to generate even valid board positions. Consequently, it cannot produce any samples suitable for puzzle quality evaluation, rendering subsequent RL optimization based on puzzle scores impossible. This finding strongly validates the critical role of supervised training in providing a foundational understanding of chess rules and structures, upon which RL can then refine generation towards specific puzzle characteristics.

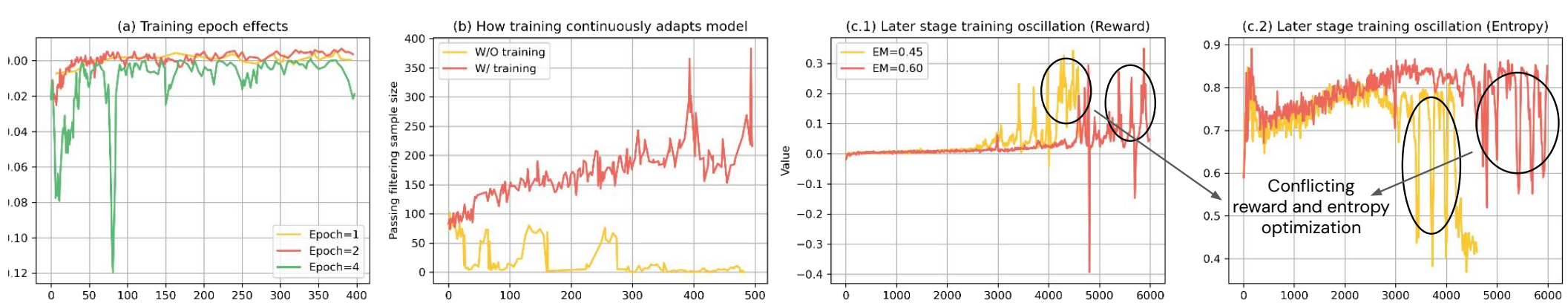

Figure 15: Other ablation studies. (a) PPO training epochs. (b) Importance of continuous training. (c) ocsillation issue at the later stage of extensive training.

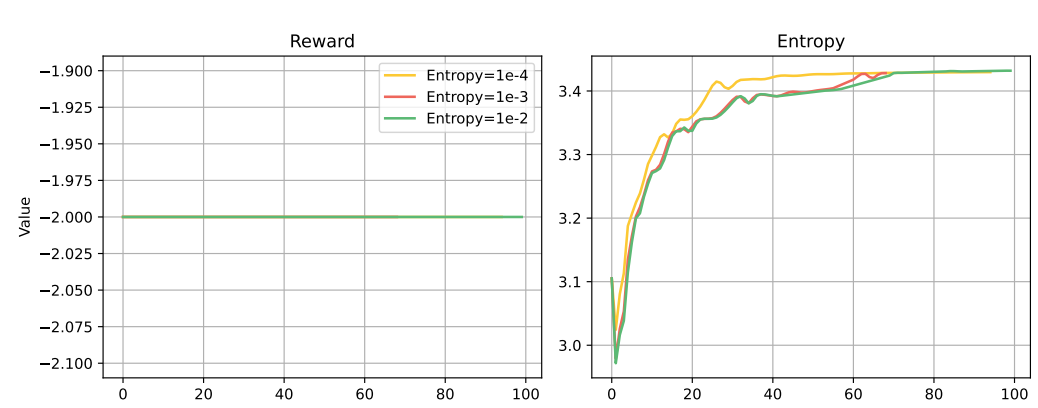

Figure 16: Zero-RL does not work

# I Aesthetics

**Theme-detecting functions.** To identify aesthetic themes derived from chess literature, we developed Python functions, similar to the Lichess Puzzler project. Each function processes a chess position and its associated solution sequence (mainline). It returns a boolean value signifying whether the specific theme is present. If the theme is detected, the function additionally outputs the particular move from the mainline that triggers the theme's identification.

**Improving quality of theme-detecting functions.** While chess literature illustrates aesthetic themes using carefully selected, expert-approved positions where the theme is central and arises naturally, our initial theme-detecting functions struggled to capture this level of nuance. They could identify patterns but often failed to distinguish genuinely creative instances from technically matching but uninteresting ones.

To enhance the quality of detection, we undertook a manual review process, specifically aiming to filter out false positives – positions where a theme was present but lacked significance (e.g., an underpromotion immediately nullified by capture or leading to a basic tactic). This iterative refinement reduced the occurrence of uninteresting identifications, but eliminating subtle false positives completely was difficult due to the challenge of encoding human intuition about creativity.

Figure 17 presents three example positions generated by our RL agent. These are false-positives that we identified while improving the Attacking-Withdrawal theme. Position (a) demonstrates an initial flaw: flagging a Rook withdrawal (Black, e5 $\rightarrow$ e7) whose actual purpose was defensive (preventing checkmate by supporting g7), not a calculated attacking retreat. Adjusting for this revealed other issues, like position (b), where a Bishop retreat (Black, h1 $\rightarrow$ e4) was purely a reaction to save material. Addressing these specific motivations led to our final implementation, capable of finding compelling examples (some of which are presented in the booklet. Yet, the challenge of capturing nuance persists. Position (c) shows a Knight withdrawal (White, $\rightarrow$ f3) that, while matching the theme's rules, isn't considered creative because the withdrawal itself isn't a key element of the position's strategic narrative. We could not improve this further, as defining and measuring a theme's 'centrality' seems elusive.

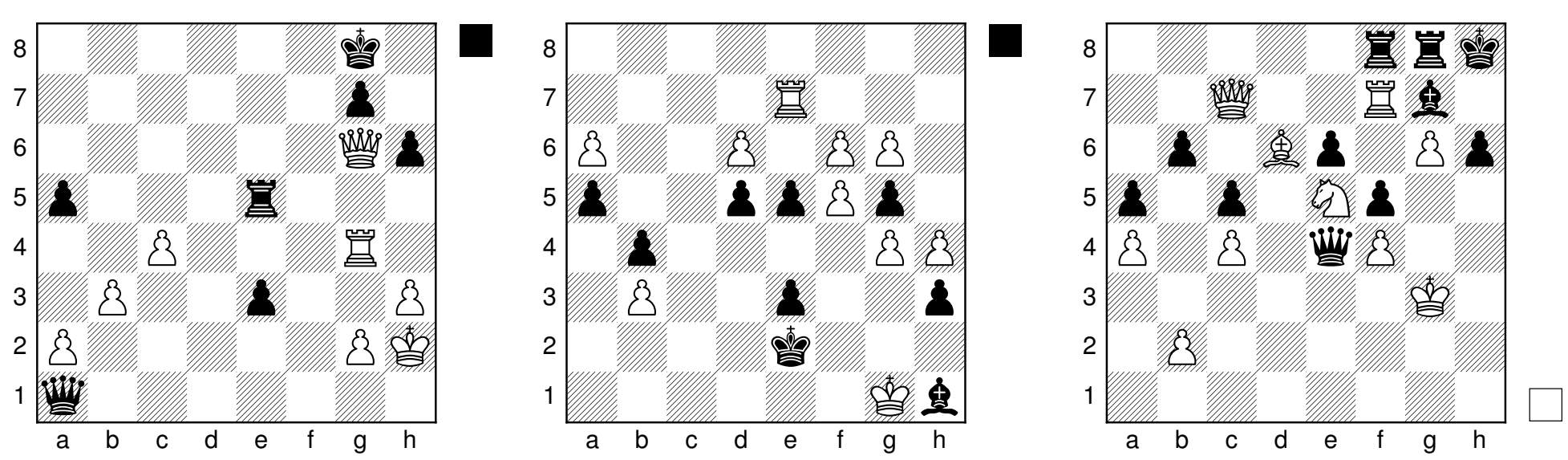

(a) Moving the Rook from e5 to e7 while looks like a withdrawal is actually defensive.

(b) Bishop retreat from h1 to e4 is a reaction to save material.

(c) Moving the Knight to f3 while a withdrawal move is not the key element in this puzzle.

Figure 17: False-positives identified while iteratively improving the Attacking-Withdrawal theme-tagging funtion.

Table 7 presents the list of themes that were used for tagging and filtering our generated puzzles. As described in Section 1, we implemented a number of themes from chess literature, namely from Levitt and Friedgood [48], Avni [2] and Persson [66]. We also included a small set of themes from Lichess Puzzler project.

Fig. 18 shows the distribution of puzzles from different puzzle sets across the aesthetic themes. The blue and orange bars corresponds to puzzles from Lichess dataset and those generated from our auto-regressive generative model. This figure shows that the generative model can produce puzzles across the different themes that are represented in the training dataset. The quantity of puzzles

| Secrets of Spectacular Chess | Creative Chess | Tiger Chaos Theory | Lichess Puzzler |
|---|---|---|---|
| Paradox-of-Material | Attacking Withdrawal | Knight on the Rim is Dim | Anastasia's Mate |
| Underpromotion | Sacrifice Pieces to Stalemate | King on Tour | Arabian Mate |
| Material Imbalance Draw | Voluntarily Entering into Pin | | Attraction |
| Winning with Limited Material | Exchange Material Down | | Backrank Mate |
| Set-Play Paradox | Unprotected Position | | Boden Mate |
| Reduces Discovered Checks | | | Castling |
| Zugzwang | | | Deflection |
| Slow Move | | | Double Bishop Mate |
| Switchback | | | Double Check |
| Fork | | | Dovetail Mate |
| Systematic Manoeuvre | | | enPassant |
| Paralysis | | | Exposed King |
| Self-Paralysis | | | Hook Mate |
| Interference | | | Intermezzo |
| Novotny | | | Mate |
| Bristol | | | Pin |
| Loshinsky's magnet | | | Skewer |
| | | | Smothered Mate |
| | | | X-Ray Attack |

Table 7: Aesthetic themes

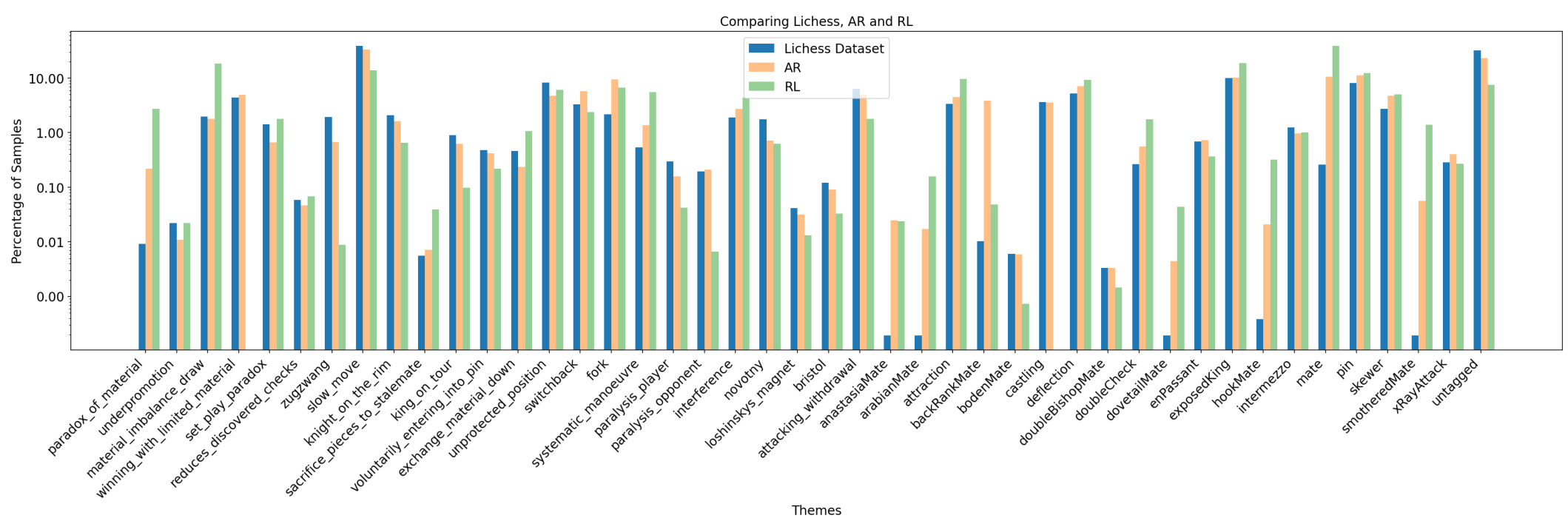

Figure 18: Distribution of aesthetic themes in unique chess positions across different datasets.

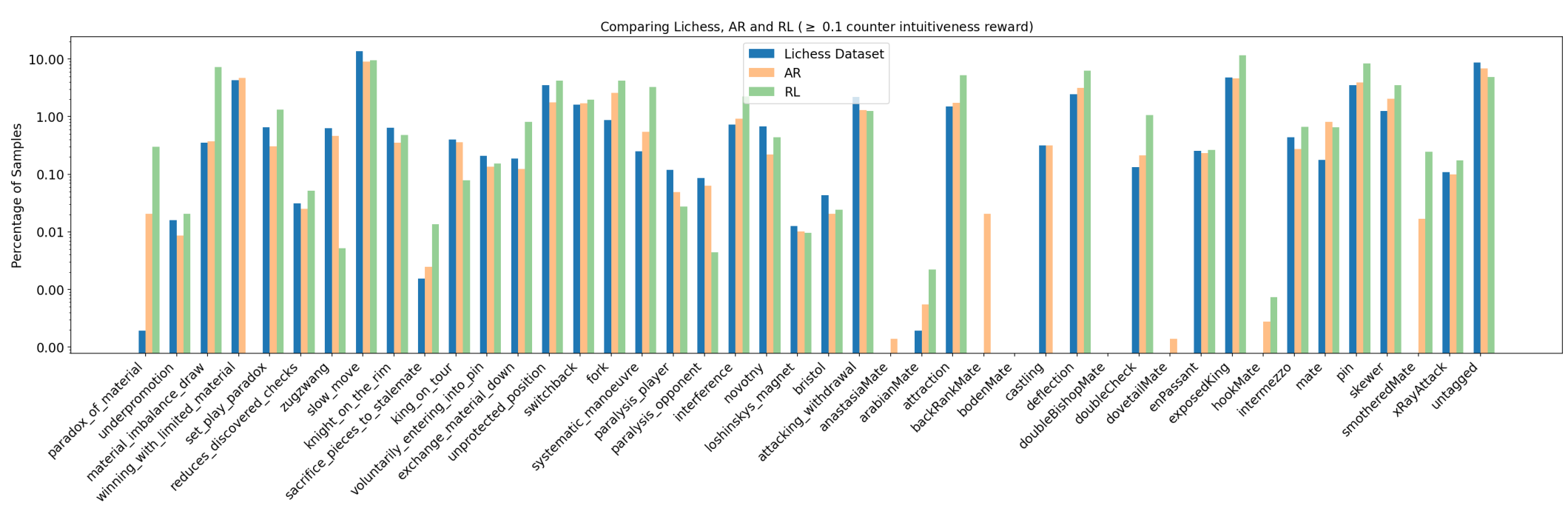

Figure 19: Distribution of aesthetic themes in unique and counter intuitive chess positions across different datasets.

**Mating patterns.** The above figures show that our puzzle generation methods don't perfectly mirror the training dataset's distribution. Specifically, certain mate themes such as Anastasia's Mate, Double Bishop Mate, and Hook Mate are missing from puzzles created by the RL agent. To understand why, we looked at the distribution of puzzles in the training data that had a counter-intuitiveness reward of 0.1 or higher (the threshold used for training the RL agent), as illustrated in Figure 19. This revealed that training examples for Anastasia's Mate, Double Bishop Mate, and Hook Mate were not rated as counter-intuitive. As a result, their effective representation in the training data used by the RL

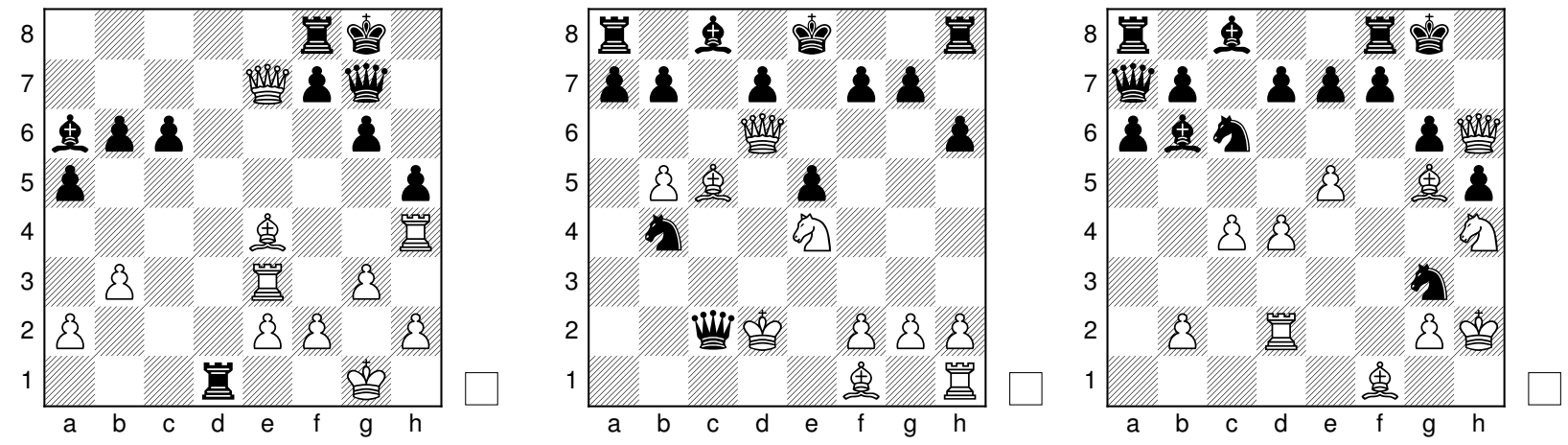

Figure 20: King on Tour puzzles from Persson [66] (left, center) and generated by AI (right).

agent was zero, which explains why the agent didn't learn to generate high-quality puzzles with these themes and why they are absent in its output.

**Creativity is subjective.** Figure 20 presents a puzzle illustrating the 'King on Tour' theme, an idea Persson [66] describes by noting, 'To take one's king for a walk is something that most chess players will shun mechanically.' This theme arises when a player moves their King far from its castle while avoiding danger. Many chess experts, including the book's authors, view such positions as creative because their solutions defy traditional chess strategies. On the other hand, some experts find this theme less engaging or uncreative, often because it demands significant calculation. For instance, in the rightmost position of Fig. 20, White has a strong winning move (Bf6). Black can only postpone defeat with a few checks, but White has a defensive option that involves taking the king on a 'tour,' a sequence that might even take Stockfish a few seconds to identify as winning. This difference in expert opinion is common across various aesthetic themes, highlighting the subjective nature of creativity in chess.

**Omitted description of selected chess puzzles.** We report the omitted descriptions for the positions *2–4* in Figure 2. *2:* Black's position at a glance seems lost as White has many more pawns. Surprisingly, Black can force a draw, starting with the `Rxh2` rook sacrifice. After the White king recaptures, Black plays `Qh6+`, followed by `Qh4` exploiting the pin on the pawn. Even though White can seemingly defend with `Rf3`, it turns out that Black can force stalemate in this position by playing the second rook sacrifice `Rxg3+` and then sacrificing the queen. The stalemate line is unique and counter intuitive as it involves a number of sacrifices and slow moves. The fact that the result is a forced stalemate is surprising and aesthetic. *3:* The unique winning move involves retreating the bishop as far away from the attack with `Bb1+`. It turns out that no matter what Black plays, the idea of creating a bishop-queen battery for White is unstoppable. The unique bishop retreat paired with the slow creation of the battery makes this puzzle counter intuitive and aesthetic to a human. *4 (right):* The puzzle is a creative example of a checkmate in 3 moves. Black sacrifices the queen with `Qg1+` with the idea of under-promoting, followed by an 'Arabian' checkmate. The combination of the Queen sacrifice, under-promotion, and Arabian checkmate pattern makes the puzzle aesthetic and counter intuitive.

# J Golden Set

Table 8: TRAIN Set

| | Positive Examples | Negative Examples |
|---|---|---|
| 1 | 4rrk1/p1q3p1/1p1p1p1p/3PnP2/5Q2/2B3R1/PP4PP/4R2K b - - 6 28 | rnbqkbnr/1ppppp2/6p1/7p/3PP2P/5p2/1PP1BPP1/RNBQK2R w KQkq - 0 4 |
| 2 | rr6/1q2kpBp/1n4p1/p1p1p1P1/2n1PP2/P7/2P1Q2P/1NKR3R b - - 0 23 | 5Q1r/1Pk3q1/p1pp2R1/1n2p3/N1B1P3/2PP4/1pKB1P2/R7 b - - 0 29 |
| 3 | 4Q3/6p1/R3pq1r/3p1k2/1P1P4/P5P1/5P2/6K1 b - - 0 33 | 5Q2/1Rp4r/2Pp1p2/4P2N/1p1q2B1/4k2p/2KN3R/5RR1 b - - 0 77 |
| 4 | 3b2k1/p7/1p4q1/2pNBb2/P1P1pP1p/3rP1P1/5QPK/5R2 b - - 1 40 | rnK1k2r/p4p1p/3bp3/2P2p2/2Pp3P/1N2P3/4Q1PP/2R2BQ1 b kq - 0 15 |
| 5 | 8/6pk/2p1p1rp/1qP1Pp2/pPbPp3/2Q1P3/6PP/R3B1K1 b - - 4 47 | 8/8/3p1K2/ppp5/PN3p2/P4P2/1b1k4/B7 w - - 2 52 |
| 6 | 2kn2rr/Rp2bp2/3p1q2/1P1Bp2p/4P1p1/2PP1P1P/5P2/2BQR1K1 b - - 1 18 | r1bqkb1r/ppp2p1p/4PPp1/3p4/8/2N3P1/1nP1PP1P/R1BQKBNR b KQkq - 1 7 |
| 7 | r2q1rk1/1b3p2/p2p1np1/1pn1p1N1/4P3/PBPP1Q2/3B1PP1/bN2K2R w K - 0 19 | r1bqkb1r/1p2pppp/p4n2/2p5/4P1PP/2PP4/PP2p1P1/RNBQKBNR b KQkq - 0 6 |
| 8 | r5k1/5ppp/1B2p1q1/3p4/1P6/4Q2P/rb3PP1/1RR3K1 w - - 3 23 | rn1qk2r/ppp2p2/8/4N3/3Pp1pp/2P1Pp2/PP3P2/RNB1QK1R b kq - 1 17 |
| 9 | 6k1/6p1/pq3pQp/4b3/P7/B1r4P/5PP1/1R4K1 b - - 3 34 | 3Q1bk1/p7/1p2p1p1/5p2/1P1P4/P3P1Pp/4KP1q/4B3 w - - 2 37 |
| 10 | 6k1/q4pp1/P2P3p/4p3/4n1n1/5N2/5PPP/R2Q2K1 b - - 0 34 | 1Q2nk2/pb2q2p/2p3p1/3r1pN1/8/2P5/PP2BPPP/4R1K1 b - - 6 24 |
| 11 | kbK5/pp6/1P6/8/8/8/8/R7 w - - 0 1 | 8/8/7p/2p1K1pk/1pP4p/pP5P/P7/8 w - - 0 1 |
| 12 | 8/8/2Q5/3B4/1K6/2P5/Nk6/2R5 w - - 0 1 | rrb5/1p3p1k/1Nn2Qpp/2B5/6P1/5PK1/2p5/8 w - - 0 1 |
| 13 | 2b3N1/8/1r2pN1b/1p2kp2/1P1R4/8/4K3/6Q1 w - - 0 1 | 2q2r1k/4b1pp/8/6N1/8/1Q6/B4PPP/6K1 w - - 0 1 |
| 14 | 5B2/8/K7/8/kpp5/7R/8/1B6 w - - 0 1 | r1r1n1k1/4RRp1/1Bp3Q1/3p4/2P5/1P4PP/Pq4BK/8 w - - 0 1 |
| 15 | 8/p4p2/Q7/3P4/1p1kB3/1K4N1/5R2/8 w - - 0 1 | 3R4/2q3nk/7p/5P2/6PP/P7/1P1Q4/1K6 w - - 0 1 |
| 16 | 8/qQ5p/3pN2K/3pp1R1/4k3/7N/1b1PP3/8 w - - 0 1 | 1r5r/ppq1n1k1/3p1ppp/3B1b2/2P2P2/1R2B3/PQ4PP/1R4K1 w - - 0 1 |
| 17 | K4BB1/1Q6/5p2/8/2R2r1r/N2N2q1/kp1p1p1p/b7 w - - 0 1 | 3r1r2/k2p3p/1p2qNp1/1P2p3/8/3B4/P5QP/2R2RK1 w - - 0 1 |
| 18 | BK6/1NP5/8/R4P2/r3k3/3Rp1P1/1q6/5Qbb w - - 0 1 | 8/3PR3/p7/3kP3/1p6/1P1b2B1/P2r4/K7 b - - 0 1 |
| 19 | 1B4n1/7N/K1pN4/5P2/R1n2k2/2P1R2P/4b1B1/2Q5 w - - 0 1 | 1b4k1/r5np/1p5B/p1p5/2q3P1/2P4P/8/4QRK1 w - - 0 1 |
| 20 | 8/2B4p/2p2R1K/1Pk3N1/1n1r1n2/NQP5/8/8 w - - 0 1 | rq5k/6pp/8/4p3/3p1N2/1B6/PPP5/1KR5 w - - 0 1 |
| 21 | 8/5N2/8/1b3k2/1P1R1P2/PK6/3p1r2/8 b - - 1 63 | 8/2bB4/5k2/p1pb3p/4p3/P1P1KpPP/1P3P1B/8 b - - 1 34 |
| 22 | 6rk/2pb2b1/4p1QP/3pP3/1n1P4/1q2B1N1/1p3PBK/8 w - - 1 39 | 6k1/7p/p1Pp3r/3P2p1/4Pp2/5Bq1/P4RP1/1R4K1 w - - 0 33 |
| 23 | 8/pp4qp/6b1/6p1/PQ4Pk/1P1p4/5PBK/8 w - - 1 42 | b4kr1/5p1p/p7/2q1BP2/8/P4PP1/6K1/1R2R3 w - - 0 33 |
| 24 | 1r4k1/5p1p/3p2p1/p1pPp1P1/P1Q1Br1P/1P3P1K/3q4/4RR2 b - - 4 32 | 5r2/2p1k1p1/Q2pP1q1/2p5/P6p/2B1nr1P/1P2R1P1/4R1KN b - - 1 30 |
| 25 | 8/3kp1b1/p7/P1R1P3/2P5/3KN3/8/4r3 b - - 2 43 | 6rk/2bP4/6RP/pp5P/p1p1BPK1/2P4N/pp3b2/4b3 w - - 2 50 |
| 26 | 4R3/8/5rpp/p2p2k1/P2q1pP1/3P1P1Q/1p3r2/1R5K w - - 1 43 | 4BQ2/3Q2P1/bp3R1B/3PpP1r/2n1n1p1/R6n/P2P1n1q/1KN2kb1 b - - 1 56 |
| 27 | 5b2/6Bk/5p1P/1p2pP1K/p1p1P3/P7/1PP5/8 b - - 9 59 | R1RN4/1P4kp/3P1r2/6pp/PRPNp2P/p1p1b3/Bb1r2P1/bKB1n3 b - - 0 37 |
| 28 | 2r2rk1/5p1p/pqn1bPp1/1pQ4p/1p1PNpP1/P2P4/R3b3/4NR1K w - - 3 30 | 5nn1/2bRQ2R/2PP3P/r2k3P/P1p5/1rp2pP1/2p2r2/q2b3K w - - 0 50 |
| 29 | 1rkn1NQ1/1nP2N2/p3p3/P3Pn1p/P5r1/6q1/1P5P/6RK w - - 0 34 | Nn1qN1Q1/4N1PR/pr1p4/2pN1Pp1/2PR3P/3P2p1/P3K3/5nk1 b - - 6 46 |
| 30 | r1b1n3/P1qp2bp/pn4pn/1p1B4/pP1NP3/q2P4/PN1k2K1/R1R4R w - - 0 21 | 2R3K1/1n2kNPR/np3p2/6p1/2p5/1pp2P2/b5pP/r2r4 w - - 0 46 |
| 31 | | 8/2P2p1N/6p1/1pP3P1/P5K1/2Ppk3/8/8 b - - 0 55 |
| 32 | | 1n6/7p/2P5/6P1/1p6/1n1p1K2/6N1/k7 w - - 7 70 |
| 33 | | 6k1/8/5Pp1/Pp1P2rB/pPpP2PP/p1K1RpP1/8/4n3 b - - 1 42 |
| 34 | | 2B2B2/n5P1/r5pp/6Nk/2p2p2/5P2/1p1pP1Kp/3R4 b - - 3 43 |
| 35 | | rrbq1rk1/pp2ppb1/3p3B/4p2Q/1nr1P3/8/PPP2PP1/R2NKR2 w Q - 3 10 |
| 36 | | 5k2/3R2p1/b2n1pB1/3K1RP1/1Ppbpr2/1p2p3/2p3br/8 w - - 3 50 |
| 37 | 2r1nr1b/2q2p1k/4p2P/p2p2QR/b2p1P2/2P5/3BN3/4R2K w - - 0 35 | 8/1Qpk1ppp/3r4/1P2r3/8/4B2q/P1P2P1P/R4RK1 b - - 0 24 |
| 38 | r7/1k4p1/p1bP4/K1PBb2p/P6P/1P6/5PP1/3R3R b - - 0 41 | 2r3k1/q2p3p/p2BpQp1/1p3p2/4P3/1PP5/P4nPP/R4RK1 b - - 6 29 |
| 39 | r3r3/ppR5/5ppk/7p/4PP2/6QP/P1P3PK/q7 w - - 0 23 | 2r5/p4pbk/3p2p1/1r1Pp2P/q3P3/2pQ1N2/PP2N3/KRR5 b - - 11 28 |
| 40 | 2r3rk/5p1p/5PpQ/1p6/p2Pq3/P2R4/1P3P2/1K4R1 w - - 0 31 | r4rk1/1pp2ppp/8/p2N4/P1B5/6q1/1PQKb3/3R4 w - - 0 26 |
| 41 | rn2r1k1/pp2bp1p/2p2Bp1/8/2p5/7Q/P1q2PPP/3RR1K1 w - - 8 22 | r4rk1/pp3ppp/4b3/1B1Nq3/8/3Q1KP1/PPP5/R7 w - - 0 23 |
| 42 | r6r/2p2p2/2qk3p/ppNp2p1/1P1b4/3B1PP1/P1Q2P2/2K1R3 w - - 0 25 | 7Q/1ppk2pp/3q4/3r4/3n4/8/PP3PPP/R1B2RK1 b - - 0 18 |
| 43 | 4r1k1/1QP2pp1/6rp/p2p4/3P4/P3PqP1/2R2P2/2B1R1K1 b - - 0 32 | 3k3r/1bq1bP1P/np1pp2p/2p3p1/2P2Pn1/1bNPBp1P/3Q2PN/5RKN w - - 1 17 |
| 44 | 6k1/p1p2r1p/1p1p4/3PP2r/1PP3b1/2Q3Pq/PB3R1P/5RK1 b - - 0 32 | n1K2k2/1n3P2/r1N2pp1/3p2p1/2PprPp1/1p1PQ3/P1P3p1/2RQ4 b - - 1 36 |
| 45 | 1R6/8/P2p2N1/3P1B2/P5p1/k3pbK1/3p2P1/8 w - - 0 50 | 4r2R/4BkPp/PN1r3p/5p2/1pP5/2r1P1p1/2P2nBn/R2RrRK1 b - - 1 27 |

Table 9: TEST Set

| | Positive Examples | Negative Examples |
|---|---|---|
| 1 | 8/8/2nrb3/2k5/4Q3/2Kp4/5N2/1R6 w - - 0 1 | 2kr2r1/pbpp1p1Q/1p3B2/2b1P1q1/2B5/6P1/PPP2P1P/RN3RK1 b - - 2 15 |
| 2 | 8/8/1p4pp/1R4pk/K6P/5PP1/8/5B2 w - - 0 1 | r1b2rk1/ppp3pp/3p4/8/4p3/2Pnq3/PPQNBRPP/5RK1 w - - 2 19 |
| 3 | r1/pk1nq2r/2n1p3/2p2p2/2PpQB2/P2P1NP1/5PB1/4R1K1 w - - 0 1 | 2k5/1b3Q1p/pqp5/3p4/5B2/8/PPP2nPP/R4RK1 b - - 4 24 |
| 4 | 2Q5/B7/1R1p1K2/r2p1B2/2bk4/8/3P4/b7 w - - 0 1 | 8/p5pp/2Q1b1k1/4Pq2/1P6/3rrPB1/P1N3PP/R1K3R1 b - - 4 29 |
| 5 | 1r3rk1/pbpn2qp/1p1p1np1/3P1pQ1/1P5N/5PPB/PB2P2P/3R1RK1 w - - 0 1 | 4nr1k/1p2Qp1p/p4Ppq/8/2b5/3R3P/PPP3P1/5R1K w - - 5 27 |
| 6 | 1r4nk/1p1qb2p/3p1r2/p1pPp3/2P1Pp2/5P1P/PP1QNBRK/5R2 b - - 0 1 | 1rb2rk1/p7/3b2Pp/q1nPp3/1p6/3B1P2/PPPQ4/K1NR4 w - - 0 24 |
| 7 | 1r5k/1p2b2p/3p3r/p1pPpK2/2P1Ppn1/5P2/PP1QNBR1/5R2 w - - 0 1 | 5rk1/5rp1/pq2pb1p/1p6/1P6/PQp1R2P/5PP1/BN3RK1 b - - 0 27 |
| 8 | 8/1KP5/8/2p5/1pP5/p7/k7/1R3R2 w - - 0 1 | r4rk1/p4ppp/8/3N4/4Q3/4PR2/q1P3PP/6K1 w - - 0 20 |
| 9 | 1K6/3B1kp1/7p/P5p1/4P3/4n1P1/3p4/8 w - - 0 1 | 7Q/8/3kp3/3p1p2/1N3P2/2P1q3/1P3nPP/R4RK1 b - - 6 40 |
| 10 | 3K4/3Rp1k1/7p/1r6/7P/6P1/2p5/8 w - - 0 1 | 3q2rk/6pp/3p1p2/2p1pP1Q/p1PnP3/Pr4R1/6PP/1R5K w - - 0 29 |
| 11 | 8/1pPK3b/8/8/8/5k2/8/8 w - - 0 1 | 8/q7/3p1Q2/1B1Pp3/1K1kPpP1/5P2/8/8 b - - 8 58 |
| 12 | 8/1n6/5Rpp/7k/8/6K1/8/8 w - - 0 1 | 2r3k1/1Q3pp1/p3b1qp/4P3/P3p3/3rB2P/1P3PPK/R4R2 b - - 2 25 |
| 13 | 4r3/2P1kpR1/7p/p4K2/P5B1/6bP/8/8 w - - 0 1 | 8/8/3p4/2pPp1p1/2N1P1Pb/6kP/5p2/5K2 w - - 3 73 |
| 14 | 8/2p2Pp1/p3pN2/2p1P3/2Pk4/3P4/2PK1Pq1/8 w - - 0 1 | 8/3r4/1k5P/1p2KP2/p7/3p4/PP5R/8 b - - 0 37 |
| 15 | 1k5r/1p5p/1B5K/8/8/8/8/7Q w - - 0 1 | 5Q2/6p1/4p1pk/1P1pq3/8/7P/6P1/7K w - - 0 37 |
| 16 | 3r4/1p3r2/R7/1R2kp2/p7/1n1P1PK1/P2P4/8 b - - 1 1 | 8/8/4rpk1/7R/5P2/3K4/8/8 w - - 2 81 |
| 17 | 2b2N2/4p3/6R1/7k/2N5/6K1/8/8 w - - 0 1 | 8/3p2k1/1Rb5/2P4P/5K2/2n5/8/8 b - - 16 80 |
| 18 | 7k/6p1/P1P5/7r/1N3P2/8/7P/5b1K w - - 0 1 | 5k2/8/7b/7P/5pP1/P2R1n2/1PK5/8 b - - 1 37 |
| 19 | 1qbn3B/8/1p5p/1Q2n3/3rPkpP/3R2N1/B3PP2/K1N2Rr1 w - - 0 1 | 2k1rR2/ppp1b1Qp/8/8/8/2PP4/P1P3PP/4qR1K b - - 0 23 |
| 20 | R7/Pp2b1p1/8/8/4p3/8/k1K3p1/8 w - - 0 1 | |

# K Broader Impact

Our results of creative chess puzzle generation bring in positive societal impacts, including:

- **Educational Enhancement:** Providing engaging tools for learning chess strategy, tactics, and creative problem-solving, which can foster cognitive skills.
- **Increased Accessibility & Engagement:** Making novel and diverse chess puzzles more widely available, potentially increasing participation and enjoyment of the game.
- **Contribution to AI Creativity Research:** The methods developed can offer insights into computational creativity, applicable beyond chess.

We believe our paper does not have potential negative societal impacts.

## L Compute Resources

We spend most of the compute over CPUs. For the final RL experiment, we run the StockFish on 28M chessboards in total with 4096 CPUs, corresponding to 175k CPU hours (each position takes 15-30s, we choose average 22.5s for calculation). Note that the full research project requires much more compute than this single run cost.

## M Creative Chess Puzzles Booklet

This booklet contains chess puzzles created using Artificial Intelligence (AI) techniques. This is a brief, non-technical summary of the methods used; for more detail, please refer to the paper.

**Generating millions of chess puzzles:** Our first method involved training generative neural networks (auto-regressive transformer, discrete diffusion and MaskGit) on a large open dataset (4M) of chess puzzles from Lichess to learn the distribution of those puzzles. The dataset was constructed by selecting positions from Lichess games using software. Each chess position was represented as a sequence in the FEN notation, and a neural network was trained to predict the distribution of the next character in the FEN string based on the characters that preceded it. The neural network was trained exclusively on this dataset; no other datasets, including chess compositions, were used. The trained network was then employed as a generative model to sample chess puzzles, starting from the first character of the FEN and iteratively sampling the remaining pieces. These techniques are commonly used in generative neural networks and language models.

The second technique used reinforcement learning to train the neural network on its generated output. Reward functions were used to select the best samples, and the network iteratively improved at generating puzzles with higher rewards. The reward function had two parts: a uniqueness check, similar to the one used in Lichess, to ensure there was only one winning move; and a counter-intuitiveness check, to ensure the position could be solved by a strong chess engine but not a weak one. There are many specific details involved in what constitutes a weak vs. strong chess engine in this study. Indeed this was a significant part of this research. However, one simple example would be to use a chess engine with a very short computational budget as the weak engine and one with “full power” settings as the strong engine.

The third method utilized an evolutionary search process. In this process, new positions were created by randomly adding or removing pieces from the board. Each iteration’s best positions, as determined by the reward, were selected and continued to evolve. This process resulted in interesting, yet unrealistic, puzzles, which are presented in a separate section.

**Selecting puzzles by reward:** For the booklet, approximately 4 million chess puzzles were generated by sampling positions from the models described above - a similar size to the Lichess database. The positions were sorted according to the reward function described above and evaluated by a group of chess players at Google. Although the quality of the positions was generally high, and the players found them to be creative, engaging and fun, they have not reached the level of creativity that can be found in chess books or chess compositions.

Some examples of shortcomings include brute calculation, very long mates, a lot of material exchange, or positions where the opponent can give a lot of checks to save time. However, the chess engines found these positions difficult (for example, it is common to see the web version of stockfish change its evaluation after thinking for a few seconds). They may be interesting to study, and therefore some examples can be found in the "Puzzles adversarial to chess engines" section.

**Improving selection of puzzles with aesthetics:** To improve our process, we took inspiration from the chess literature, studied chess aesthetics and developed theme detectors. Although these detectors were not flawless and often classified uninteresting or poor occurrences of the themes, we were able to identify creative positions by reviewing the top 50 reward samples per theme and consulting with 2200-2300 FIDE rated players.

Our research suggests that an AI can potentially create interesting chess puzzles, but it still lacks a complete understanding of the nature of creativity in chess puzzles and does not provide a definitive answer to this question.

**Outline:** The selected puzzles are presented across three sets. The first set is organized into twelve sections: eleven highlight specific aesthetic chess themes, and the last features puzzles without a dominant theme. To allow for quality comparison, the thematic sections first show examples from existing literature (when possible), followed by our generated puzzles.

The second set comprises puzzles created through an evolutionary search process. The third set features puzzles observed to be particularly challenging for the Stockfish chess engine.

While many of our chess puzzles have annotated solutions, those from our recent experiments currently do not have comments or solutions. We plan to annotate those newer puzzles as time allows.

For the learning methods, which used the lichess data for training, we also present the three closest puzzles from the training dataset (this includes the auto-regressive pre trained model and the model that was trained with reinforcement learning, but not the evolutionary search method). Closeness was determined by calculating the edit distance between each generated puzzle and every puzzle in the training set; the three with the smallest distance were selected and reported in the booklet. Inspecting the differences between the gener-

ated positions and the closest puzzles may help to asses the novelty of the generated position.

The final section of this booklet presents all of our generated puzzles as FENs from the three sets. This table excludes solutions and comments, allowing readers to analyze and solve the puzzles without spoilers.

We hope you enjoy our puzzles, and we would be curious to hear any feedback or questions you might have.

### M.1 Sacrifice

**Book example:**

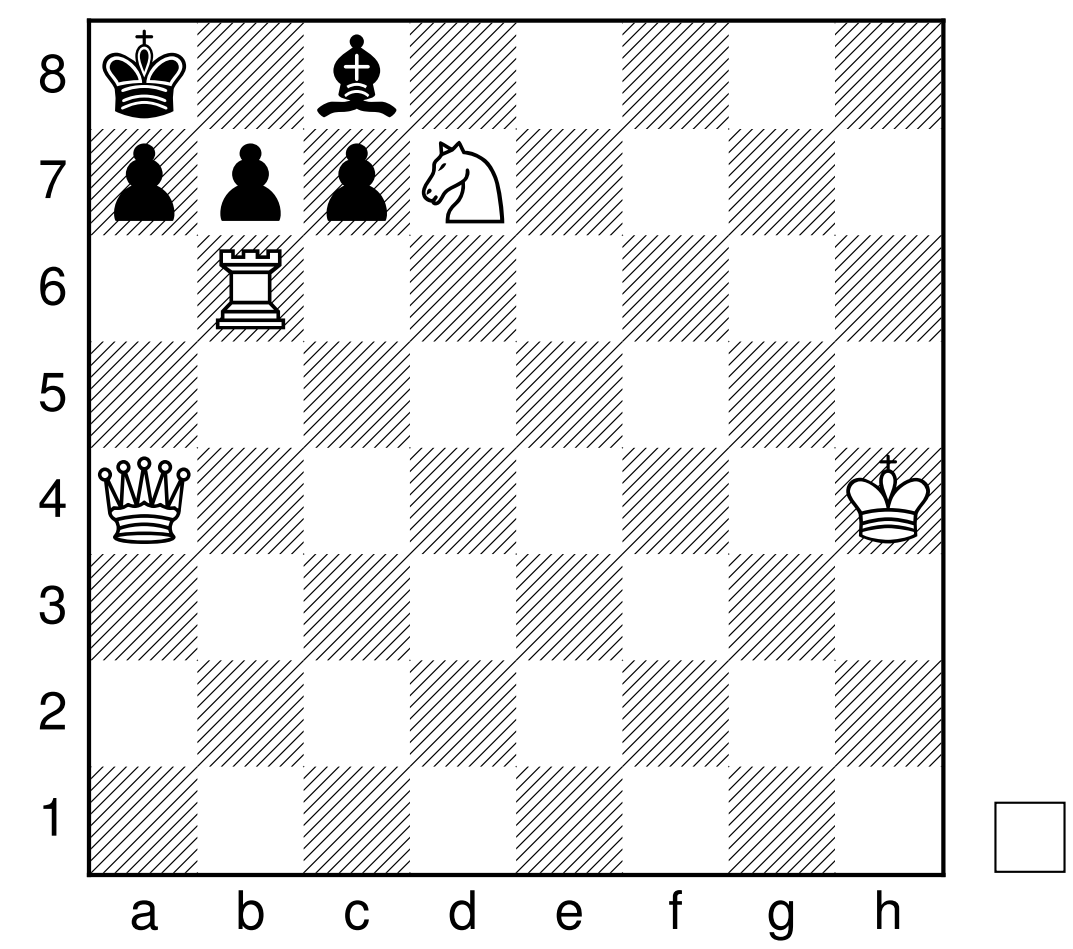

**[Analyse on Lichess]**

**Selected puzzles:**

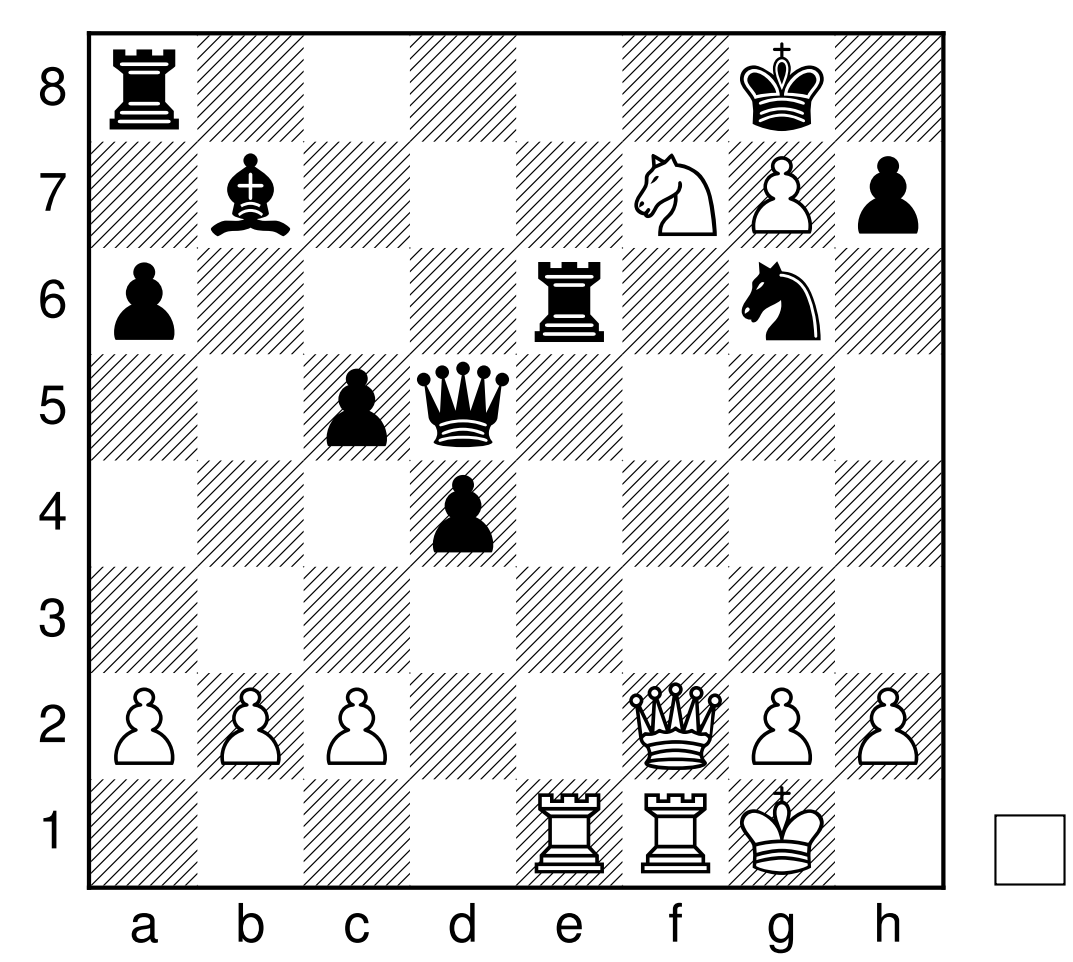

**[Analyse on Lichess]**

**Closest FENs - [1], [2], [3]**

1. Nd8! **The knight is sacrificed two ways, but cannot be captured due to Qf7#. Black is forced to give up material.** Qxg2+ 2. Qxg2 Bxg2 3. Rf8+! **An intermezzo sacrifice that aims to liquidate.** Nxf8 4. gxf8=Q+ Kxf8 5. Nxe6+ Kf7 6. Kxg2 **White plays the endgame up a piece.**

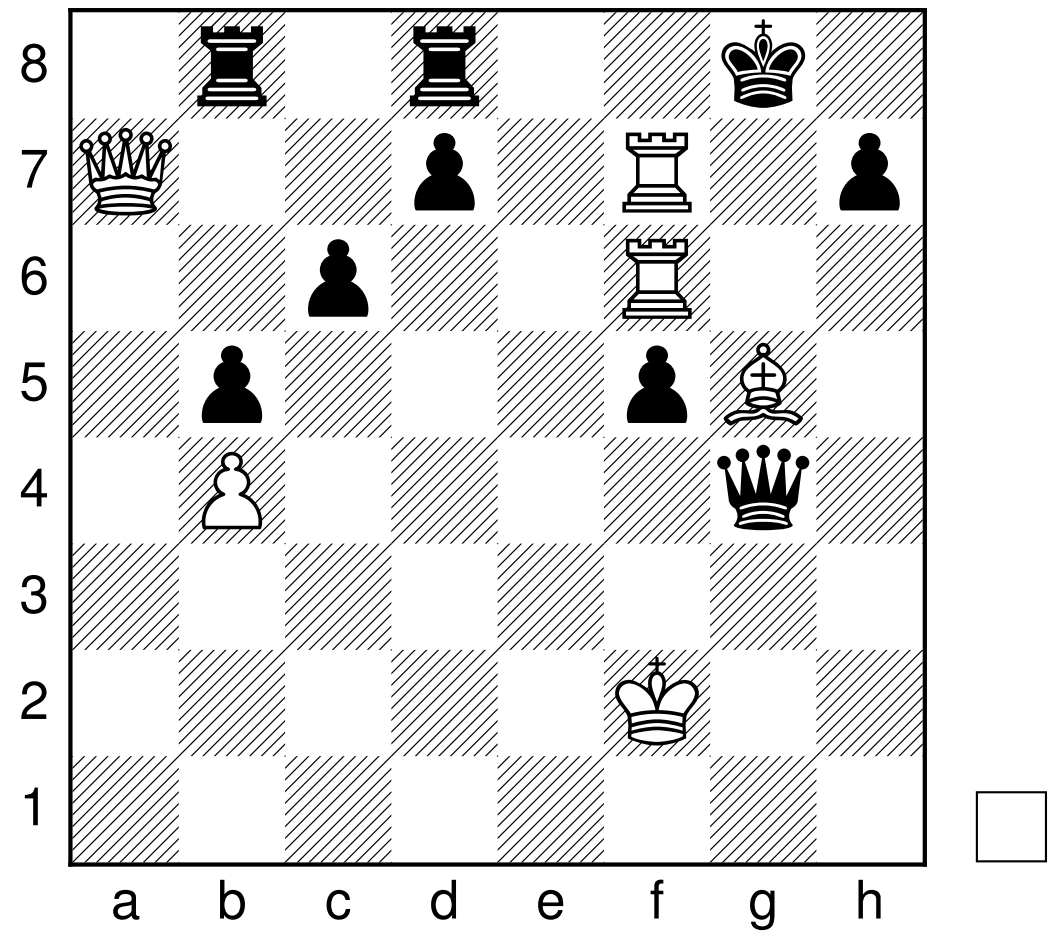

[Analyse on Lichess]

**Closest FENs - [1], [2], [3]**

1. Rg6+! **White gives up both rooks to open up the a1-h8 diagonal. Capturing either rook eventually transposes to the same position.** hxg6 2. Qa1! Kxf7 3. Qf6+ Kg8 4. Bh6 **and White covers all the checks**.

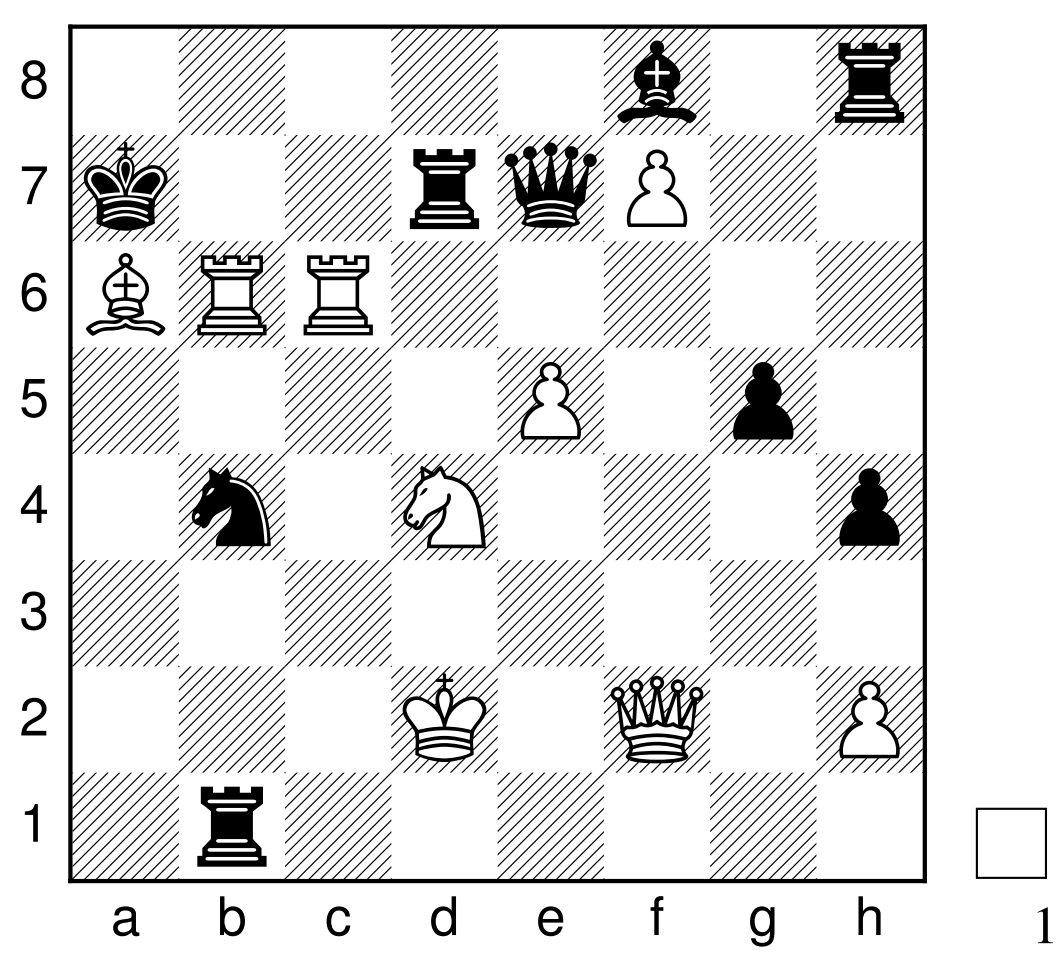

[Analyse on Lichess]

**Closest FENs - [1], [2], [3]**

1. Rb7+! **White gives up the rook to open the diagonal for the queen.** Rxb7 2. Nb5+ Kb8 3. Qa7+ **sacrificing the queen to finish the game.** Rxa7 4. Rc8#.

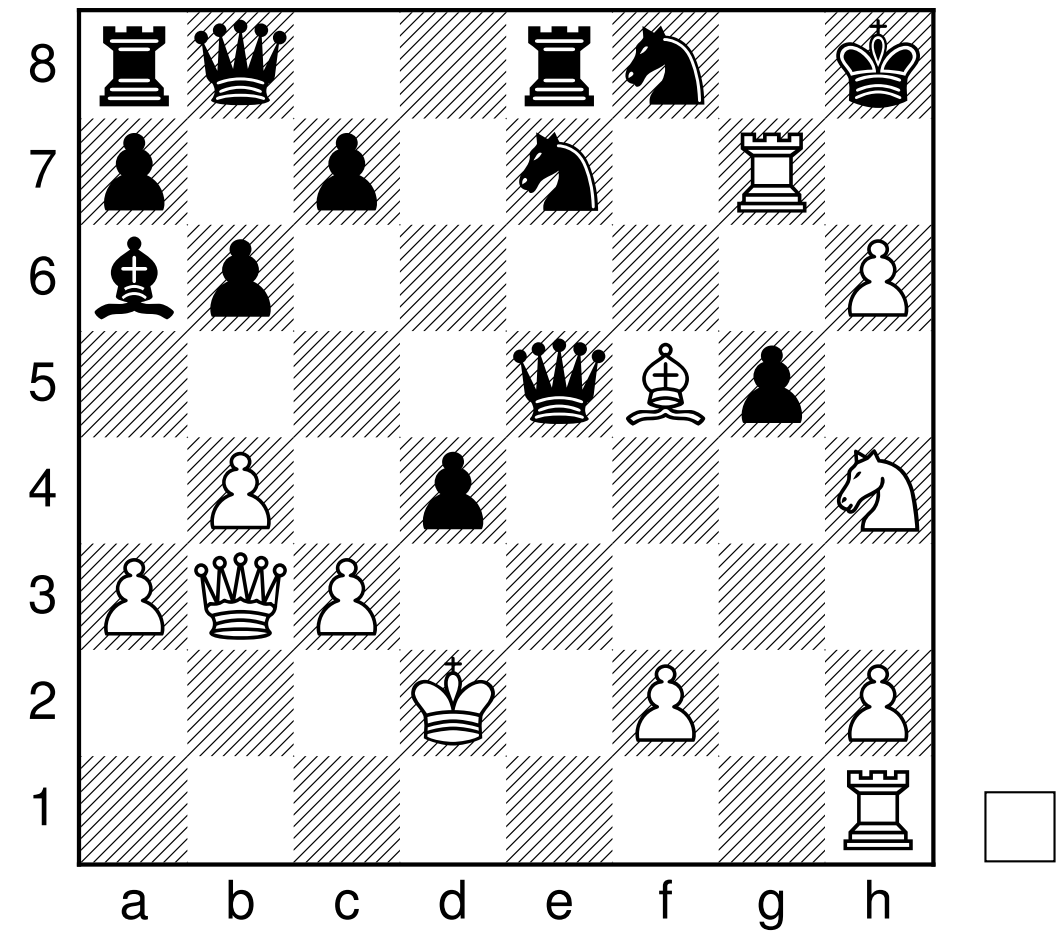

[Analyse on Lichess]

**Closest FENs - [1], [2], [3]**

1. Qg8+ Nxg8 2. Rh7+ Nxh7 3. Ng6# **A creative example of a smothered mate.**

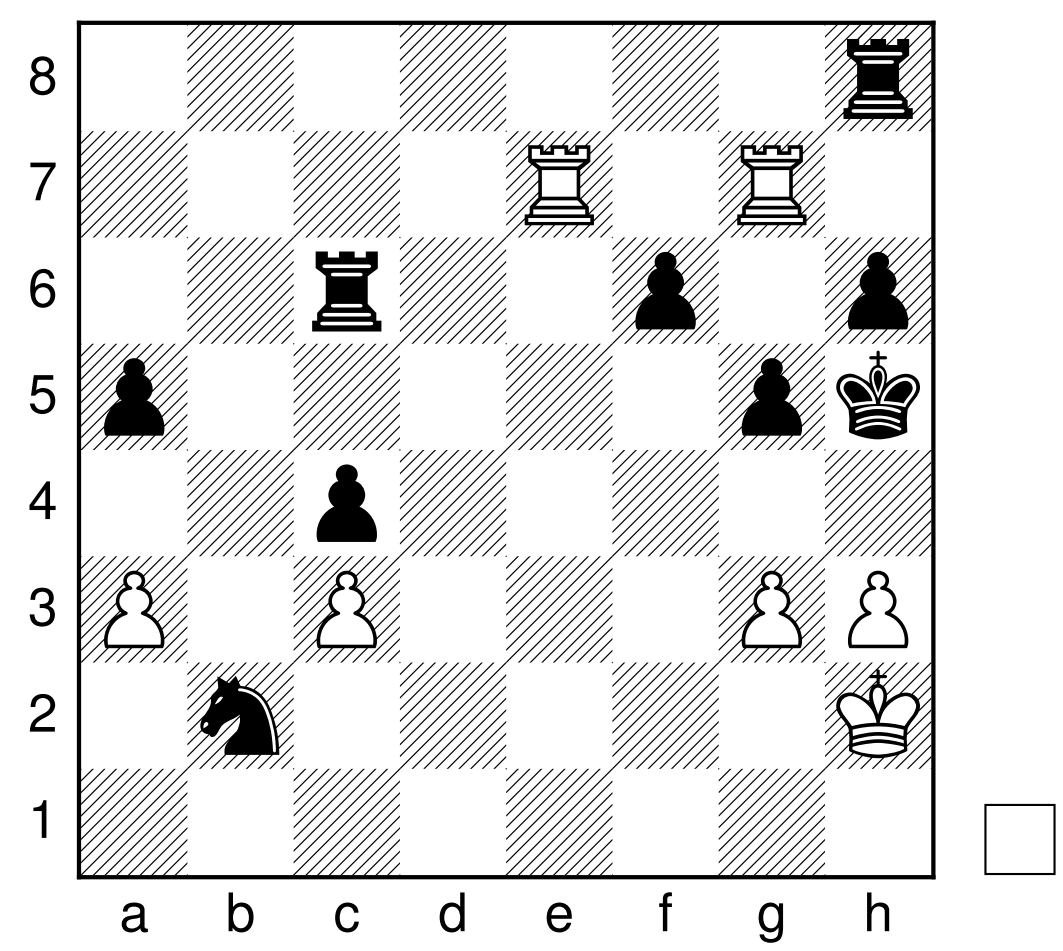

[Analyse on Lichess]

**Closest FENs - [1], [2], [3]**

1. Re4 **Setting up a mating net.** f5 2. Rh4+! gxh4 3. g4+ fxg4 4. hxg4#.

### M.2 Underpromotion [48]

**Book example:**

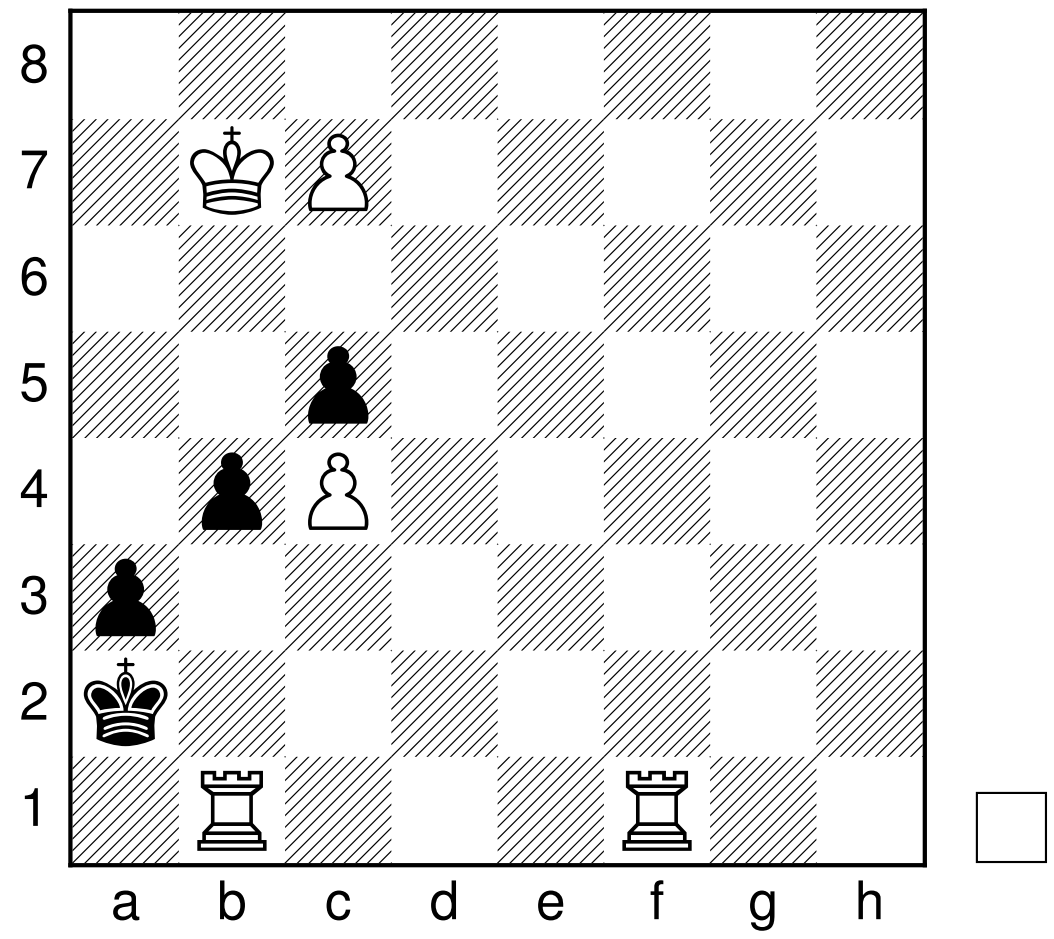

[Analyse on Lichess]

**Selected puzzles:**

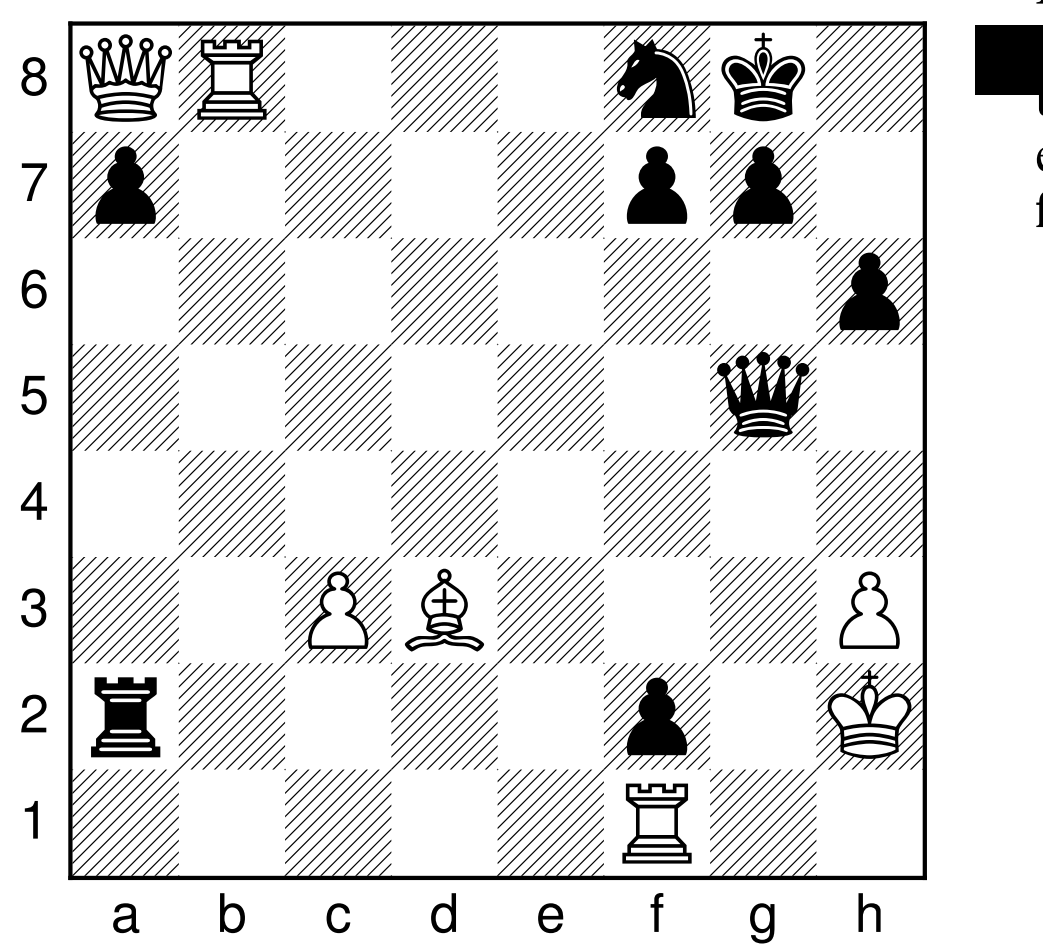

[Analyse on Lichess]

**Closest FENs - [1], [2], [3]**

1... Qg1+! **Black gives up the queen to set up the underpromotion and mate.** 2. Rxg1 f1=N+! 3. Kh1 Rh2#.

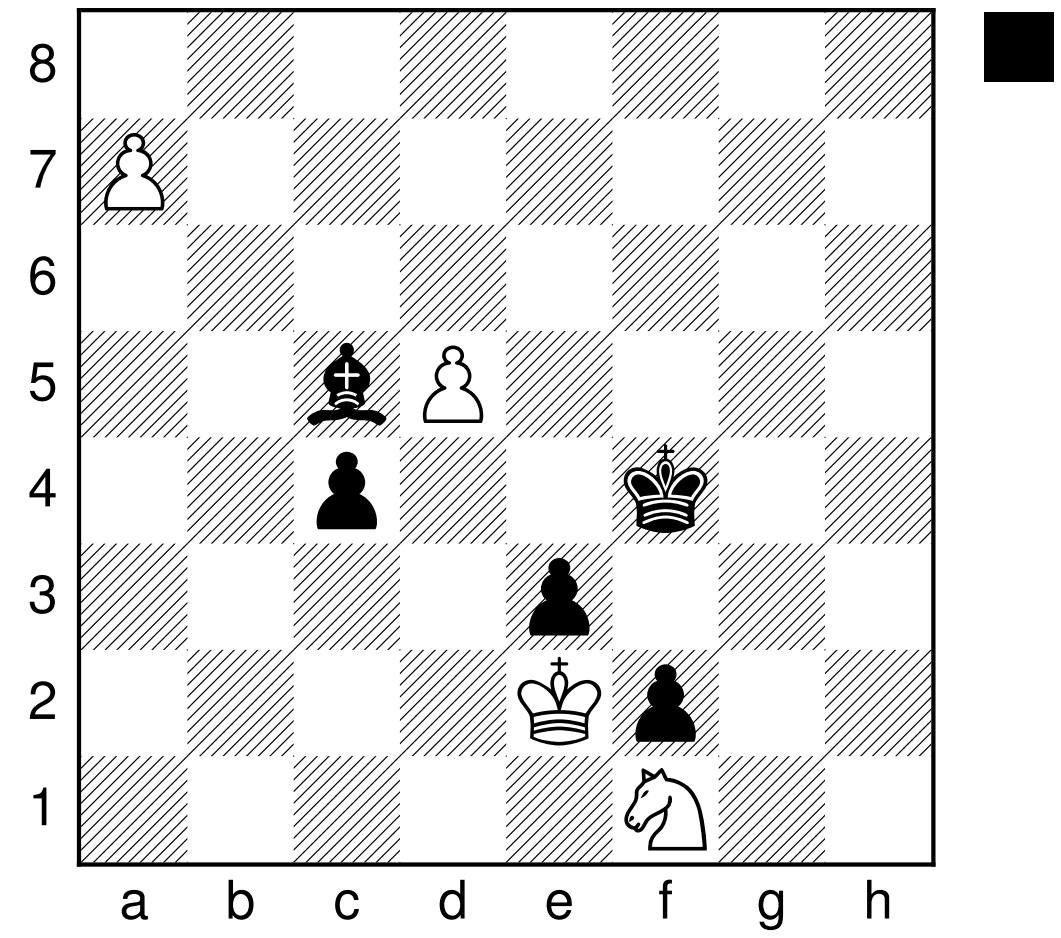

[Analyse on Lichess]

**Closest FENs - [1], [2], [3]**

1... Bxa7 2. d6 c3 3. d7 **The position looks like it is heading for a draw as it seems like the black bishop has to stop White's pawn. Black however has another idea in mind.** c2! 4. d8=Q c1=N! **The only winning line! Black underpromotes with tempo.** 5. Kd1 e2+ 6. Kc2 exf1=Q **Black can eventually escape the checks finding shelter on g1 covered by the new queen.**

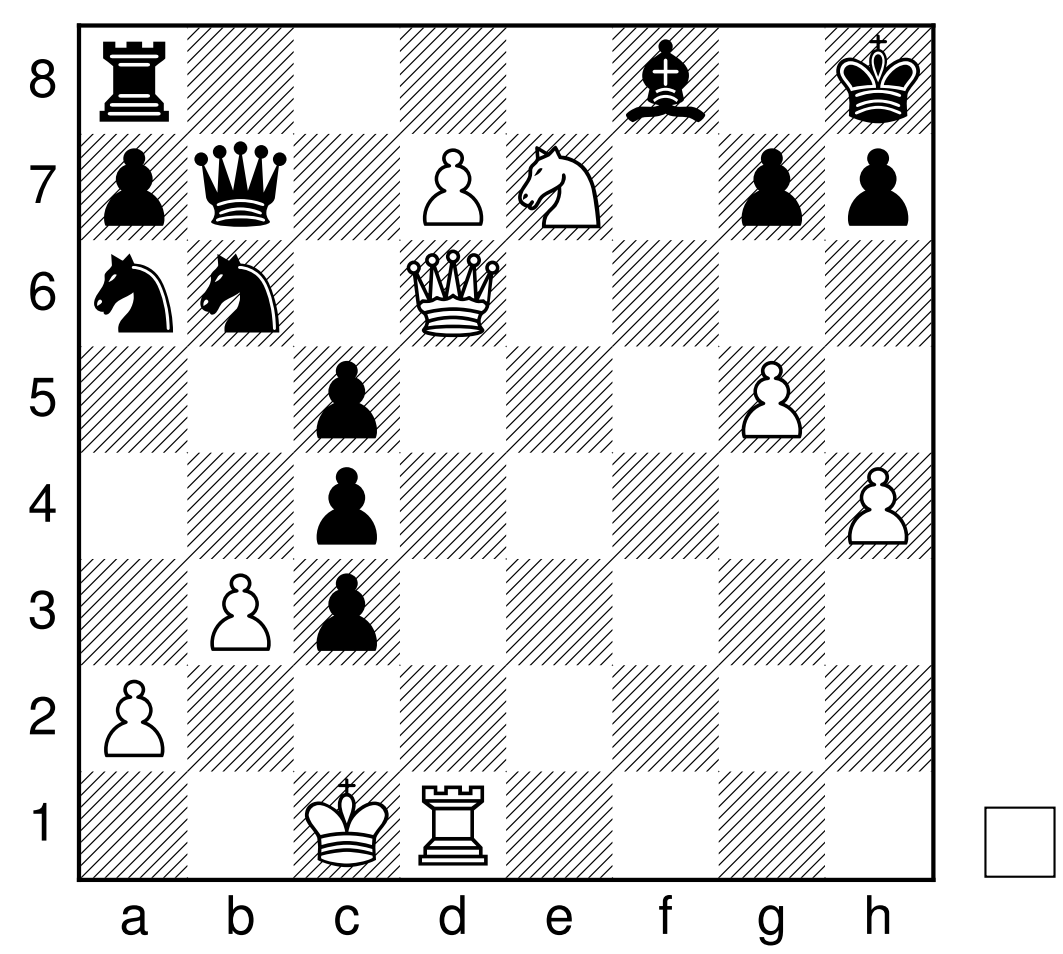

[Analyse on Lichess]

**Closest FENs - [1], [2], [3]**

1. Qe6 **Threatening mate.** Bxe7 2. d8=N! **Underpromoting to threaten a smothered mate. White's best option is to give up material.** Qf3 3. Nf7+ Qxf7 4. Qxf7

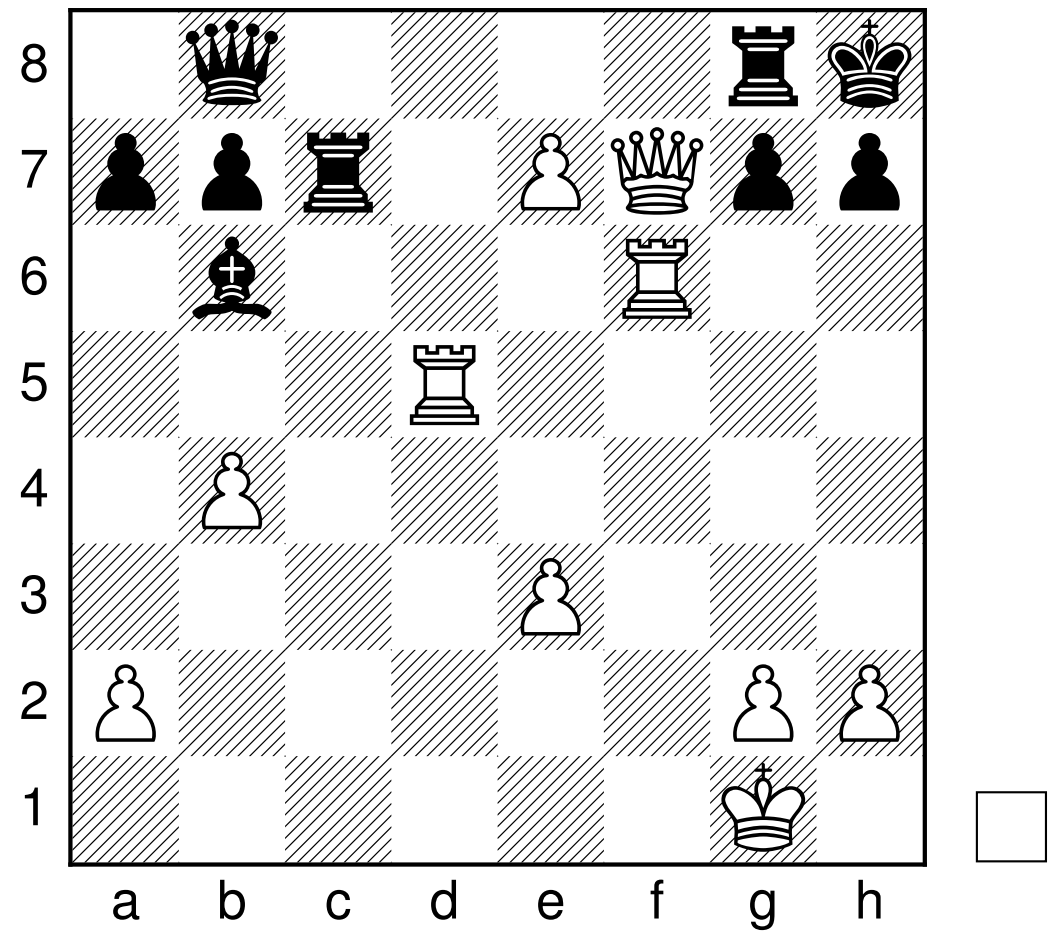

[Analyse on Lichess]

**Closest FENs - [1], [2], [3]**

1. Rd8! **White exploits Black's back-rank issues.** Qxd8! 2. exd8=N! **Underpromoting to a knight is the only move that wins!** Rc1+ 3. Kf2 Bxd8 4. Re6 **White is up a material with a winning position.**

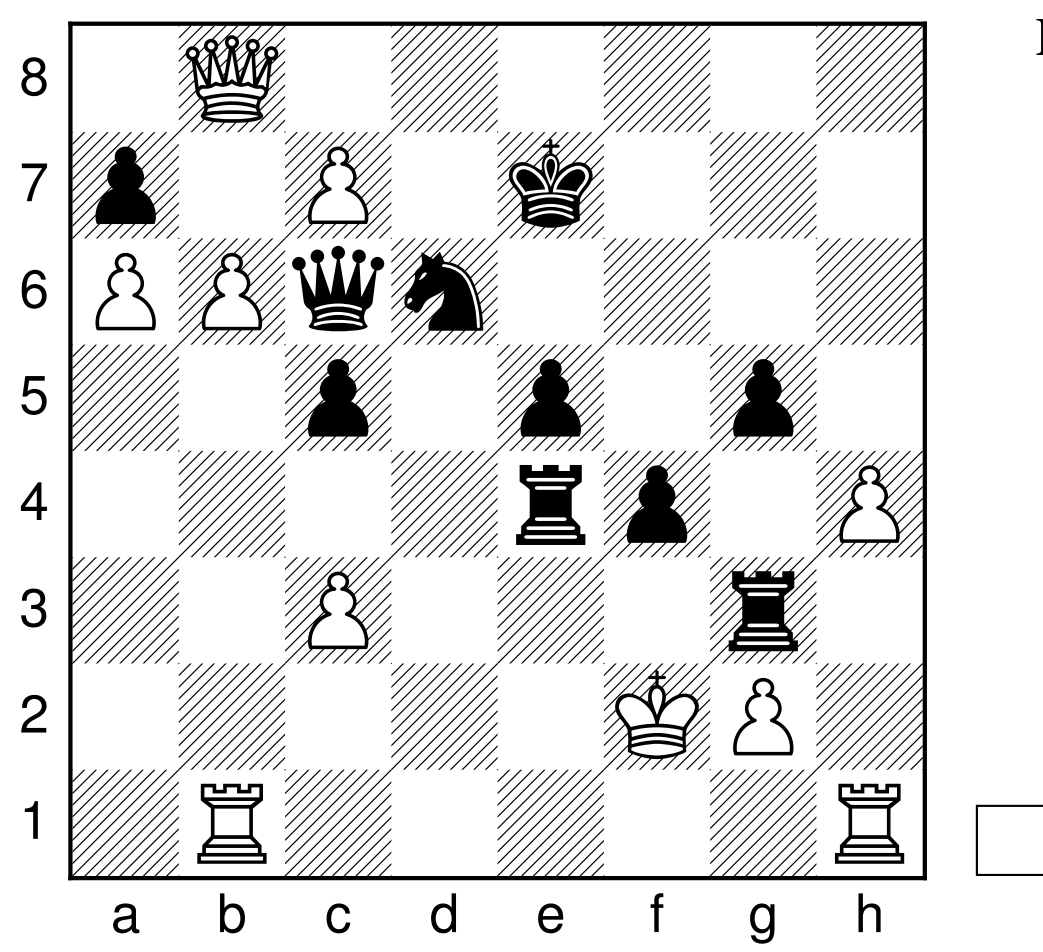

[Analyse on Lichess]

**Closest FENs - [1], [2], [3]**

1. c8=N! **White underpromotes to a knight.** Nxc8 **if 1... Qxc8 2. Qxc8 Nxc8 3. b7 or if 1... Ke6 2. Qxd6 Qxd6 3. Nxd6** 2. Qb7+! **Forcing matters.** Kd6 3. Qxc6 Kxc6 4. b7.

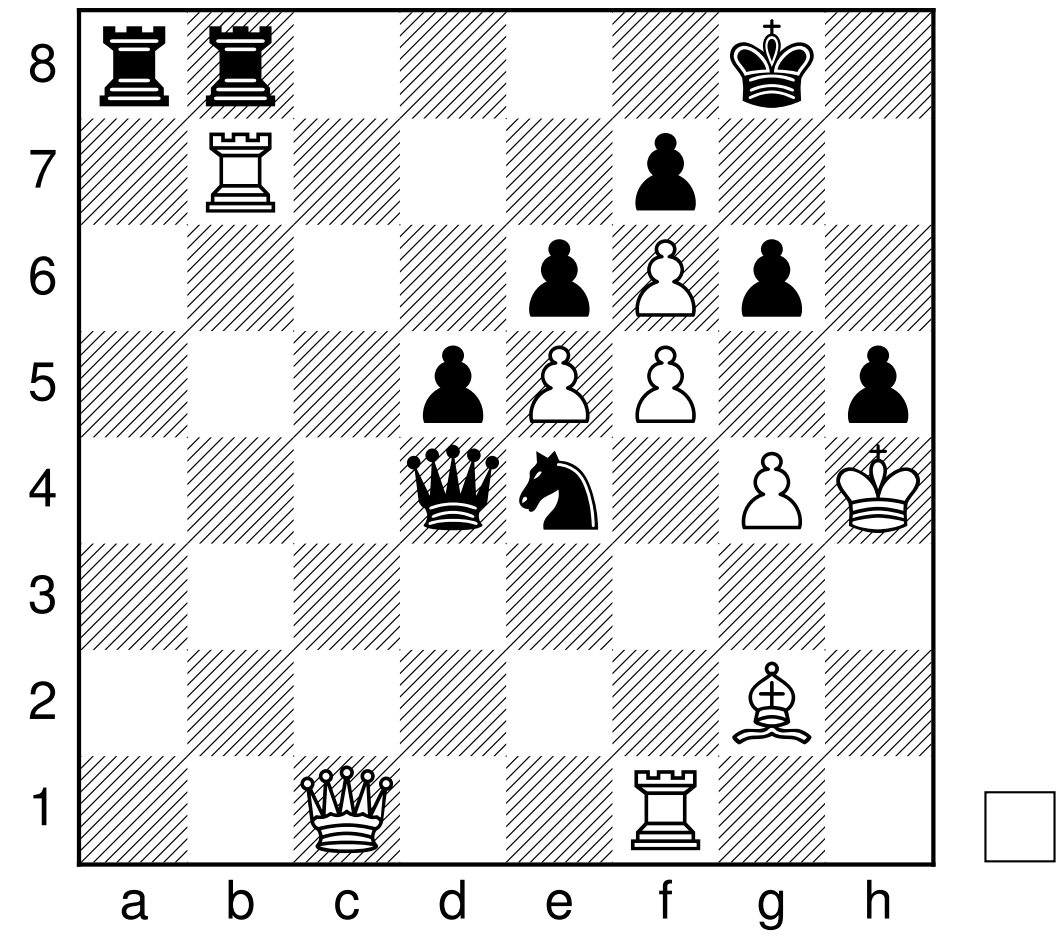

[Analyse on Lichess]

**Closest FENs - [1], [2], [3]**

1. Rxb8+ Rxb8 2. fxg6 Nd2 **If 2... fxg6 3. Qc7 wins.** 3. gxf7+ Kh7! 4. f8=N+! Kh6! 5. Be4! **An underpromotion followed by a stunning bishop sacrifice. Black cannot capture the bishop.** Qe3 6. Nxe6 **White ends up with a large material advantage and should convert quickly with careful play.**

### M.3 Attacking Withdrawal [2]

**Book example:**

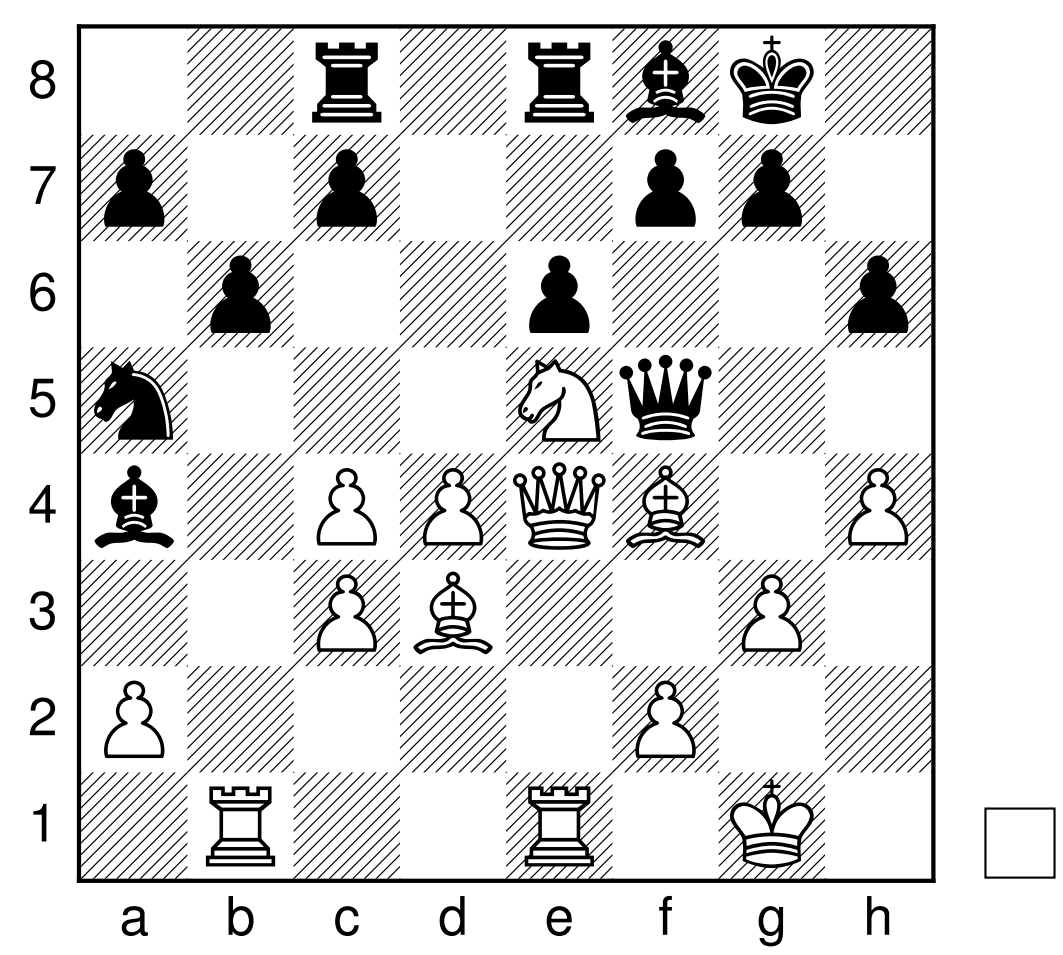

[Analyse on Lichess]

**Selected puzzles:**

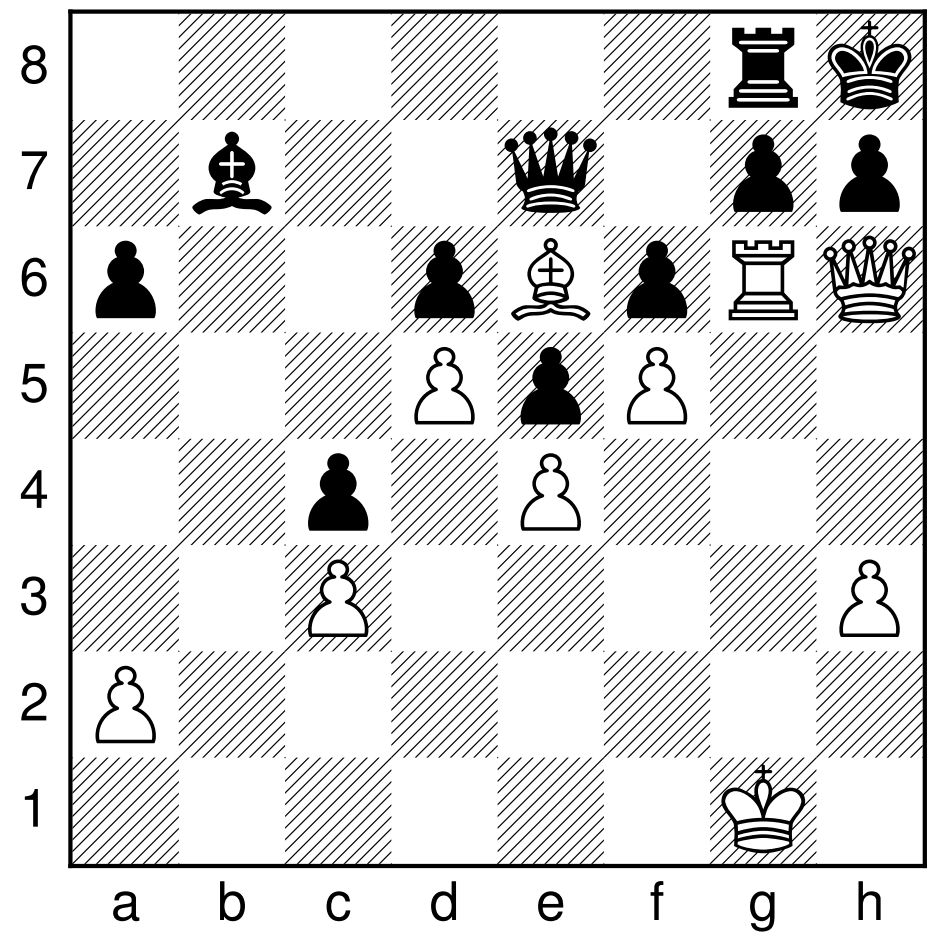

[Analyse on Lichess]

**Closest FENs - [1], [2], [3]**

1. Rg4 **White retreats the rook and sets up the threat of 2. Qxh7+ Kxh7 3. Rh4#.** g5 2. Bxg8 Kxg8 3. h4 **and White is quickly breaking through.**

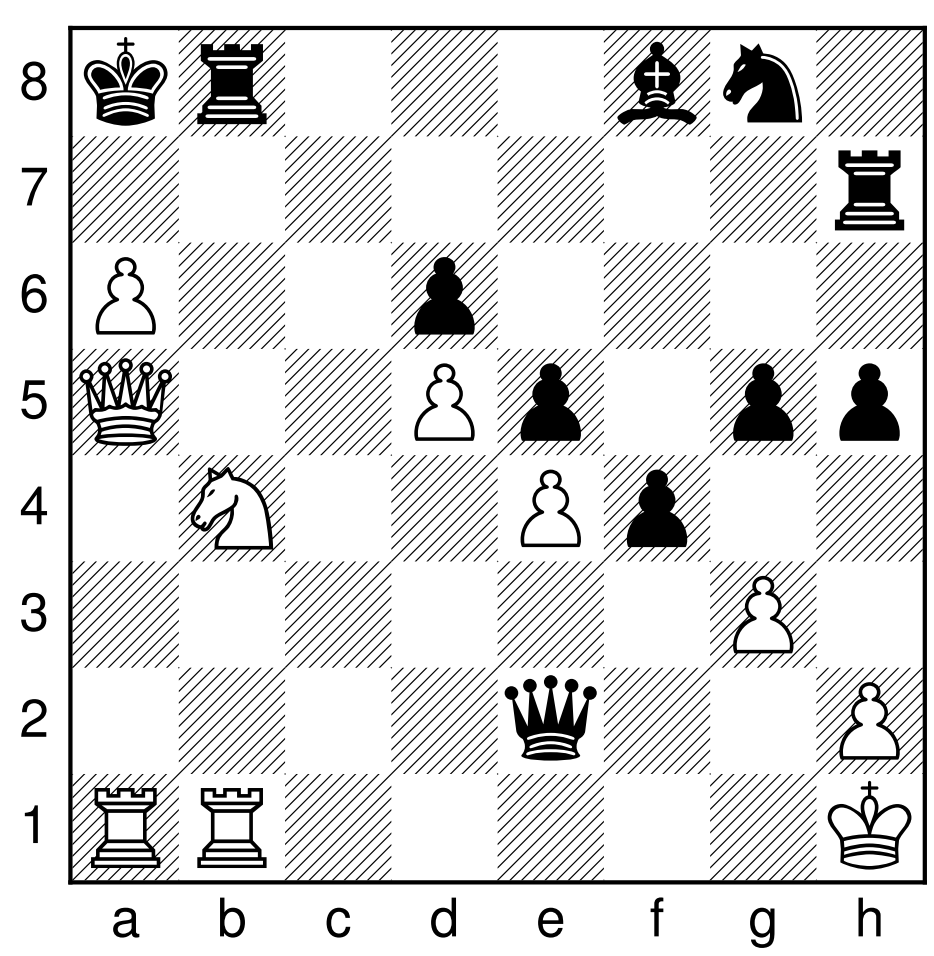

[Analyse on Lichess]

**Closest FENs - [1], [2], [3]**

1. Nc2! **The white knight retreats and sacrifices itself to gain valuable time.** Qxe4+ 2. Kg1 Qxc2 3. Rxb8+ Kxb8 4. a7+ Rxa7 5. Qxa7+ **and White has an unstoppable attack.**

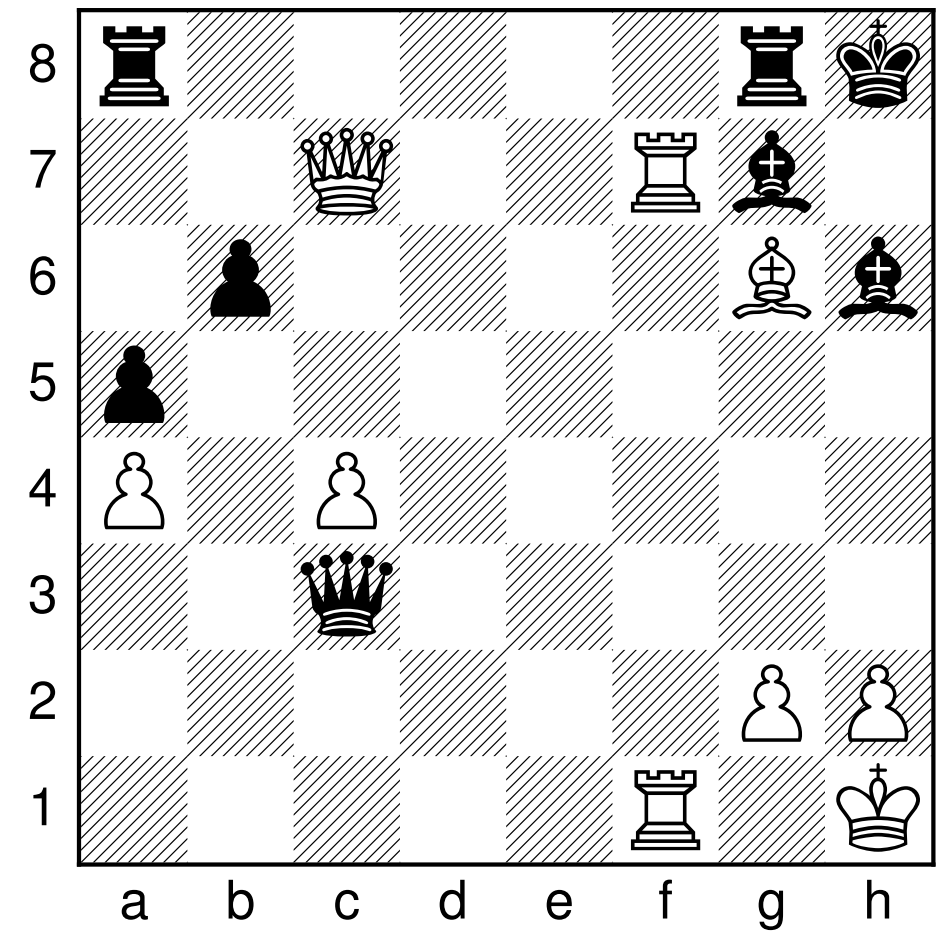

[Analyse on Lichess]

**Closest FENs - [1], [2], [3]**

1. Bb1! **White withdraws the bishop from the attack with the goal of setting up a bishop-queen battery. It turns out that this plan is largely unstoppable.** Qb4 **Engine also gives Bg5, but after 2. R7f3 Black has to depart with the queen and is defending with less material.** 2. Qc6 **Threatening both 3. Qg6 and 3. Qxh6!. Black cannot defend.** Rgd8 3. Qg6 Qxb1 **Black has to give up the queen.**

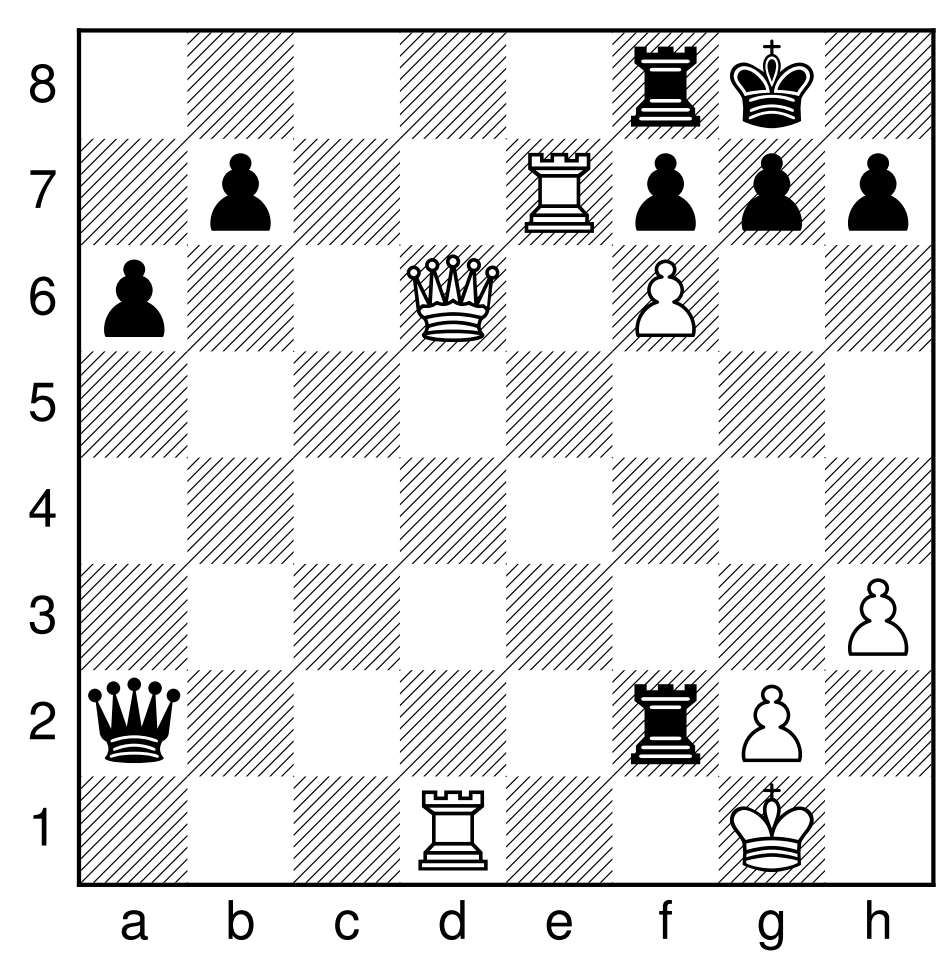

[Analyse on Lichess]

**Closest FENs - [1], [2], [3]**

1. Re3! **The onyl winning withdrawing move.** Rxg2+ 2. Kh1 gxf6 **If 2... h6 3. Qxf8+!** 3. Rg3+ Rxg3 4. Qxg3+ Kh8 5. Qd6! **Black cannot defend the hanging rook and pawn on f6.**

### M.4 Knight on the Rim is Dim [66]

**Book example:**

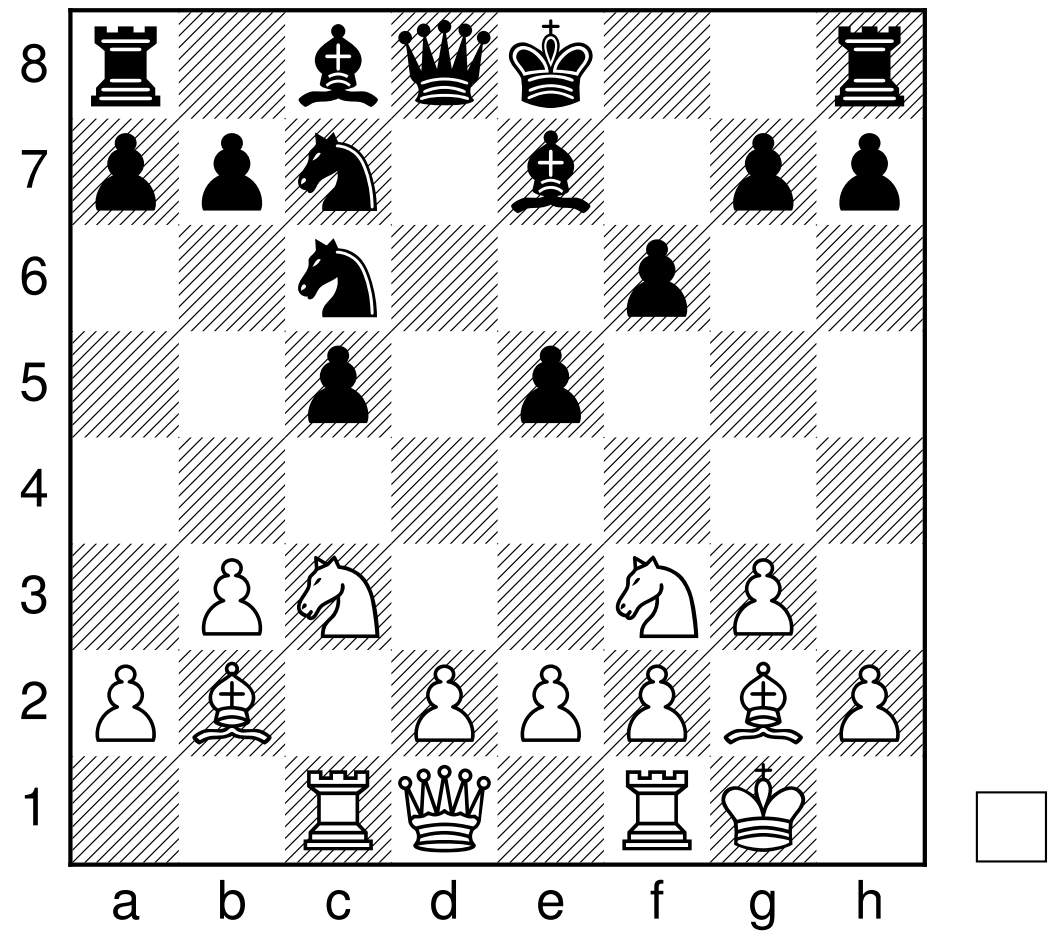

[Analyse on Lichess]

**Selected puzzles:**

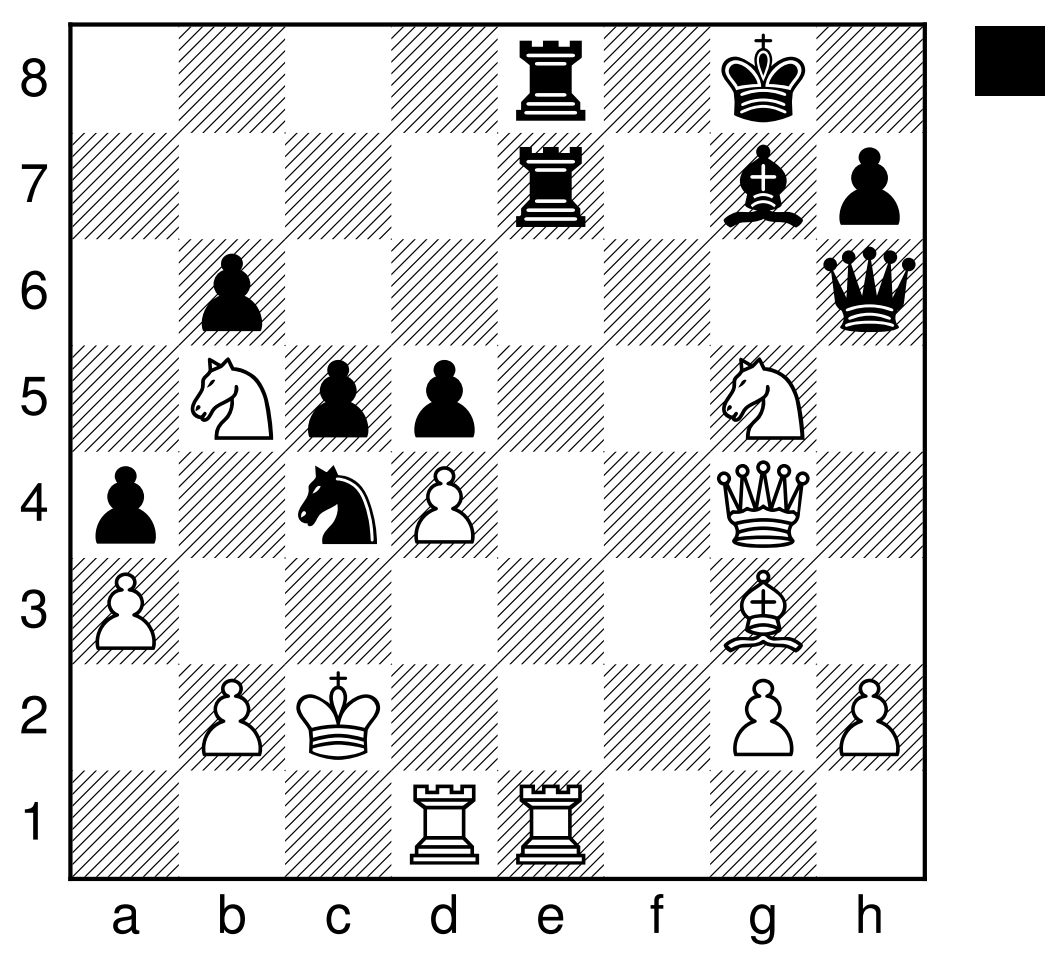

[Analyse on Lichess]

**Closest FENs - [1], [2], [3]**

1... Qg6+ 2. Kc1 Na5! **Black places the knight on the rim, threatening checkmate.** 3. b3 axb3 **White surprisingly has no good way to defend the threat of Qc2**.

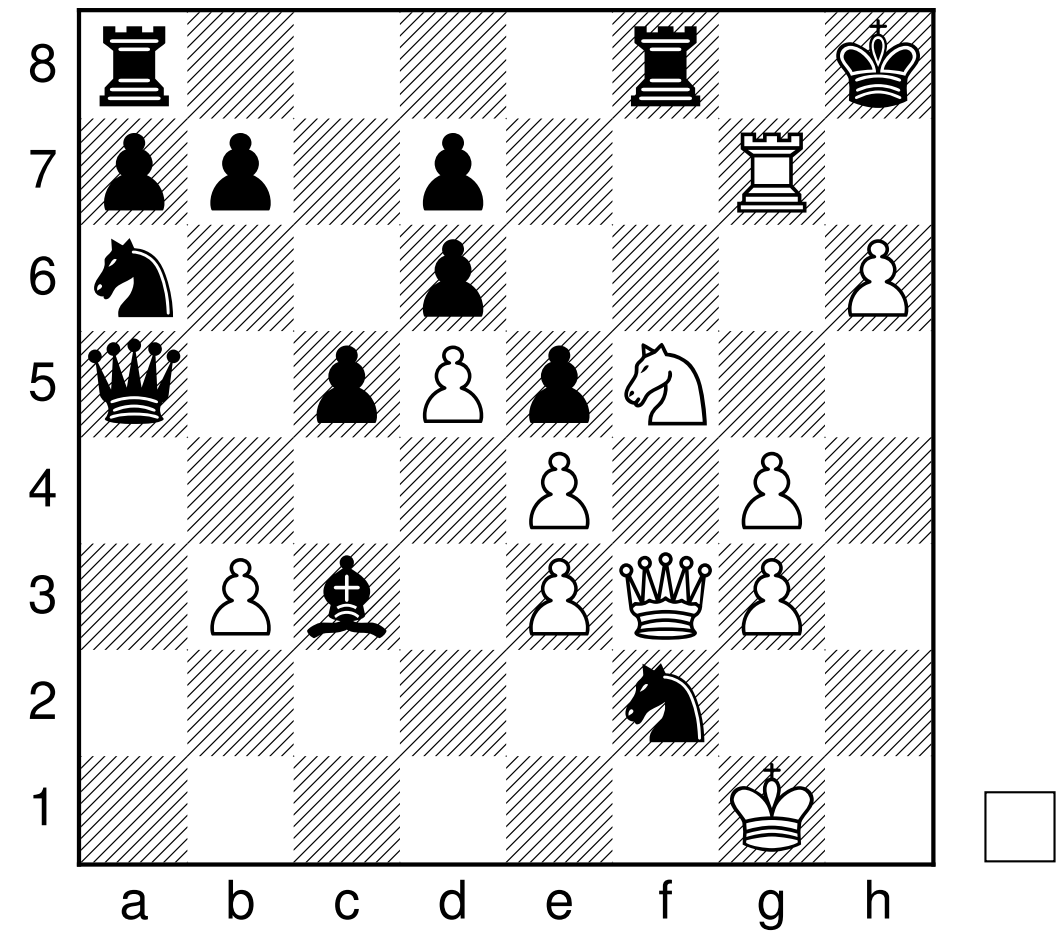

[Analyse on Lichess]

**Closest FENs - [1], [2], [3]**

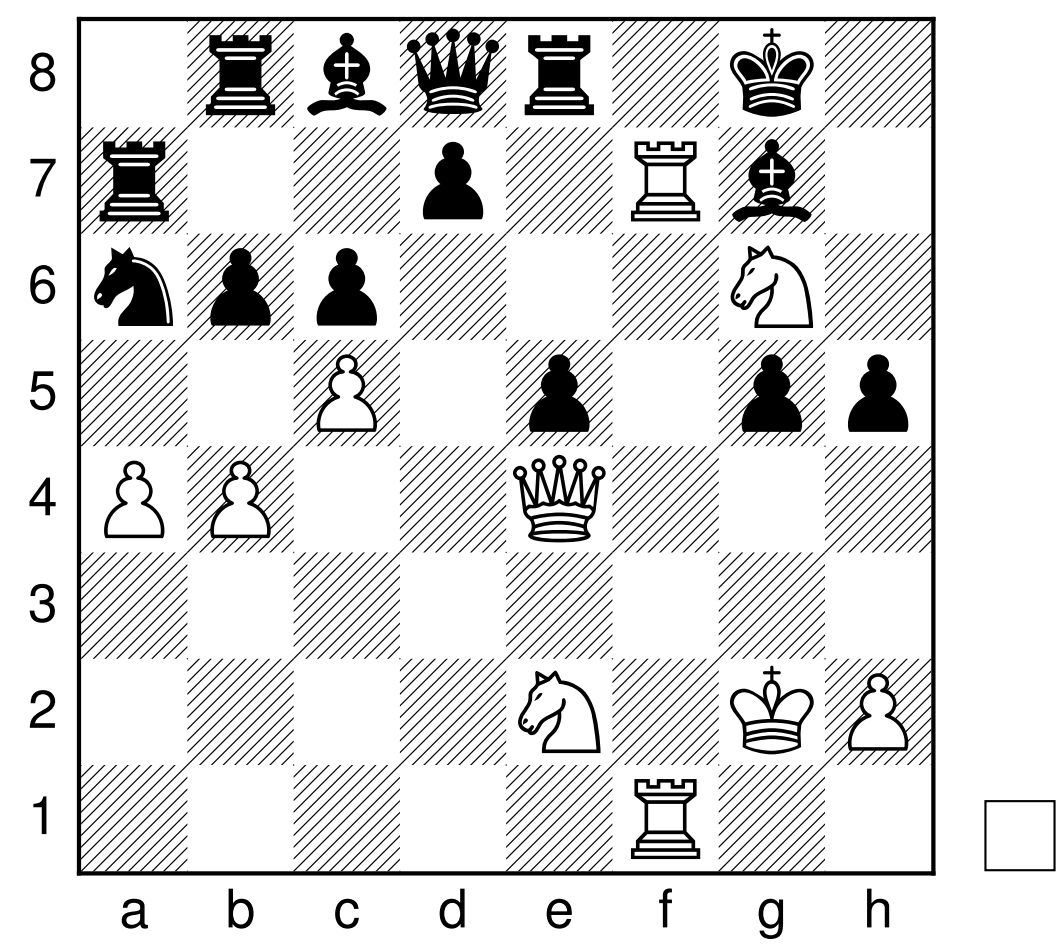

[Analyse on Lichess]

**Closest FENs - [1], [2], [3]**

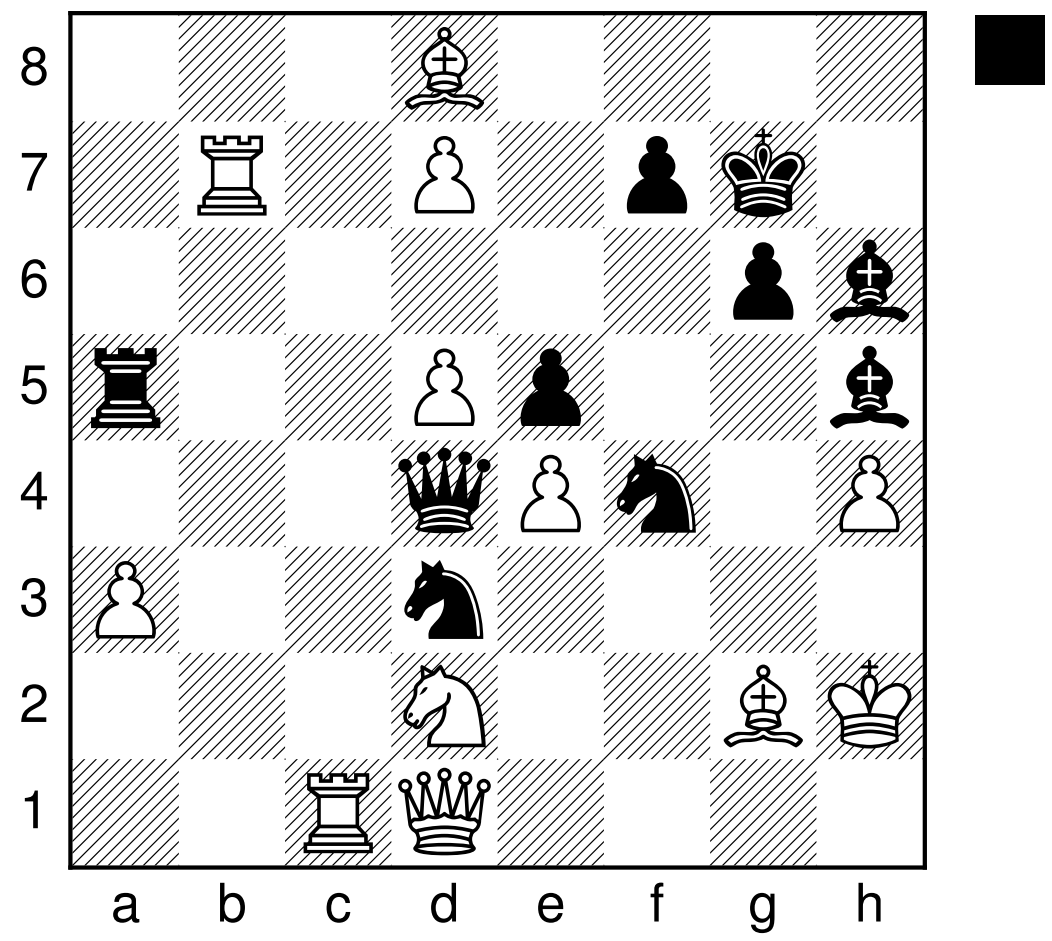

[Analyse on Lichess]

**Closest FENs - [1], [2], [3]**

## M.5 Sacrifice Pieces to Stalemate [2]

**Book example:**

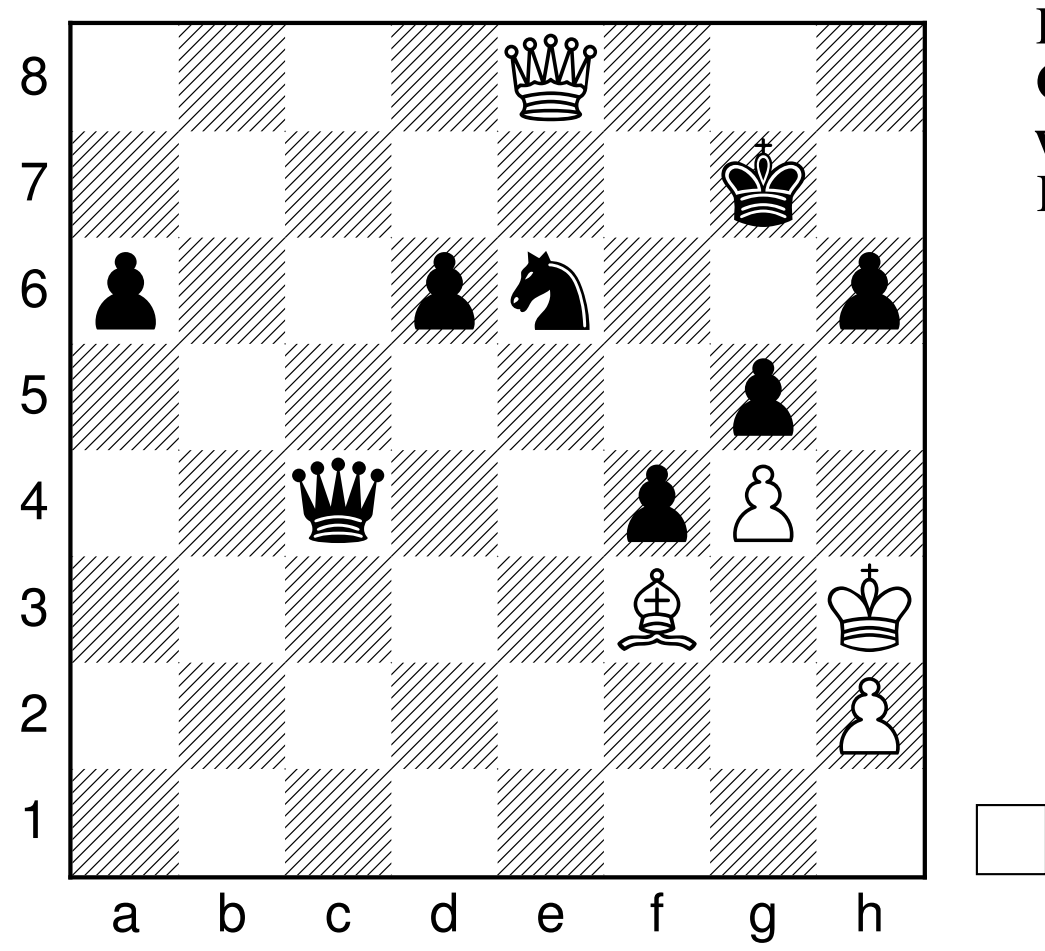

[Analyse on Lichess]

**Selected puzzles:**

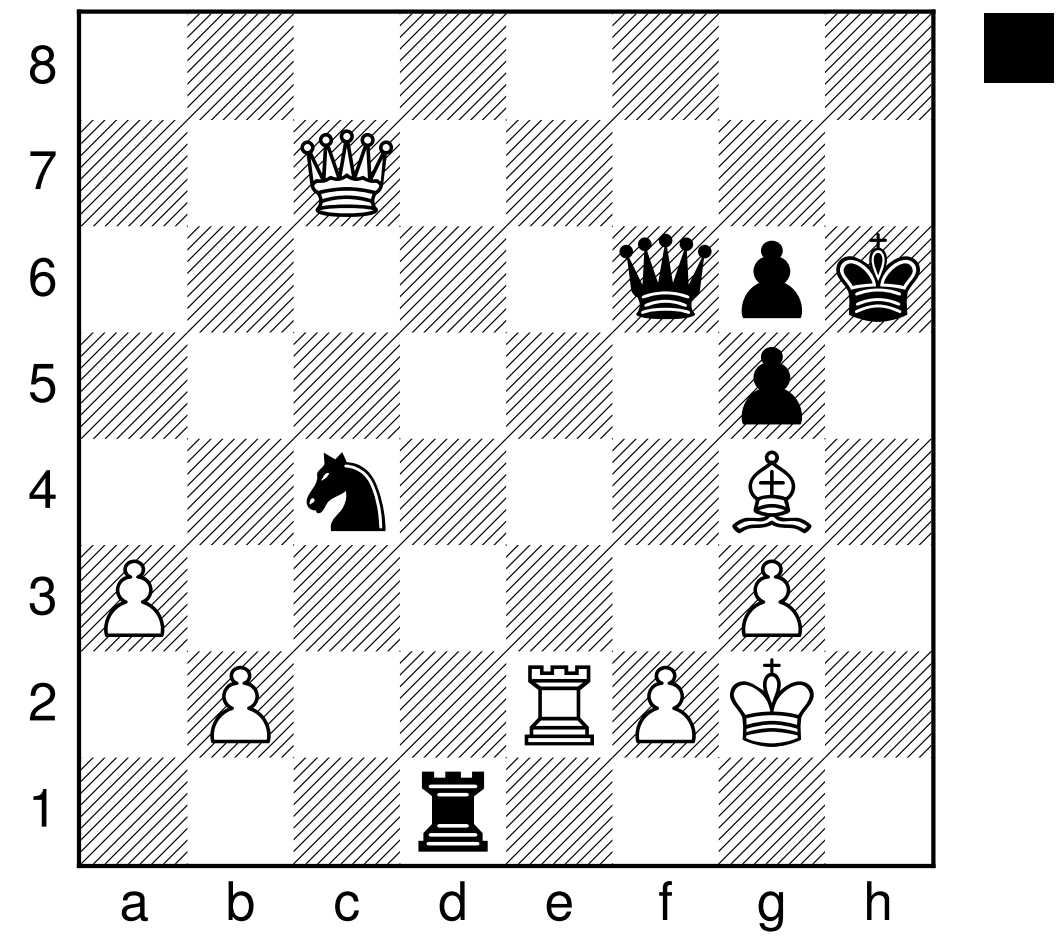

[Analyse on Lichess]

**Closest FENs - [1], [2], [3]**

1... Ne3+! 2. Rxe3 **White is forced to accept the sacrifice due to the bishop hanging on g4.** Rg1+! **A second sacrifice!** 3. Kxg1 (**3. Kh3 Qxf2 and White is forced to give up the rook with Qb7 to stop mate.**) Qxf2+ 4. Kh1 Qg1+ 5. Kxg1 **Stalemate**.

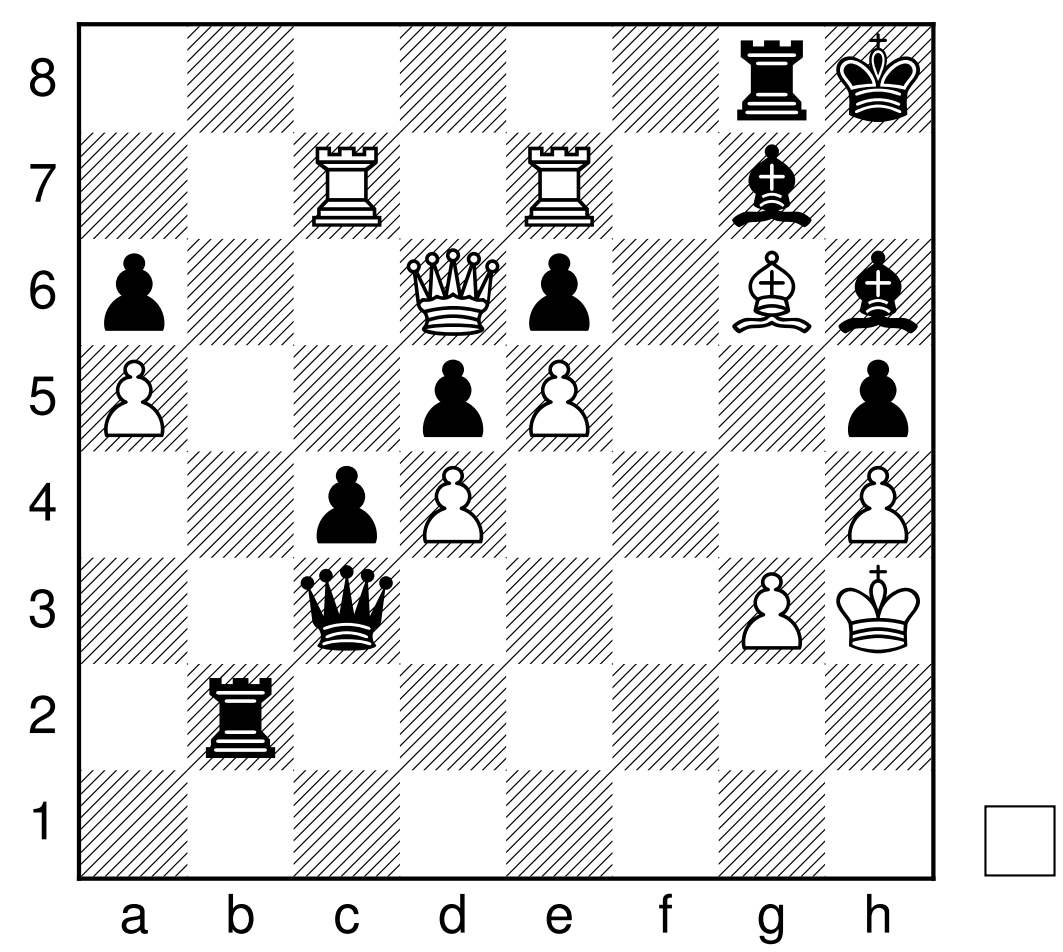

[Analyse on Lichess]

**Closest FENs - [1], [2], [3]**

1. Rxg7! **White starts a chain of sacrifices that surprisingly force a draw.** Bxg7 (**1... Rxg7?? 2. Qf8+! Rg8 3. Qxh6#**) 2. Rxg7! Kxg7 3. Qe7+ Kxg6 4. Qh7+ Kxh7 **Stalemate.**

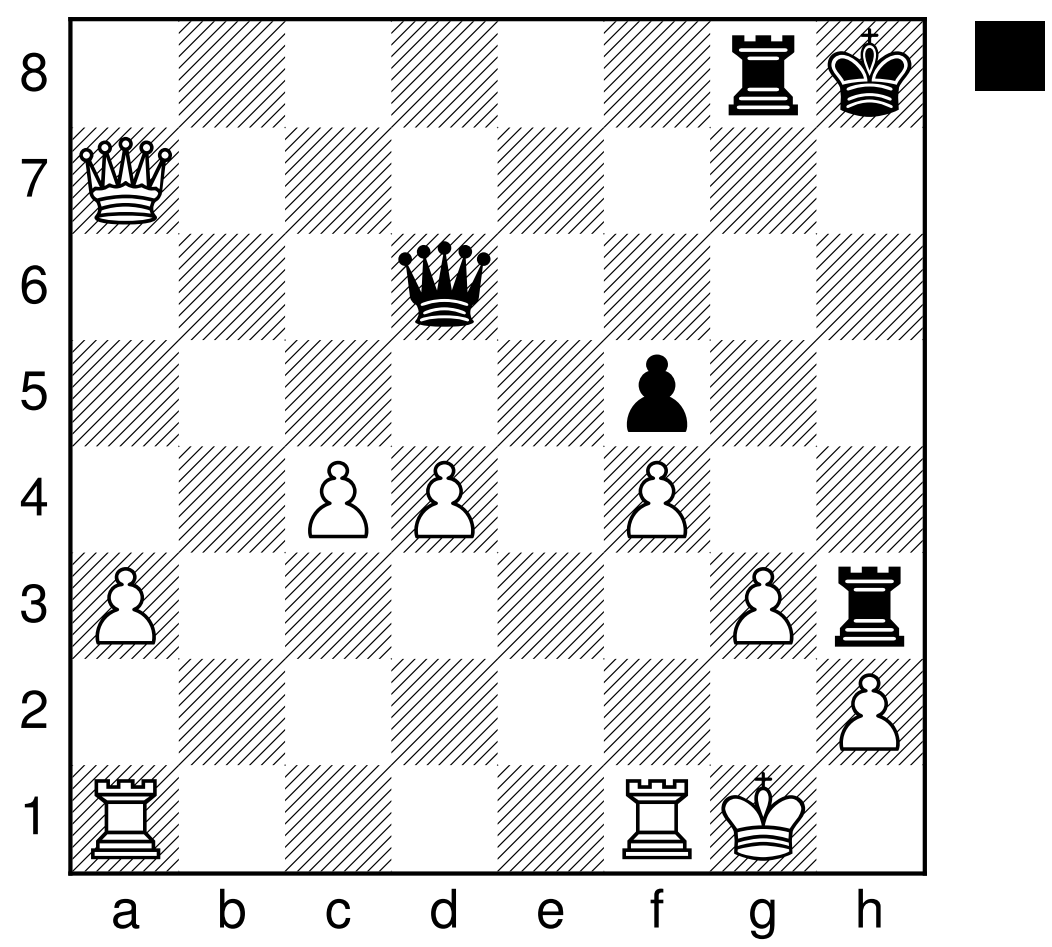

[Analyse on Lichess]

**Closest FENs - [1], [2], [3]**

1... Rxh2! 2. Kxh2 Qh6+ 3. Kg2 Qh4! **Black sets up strong mate threats that White has to address.** 4. Rf3 (**4. Rh1?? Rxg3 5. Kf2 Rh3+ and Black wins.**) Rxg3+! 5. Rxg3 **Black has managed to set up the stalemate with the help of White's rook.** Qh2+! 6. Kf3 Qe2+ 7. Kxe2 **Stalemate.**

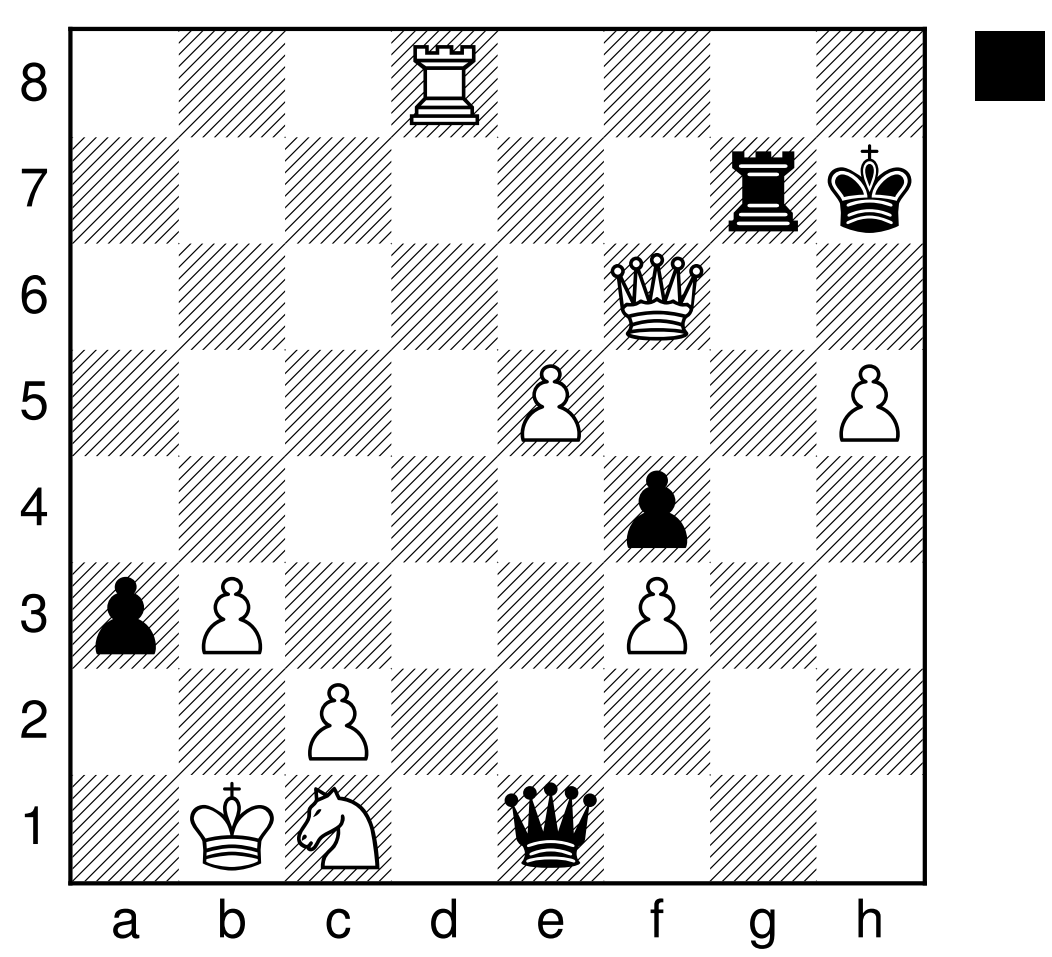

[Analyse on Lichess]

**Closest FENs - [1], [2], [3]**

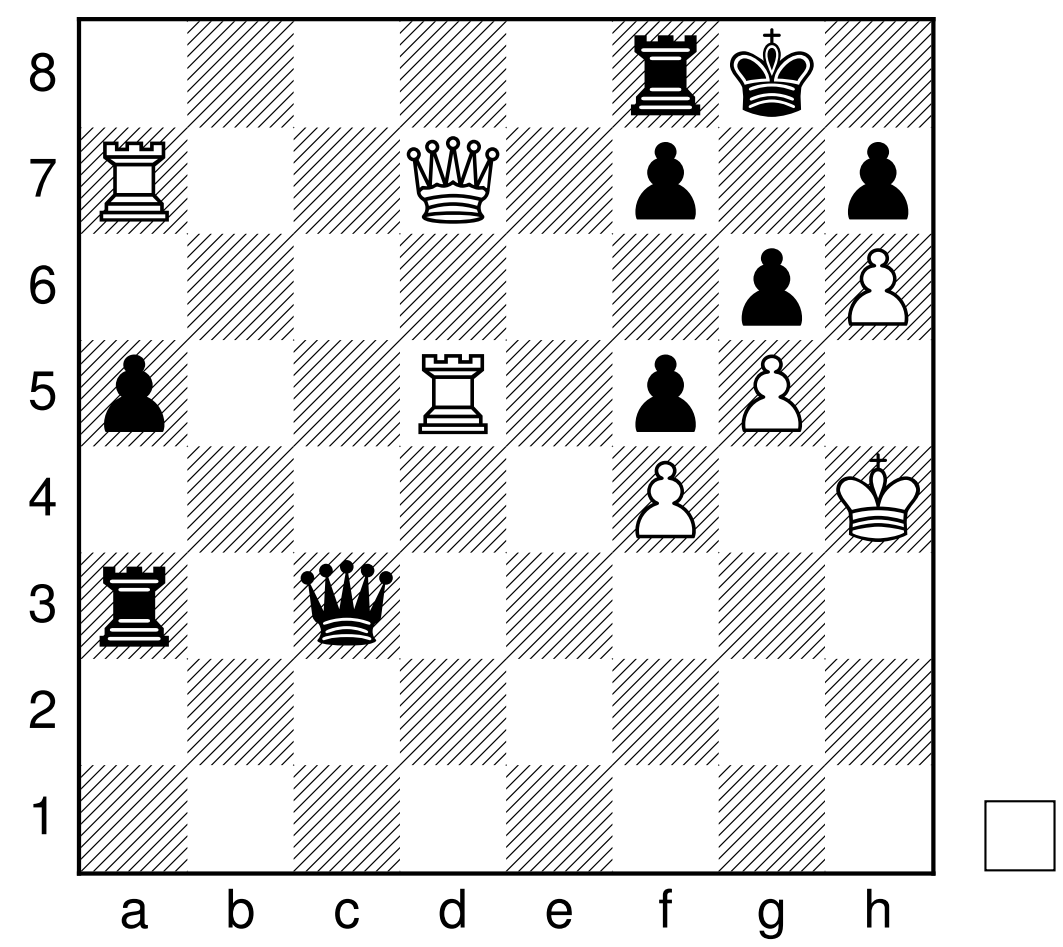

[Analyse on Lichess]

**Closest FENs - [1], [2], [3]**

### M.6 Novotny [48]

**Book example:**

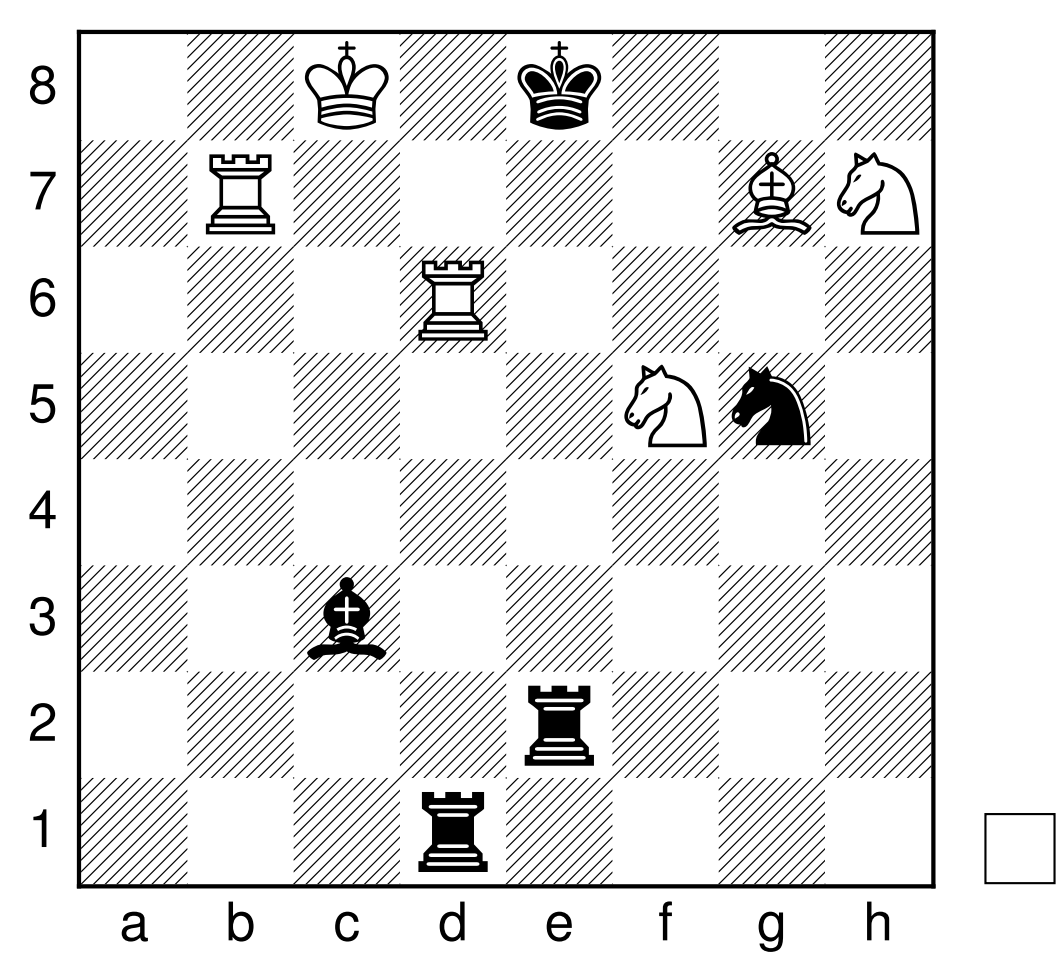

[Analyse on Lichess]

**Selected puzzles:**

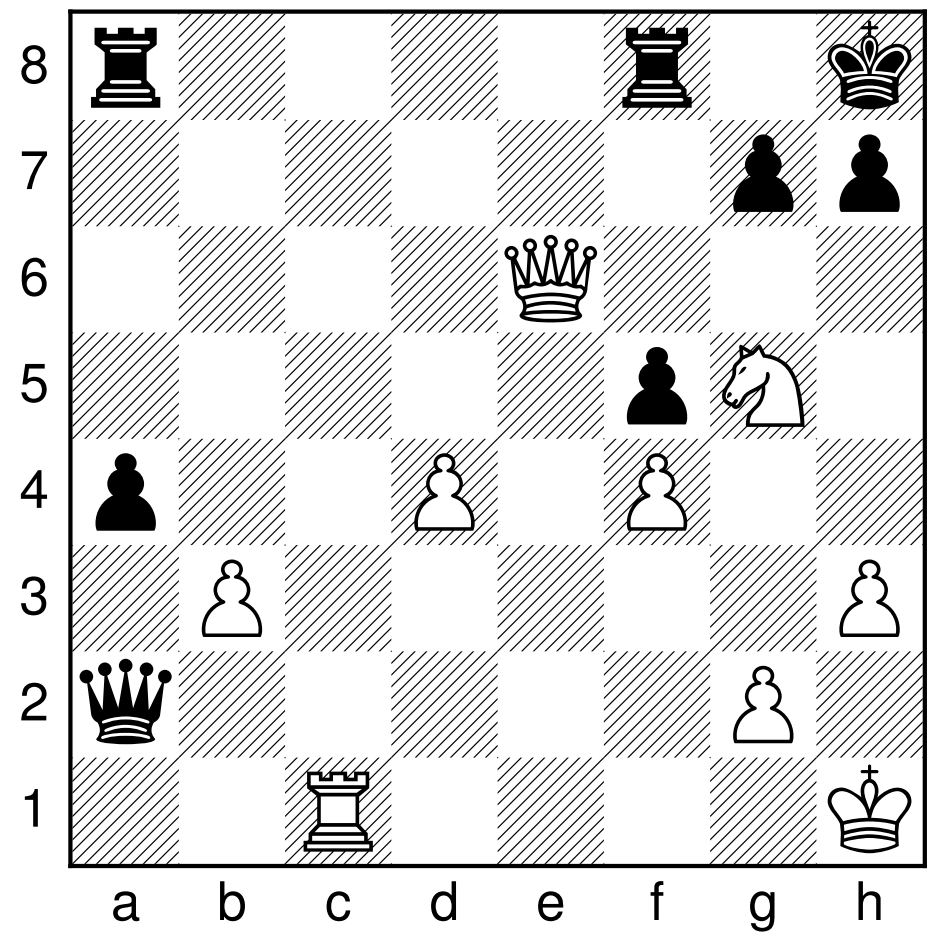

[Analyse on Lichess]

**Closest FENs - [1], [2], [3]**

1. Rc8! **White sacrifices the rook two ways!** Qxb3 (**1... Raxc8 Nf7+ and similarly 1... Rfxc8 Nf7+**) 2. Nf7+ d5! **Blocking the queen trade.** 3. Qd1+ Kh2 4. Qh5 Nd8+! 5. Kh8 Rxa8 **White is up a rook and quickly ending the game.**

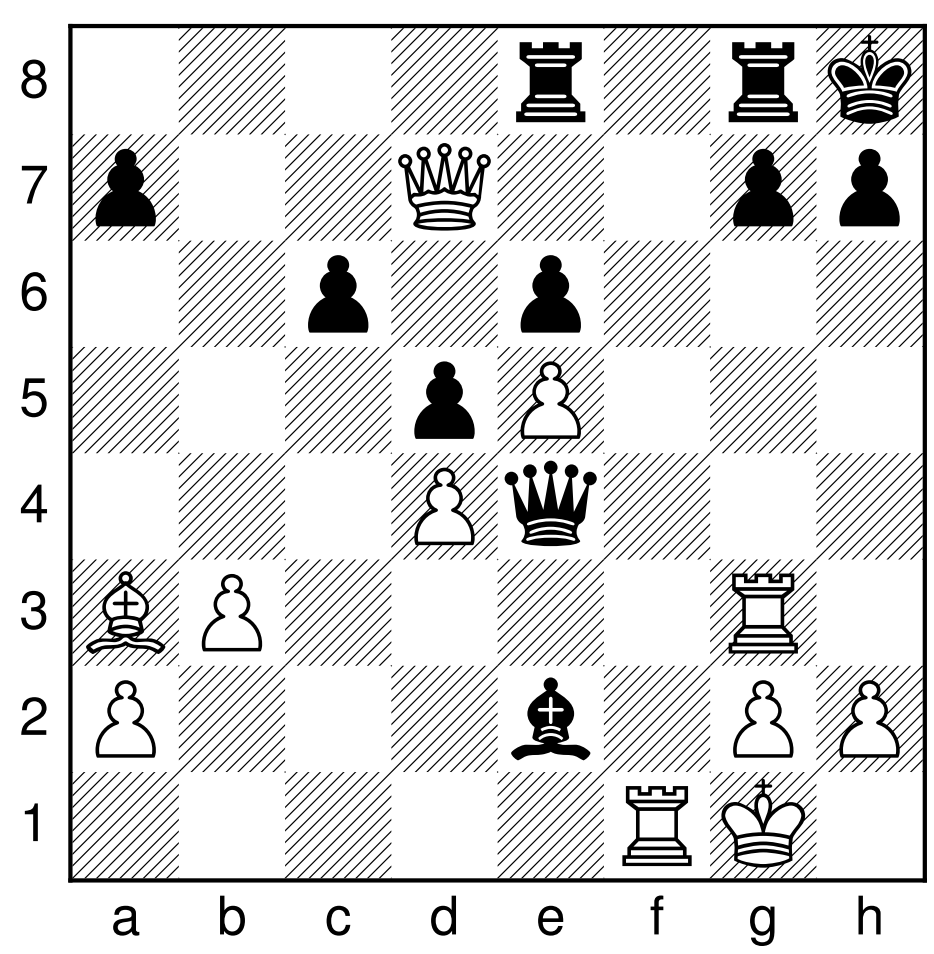

[Analyse on Lichess]

**Closest FENs - [1], [2], [3]**

1. Bf8! **White sacrifices the bishop two ways!** g6 (**1...Rexf8 2. Qxg7+! Rxg7 3. Rxf8+ Rg8 Rfxg8#**) 2. Qxe8 Bxf1 3. Qf7! Qf5 4. Rf3! Qxf7 5. Rxf7 **Black ends the combination with a dominating position.**

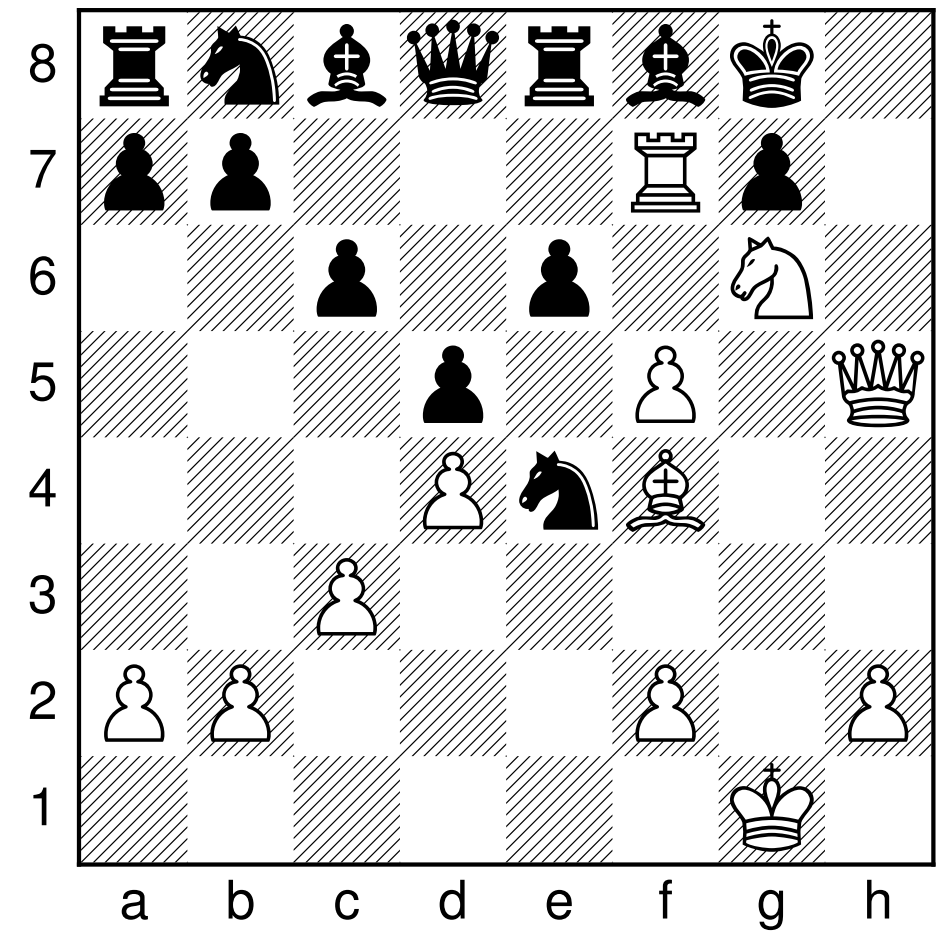

[Analyse on Lichess]

**Closest FENs - [1], [2], [3]**

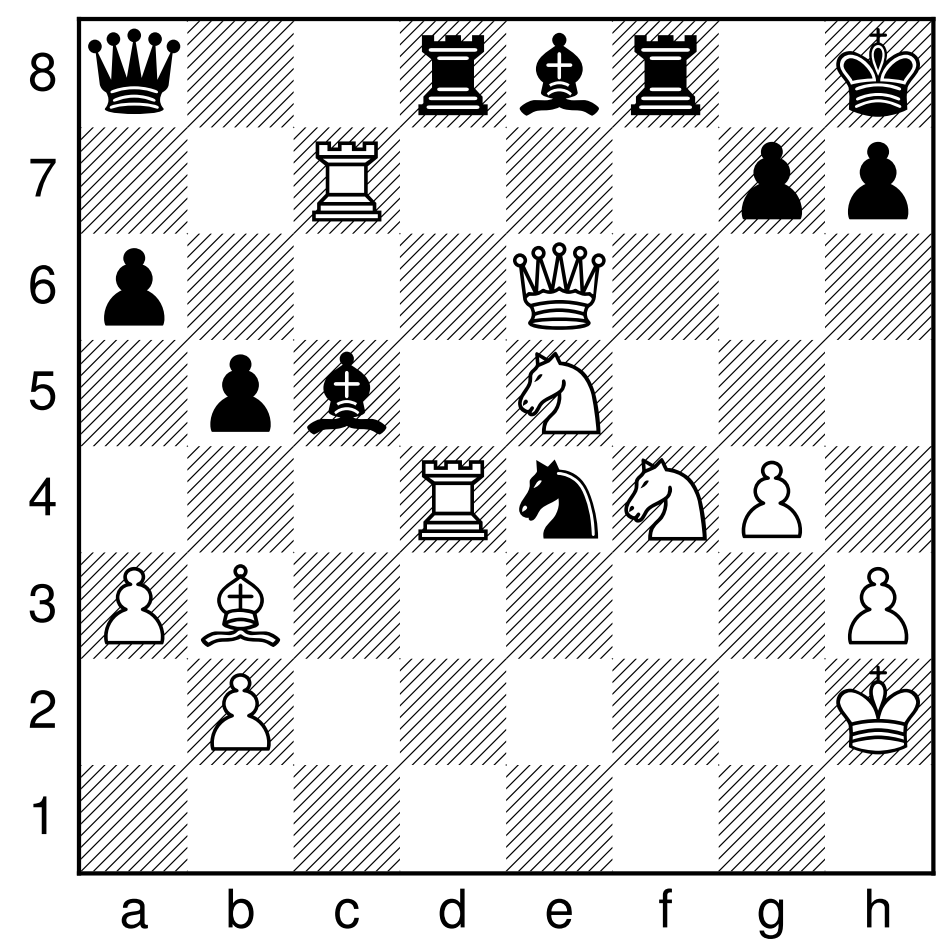

[Analyse on Lichess]

**Closest FENs - [1], [2], [3]**

## M.7 Interference [48]

**Book example:**

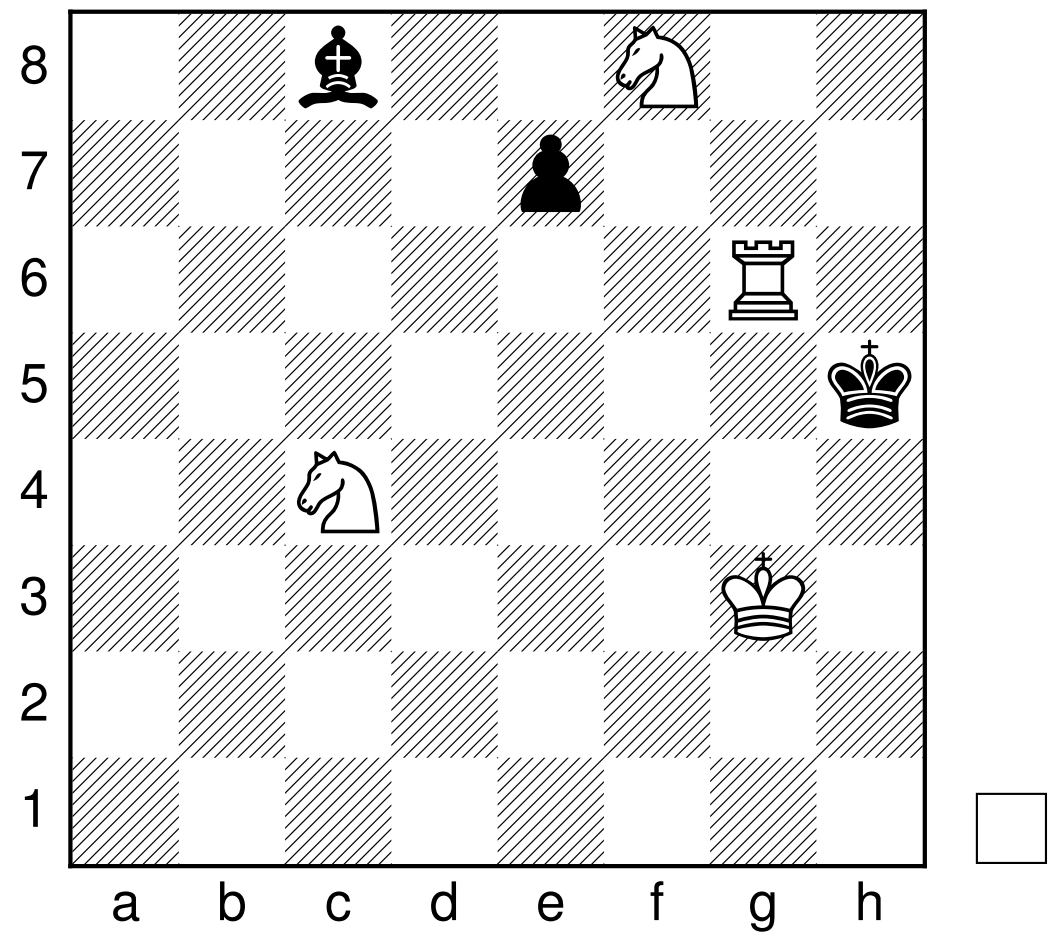

[Analyse on Lichess]

**Selected puzzles:**

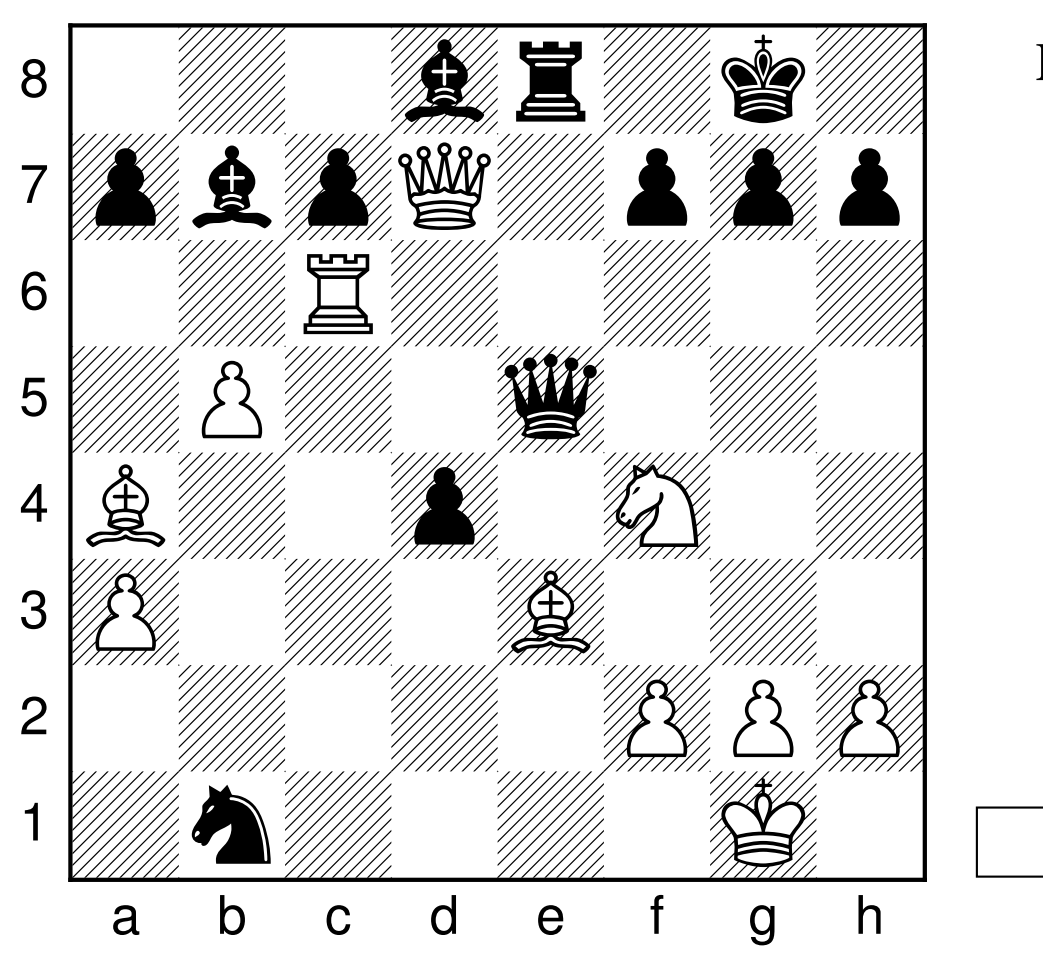

[Analyse on Lichess]

**Closest FENs - [1], [2], [3]**

1. Re6! **White interferes with the coordination of Black's queen and rook.** Rxe6 2. Nxe6 **Black cannot recapture due to back-rank issues.** h6 3. Qxd8+ Kh7 4. Nf8+ Kg8 5. Ng6+ Kh7 6. Nxe5 **White ends the combination with overwhelming material.**

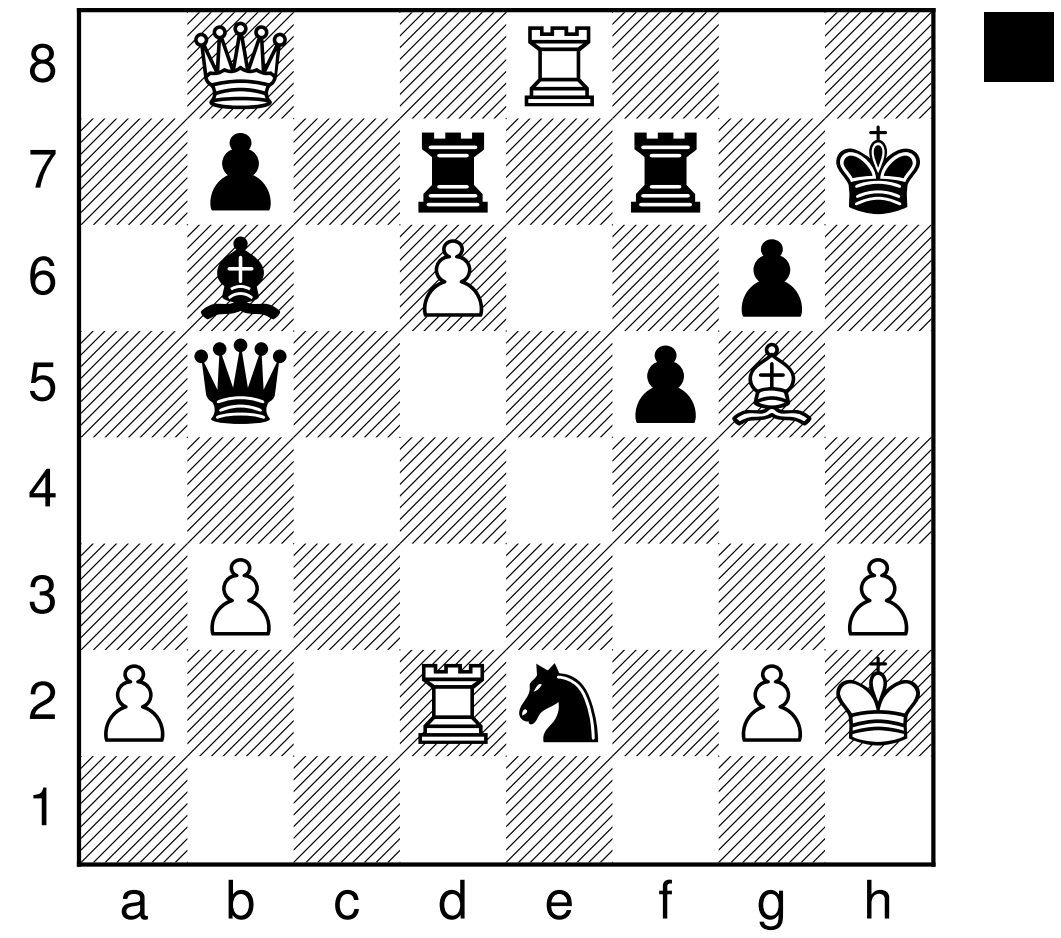

[Analyse on Lichess]

**Closest FENs - [1], [2], [3]**

1... Rd8! **Black sacrifices the rook to interfere with White's pieces.** 2. Bxd8 **If instead 2. Rxd8 Qe5+ is mating quickly and if 2. Qxd8 simply Bxd8** Qxe8 3. Rxe2 Qxd8 **Black remains up a piece and should win comfortably with correct play.**

## M.8 Unprotected Position [2]

**Book example:**

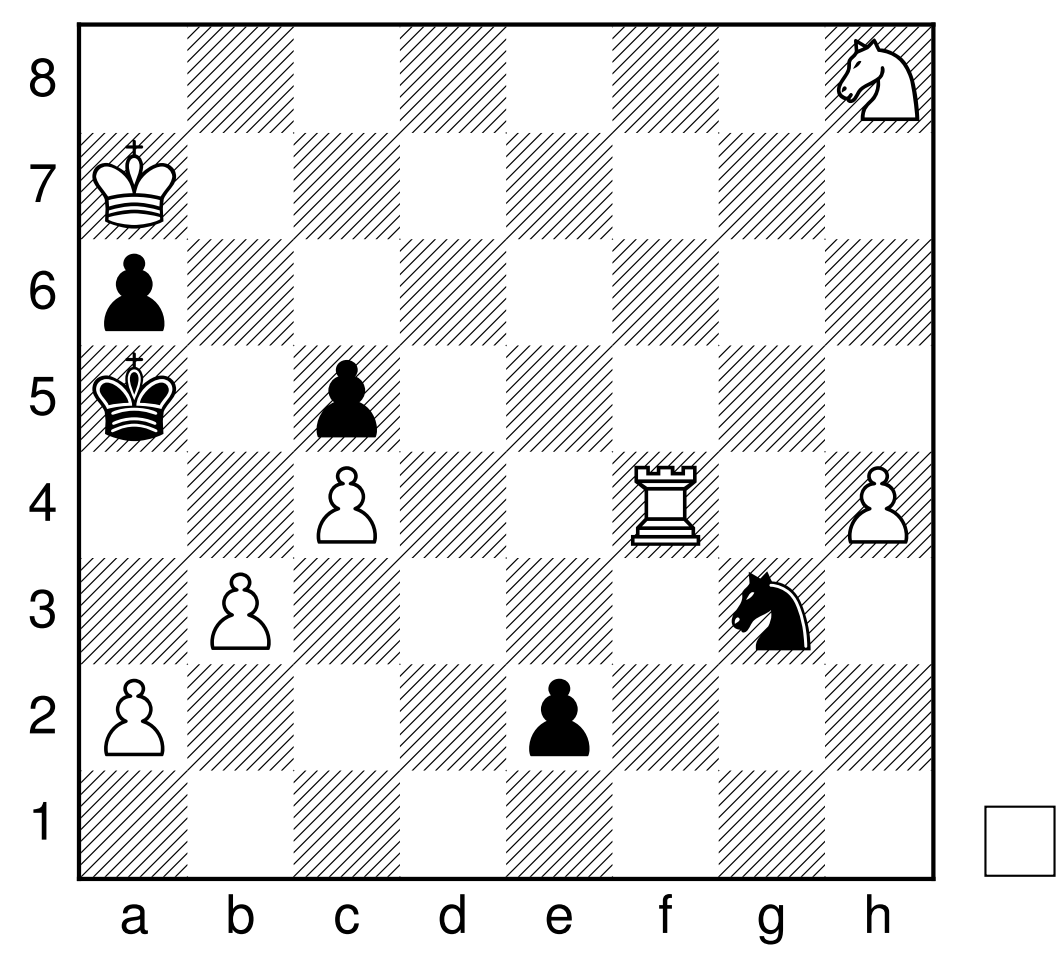

[Analyse on Lichess]

**Selected puzzles:**

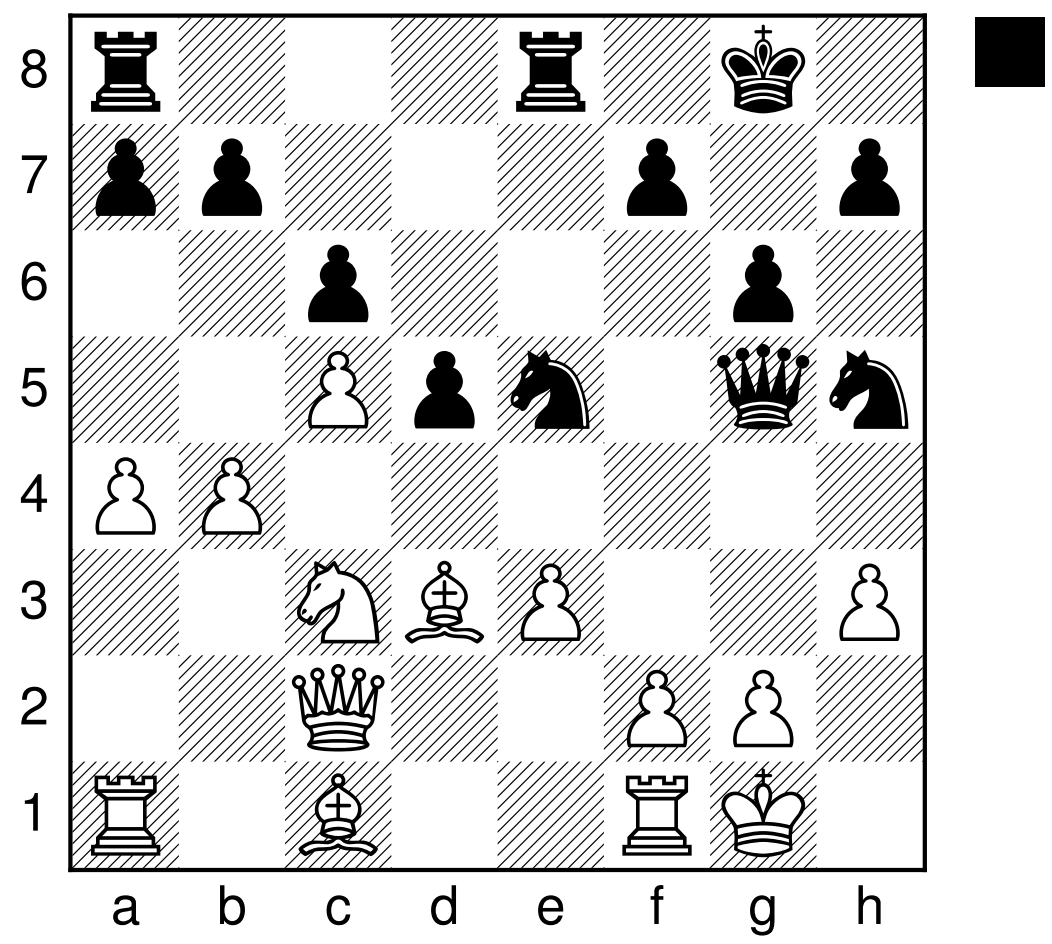

**[Analyse on Lichess]**

**Closest FENs - [1], [2], [3]**

1... Nf3+ 2. Kh1 Qg3! 3. gxf3 Qxh3+ 4. Kg1 Re5 **Ending the game with a rook lift.** 5. f4 Nxf4! 6. exf4 Qg4+ 7. Kh1 Rh5#.

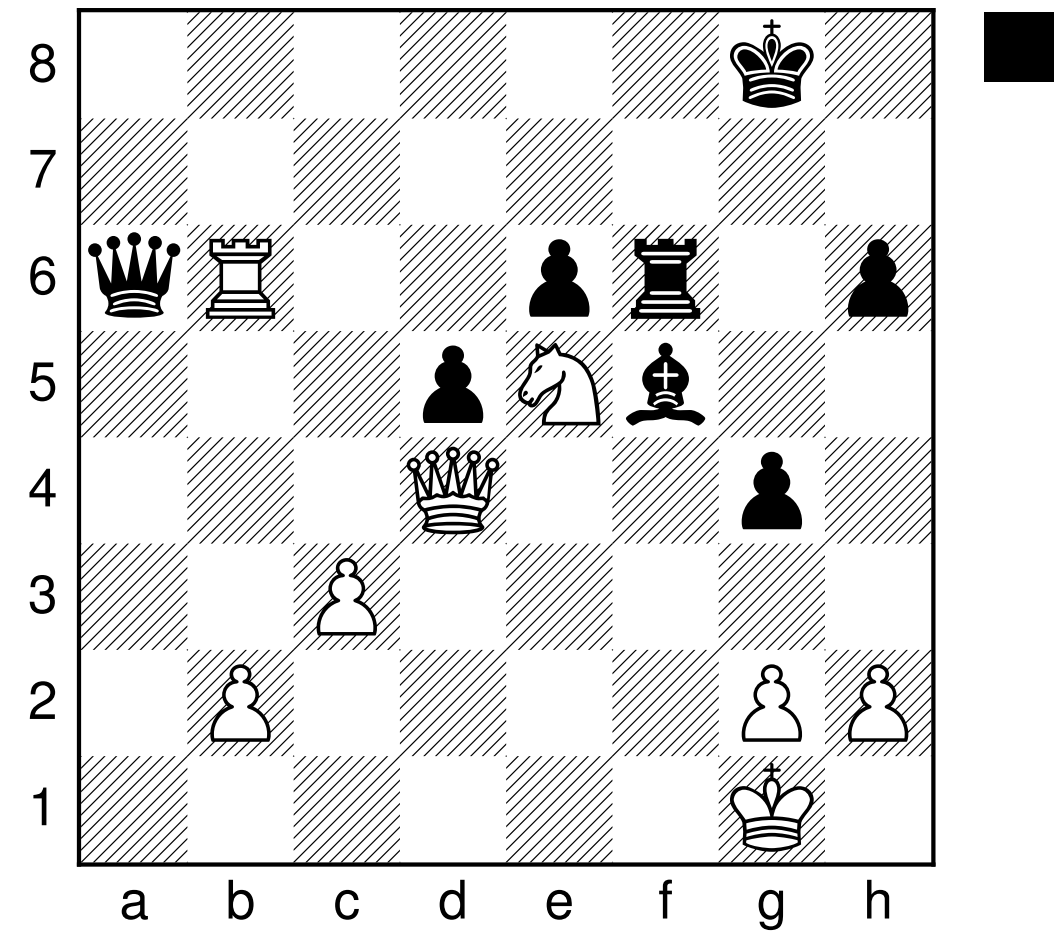

**[Analyse on Lichess]**

**Closest FENs - [1], [2], [3]**

1... Bd3! **Black leaves the queen hanging, but the queen cannot be captured due to the mate threat on f1.** 2. Rb8+ **If 2. h3 g3! 3. Rb8+ Kg7 4. Qg4+ Bg6 5. Qxg3 Qa7+ and Black picks up the rook.** Kg7 3. Qxg4+ Bg6 4. h3 Ba7+ **Black picks up the rook.**

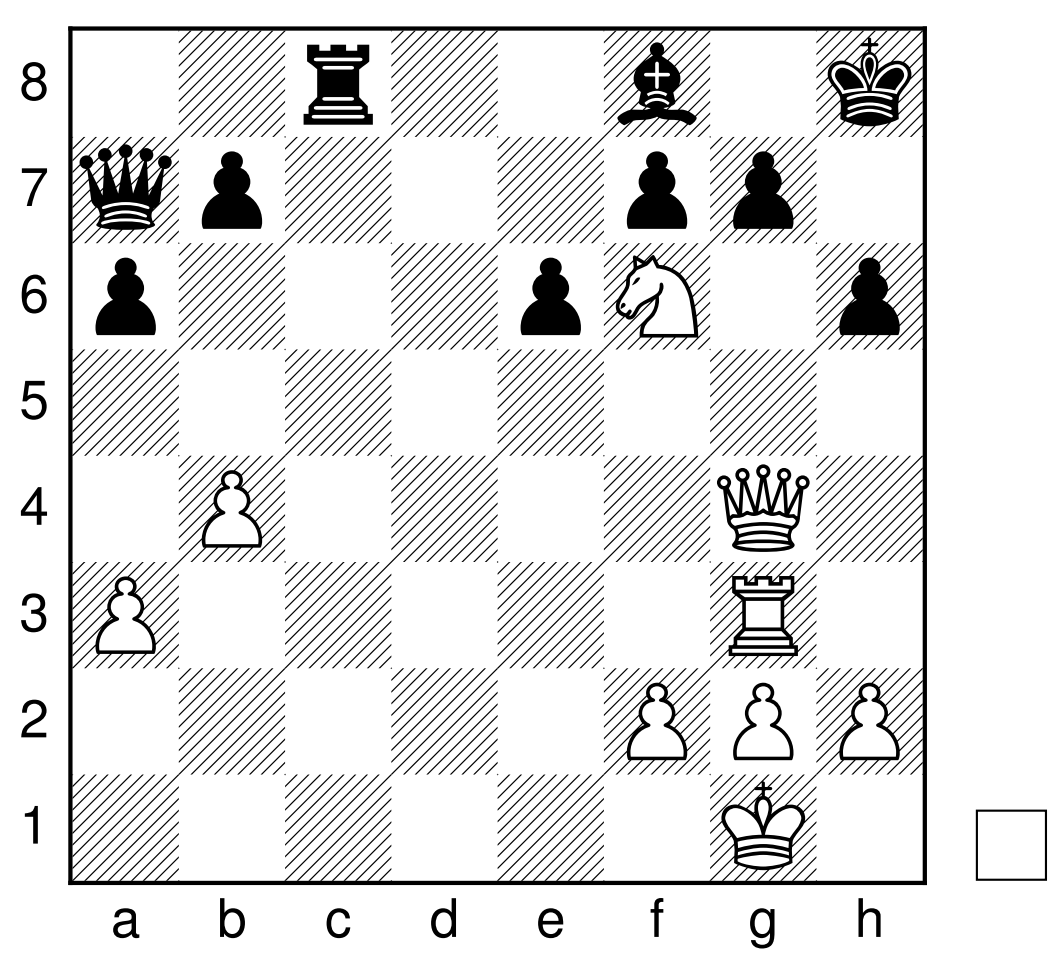

**[Analyse on Lichess]**

**Closest FENs - [1], [2], [3]**

1. Qg5! **Defending c1 and threatening 2. Qxg6+ gxh6 3. Rg8#. Black tries to hold the position.** g6 2. Rh3 Kg7 3. Rxh6 Rc1+ 4. Qxc1 **Black is forced to give up material to defend checkmate.**

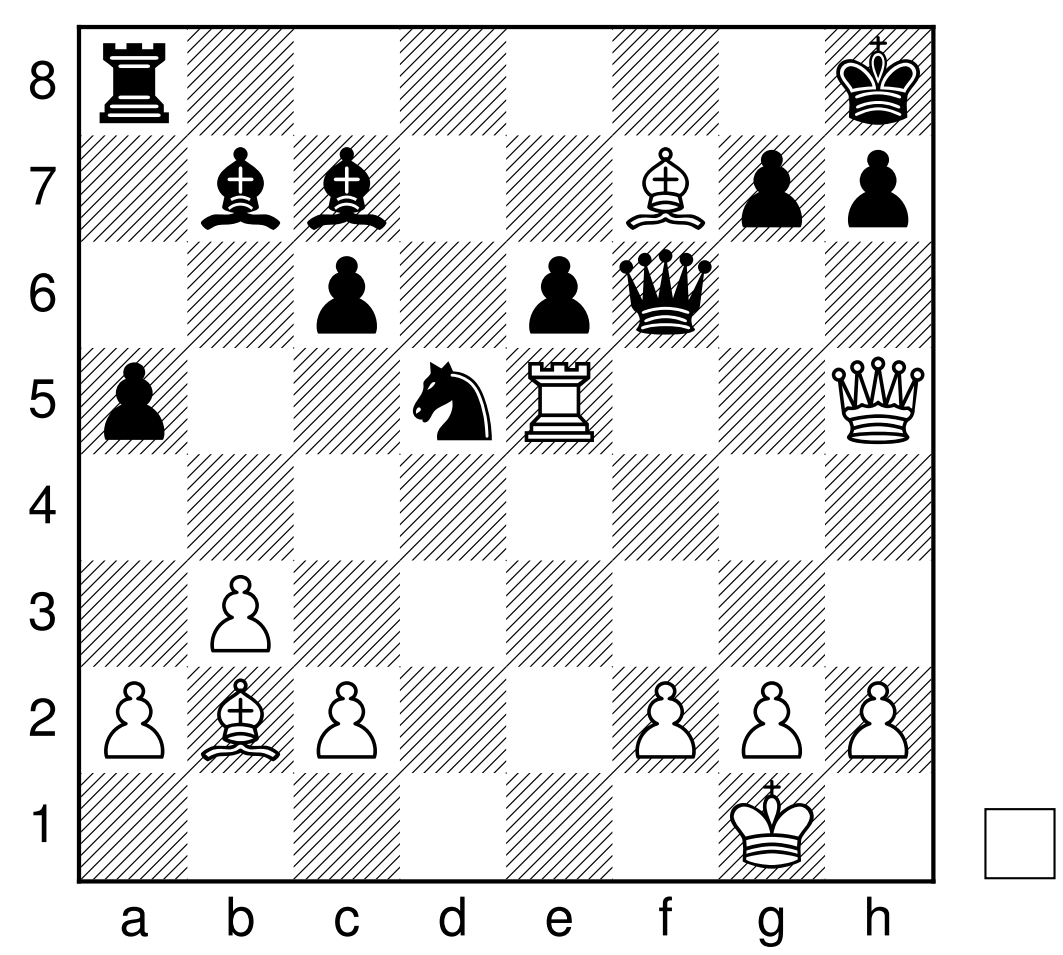

**[Analyse on Lichess]**

**Closest FENs - [1], [2], [3]**

1. Rxd5! e5 **If Qxb2 2. Qxh7+! Kxh7 3. Rh5#** 2. Bxe5 Bxe5 3. Rxe5 g6 4. Qg5! **The critical move that makes this variation work.** Qxf7 5. Re7 Qf8 6. Qe5+ Kg8 7. Rxb7 Re8 **White seems in trouble due to the back-rank issues, but there is a beautiful finishing move here.** 8. Rg7+! Qxg7 9. Qe8+ Qf8 10. Qxf8 **After the dust has settled, white remains up 2 pawns and easily wins the pawn endgame.**

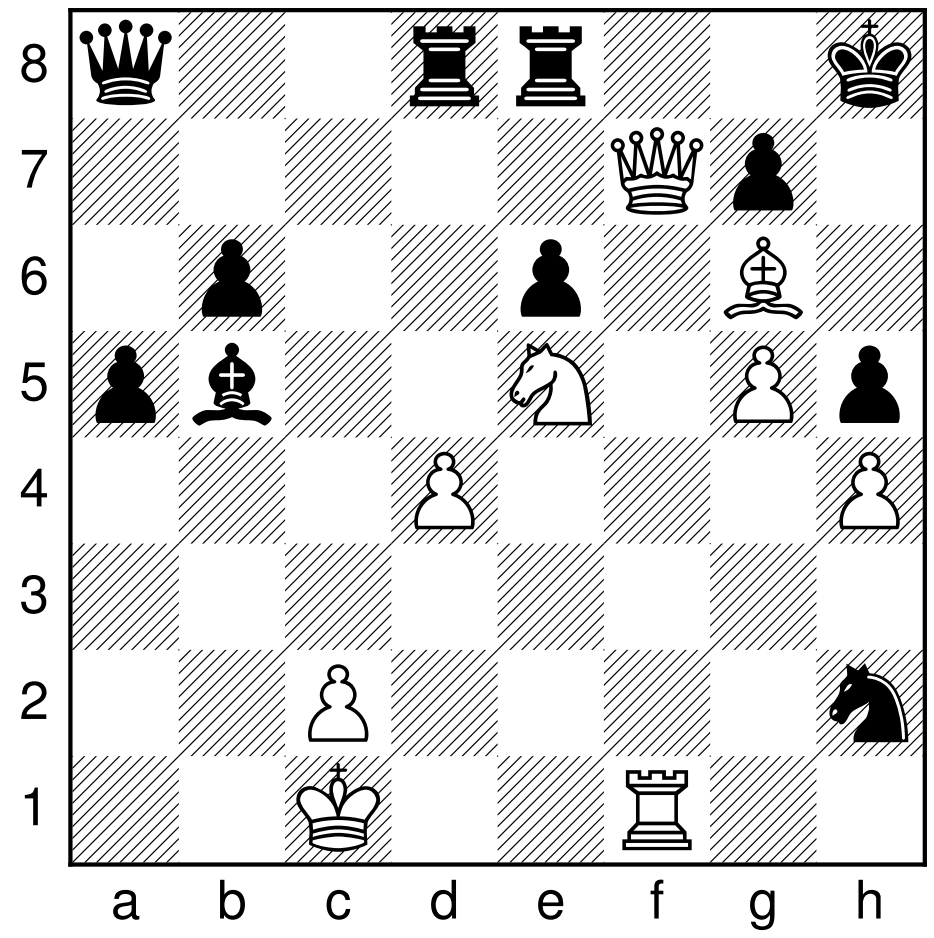

**[Analyse on Lichess]**

**Closest FENs - [1], [2], [3]**

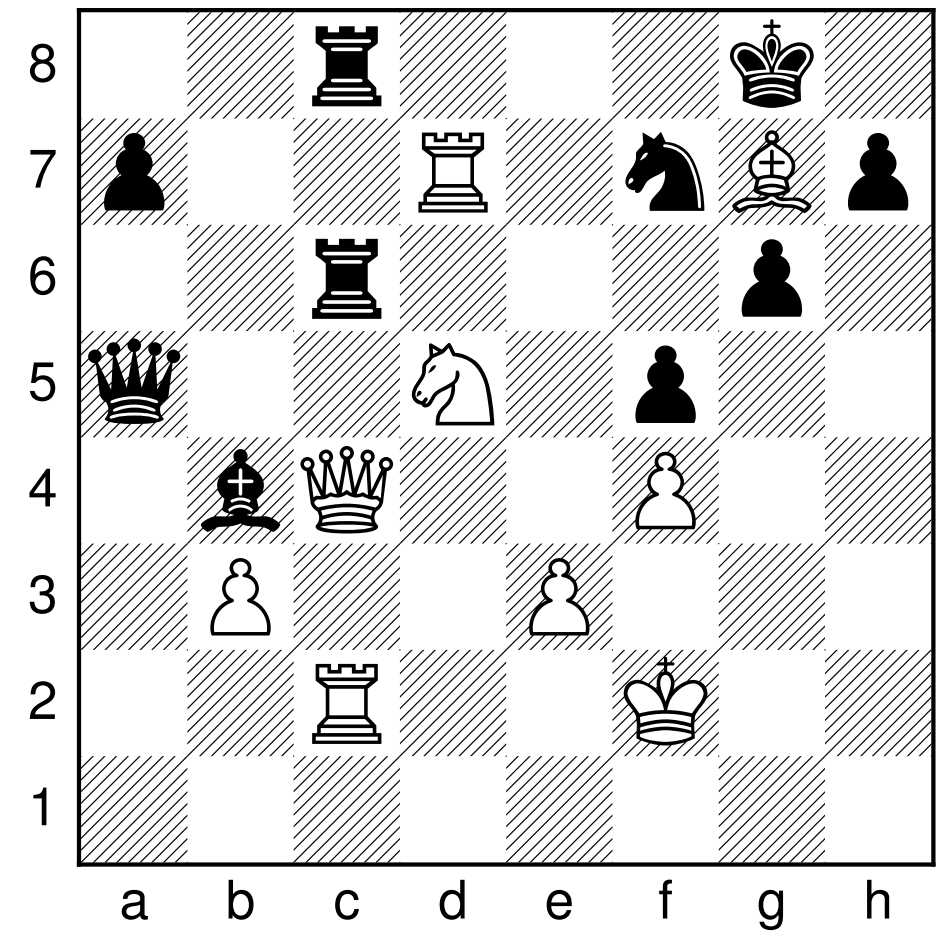

**[Analyse on Lichess]**

**Closest FENs - [1], [2], [3]**

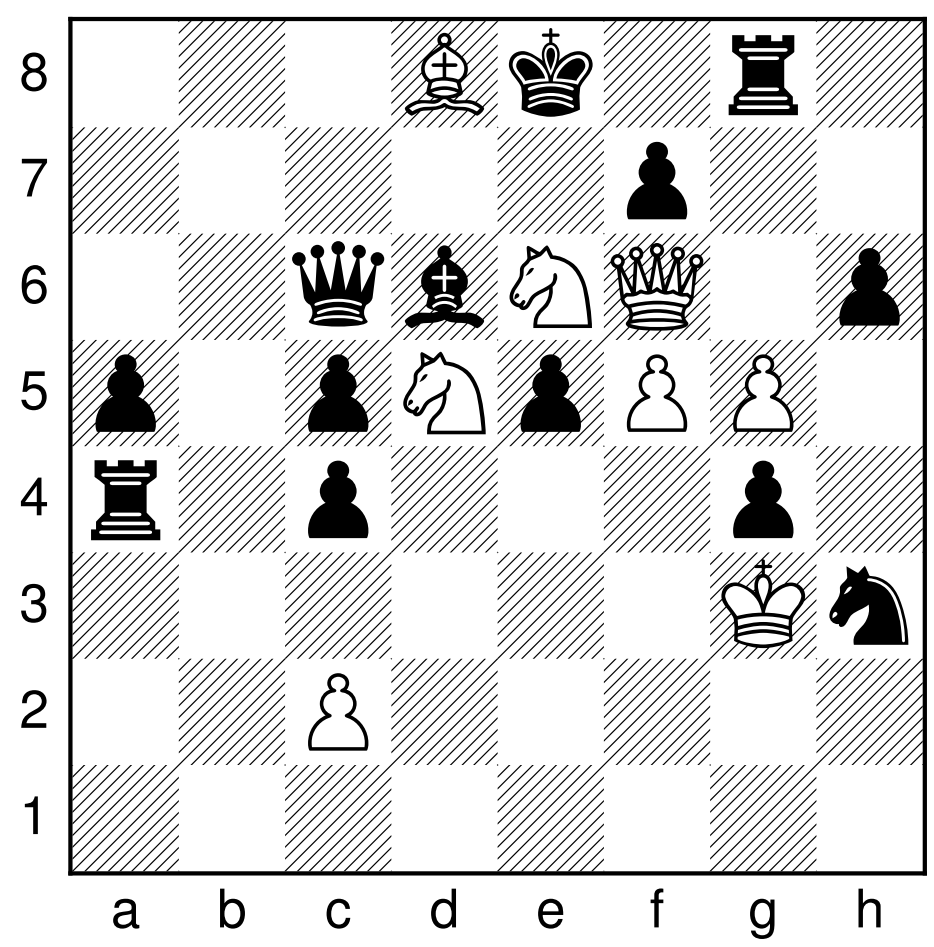

**[Analyse on Lichess]**

**Closest FENs - [1], [2], [3]**

### M.9 XRay Attack

**Selected puzzles:**

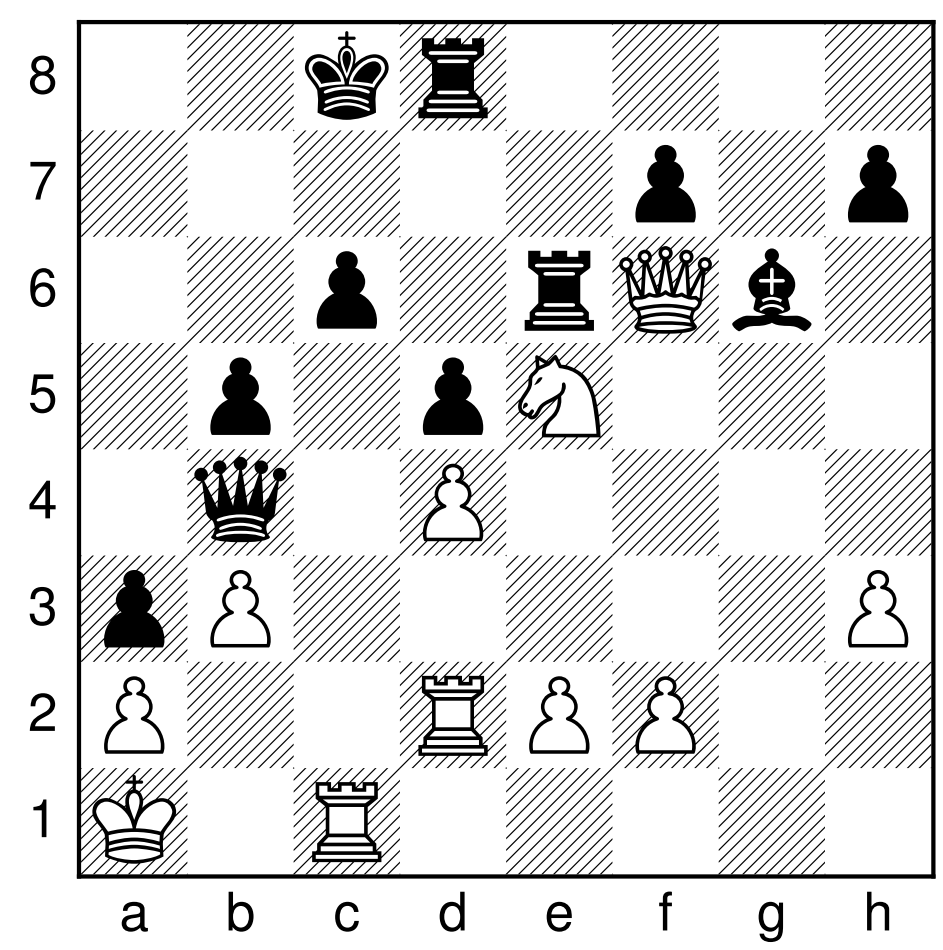

**[Analyse on Lichess]**

**Closest FENs - [1], [2], [3]**

1. Rc6+ **The queen defends the rook via an x-ray.** Rxc6 **The immediate Kb7 transposes to the main line.** 2. Qxd8+! Kb7 3. Qb8+ **White insists on the sacrifice. Black is now forced to accept.** Kxb8 4. Nxc6+ **Winning the queen in the next move. White finishes the variation up a rook.**

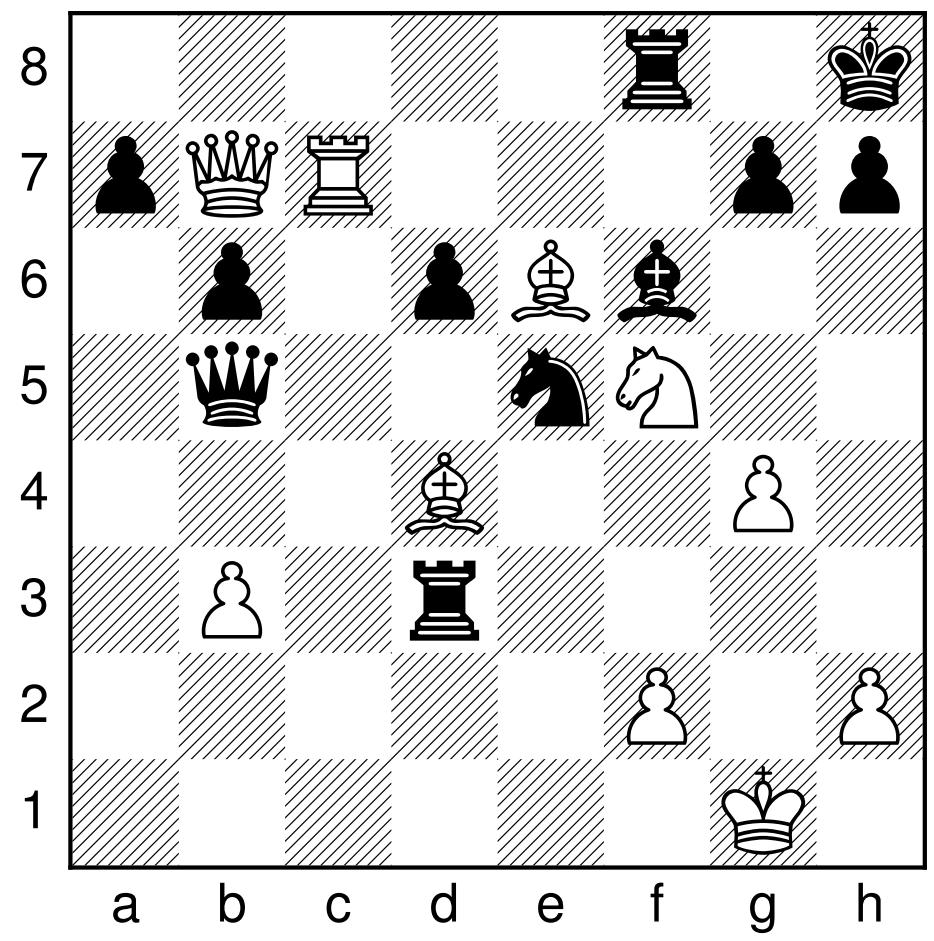

[Analyse on Lichess]

**Closest FENs - [1], [2], [3]**

1. Bc4 Nf3+ **If Rd1+ instead, then 2. Kg2 Qb4 3. Rxg7! and White wins quickly.** 2. Kg2 Nh4 3. Nxh4 Qg5 **Black attempts a clever maneuver, rerouting the queen towards a better attacking square hoping to complicate matters.** 4. Rxg7! Qxg7 5. Qxg7 Bxg7 8. Bxg7 **The x-ray attack of White's bishop results in massive liquidation.** Kxg7 9. Bxd3 **White ends the combination with a bishop and knight for a rook, in a technically winning endgame although it will require careful play.**

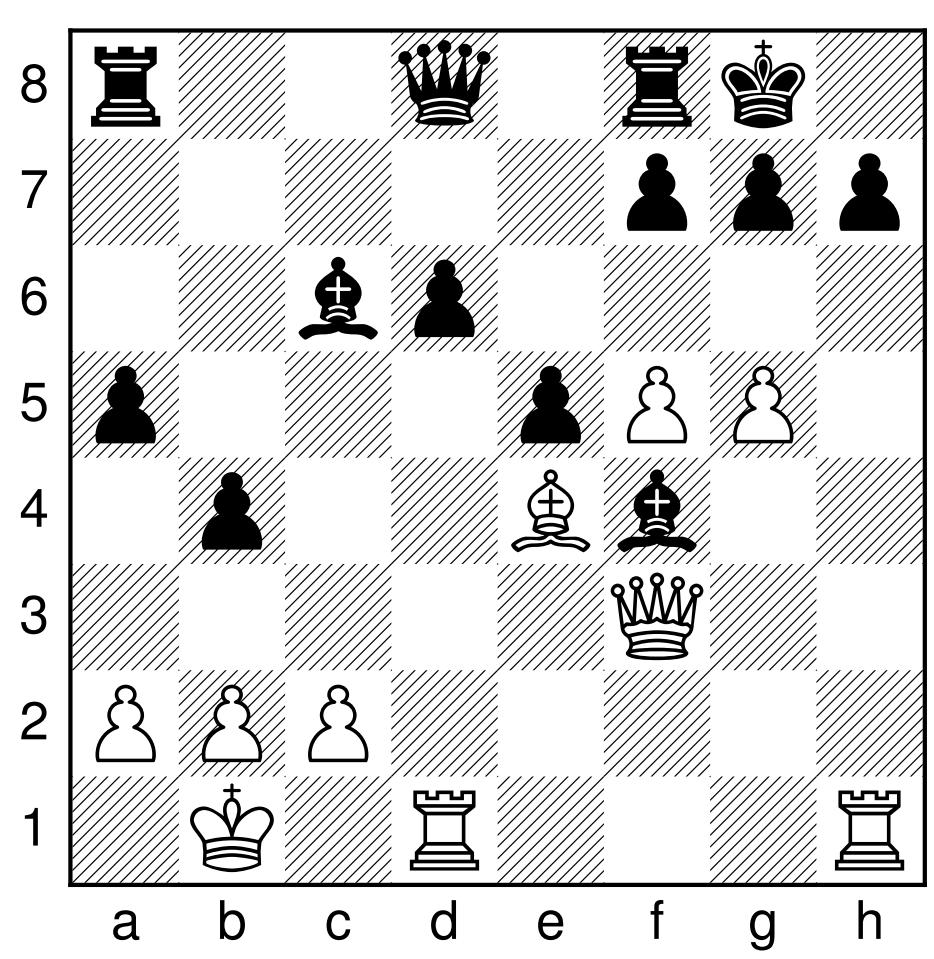

[Analyse on Lichess]

**Closest FENs - [1], [2], [3]**

### M.10 Paralysis [48]

**Book example:**

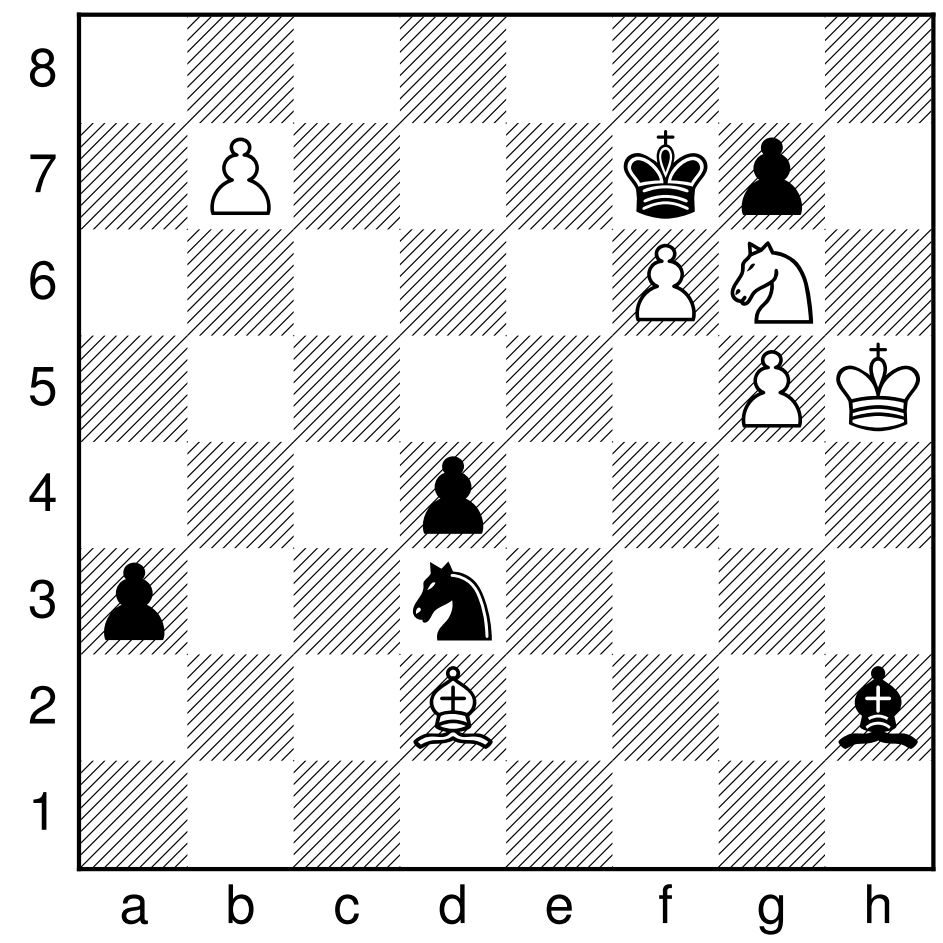

[Analyse on Lichess]

**Selected puzzles:**

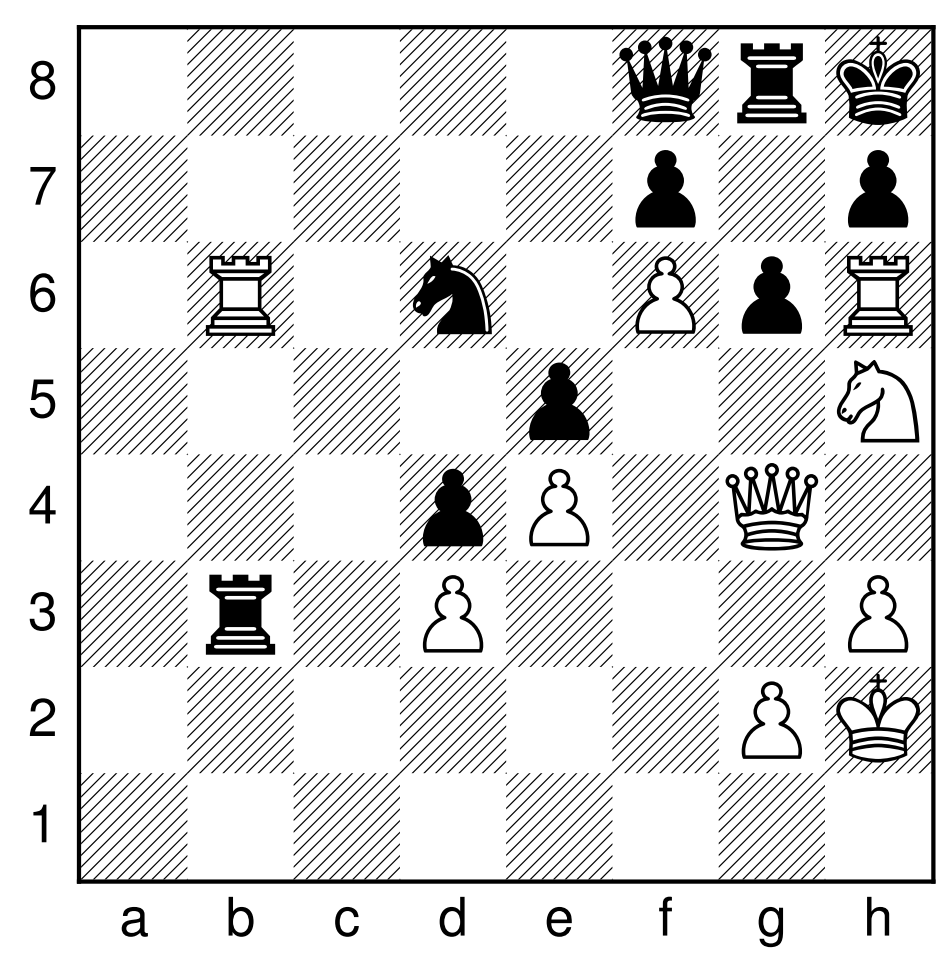

[Analyse on Lichess]

**Closest FENs - [1], [2], [3]**

1. Ng7 **White sets up 2. Rxh7+ 3. Qh4#. Black is forced to act quickly.** Rxg7 2. Rxb3! **White ignores the situation on the king side and instead aims to suffocate Black's pieces.** Ne8 3. Qh4 **Black's position is completely paralysed. We show a line that highlights the theme.** Qc5 4. Rb8! Qc6 5. Qg5! **Black is never able to save the rook due to mate threats with Rxh7+.** Kg8 6. Qxe5! Kh8 7. Rxe8+ Rg8 8. Rxg8+ Kxg8 9. Qb8+ Qe8 10. Qxe8#.

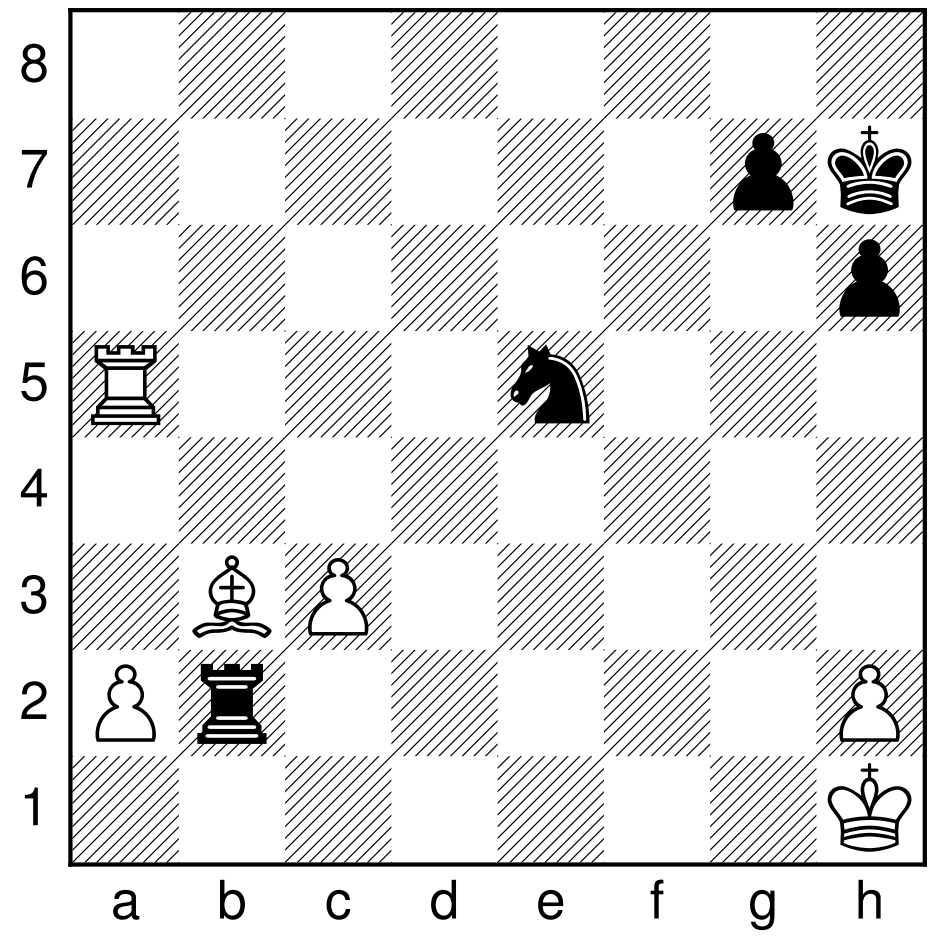

[Analyse on Lichess]

**Closest FENs - [1], [2], [3]**

1... Nf3 2. Rh5 g5! **Black aims to trap the White rook.** 3. Bd5 Nh4! **If 3. Rh3 g4! 4. Rh5 Kg6 and White loses the rook and is getting mated quickly.** 4. Be4+ Kg7 5. Kg1 Rxa2 **White's rook is paralysed and Black should be able to eventually win this position without much resistance.**

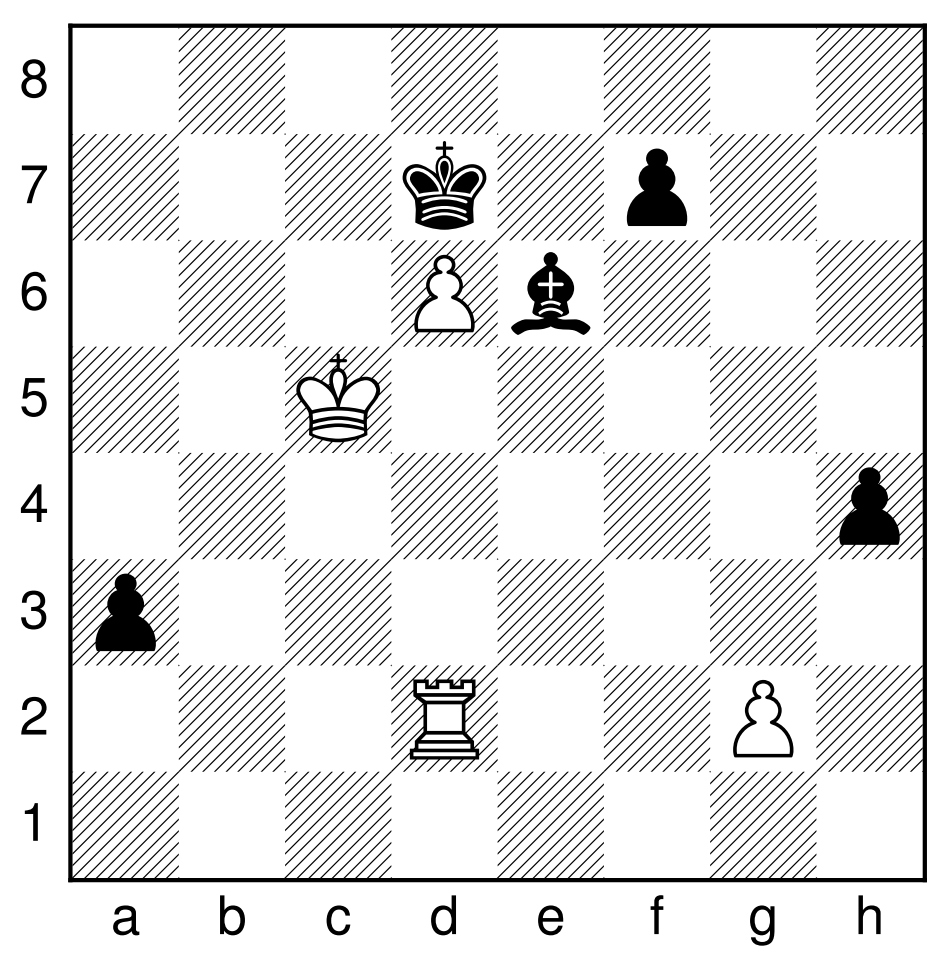

[Analyse on Lichess]

**Closest FENs - [1], [2], [3]**

1... a2 2. Rd1 Bf5 **Black threatens Bb1 which would secure the promotion.** 3. Rf1 Kd8! **Black cannot immediately play Bb1?? due to Rxf7+. The slow Kd8 instead wins!** 4. Ra1 Bb1 **White is now completely paralysed. Black has the simple plan of pushing the f and h pawns and wins this position without much effort.**

**M.11 Bristol [48]**

**Book example:**

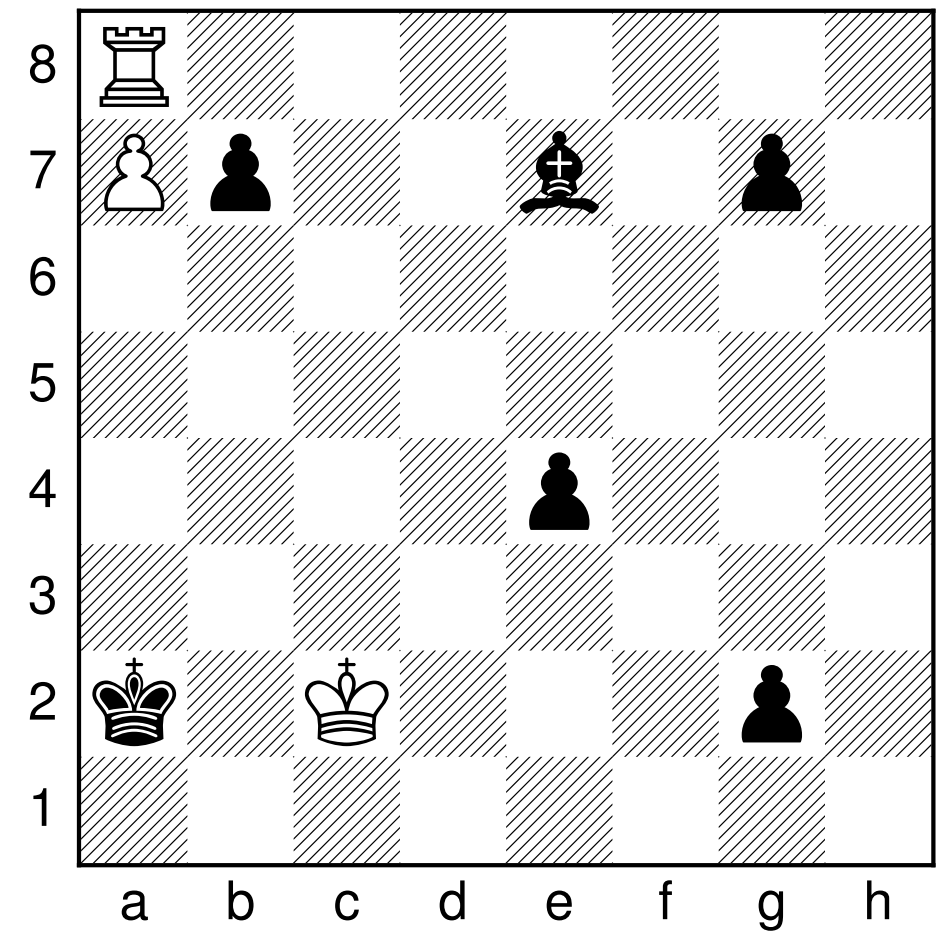

[Analyse on Lichess]

**Selected puzzles:**

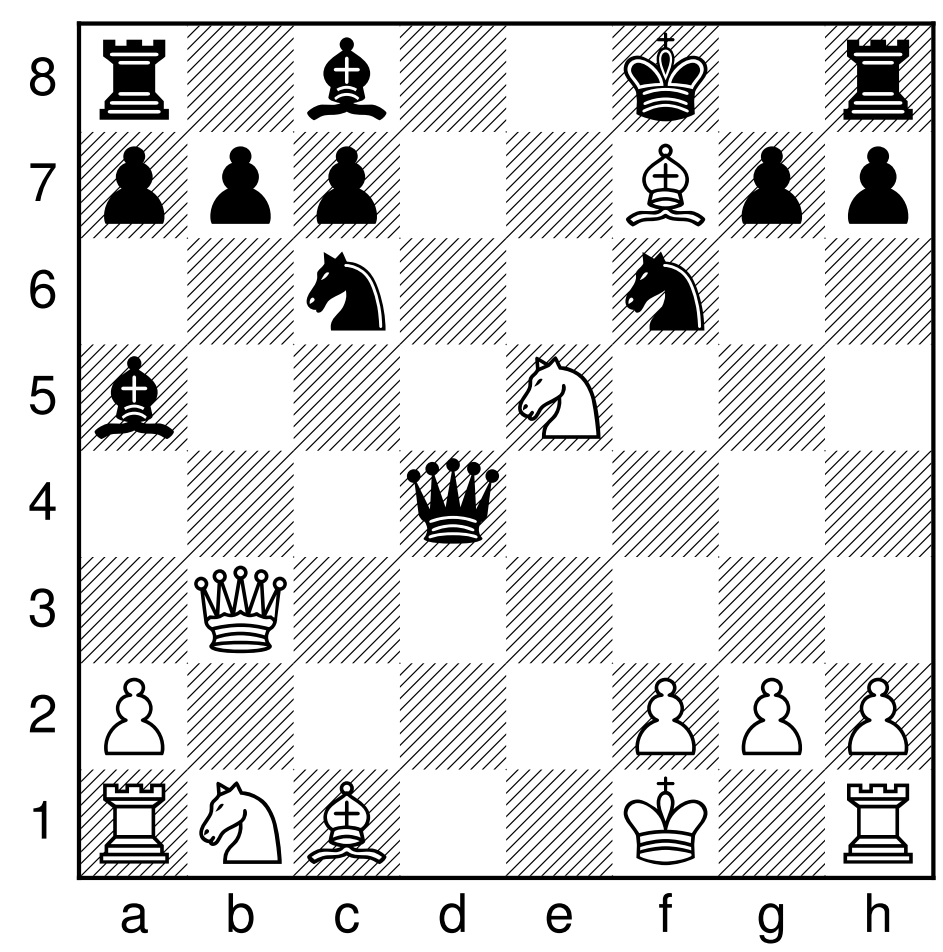

[Analyse on Lichess]

**Closest FENs - [1], [2], [3]**

1. Ba3+ Nb4 **1... Ne7 is again met by 2. Bg8 and 1... Bb4 is followed by simply 2. Nxc6.** 2. Bg8! **The Bristol move, making space for the White queen.** Be7 3. Bb2 **White does not hurry with Bf7+ and first plays an intermezzo to solidify their position.** Qc5 4. Qf7+ Kd8 5. Nc3 **White enjoys a decisive advantage with a safer king and very active pieces.**

### M.12 King on Tour [66]

**Book example:**

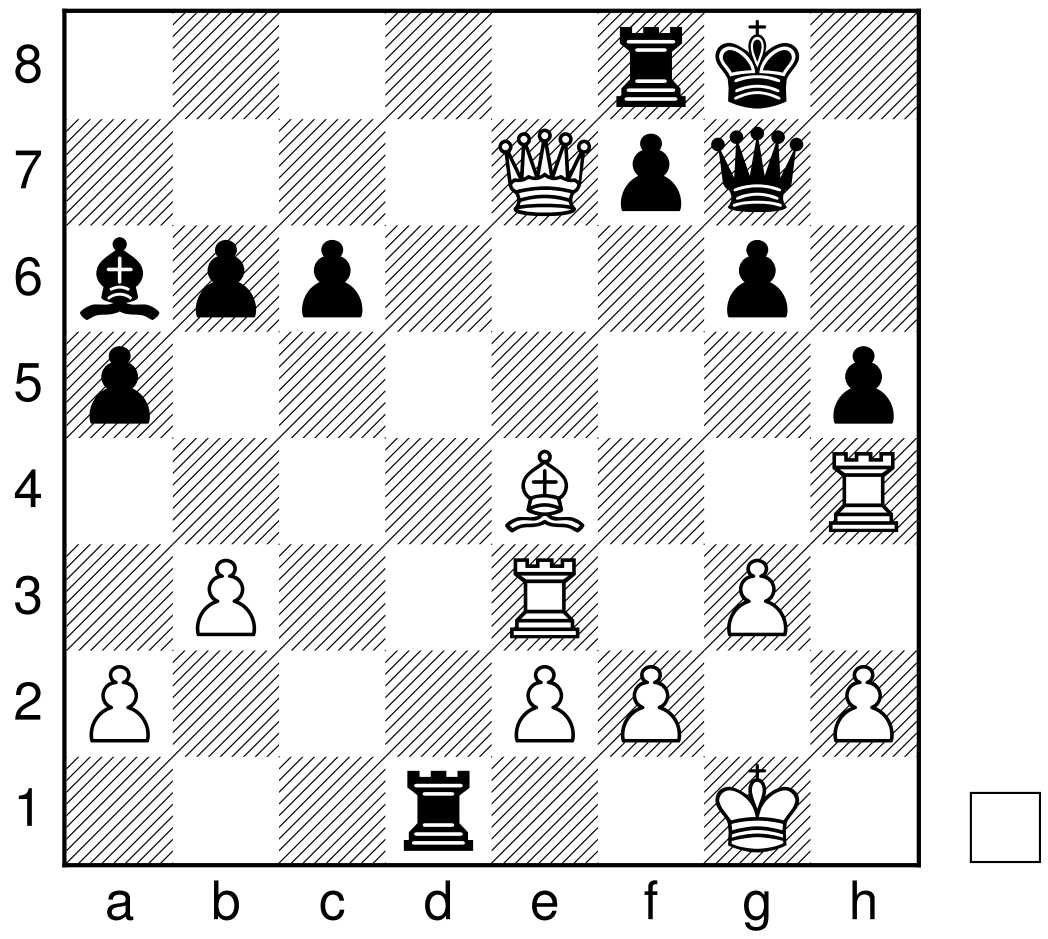

[Analyse on Lichess]

**Selected puzzles:**

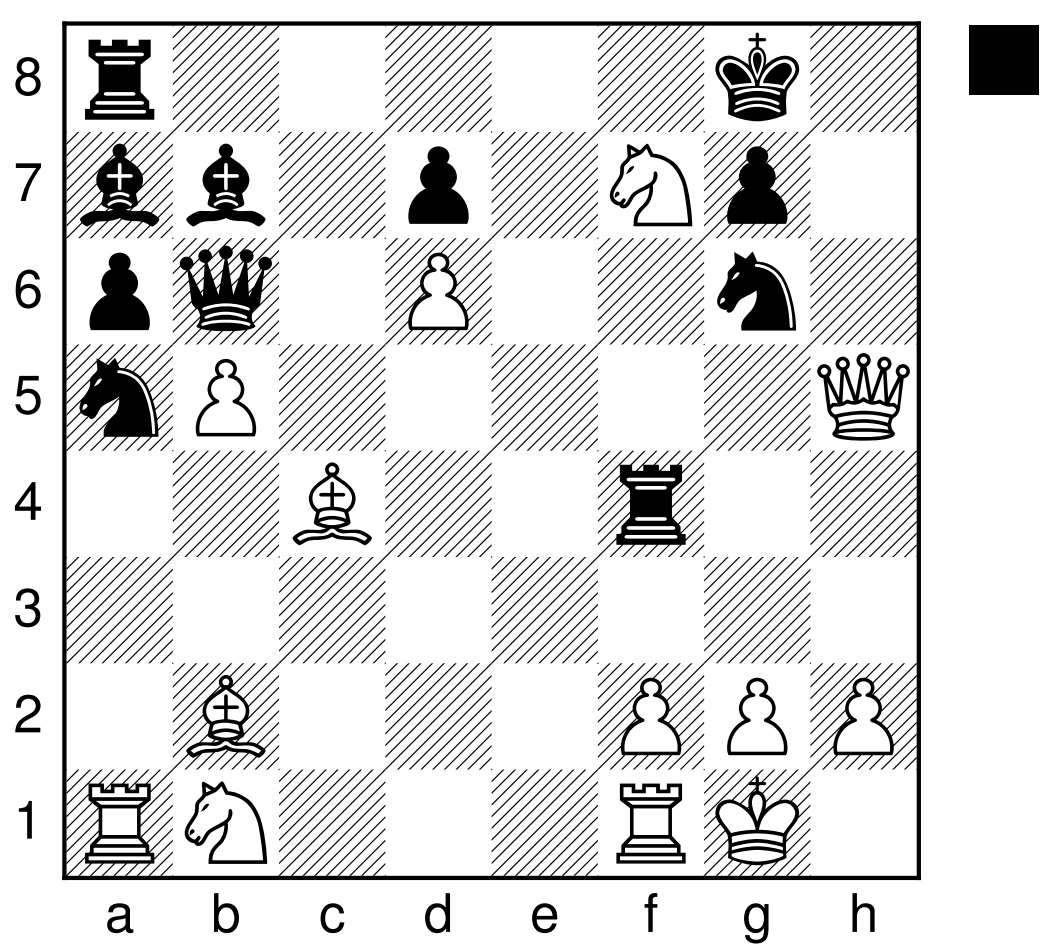

[Analyse on Lichess]

**Closest FENs - [1], [2], [3]**

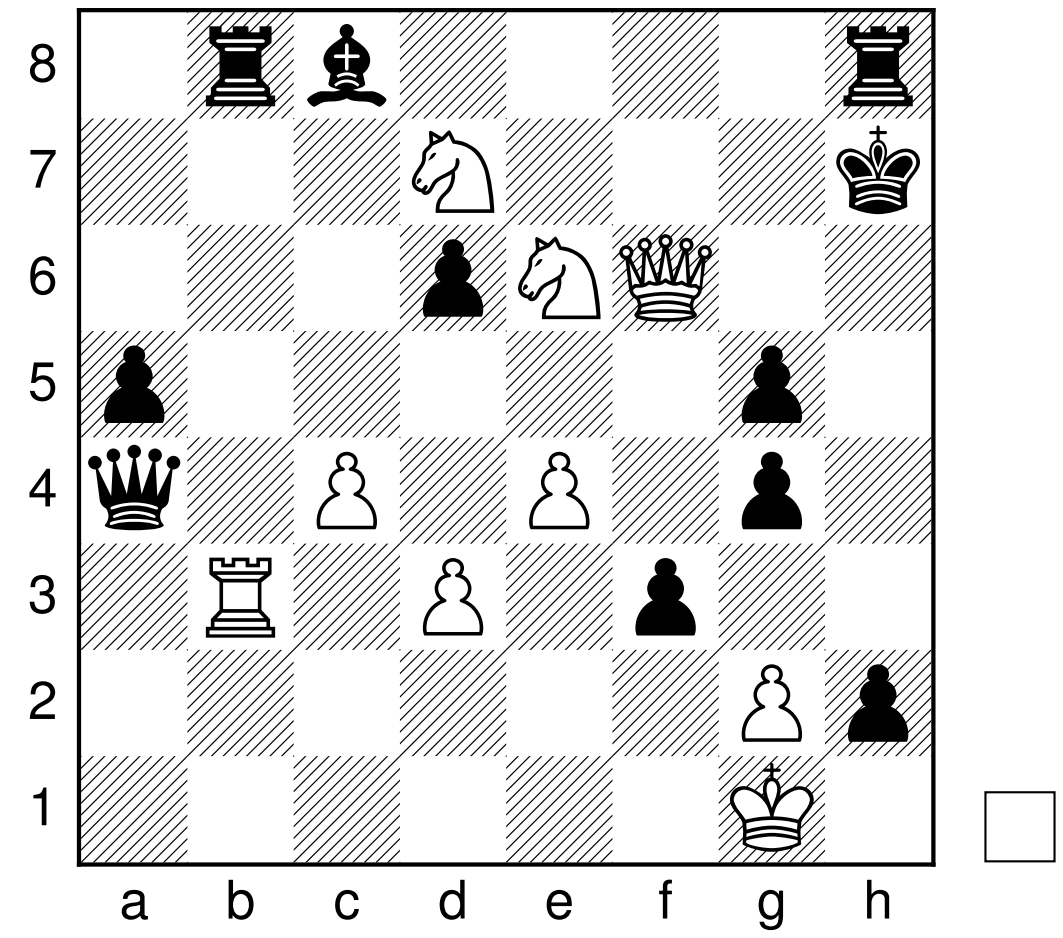

[Analyse on Lichess]

**Closest FENs - [1], [2], [3]**

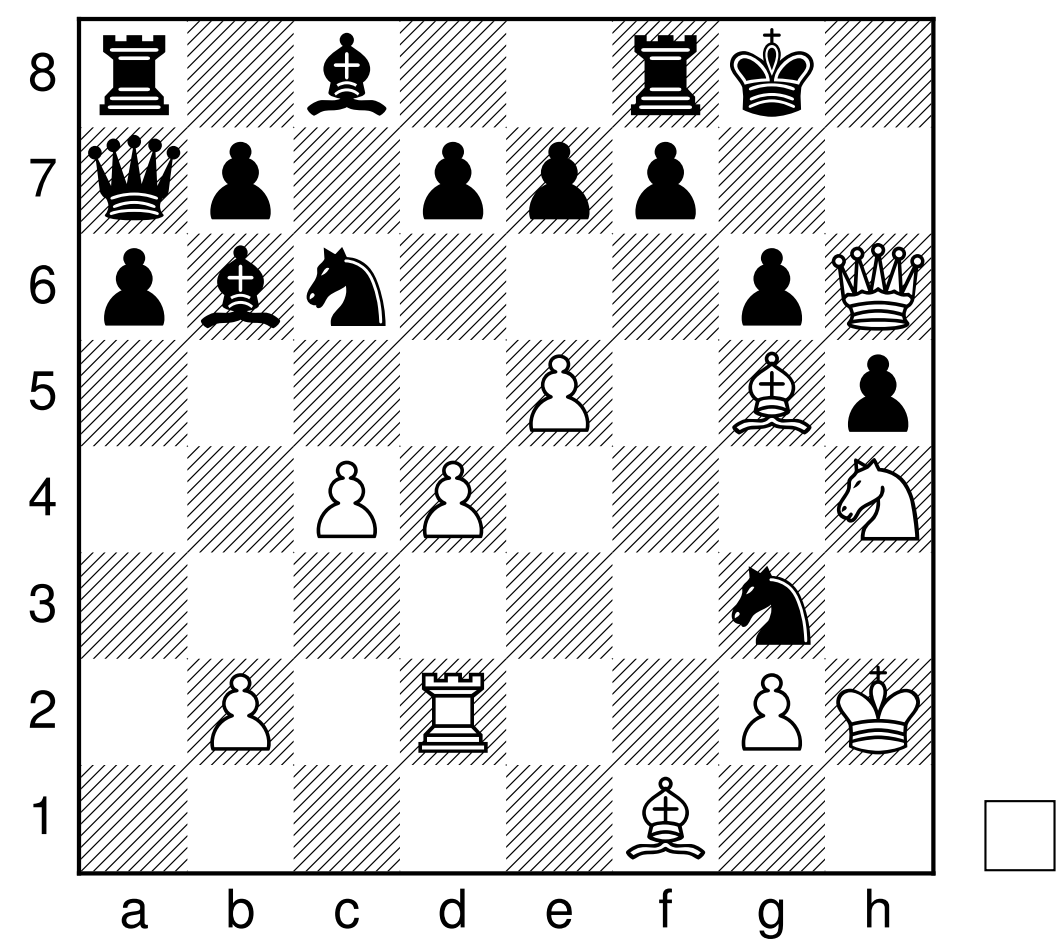

[Analyse on Lichess]

**Closest FENs - [1], [2], [3]**

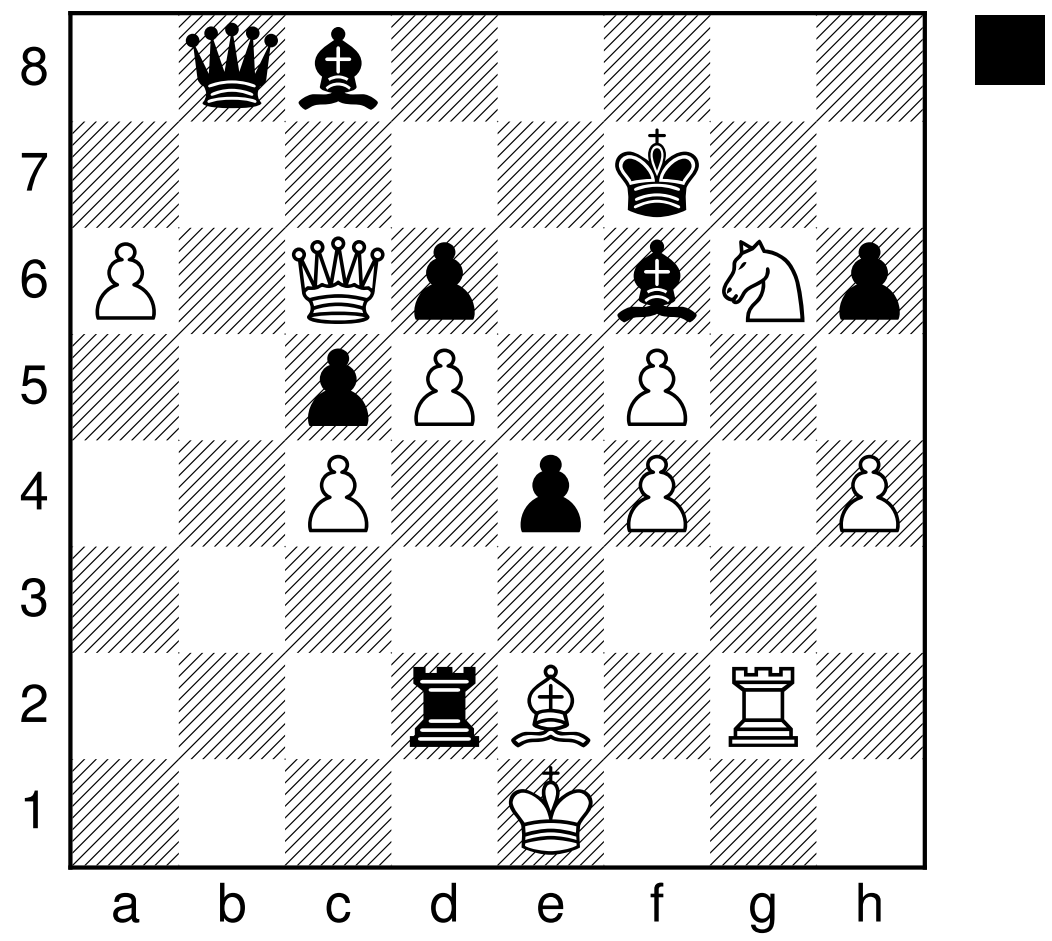

[Analyse on Lichess]

Closest FENs - [1], [2], [3]

### M.13 Switchback [48]

**Book example:**

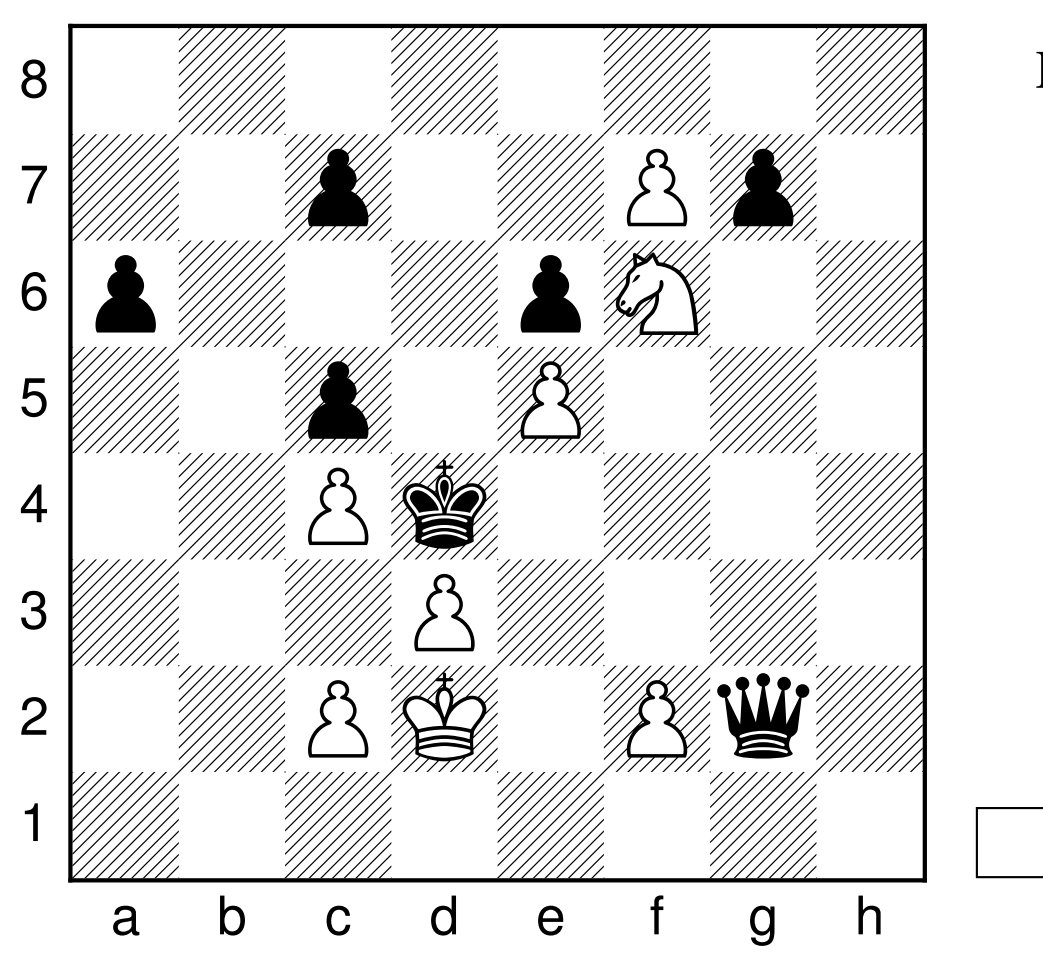

[Analyse on Lichess]

**Selected puzzles:**

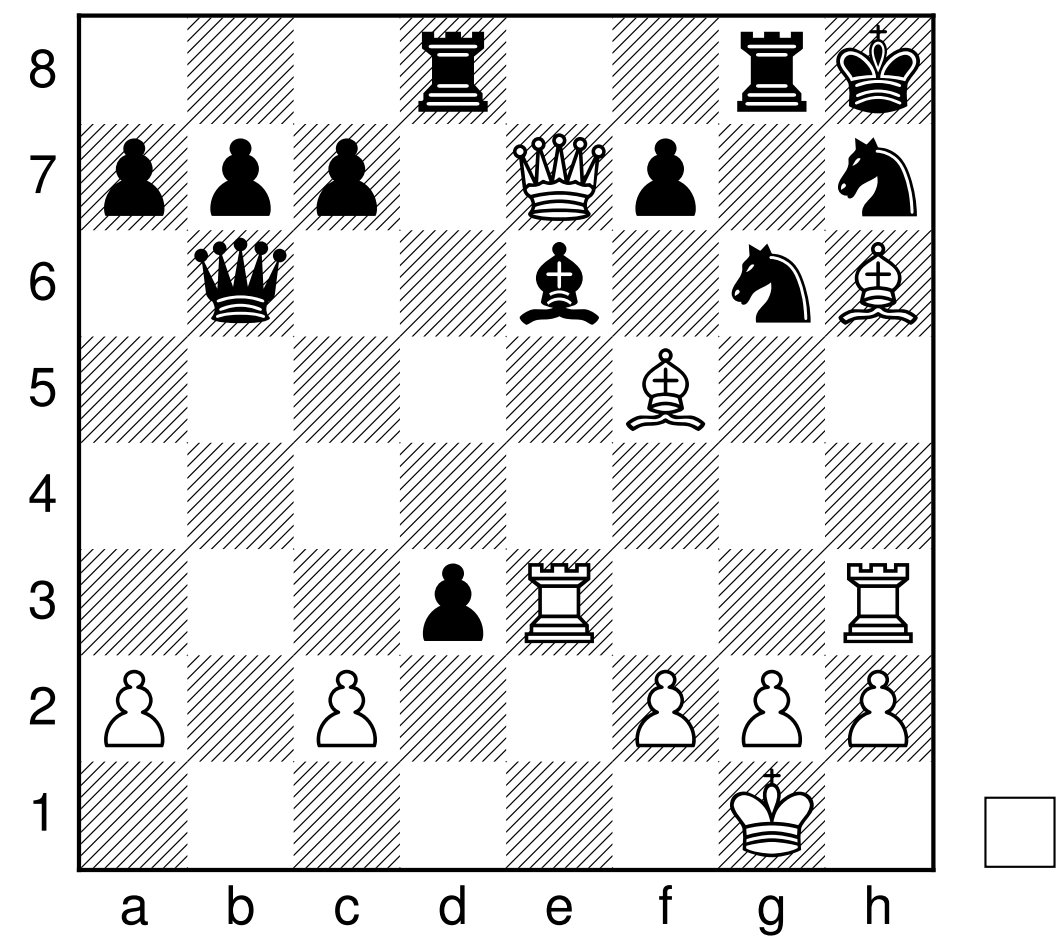

[Analyse on Lichess]

Closest FENs - [1], [2], [3]

### M.14 Uncategorized

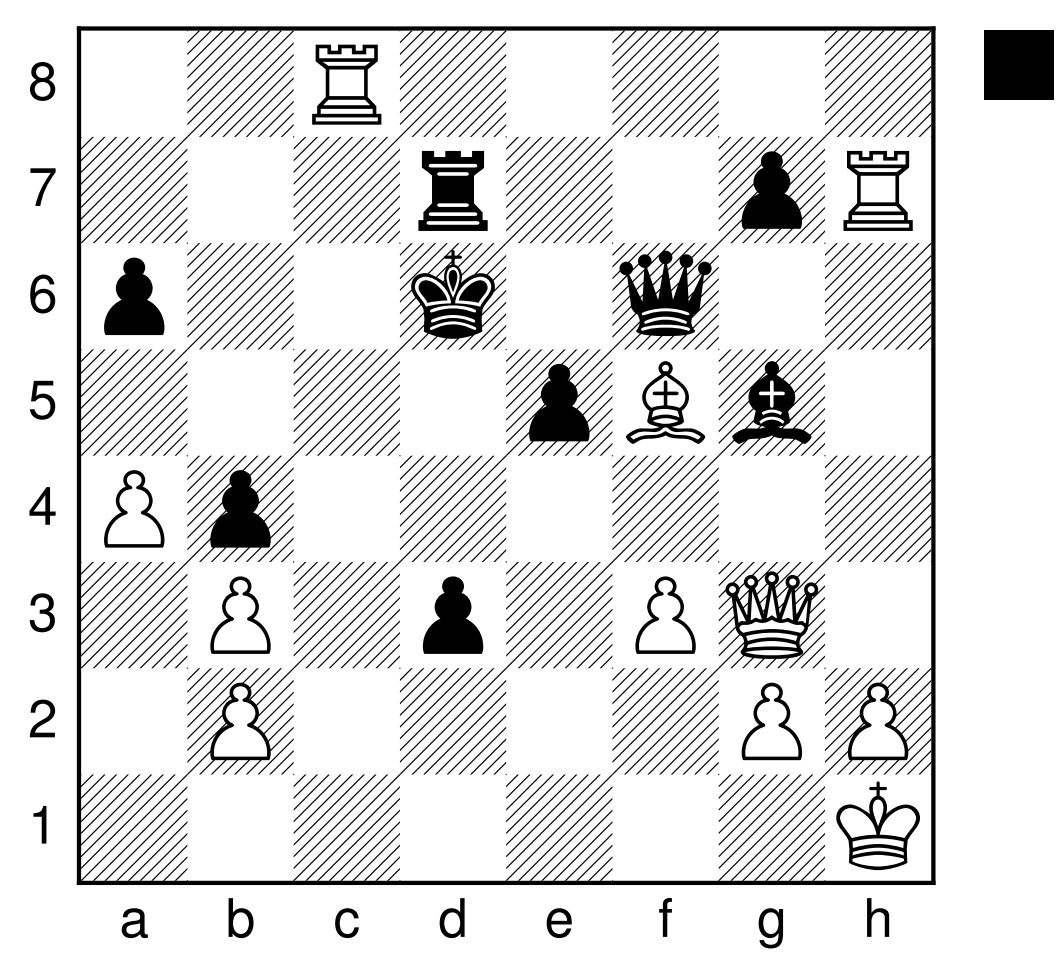

[Analyse on Lichess]

Closest FENs - [1], [2], [3]

1... d2 2. Bc2 Qf5! **The only winning idea. Black sacrifices the queen to try to promote the pawn.** 3. Bd1 Qc1 **White's pieces are not coordinated and there is no good response.**

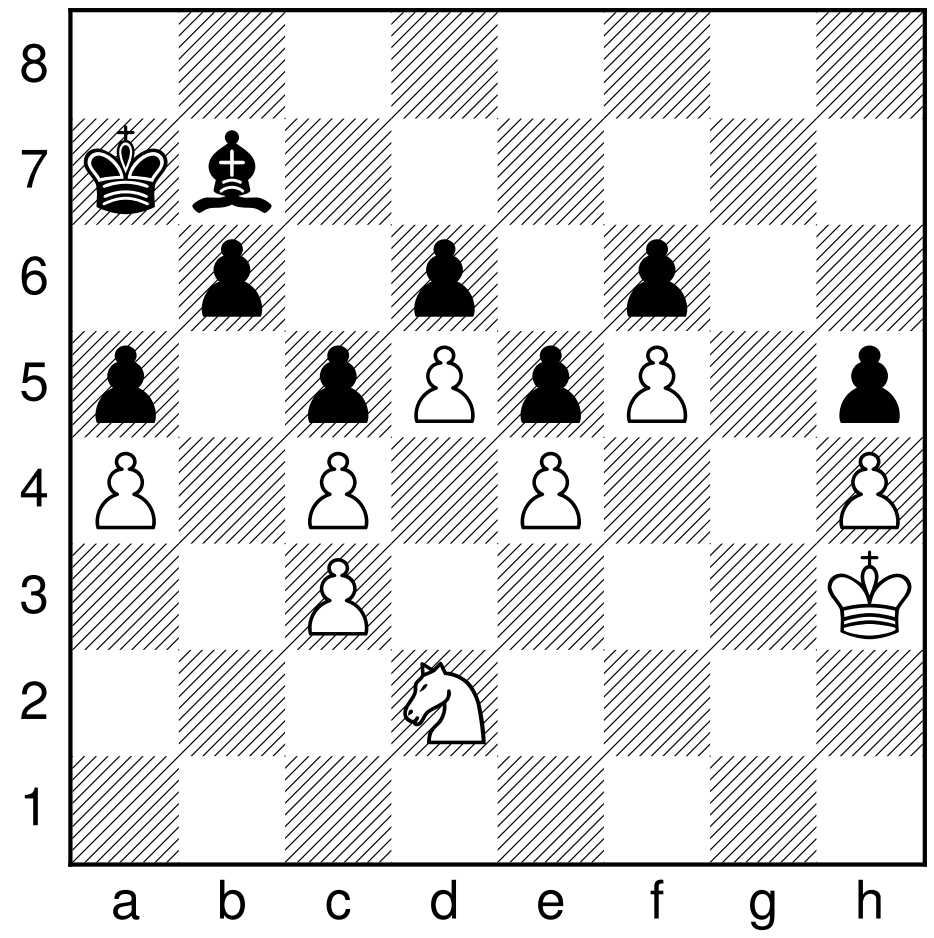

[Analyse on Lichess]

**Closest FENs - [1], [2], [3]**

# N Puzzles generated with evolutionary search

The following puzzles are selected from the ones generated with evolutionary search.

As briefly discussed in the Introduction, this method contrasts sharply with our generative modeling approaches. Instead of learning from data, this method relies on applying random mutations and perturbations to a population of chess positions, and directly optimizes for counter intuitiveness check. The optimization process does not enforce realism constraints, which is a feature of our generative approaches and so the puzzles produced with this method often diverges significantly from expert-level games. The allows for generating beyond what is present within the training data, demonstrating a powerful alternative for creative chess puzzle generation.

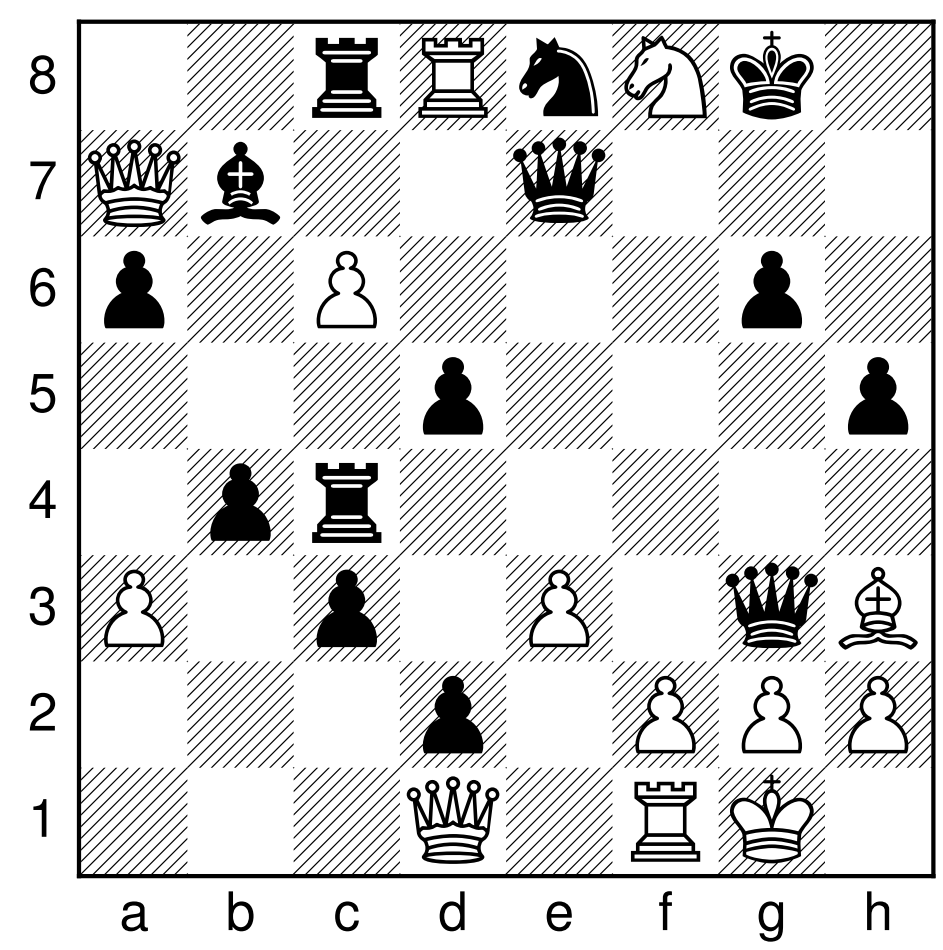

[Analyse on Lichess]

1... Qxh3 **Giving up the queen for the bishop is the right decision here. The other moves fail tactically. For example 1...Qc7 2. Rd7 or 1... Qb8 2. Qxb8 or 1... Qg5 2. cxb7 Rxd8 3. Ne6 Qf6 4. Nxd8 Rc7 5. Qa8 is winning for White** 2. gxh3 Rxc6 3. Rxc8 Qg5+ 4. Kh1 Bxc8 5. Qf3 Qf5 6. Qxf5 Bxf5 7. Qd4 Kxf8 8. Qxb4 Nd6 **And the pawns on the queenside can't be stopped.**

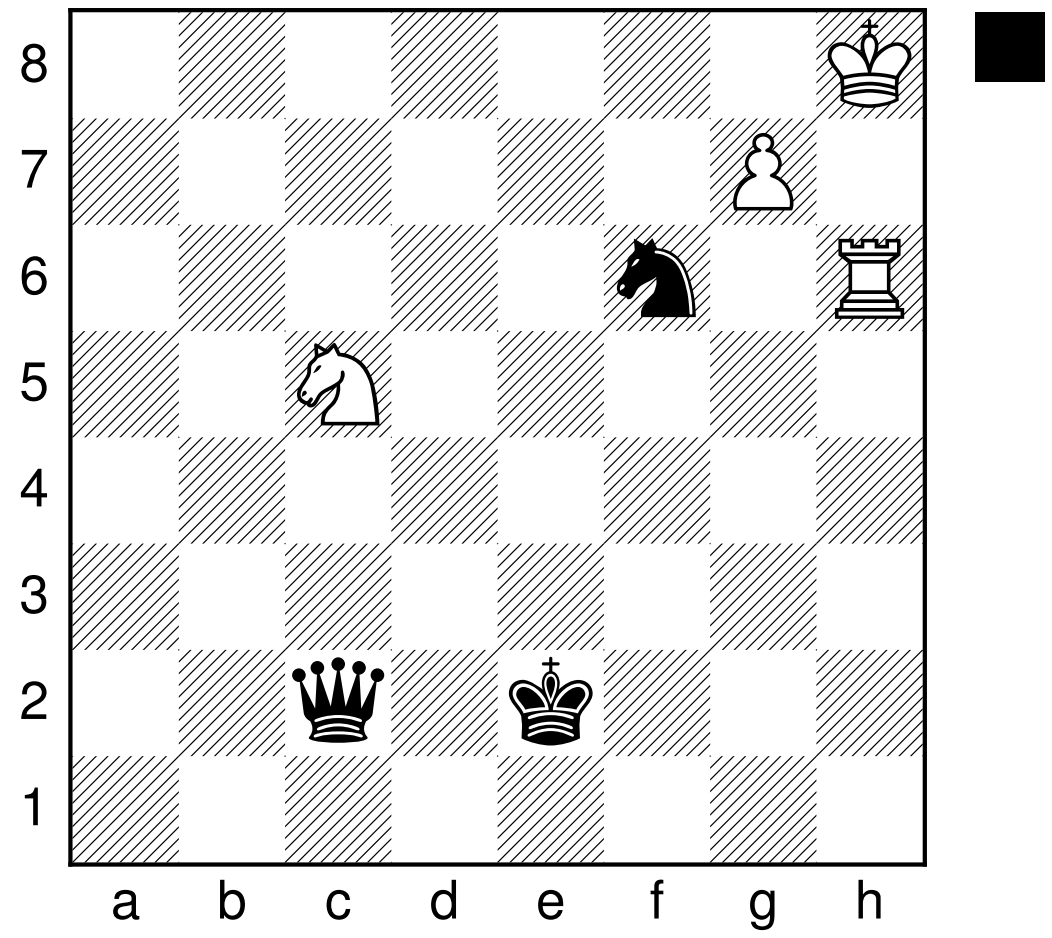

[Analyse on Lichess]

1... Qc4! **Not 1... Qxc5 2. Rxf6 Qh5+ 3. Kg8 and while Black can hold the draw, there is no path to a win here. Other tries don't work either, for example 1... Qf5 2. Nd7! Qxd7 3. Rxf6 Qh3+ 4. Kg8 Qc8+ 5. Rf8 and this is a draw. Another try would be 1... Qc3, but then White can just play Ne6 and a draw would ensure shortly.** 2. Rxf6 Qh4+ 3. Kg8 Qxf6 4. Nd7 Qf5 5. Nf8 Ke3 **and Black can start bringing the king forward. If White tries 6. Kh8 Qe5 7. Kg8 Qh5 8. Ne6 Ke4 and the king keeps approaching.**

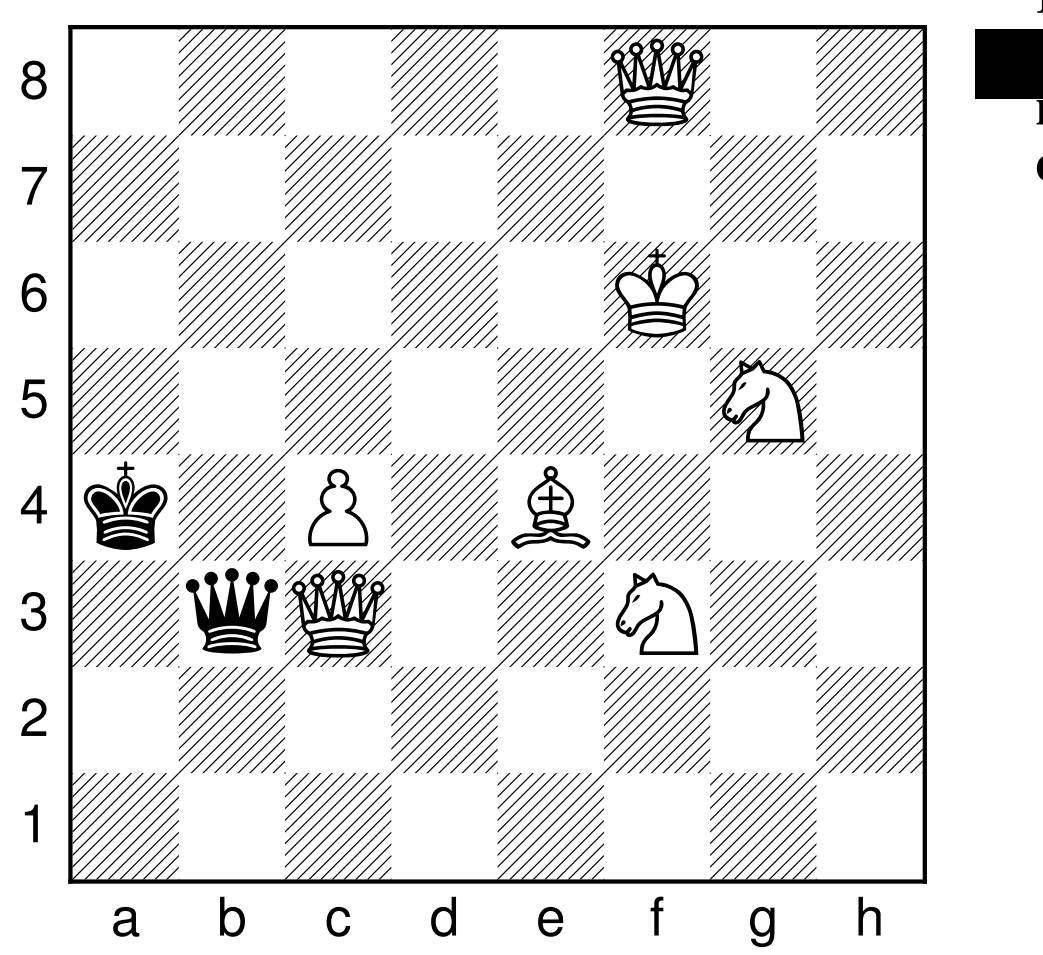

[Analyse on Lichess]

**Despite White's overwhelming material advantage, Black manages to salvage a draw.** 1... Qb6+ 2. Ke5 Qf6+ 3. Kd5 Qd6+ 4. Kxd6.

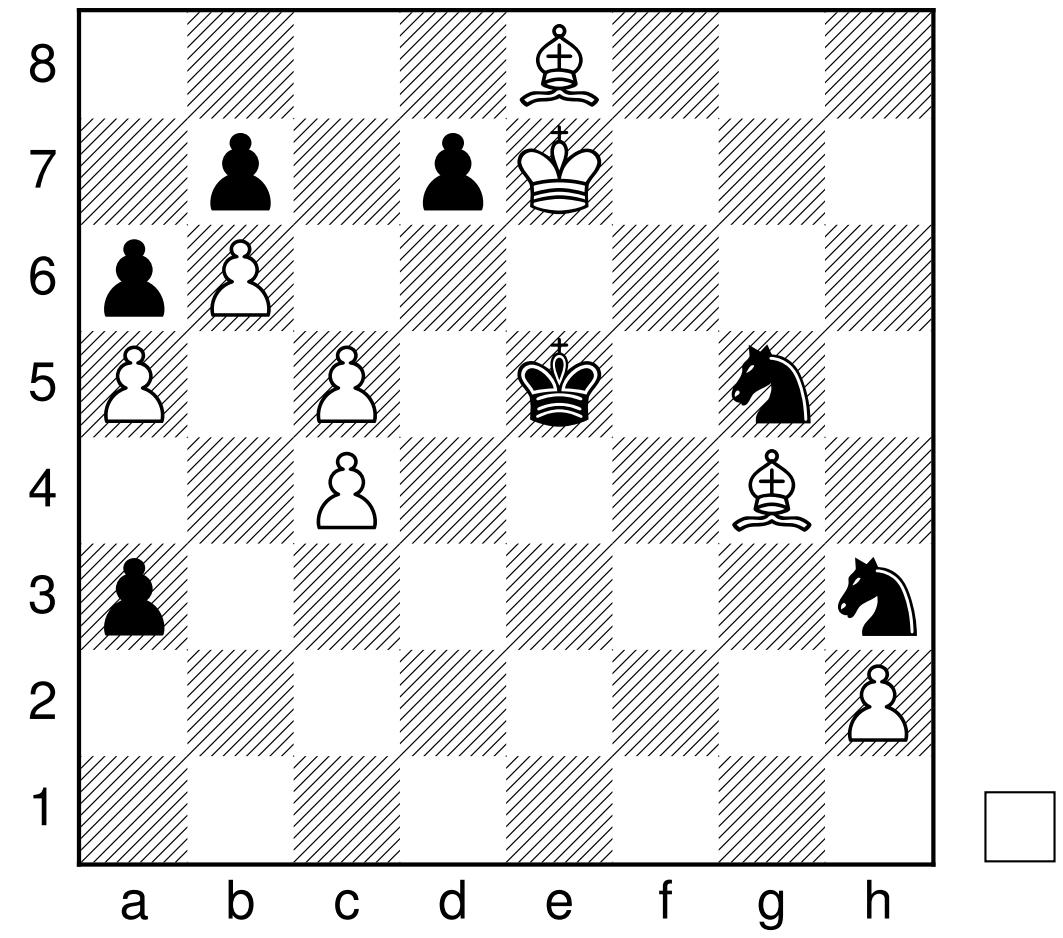

[Analyse on Lichess]

**Among the 3 possible captures on d7 (and other alternatives) only one is correct.** 1. Bgxd7 a2 2. c6 a1=Q 3. cxb7 Qa3+ 4. Kd8 Ne6+ 5. Kc8 Qc5+ 6. Bc6 Qe7 7. b8=Q+ **wins for White - as an example line that follows. If White were to instead play 1. Kxd7, then Black would lose if following up with 1... a2, but wins in case of 1. Kxd7 Ne4 2. c6 Nc5+ 3. Kc7 bxc6 4. Kxc6 Kd4 5. Bxh3 a2 6. b7 Nxb7 7. Kxb7 a1=Q. It's important to note the following variation: 1. Bgxd7 Ne4 2.. c6 Nd6 3. c5, which is why Ne4 can't save Black in the solution. Finally, in case of the other bishop capture: 1. Bexd7 a2 2. c6 a1=Q and now 3. cxb7 wouldn't work (in contrast to the solution) 3... Qa3+ 4. Kd8 Nf7+ 52. Kc7 Qd6+ as an example line - the knight check on f7 is possible as the bishop is no longer on e8 - other moves don't work either (c7 or alternative king moves).**

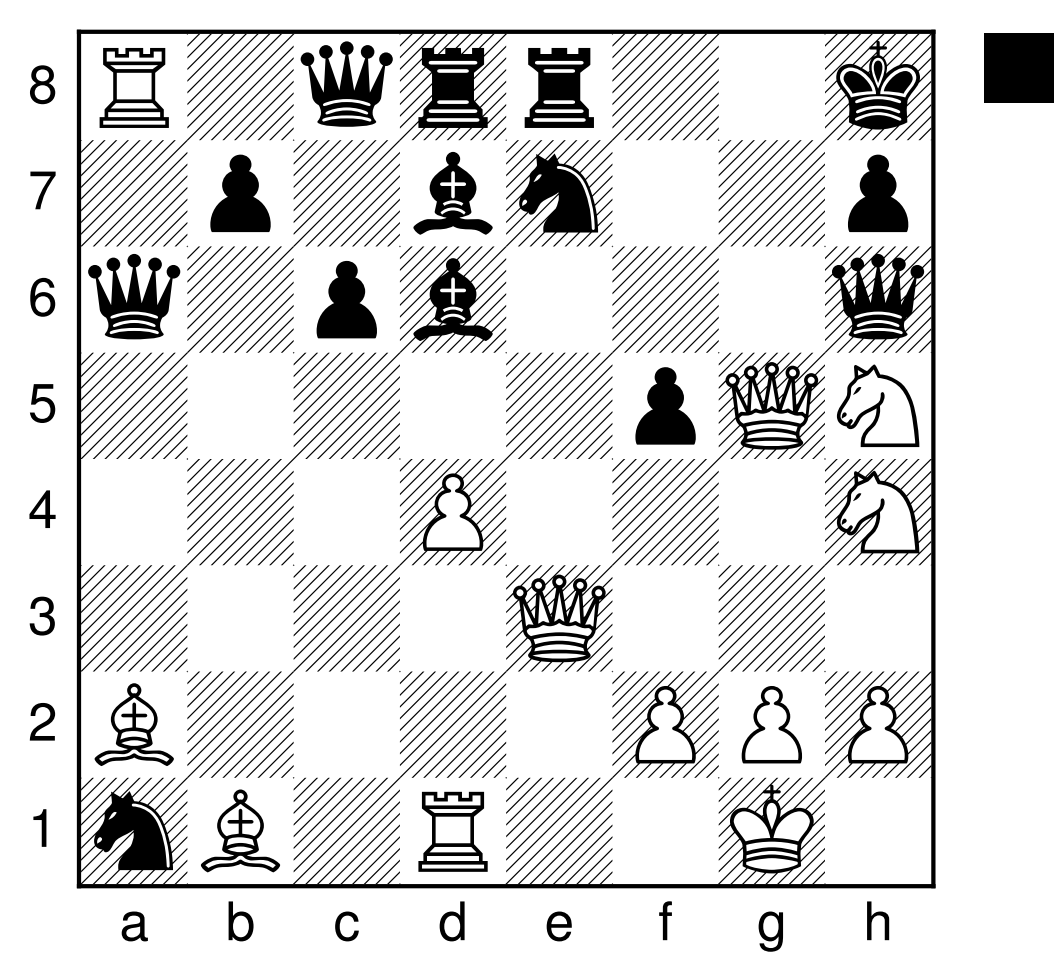

[Analyse on Lichess]

1... Qe2! **Moving the queen from one square where it can be captured to another.** 2. Qxe2 Qxg5 3. Rxc8 Qxh4 4. g3 Nxc8 **and Black is winning.**

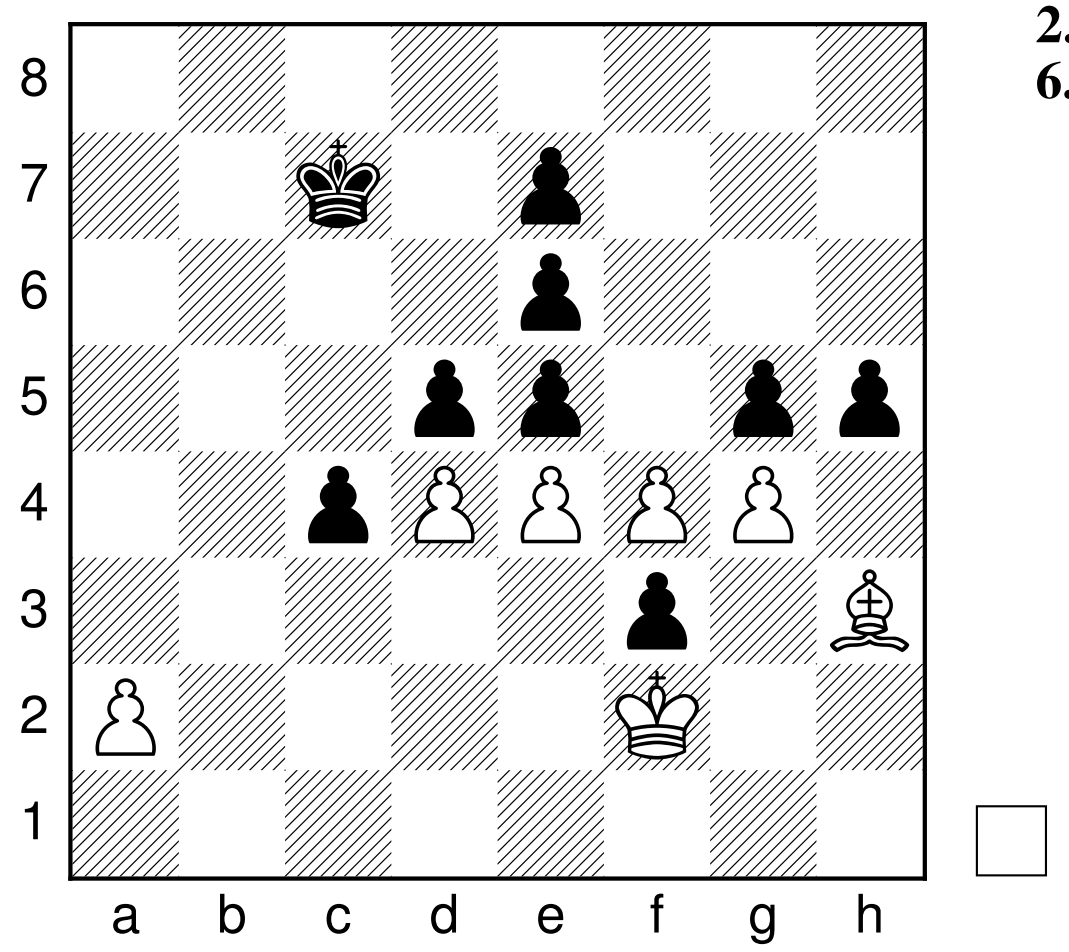

[Analyse on Lichess]

**In this position, there are many moves to consider, and precise calculation is needed to establish that only one of them wins for White.** 1. Kxf3 hxg4+ 2. Bxg4 dxe4+ 3. Kxe4 c3 4. Kd3 exf4 5. Bxe6 **Let's consider the alternatives. 1. dxe5 dxe4 2. fxg5 c3 3. Ke3 f2 4. Bf1 h4 5. g6 c2 6. Kd2 h3 7. Bxh3 e3+ 8. Kxc2 e2 9. g7 e1=Q 10. g8=Q Qxe5 is a draw instead. 1. gxh5 c3 2. h6 c2 3. h7 c1=Q 4. h8=Q Qd2+ 5. Kxf3 dxe4+ is winning for Black, because 6. Kxe4 is impossible due to the threat of 6... Qe2# For the same reason 1. fxg5 fails. Finally, 1. exd5 c3 2. Ke3 exf4+ 3. Kd3 hxg4 4. Bxg4 f2 5. Be2 g4 is winning for Black.**

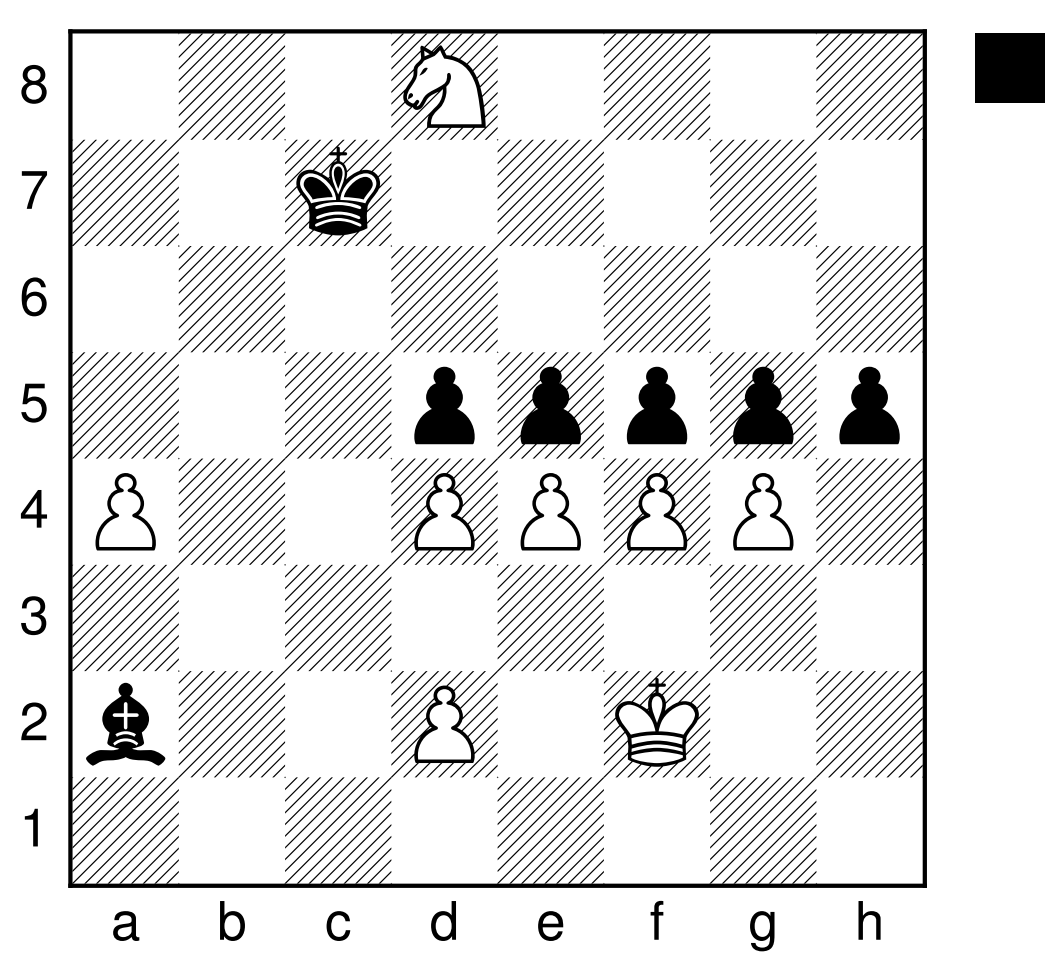

[Analyse on Lichess]

**Of all the possible pawn captures, only one is winning for Black**. 1... fxg4 2. Ne6+ Kd6 3. Nxg5 exf4 4. e5+ Ke7 5. a5 Bc4 **and White can't stop the Black pawns. Interestingly, capturing the hanging White knight on the first move would have lost the game for Black. 1... Kxd8 2. gxh5 Ke7 3. exd5 exf4 4. h6 Kf6 5. h7 Kg7 6. d6 Be6 7. a5 Bc8 8. a6.**

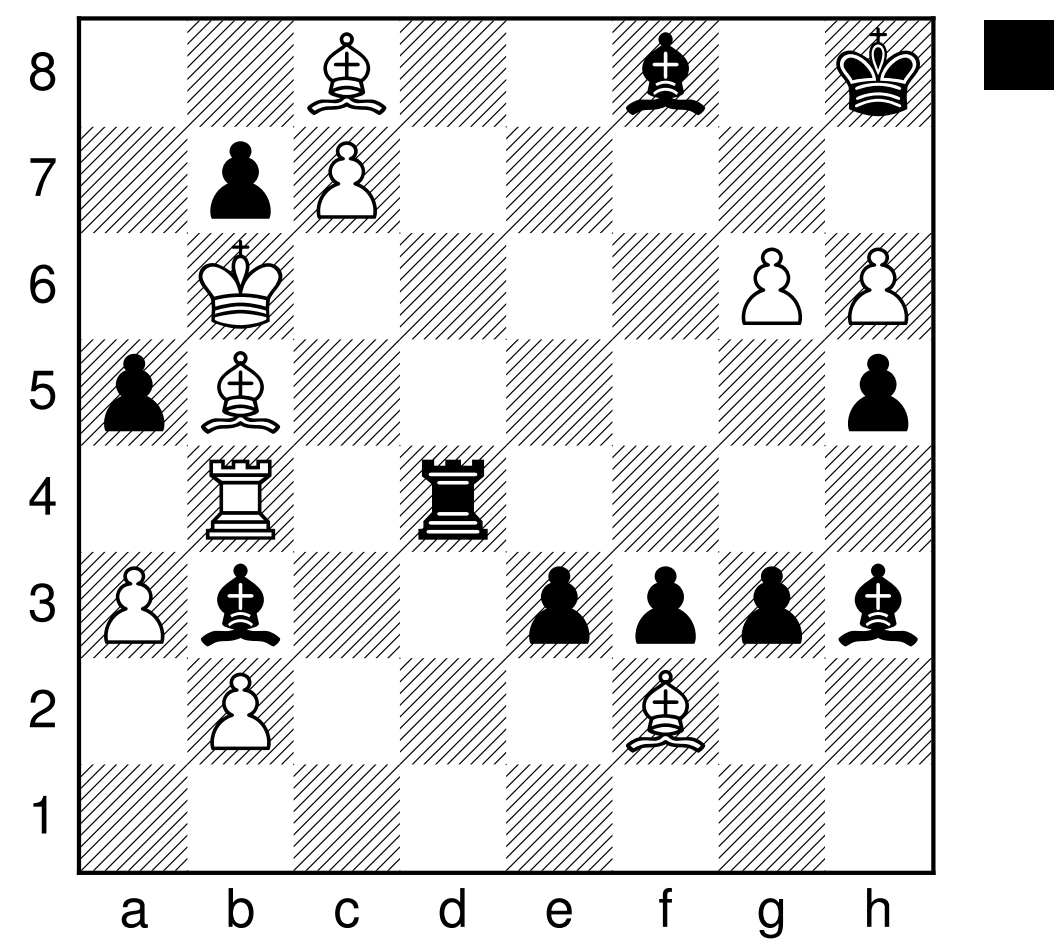

[Analyse on Lichess]

**In what is a very messy position, there is a narrow path to a win.** 1... Rd6+ (1... axb4 2. Bxh3 Rd6+ 3. Ka7 Bxh6 4. c8=Q+ Bg8 5. Bxg3 Rxg6 6. Be5+ **would be winning for White instead. The other captures on move 1 fail as well.**) 2. Ka7 (2. Kxb7 Bd5+ 3. Ka7 Bxc8) Bxc8 3. Bxg3 Rxg6 4. Be5+ Kh7 5. Rxb3 Bc5+ 6. Kb8 Rg8 7. Rd3 Bb6 8. Rd8 Rxd8 9. cxd8=Q Bxd8 10. Kxc8 Kxh6 11. Kxd8 Kg5.

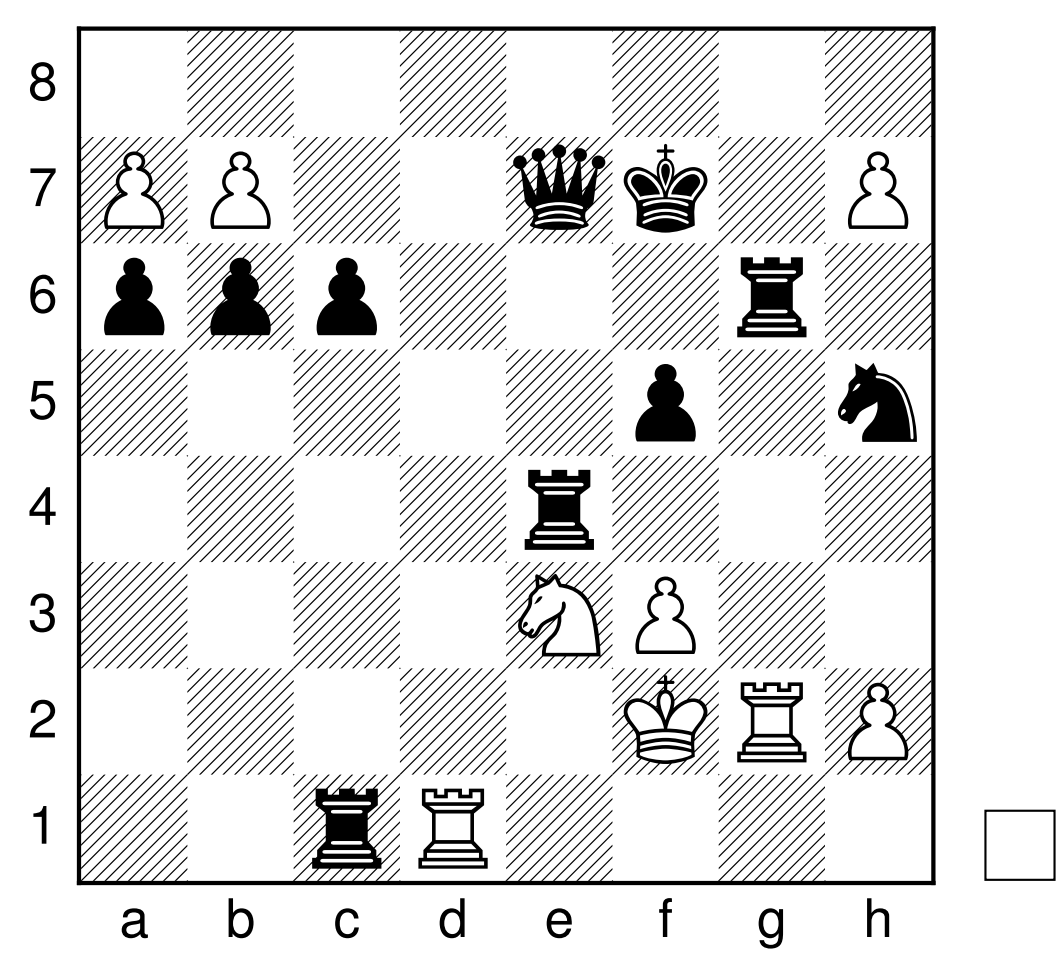

[Analyse on Lichess]

**In what is aesthetically a very pleasing position, there are three different pawns that can be promoted on the next move, and multiple additional promising captures to calculate as well - for example, the hanging Black rook on c1. However, it turns out that there is only one winning move, and it is an under-promotion!** 1. h8=N+ Kg7 2. Rxg6+ Kh7 3. b8=Q Rc2+ 4. Kf1.

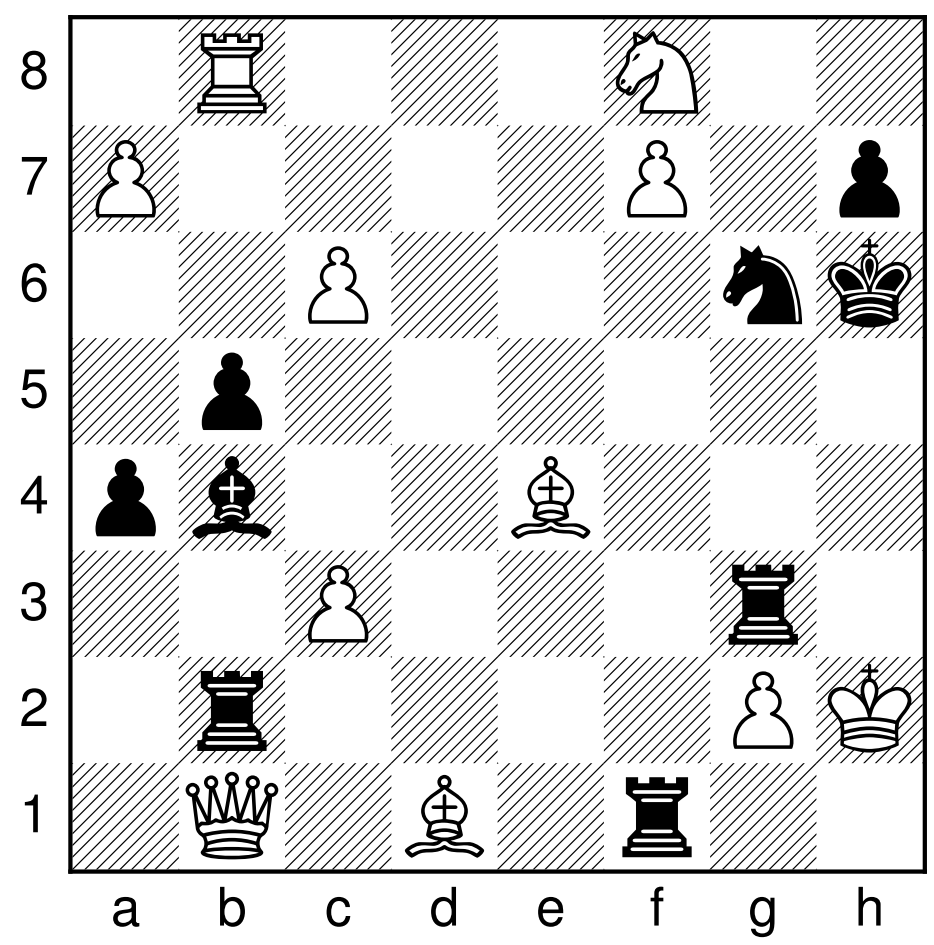

**[Analyse on Lichess]**

**Despite the White queen hanging on b1, Black needs to find a different motif to win in this position.** 1... Bd6 2. Qxb2 Rgf3+ 3. g3 Bxg3+ 4. Kh3 Rh1+ 5. Kg2 Nh4+ 6. Kxh1 Rf1#.

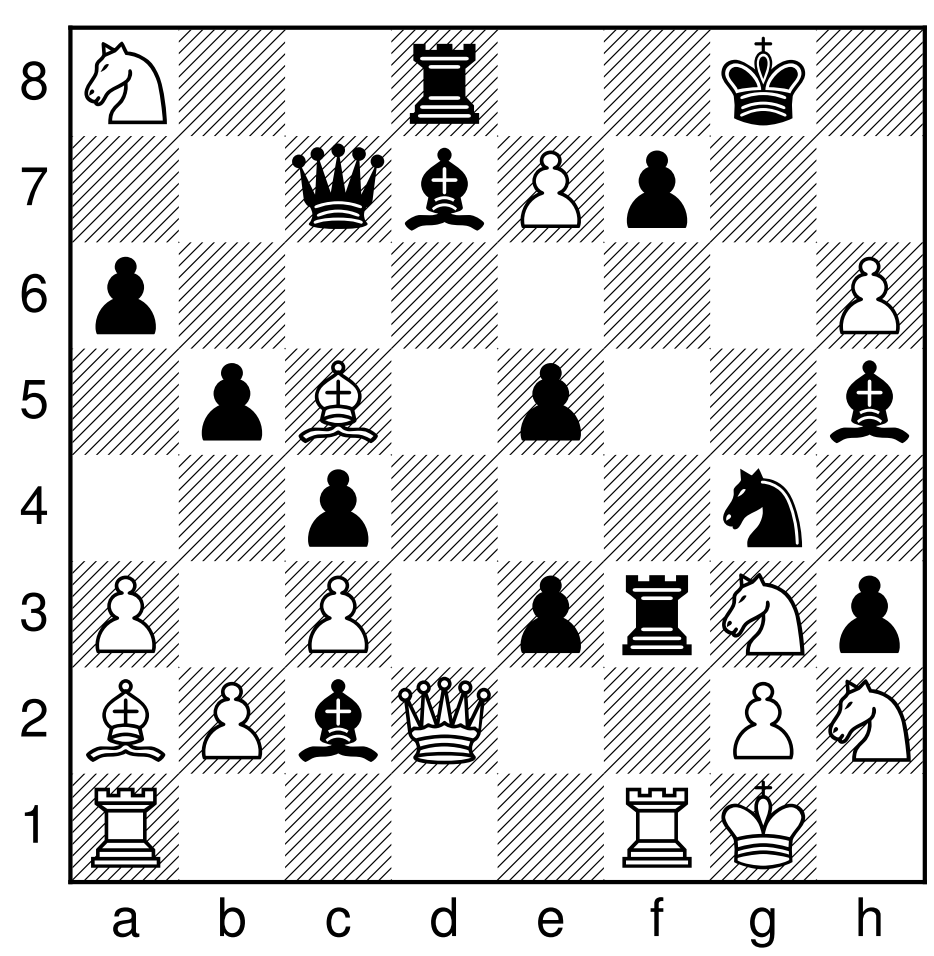

**[Analyse on Lichess]**

**In this astoundingly chaotic and complicated position with many possible captures, there is only one move that secures the win. Perhaps more interestingly, interjecting a check on f1 would throw away the advantage, which is rarely the case! This is especially puzzling at the first glance since the rook is otherwise hanging on f3. Yet, there are specific tactical reasons for why this resource is not available. The correct capture is the capture on a8, one example line being:** 1... Rxa8 2. Qd5 Rxg3 3. Nxg4 Bc6 4. Nf6+ Kh8 5. Nxh5 Bxd5 6. Nxg3 Qxc5 **where Black is winning. Obviously this line is not forced, but the alternatives don't affect the outcome. So, let's look at why Rxf1 fails specifically on the first move of the problem: 1... Rxf1+ 2. Rxf1 Rxa8 3. Bxe3 Nxe3 4. Qxe3 Qc6 5. Qg5+ Bhg6 6. Nh5 would be winning for White. In contrast, with the rook still there, Black is able to generate simultaneous threats against e3 and g2 in the same variation, under the main line of the solutions: 1... Rxa8 2. Bxe3 Rxg3 3. Nxg4 Bdxg4 4. Qxc2 Rxg2+ 5. Qxg2 hxg2.**

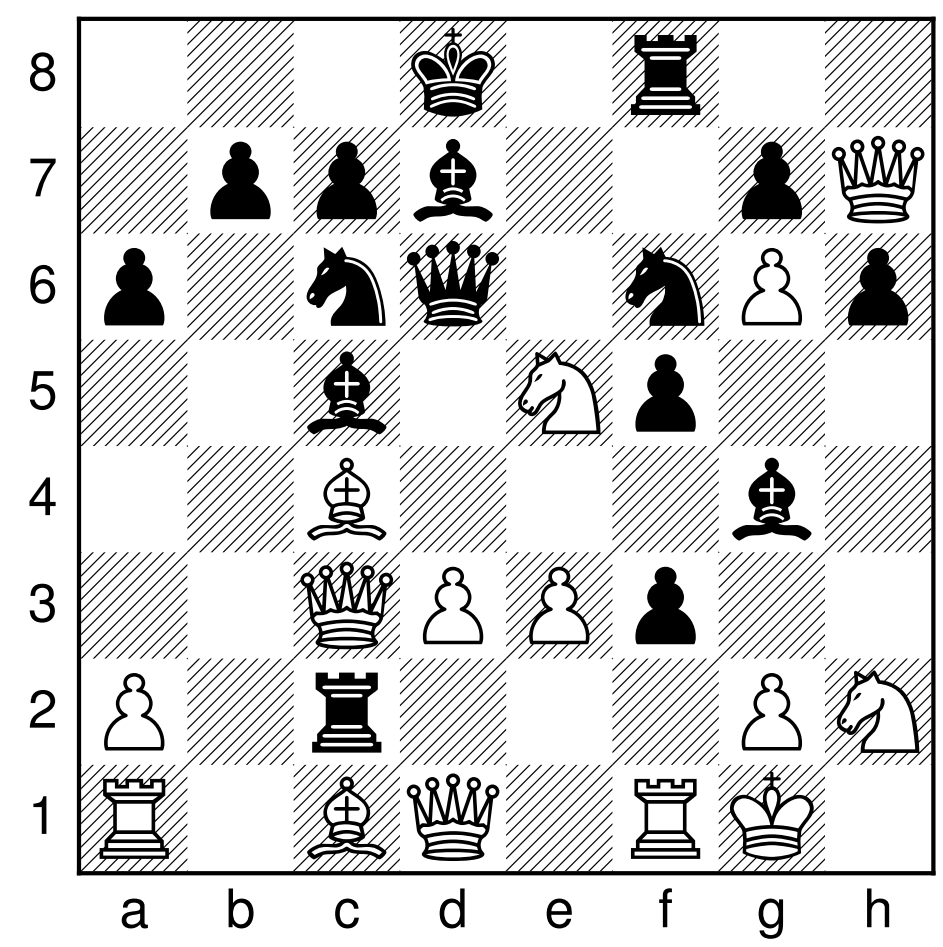

**[Analyse on Lichess]**

**It is possible to capture either of the two White queens on c3 and h7 respectively, but neither is the best move - the best move involves ignoring the opportunity and giving a check instead.** 1... Rxg2+ 2. Kh1 Nxe5 3. Qxg7 Nxg6 4. Nxf3 Qg3 **is the winning recipe. If Black were to have been tempted by material instead: 1... Rxc3 2. Nxd7 Qxd7 3. gxf3 Bh5 4. Bb2 Nxh7 5. gxh7 Rxc4 6. dxc4 Rh8 7. Bxg7 Rxh7 8. Qxd7+ Kxd7 9. Bd4 Nxd4 10. Rfd1 Bd6 11. Rxd4 Kc6 12. f4 Bc5 13. Rd3 Be2 14. Rc3 is only a draw - as an example line that may follow from the capture.**

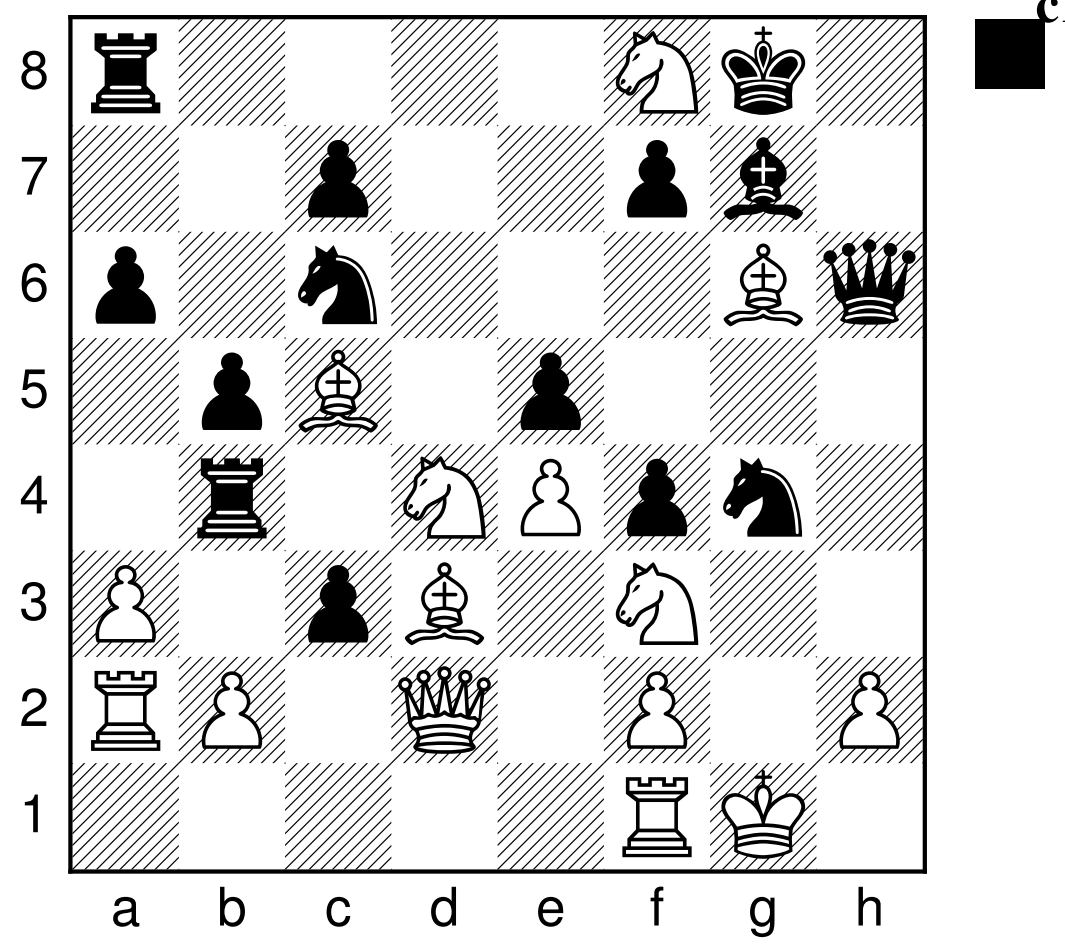

[Analyse on Lichess]

**Black needs to find the correct capture, among the available options.** 1... Nxd4 2. Bxd4 cxd2 3. axb4 exd4 4. Bf5 Qh3 **is winning for Black, unlike the alternatives. If instead: 1... Rxd4 2. bxc3 Rxd3 3. Qxd3 Bxf8 4. Bxf7+ Kxf7 5. Qd5+ Kg7 6. Qd7+ Kh8 7. Qxg4 or 1... cxd2 2. Nf5 Qh3 3. Bxf7+ Kxf7 4. Ng5+ or another example line (with possible deviations at several points): 1... exd4 2. bxc3 fxg6 3. Ne6 Nce5 4. Neg5 dxc3 5. Qd1 Bf6 6. Bxb4 Bxg5 7. h4 Bxh4 8. Bxc3.**

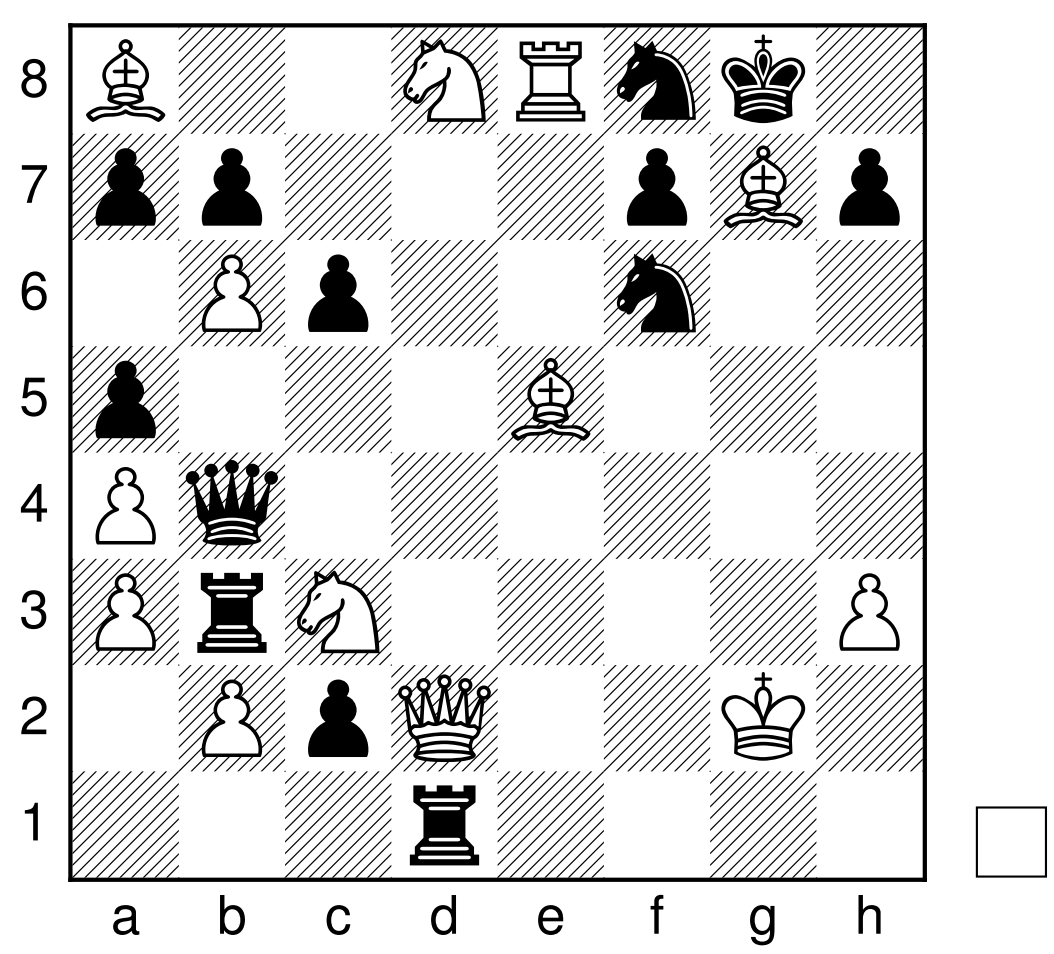

[Analyse on Lichess]

**Of several possible first moves, only one wins. One possible continuation would be:** 1. Qf2 c1=Q 2. axb4 Qg5+ 3. Bg3 Kxg7 4. Nxd1 Qd5+ 5. Kh2 Nxe8 6. bxa7 Qxd8 7. Bxb7 Nc7 8. Ne3 Kg8 9. Ng4 Rxg3 10. Qxg3 Nfe6 11. Nh6+ Kf8 12. Qg8+ Ke7 13. Qxf7+ Kd6 **Moving the queen to e2 instead fails due to: 1. Qe2 Rg1+ 2. Kxg1 c1=Q+ and if 1. Qg5 c1=Q.**

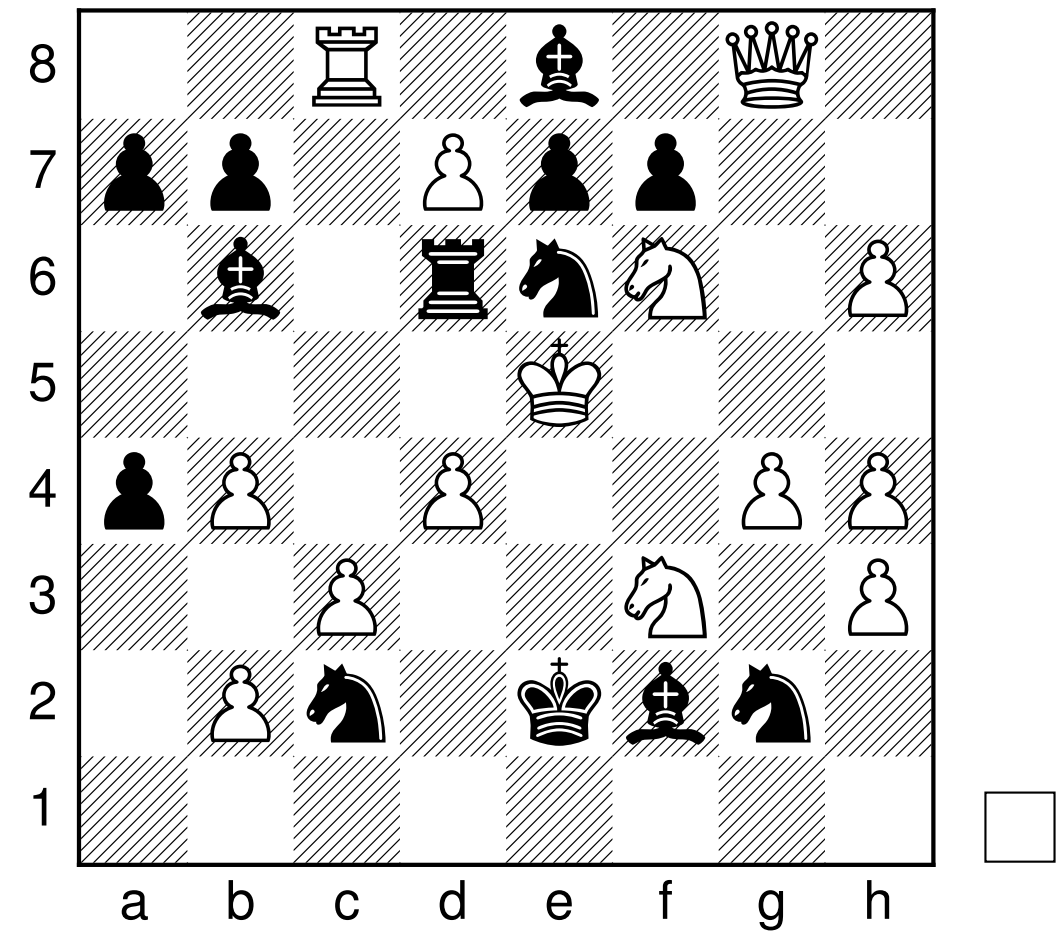

[Analyse on Lichess]

**To win, amidst the chaos, White needs to play a quiet king move in the centre of the board.** 1. Ke4 Bxd7 2. Nxd7 **If instead 1. Qh7 Bg3+ 2. Ke4 Nc5+ 3. Rxc5 Re6+ 4. Re5 Rxf6 5. Ng1+ Kd2 6. Nf3+ Ke2 7. Ng1+ the threats against the White king secure Black a draw.**

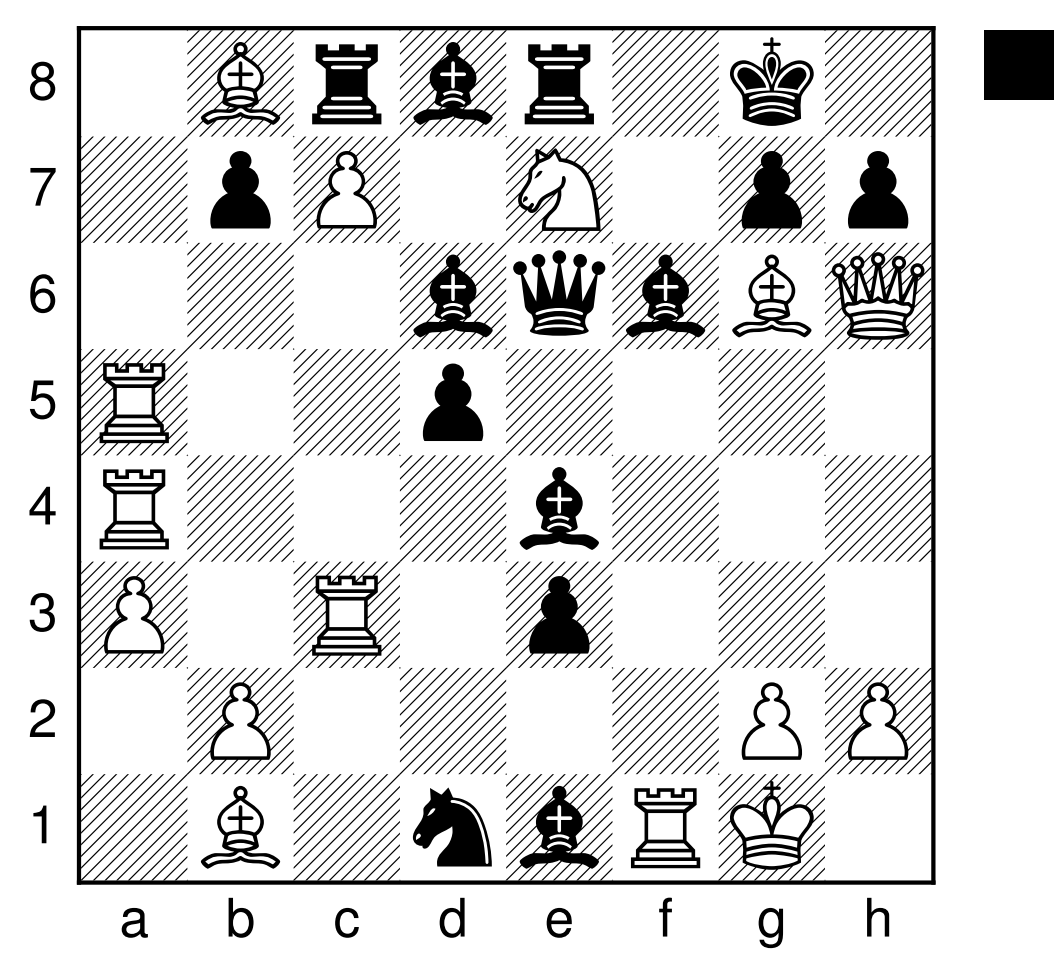

[Analyse on Lichess]

**Black has 5 different ways of capturing the knight on e7 that is giving check. However, the only correct decision is not to capture it at all!** 1... Kf8 2. Qxh7 Bf2+ 3. Kh1 Kxe7 4. cxd8=Q+ Rexd8 5. Bxd6+ Rxd6 6. Rxc8 e2 7. Re8+ Kd7 8. Bd3 Bxd3 9. Bxd3 exf1=Q+ 10. Bxf1 Qxe8 11. Qd3 Ne3 12. Qb5+ Ke7 13. Qxb7+ Qd7 **seems to be holding on, in an example continuation. Alternatively, 1... B8xe7 2. Qxh7+ Kf8 3. Rxe4**

**dxe4 4. Qh8+ Qg8 5. Qxg8+ Kxg8 6. Ba2+ Kf8 7. Rh5 is winning for White. Rxe4 comes up as a motif in some of the other lines, for alternative first-move captures.**

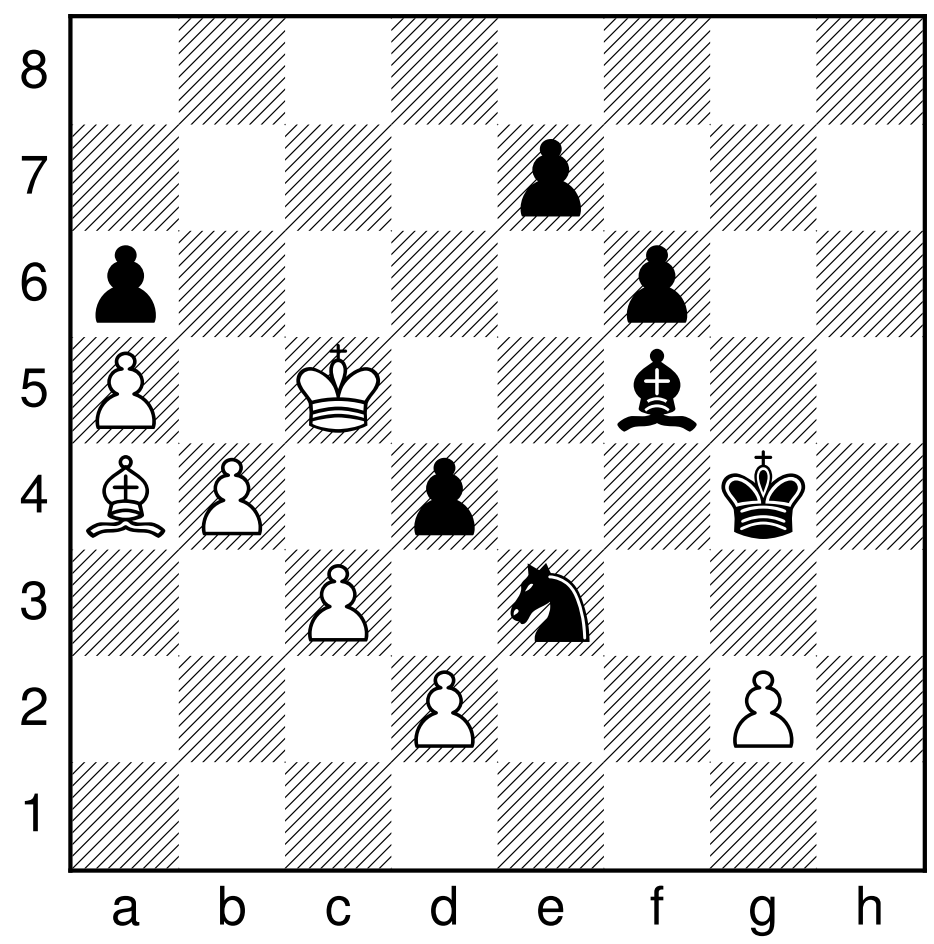

**[Analyse on Lichess]**

1. cxd4 Nxg2 2. Kb6 Bc8 3. Bc6 Nf4 4. Bb7 Bxb7 5. Kxb7 Nd5 6. b5 Nb4 7. bxa6 Nxa6 8. Kxa6 f5 9. Kb6 f4 10. a6 f3 11. a7 f2 12. a8=Q f1=Q 13. Qg8+ **and White will pick up the e7 pawn and have a winning position. Alternative first-move captures don't work for White, for example: 1. Kxd4 Nxg2 2. Kc5 Nf4 3. b5 Nd3+ 4. Kb6 Nb2 5. Bb3 axb5 6. a6 Nc4+ 7. Kxb5 Nd6+ 8. Kc5 Nc8 9. Bd5 Bd3 10. Bb7 Bxa6 11. Bxa6 Nd6 is a draw, and capturing the knight instead loses: 1. dxe3 dxc3 2. Kb6 Bd3.**

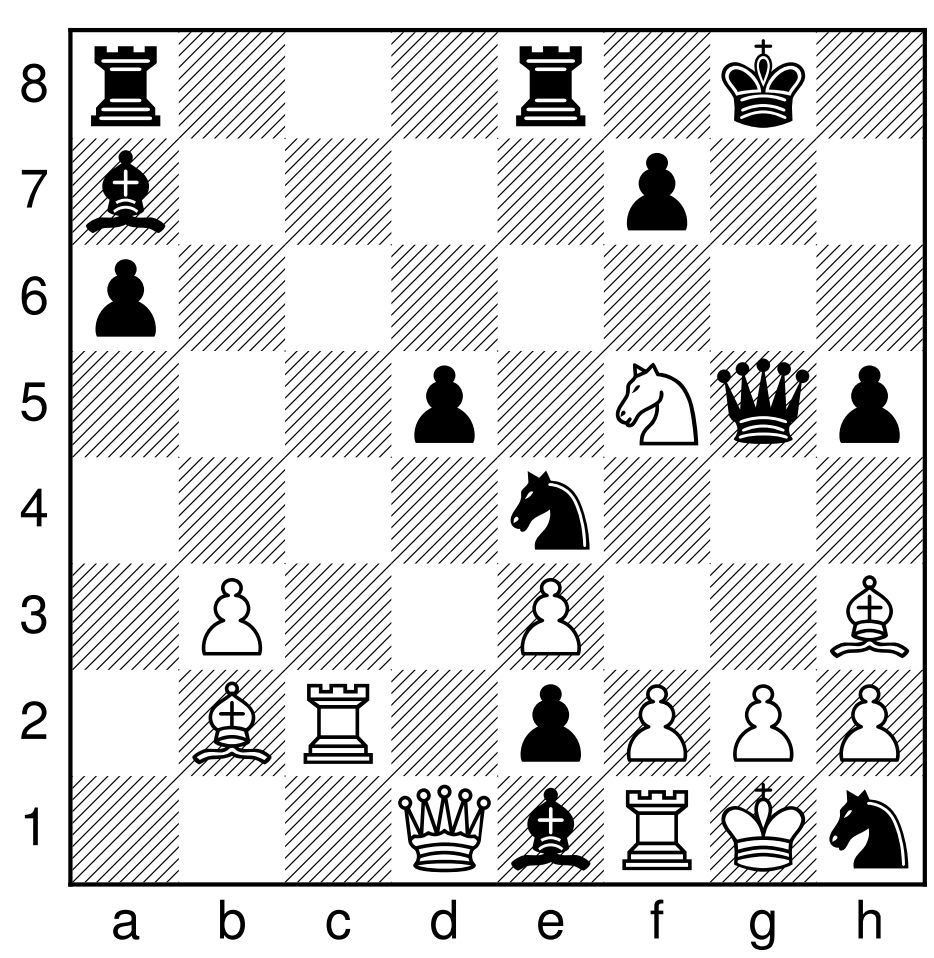

**[Analyse on Lichess]**

1. Rxe2 Nhxf2 2. Qxe1 Nd3 3. Qd1 Nxb2 4. Rxb2 Bxe3+ 5. Nxe3 Nc3 6. Qf3 Qxe3+ 7. Rbf2 Ra7 8. Qxe3 Rxe3 9. Rf5 **With equality. This is ultimately a choice between two possible captures on e2 - with interesting ideas in both lines. If (the wrong move) 1.Qxe2 then after 1...Nhxf2 2. Qxe1 Nxh3+ 3.Kh1, Black plays the beautiful move 3...Nf4! where after 4.Rxf4 Qxf4! (Another nice sacrifice) 5. exf4 Nf2+ 6. Qxf2 Bxf2 Black is suddenly up material. But the reason why Rxe2 works instead is even more subtle. So, let's look at that same line, just with the rook on e2. 1.Rxe2 Nhxf2 2. Qxe1 Nxh3+ 3. Kh1 Nf4 4. Rxf4 Qxf4 and now instead of exf4, White has a surprising option 5. Ne7+! Rxe7 6. exf4 Nxf2+ 7. Qxf2 Bxf2 8. Rxe7 - because the rook on e7, where it captured the knight, is not defended by the rook on a8, White doesn't end up down an exchange.**

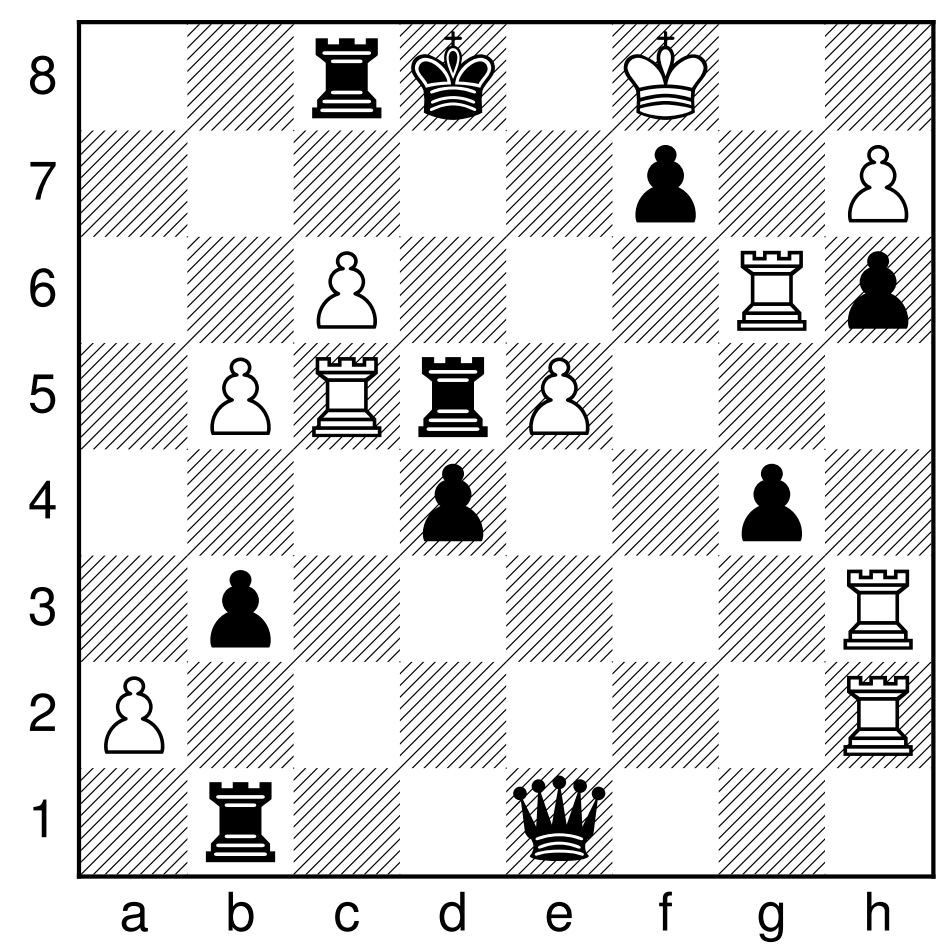

**[Analyse on Lichess]**

**The Black rook on d5 is hanging with check, but capturing it is, surprisingly, not best.** 1. b6 Qb4 2. Rd6+ Rxd6 3. h8=Q Qxb6 4. exd6 **With mate to follow. If instead 1. Rxd5+ Kc7+ 2. Kxf7 Qf1+ 3. Rf6 Qc4 4. Rd6 gxh3 5. Ke7 Rg1 it would be Black who is winning.**

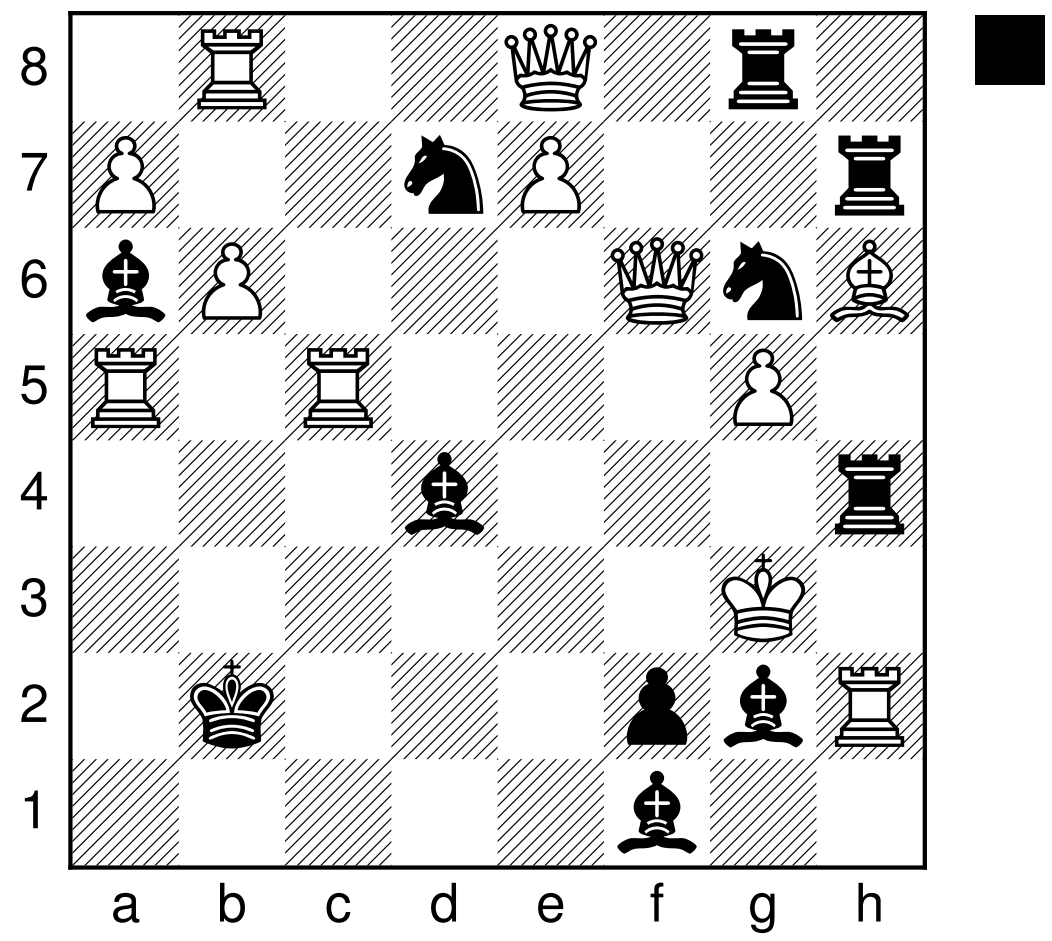

[Analyse on Lichess]

**There are many possible moves to consider in this chaotic and highly unrealistic board setup. Yet, there is only one solution, and it involves a temporary forced rook sacrifice!** 1... Rg4+ 2. Kxg4 Nxf6+ **And now recapturing doesn't work because 3. gxf6 Bfe2+ 4. Kg5 Nf8+ 5. Bg7 Be3+ 6. Kf5 f1=Q+ 7. Ke5 Qf4#** 3. Kg3 Ne4+ 4. Kg4 Bfe2+ 5. Kf5 f1=Q+ 6. Ke6 Nxc5+ 7. Rxc5 Bg4+ 8. Rf5 Qxf5+ 9. Kd6 Qe5#.

## O Puzzles adversarial to chess engines

The following puzzles were generated with reinforcement learning. These positions are computationally demanding, and often requires significant search-time for Stockfish to determine the optimal line. This is likely because of many pieces on the board, which causes Stockfish to evaluate a number of lines from a large search tree.

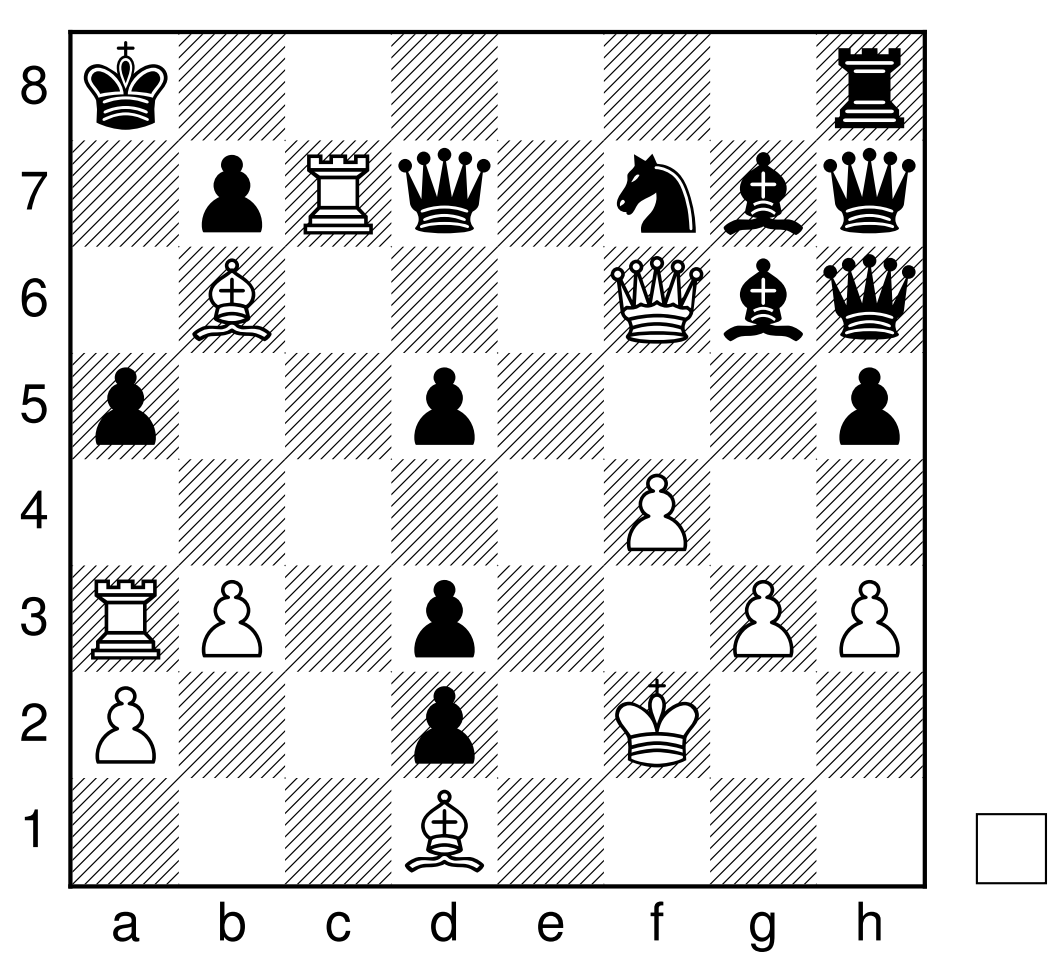

[Analyse on Lichess]

textbfClosest FENs - [1], [2], [3]

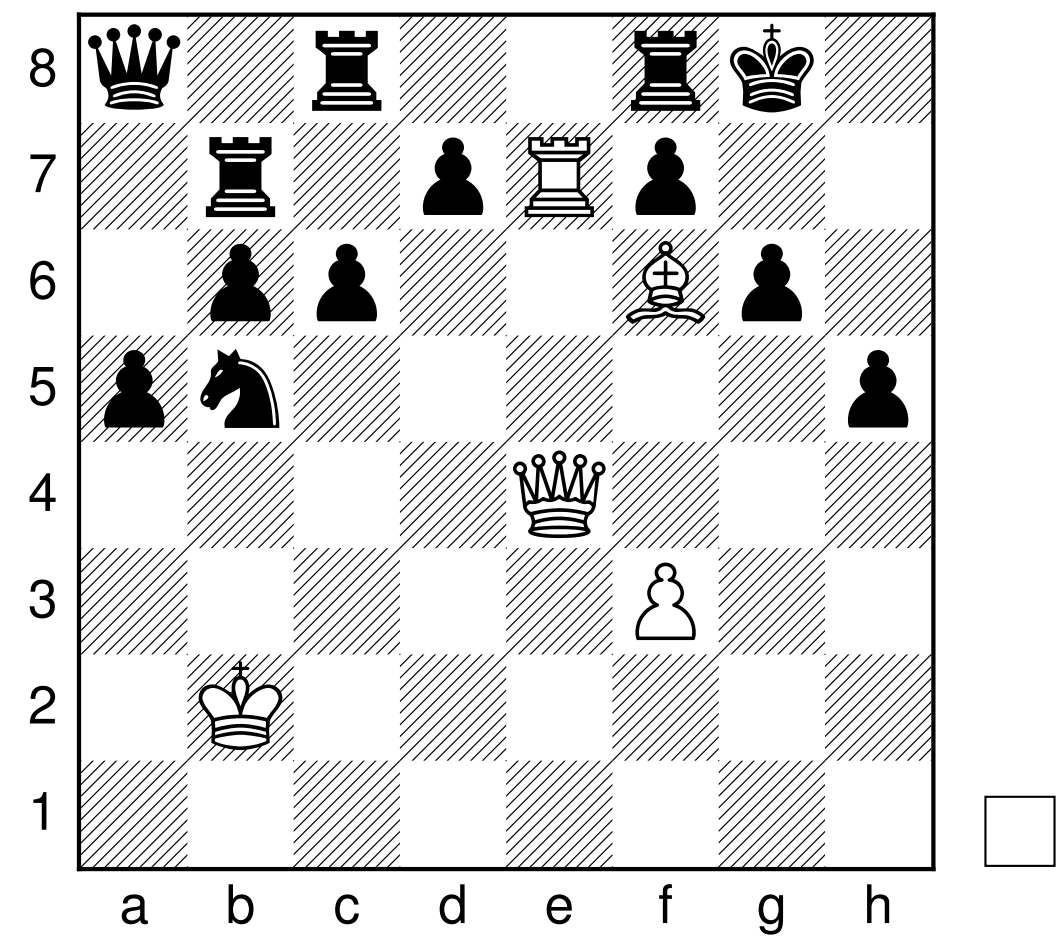

[Analyse on Lichess]

**Closest FENs - [1], [2], [3]**

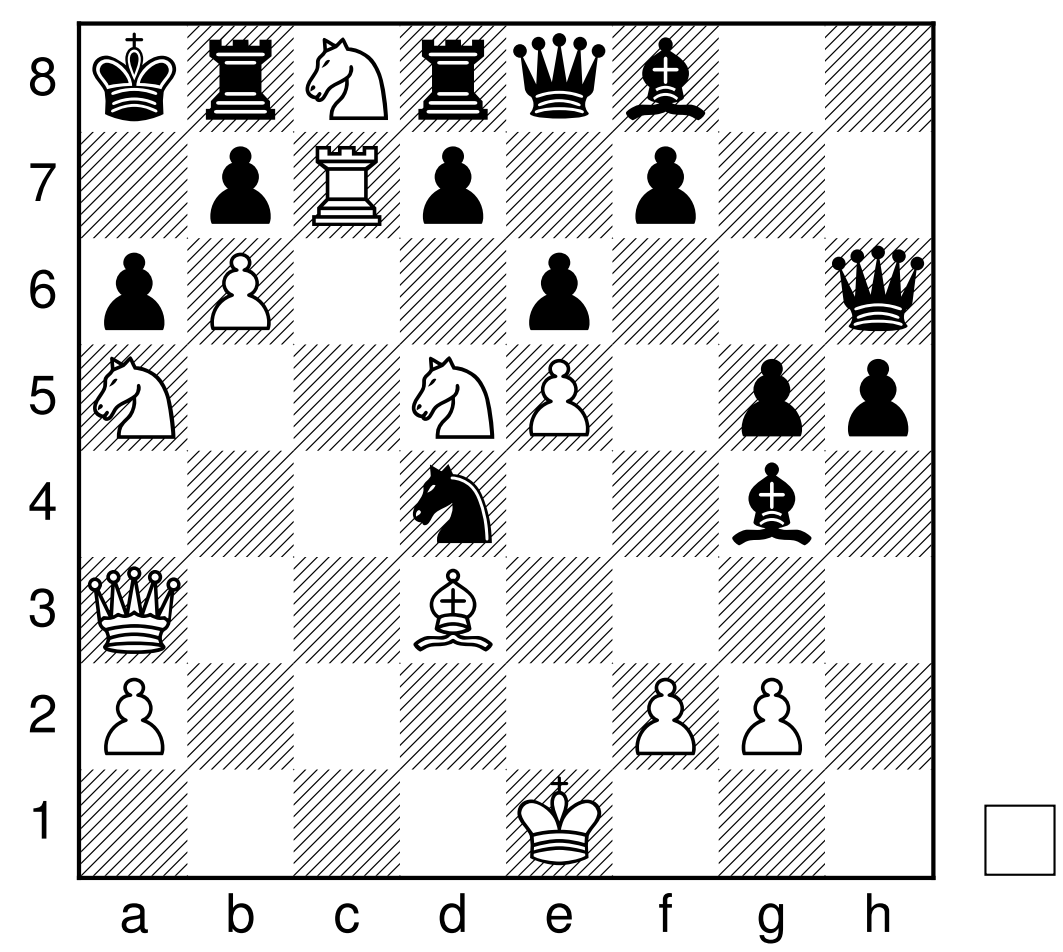

[Analyse on Lichess]

**Closest FENs - [1], [2], [3]**

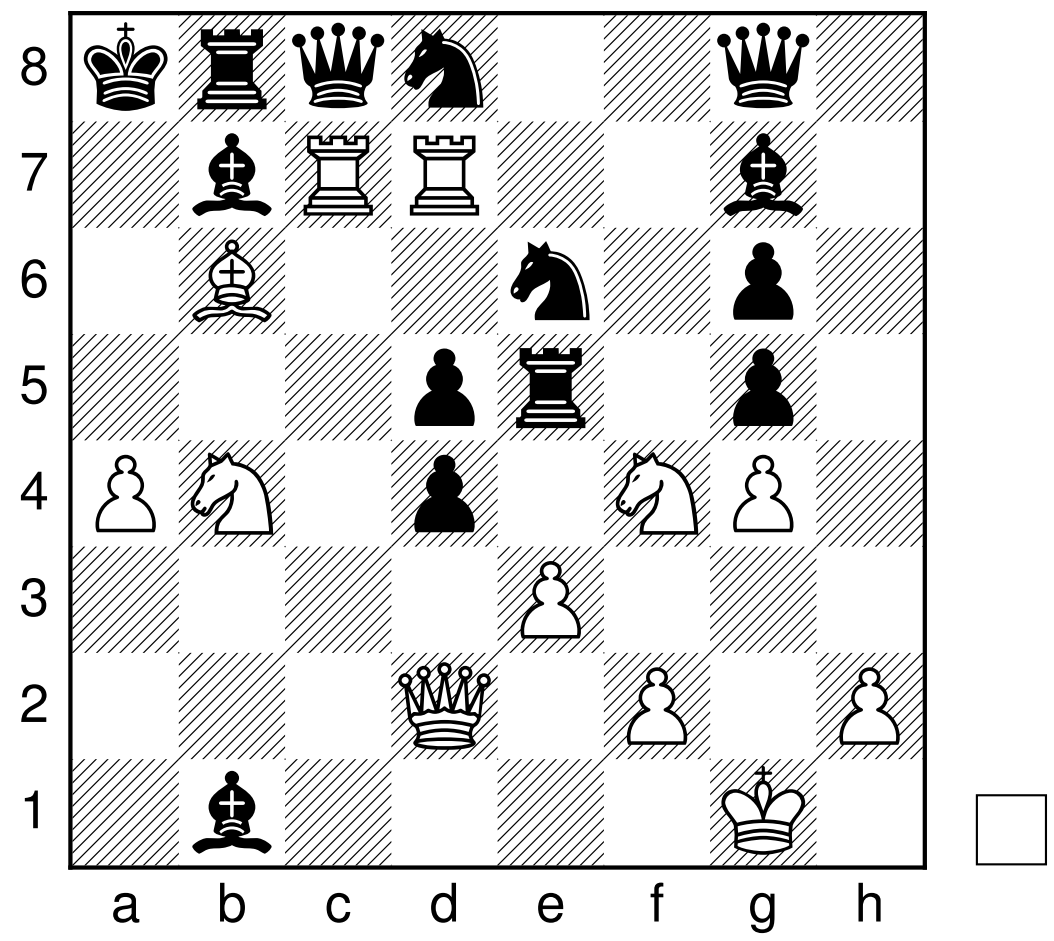

[Analyse on Lichess]

**Closest FENs - [1], [2], [3]**

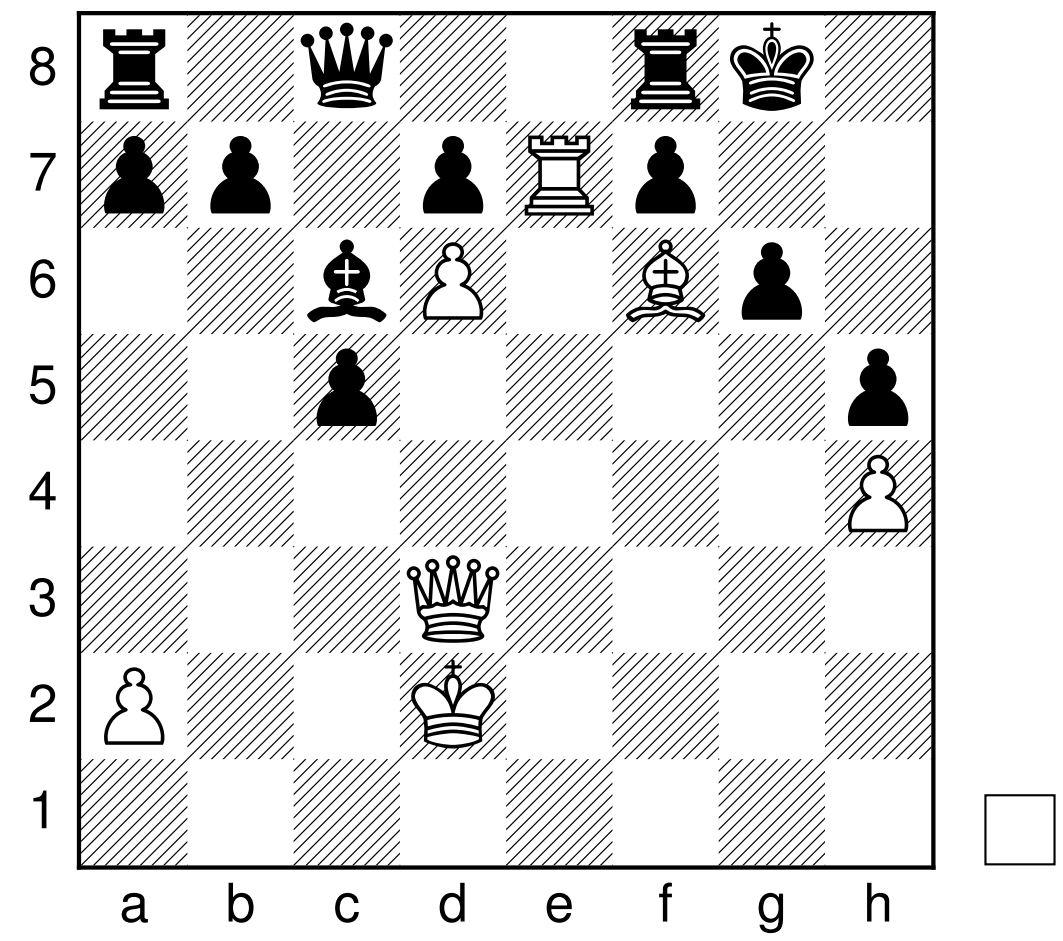

[Analyse on Lichess]

**Closest FENs - [1], [2], [3]**

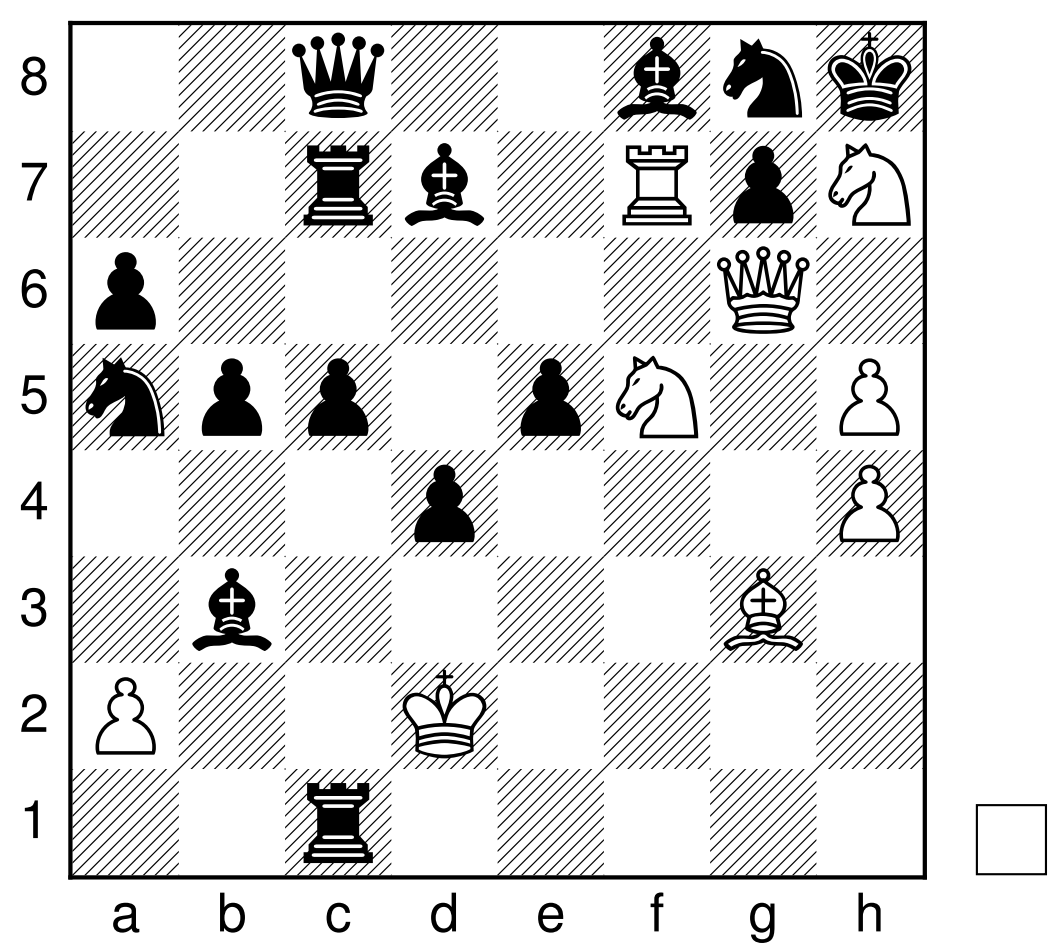

[Analyse on Lichess]

**Closest FENs - [1], [2], [3]**

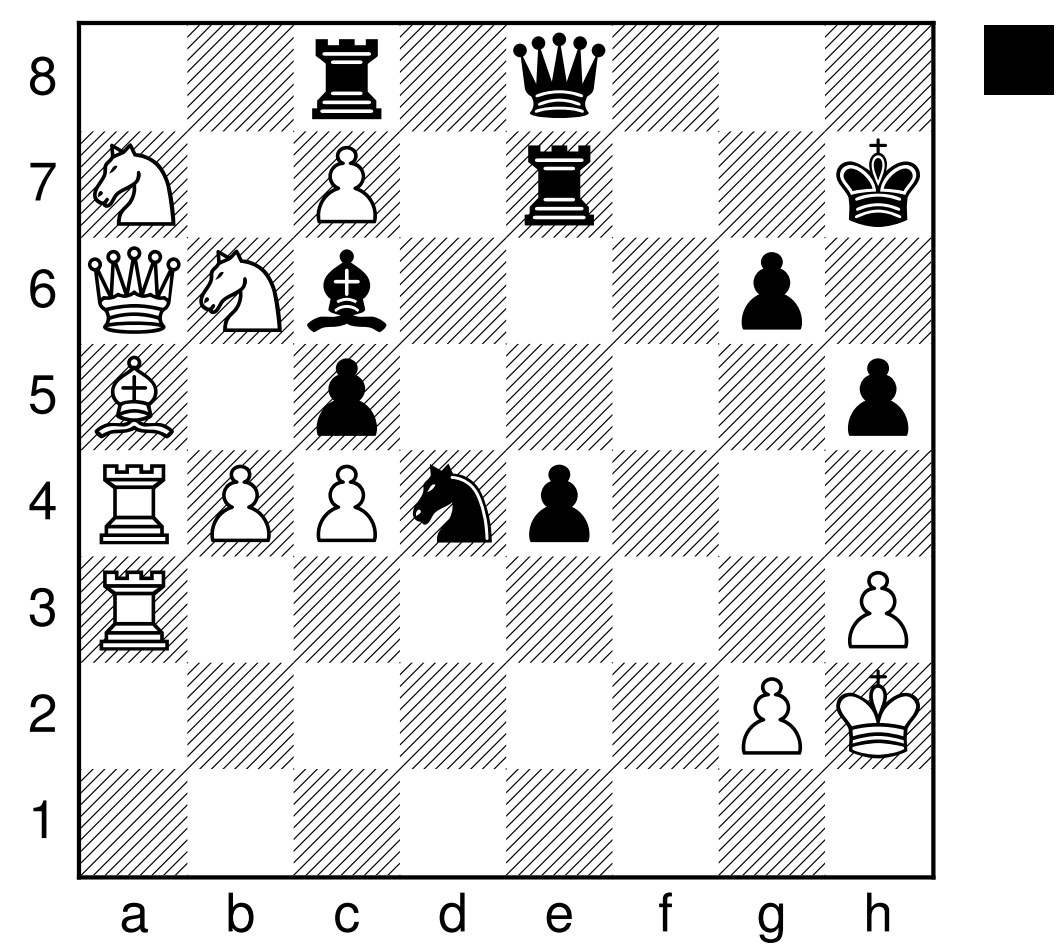

[Analyse on Lichess]

**Closest FENs - [1], [2], [3]**

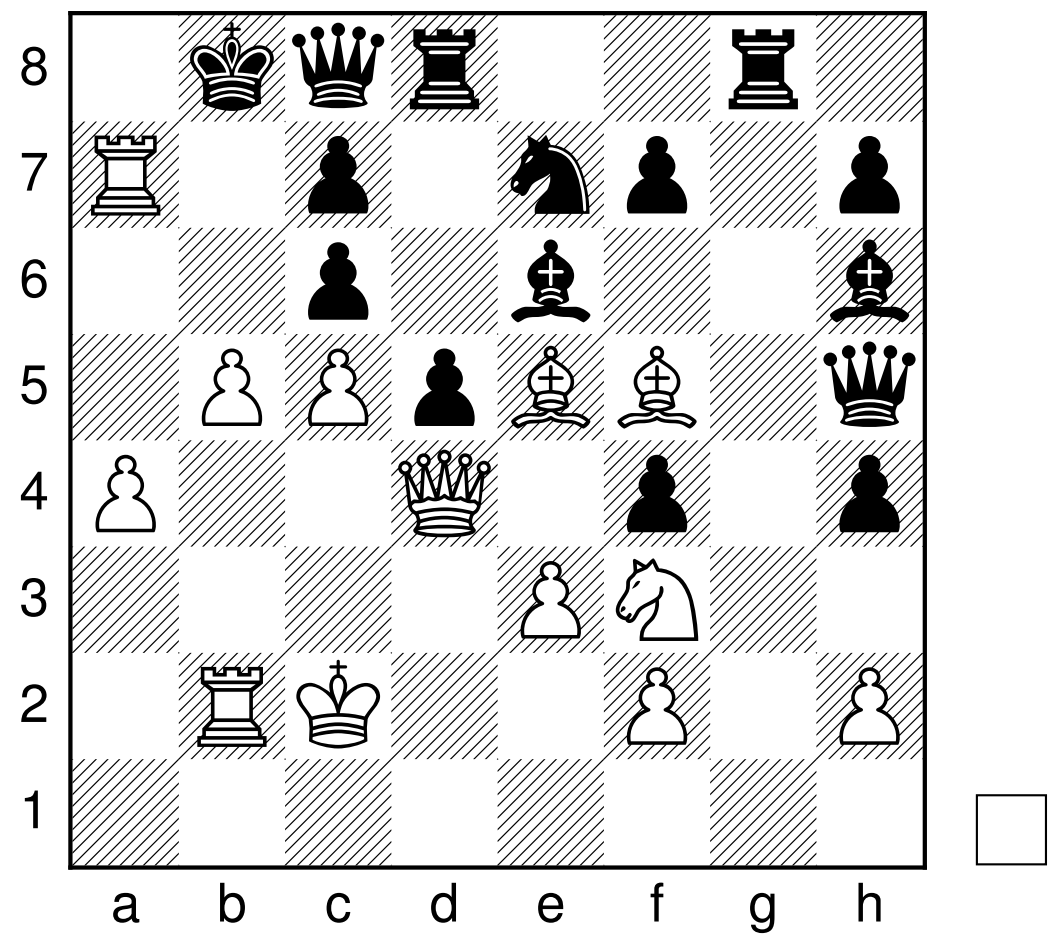

[Analyse on Lichess]

**Closest FENs - [1], [2], [3]**

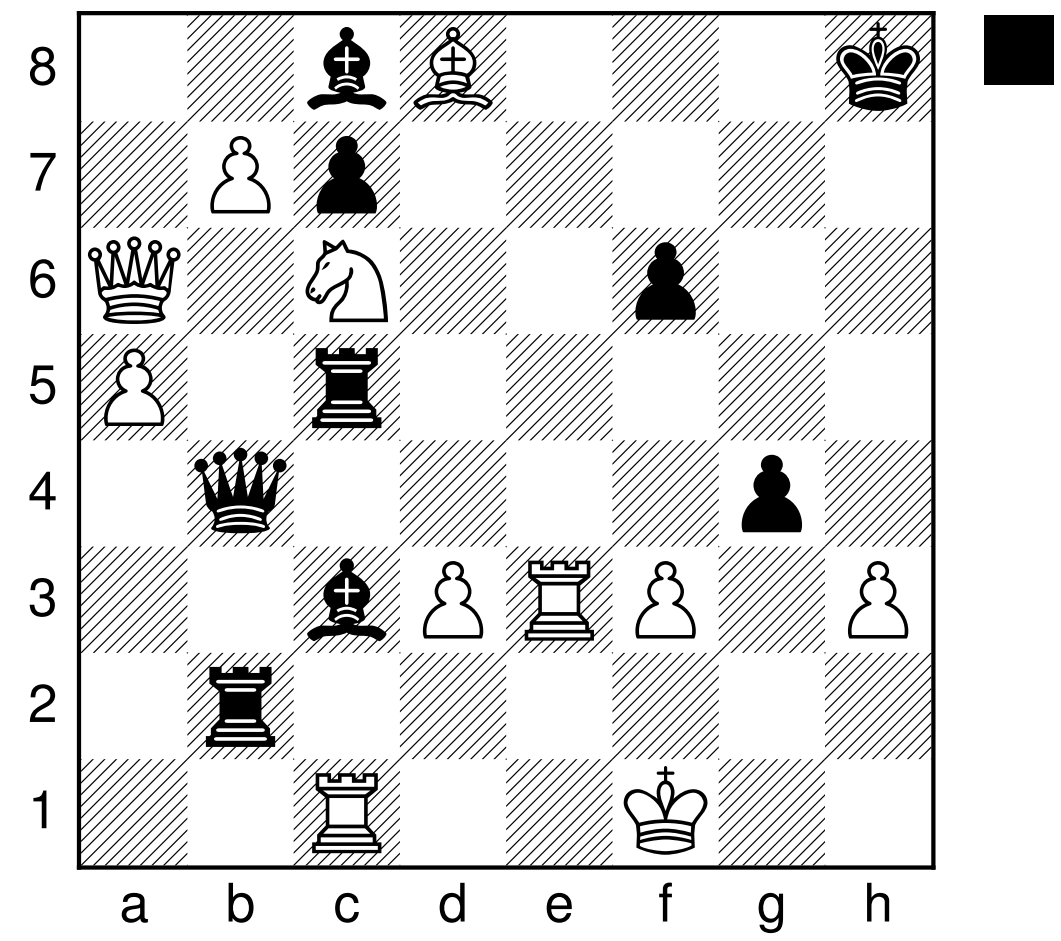

[Analyse on Lichess]

**Closest FENs - [1], [2], [3]**

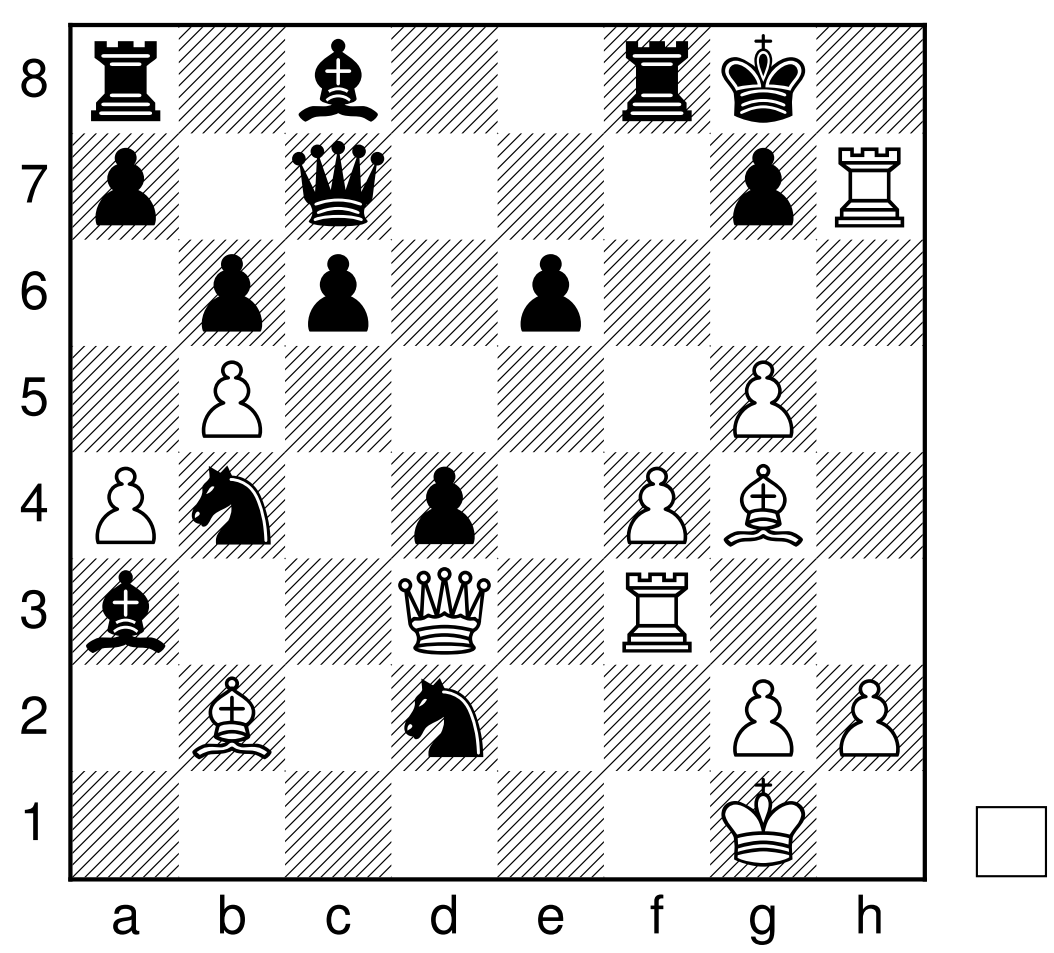

[Analyse on Lichess]

**Closest FENs - [1], [2], [3]**

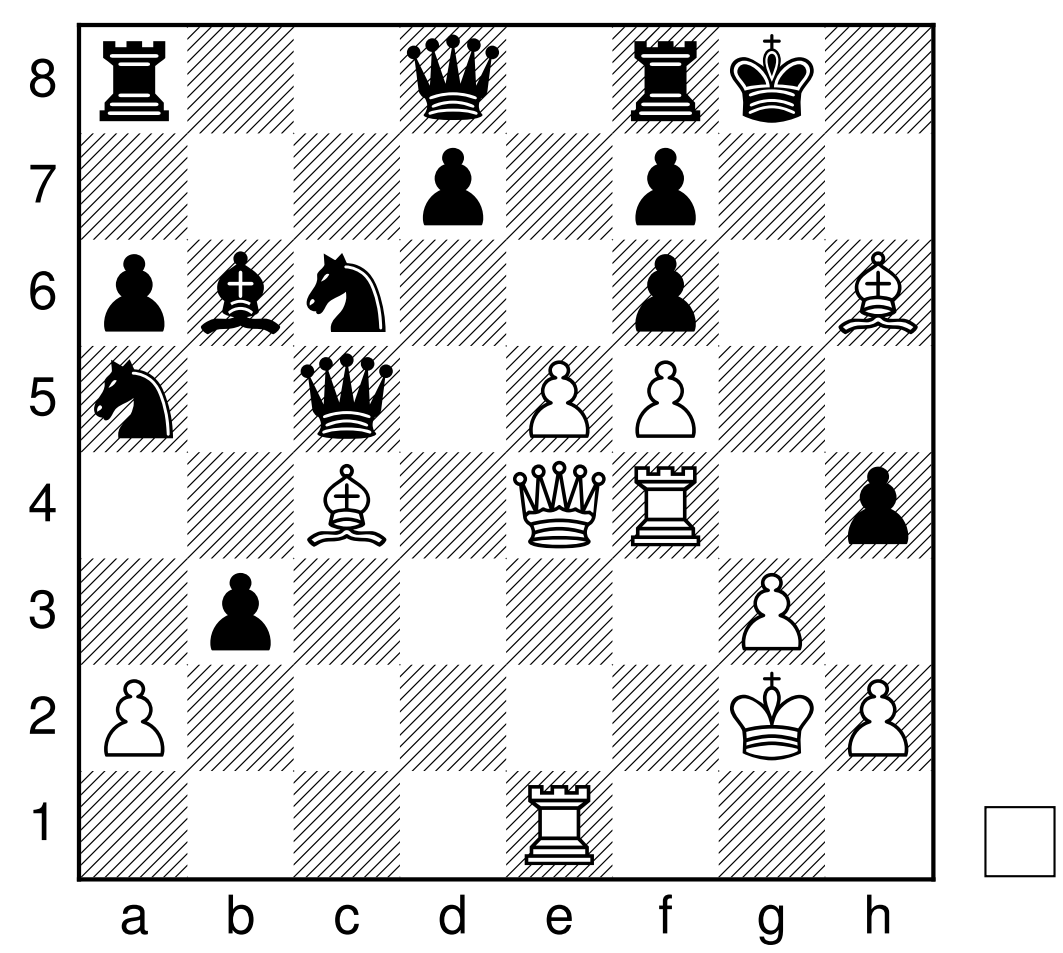

[Analyse on Lichess]

**Closest FENs - [1], [2], [3]**