# OpenReview forum: "Generating Creative Chess Puzzles"
_NeurIPS.cc/2025/Conference — NeurIPS 2025 poster_

### Official Review · Reviewer_EyYA · 2025-06-25

**Clarity:** 1
**Significance:** 2
**Originality:** 2
**Rating:** 4
**Confidence:** 4

**Summary:**

This paper introduces a methodology for the generation of creative chess puzzles, combining the definition of quality metrics and an RL pipeline. There are extensive method descriptions and I like the human evaluation study.

But overall I find this paper has several problems:
* It does not clearly specify what the contributions are and does not relate the approach to existing work (no related work section in the main paper)
* It uses a lot of jargon, and stays vague or imprecise on several key technical details.
* It is very applied to chess puzzle generation, and does not seem to introduce any technical novelty, except perhaps the definition of the counter-intuitiveness measure, which is again very domain-specific.

**Questions:**

see above

**Ethical Concerns:**

["NO or VERY MINOR ethics concerns only"]

**Final Justification:**

n/a (just edited for formatting for now)

**Limitations:**

see above

**Paper Formatting Concerns:**

Formatting:
* The figure and by equations 2–5 doesn’t have any caption or number.
* The use of three different colors for links doesn’t look very nice, and the extensive use of external web links instead of citations is not standard. Citations can be used to route towards websites where needed. Linking to an external website shouldn’t replace defining the concept in the paper itself, (eg FEN, Lichess)
* Some of these link trigger downloads of files — this is not OK please don’t do that e.g. “standard game dataset”
* Fig 3 could be made larger, it’s hard to read right now.

**Quality:**

2

**Strengths And Weaknesses:**

The paper proposes to solve chess puzzle but these are not really defined anywhere in the paper. Definitions themselves use a lot of technical jargon: eg ”excluding puzzles with multiple, solutions, non-standard elements, special stipulations, retrogrades, missing information, shortest mates, or mathematical aspects.” Do all chess puzzle have the same objective (check mate)?

Metrics definitions

The section dedicated to give definitions for the different metrics does not completely achieve its aim.
* Uniqueness: this definition is rather unclear, are we talking about the uniqueness of solutions or the uniqueness of moves? Is a unique solution a solution where each move is unique? And does the proposed criterion (Eq 1) really translates in qualifying uniqueness of a solution? How does the Stockfish engine work, and can we rely on it? If the second best move has a chance of leading to win >0, then it means there might be another solution right there?
* Creativity: this paragraph doesn’t offer any precise metric, just some discussion about the difficulty of evaluating creativity.
* Novelty:
  * What are FEN strings? Why is it a good choice? What is the principal variation measure?
  * about the entropy of the board representation: what is the model used? how is it trained?
  * I’m not sure the Levenshtein distances capture any kind of semantic distances between board positions — pieces could be differently placed but the abstract configurations may be similar, and inversely they could be placed very similarly but the abstract configurations might be very different
* Counter-intuitiveness: This is interesting, but here again a bit vague: this paragraph starts to define it, but stops before giving a final definition
* Aesthetic: here again a lot of discussion but no precise definition

Given how central these terms are, it would perhaps be good to have a concise, descriptive descriptions of these features right from the introduction. This is done to some extent but using very vague terms: eg “Counter-intuitive solutions are those that, at first glance, seem are terrible or out-of-mind, but are, in fact, brilliantly effective.” I think having a more technical, practical definition would help here. The abstract also appears to use vague terms and technical jargon.

Optimization
* Wouldn’t the use of a reverse KL divergence loss limit the creativity of the policy by forcing it to only generate puzzle within the distribution of the base model?
* I’m not sure I understand, is the policy pretrained with the lichens dataset with supervised finetuning, or are these 100k examples only added to the buffer at the beginning of the PPO phase? Does PPO even have a replay buffer? It’s an online policy, so not sure this would work so well to add completely offline data? This doesn’t seem to significantly help either (Fig 4), so not sure why it is introduced.
* Is there any reason why we should apply RL here instead of doing supervised fine-tuning on positive examples: diverse set of unique and counter-intuitive solutions?
* It would be interesting to see the ratio of the number of solutions accepted to the replay buffer as a function of time? If diversity crashes we’d have fewer and fewer novel-enough solutions to enter the replay buffer

Counter-intuitiveness measure:
* This seems like the main contribution of the paper, and it takes only a limited part of the paper and is not very clearly laid out.
* The fact that it captures well human-judgement on counter-intuitiveness is good but does not prevent the measure from being hacked by the policy during RL training. I would like to see an analysis of highly counter-intuitive models and some evidence that the measure is not being completely hacked by the policy in some strange way — eg by adding elements that would confuse the short-budget search (eg more pieces = higher branching factor = slower search) more than the high-budget search.

Novelty:
* “Inspecting Table 3 indicates that the datasets sampled from the generative models exhibit both novelty (when compared to the Lichess data) and internal diversity (when compared to other samples from the same model)” —> I find it hard to interpret these measures in an absolute sense, what’s the threshold that allows us to say these are definitely “novel” vs “not novel” here? Using the Lichess dataset as a baseline needs to be a bit justified, how do we know this dataset is diverse? If it’s a collection of boards people have played, maybe people don’t play such diverse games? The examples in the booklet help with that, thanks for including these.

Aesthetic:
* From Fig 3 it’s hard to say if anything special is going on. There is some aesthetic distribution in the training data, and some in the fine-tuned model and some in the RL-trained model. There is not obviously more in one or the other, it seems more or less random. Of course, it wasn’t optimized for. Is there a reason why not?

Conclusion:
* It's not clear what the contribution and novelty of this paper are compared to previous work
* The main contribution seems to be the definition of evaluation metrics / reward functions, which have a rather limited scope --- maybe this work belongs to a more specialized venue?
* The text is not always clear, it's dense and uses a lot of jargon, while missing key details. A lot is sent back to a very long appendix.

---

> ### Author Rebuttal · Authors · 2025-07-31
>
> We thank reviewer EyYA for your helpful feedback.
>
> ## **New human eval result with chess grandmasters (GM)**
> We refer the reviewer to other rebuttals for new evaluation results from GM, which further affirms the value of our work.
>
> ## **Rebuttal**
> **Q1: The section dedicated to give definitions for the different metrics does not completely achieve its aim.**
>
> We thank the reviewer for this feedback, which allows us to clarify our metric definitions. We will incorporate these clarifications into the main paper to enhance readability.
>
> - **Uniqueness**: We recursively apply the check to principle variation (line 78) – a solution is defined as unique if every move within its principal variation is the unique move. As formalized in Eq. 1, a move is considered "unique" only if it is substantially better than the second-best alternative. We provide Appendix C for a much detailed description of the check.
>
> - **StockFish**: StockFish is the leading chess engine in the world, widely accepted by experts as a superhuman "ground truth" for chess analysis.
>
> - **Creativity**: In general, when dealing with creativity there is not a single go to definition. Yet, there is a consensus in the chess community that we followed [2, 40] and used clear definitions throughout the paper. We choose the following three metrics as a more concrete representation (line 91, 115, 129, 130)
>
> - **Novelty**: FEN represents Forsyth-Edwards Notation (a commonly used notation to represent a board, line 141). And we indeed have a semantic measurement with model-based entropy (line 105-109). This is mainly used during the RL process using the AR model itself to filter low-diversity samples (line 197-199). In Appendix H.2 we demonstrate its necessity, showing how board-distance only is susceptible to reward hacking and why entropy-based filtering is essential (Figs 9 & 10).
>
> - **Aesthetics**: Our aesthetic metrics are based on established heuristics drawn from respected chess literature [2, 40]. Detailed definitions for each aesthetic theme, such as sacrifices, are provided in Appendix I.
>
> - **Counter-intuitiveness**: The intuitive explanation in our introduction is formalized by the definitions in Equations 2-5 (Line 122). These equations quantify counter-intuitiveness by comparing shallow vs. deep engine search evaluations. Crucially, this reward is then fine-tuned using our expert human preference data (Sec 3.1).
>
> **Q2: Optimization details.**
>
> - **KL Divergence and Creativity**: The reviewer raises an important point about the trade-off between creativity and realism. Generally, from our human evaluator and GMs’ feedbacks, players always favors creativity only if it is realistic. Thus, our KL penalty is a conscious and critical choice for realism (Lines 172-186). Without this constraint, the model is prone to generating 'out-of-distribution' (OOD) positions that, while potentially novel, are often unrealistic or nonsensical to human players. We detail more in Appendix H.3 and show OOD samples in Figure 8.
>
> - **PPO, Replay Buffer, and Offline Data**: We utilize a critic-free PPO variant with Monte-Carlo value estimate. We further tuned the pretrained model in the RL process (line 149), and show Zero-RL (RL with no pretraining) does not work (Figure 15). The replay buffer is mainly for enabling diversity filtering (Lines 181-183), a crucial mechanism to prevent entropy collapse. Regarding the 100k high-quality Lichess examples, their purpose is to anchor the model and mitigate KL divergence by seeding the training process with realistic puzzles (Figure 13(b)). There is a trade off here that it may bring in less puzzle score but more human-like positions. Indeed this data is off-policy, but you can treat it as auxiliary SFT co-training.
>
> - **RL vs. Supervised Fine-tuning on positive samples**:  we choose RL to further leverage negative samples. First, negative samples also contribute to policy learning [Zhu 2025]. Second, by treating non-novel puzzles as negative samples, we are explicitly preventing the model from entropy collapse or generating similar puzzles (Appendix H). SFT, which only learns from positive data, cannot directly address this.
>
> Zhu, Xinyu, et al. "The surprising effectiveness of negative reinforcement in LLM reasoning." arXiv preprint arXiv:2506.01347 (2025).
>
> - **Ratio of Accepted Solutions over Time**: We thank the reviewer for this excellent suggestion. We did, in fact, analyze this phenomenon in two experiments. Figure 11 demonstrates how entropy correlates with the sample pass ratio and the ratio keeps decreasing with more training steps. Figure 14.b demonstrates that without continuous training updates to the policy ('W/O training'), the number of novel samples accepted into the replay buffer rapidly diminishes. This confirms the reviewer's intuition and underscores the necessity of continuous adaptation to prevent the model from exhausting the novelty.
>
> **Q3: Counter-intuitiveness measure and reward hacking:**
>
> The potential for reward hacking is a critical concern in any RL framework, and we commend the reviewer for raising this point. We anticipated this issue and incorporated several mechanisms to mitigate it.
>
> - **Explicit Acknowledgment**: Our "Discussion and limitation" (Lines 368-383), explicitly acknowledges this tendency, including the example of "exploiting chess engine weaknesses (making the position artificially more complicated for engines...)." To counter this, we introduced realism constraints, including the KL divergence penalty and a piece count penalty (Appendix H.3, Figure 13), which prevents the model from adding excessive material to confuse the engine.
>
> - **Human-in-the-Loop Validation**: The final arbiter of quality is human evaluation. Our human expert study (Section 3.5, Table 4) confirms that the generated puzzles are not only rated as highly counter-intuitive but also as creative and fun, surpassing even human-composed book puzzles in these aspects. This provides strong evidence that our measure successfully captures human-perceived quality rather than being exploited in a meaningless way.
>
> **Q4: Clarification on Novelty.**
>
> We thank the reviewer for this insightful question, which allows us to further justify our choice of baseline and the interpretation of our novelty metrics. We agree that interpreting novelty metrics in an absolute sense is challenging. This is why our evaluation is twofold:
>
> - **Quantitative Interpretation and Baseline Justification**: Lichess puzzle dataset is the largest publicly available dataset of its kind, with its puzzle collection derived from an immense pool of over 600 million human games. Given this massive scale, the internal diversity of the Lichess dataset serves as a reasonable human standard of diversity. Quantitatively, a board distance value around 10 (at least 10 piece edits, Line 282) is considered novel.
>
> - **Qualitative Validation**: To move beyond abstract numbers and directly address the reviewer's point, our curated booklet presents each generated puzzle alongside its three closest Lichess matches. This allows for direct visual assessment of its distinctiveness. Expert feedback reinforces this; GM A noted one puzzle as "an elegant combination of themes... not something I've seen before."
>
> **Q5: Aesthetic clarification.**
>
> This was a deliberate choice. Our primary goal was to first establish a framework for generating puzzles that are sound (unique) and interesting (counter-intuitive). We deliberately chose not to hard-code human definitions of aesthetics, but instead to let the model discover on its own. The key finding (Fig 3) is that a diverse range of aesthetic themes emerged naturally from a process that did not explicitly optimize for them. This suggests our core metrics for counter-intuitiveness and realism are well-correlated with human preferred properties.
>
> **Q6: It's not clear what the contribution and novelty of this paper are compared to previous work**
> To clarify, our key contributions are:
>
> - A formal framework to quantify and optimize creative qualities in chess, refined with expert human data. (section 1, section 3,1).
> - Engineering design for puzzle generator (Section 2, section 3.2, 3.3)
> - Diversity-filtering RL techniques. We present key designs and the rationale of our RL framework, and identify and tackle the entropy collapse issue with diversity filtering, which is crucial for stable RL training. (Section 2.1, 3.4). We put the full exploration journey in Appendix H.
> - The final booklet and human study, which validates our methods and showcases the potential human-AI collaboration  – AI provides coarse generation and selection which largely decreases the human efforts for book-level samples.
>
> We refer you to our response to Q1 for Reviewer 2cAd to clarify novelty and generalizablity. Sorry that we can't include it due to space limits.
>
> Our work significantly extends prior art like Chesthetica [30-33], which focused solely on aesthetics over a decade ago, by tackling a multi-objective problem with a modern deep generative RL approach.
>
> **Q7: The main contribution seems to be the definition of evaluation metrics / reward functions, which have a rather limited scope --- maybe this work belongs to a more specialized venue...**
>
> We respectfully position our work as a Creative AI application, a contribution category explicitly welcomed by the NeurIPS call for papers. We argue that for a nuanced topic like AI creativity, a deep dive into a single, rich domain like chess is not a limitation but a methodological necessity. It allows for rigorous analysis and avoids the pitfalls of shallow evaluation seen in prior work. Chess, with its profound history and established aesthetic principles, provides the ideal testbed for drawing sharp, meaningful conclusions about the intersection of computation and creativity.
>
> **Q8: Formatting issues.**
>
> We will correct all formatting issues, including captions, external links, and figure readability.

---

> > ### Comment · Reviewer_EyYA · 2025-08-04
> >
> > Thank you for this detailed response.
> >
> > The response clarifies the majority of concerns I had about technical aspects and clarified a number of methodological questions. This said, the current version of the paper makes the whole approach hard to parse and understand as a whole and I think the presentation could be considerably improved by working on making the points raised in my review clearer and more precise to non-chess experts.
> >
> > I still believe that it does not make much technical contributions, at least not generally transferable ones: eg most metrics are chess-specific and the training pipeline is complex and does not present novel ideas. I acknowledge that it does fit the box for "creative AI applications" and is thus relevant to NeurIPS, although I expect interest will be limited primarily to the "chess x AI" community and will not extend much to other ML communities due to the number of chess-specific components in the method.
> >
> > To reflect these points I will raise my score to a 4.

---

### Official Review · Reviewer_MyRD · 2025-06-30

**Clarity:** 3
**Significance:** 2
**Originality:** 3
**Rating:** 4
**Confidence:** 4

**Summary:**

This paper trains generative models to synthesize chess puzzles. It first analyzes various kinds of generative models using dataset of chess puzzles and then fine-tunes the autoregressive generative model using RL for improved performance. The results show that the final RL-tuned model surpasses the existing puzzle dataset in various metrics (e.g. counter-intuitiveness). Another tangible outcome of the project is a manually curated booklet of 50 AI-generated puzzles.

**Questions:**

The results in Table 2 are rather surprising. One would expect a well-trained generative model to mirror the statistics of the training data, but the models here significantly underperform the original dataset in all metrics. Is there any reason why the generative models are so far off compared to their training data?

**Ethical Concerns:**

["NO or VERY MINOR ethics concerns only"]

**Final Justification:**

I'd like to thank the authors for their detailed response. While the concerns about contribution still somewhat persist, I agree that this is indeed a novel and creative application of AI and would be interesting for the broader community to see. I also think the additional GM study further highlights the value of the system, and would like to update my rating to lean towards acceptance.

**Limitations:**

Yes

**Quality:**

3

**Strengths And Weaknesses:**

Strengths:

- This project represents a useful, creative and unique application of generative AI tools. To my knowledge, while there have been numerous works which develop chess-playing systems, generative approaches for creating puzzles have not been explored.

- The paper takes a pretty sensible approach — training a generative model and finetuning with RL. The results demonstrate clear benefits from the second stage of RL-based training.

- The final booklet of puzzles could be a valuable outcome on its own as the quality is comparable to puzzles found in typical chess books.

Weaknesses:

- The technical contributions of this work are somewhat limited. Beyond the interesting choice of the setup tackled, the approach is to simply train typical generative models on an existing dataset and finetune the auto-regressive one using RL (mirroring standard practices in training LLMs e.g. for code generation).

- While the setup tackled in this work is unique, the final system that can generate puzzles is somewhat limited in scope. In particular, unlike chess engines which can start from custom positions, the puzzle generation system lacks the flexibility to help coach a player for specific scenarios. For example, this system can only be queried to generate a puzzle, but cannot be instructed to generate one of a specific kind “e.g. generate an opening position puzzle reflecting a defensive scenario”.

- The performance of the final system, while comparable to puzzles in books, is not ‘superhuman’, and in fact requires manual effort in curation etc (for example the booklet required significant manual effort). Moreover, the uniqueness of the puzzles from the RL-based system is limited. As such, it is unclear whether random samples from the learned model are of the desired quality.

---

> ### Author Rebuttal · Authors · 2025-07-31
>
> We thank the reviewer MyRD for your constructive comments and feedback.
>
> ## **New human eval result with chess grandmasters (GM)**
>
> Before the rebuttal, we’d like to add a new human evaluation result with chess GM on our booklet. The following excerpts are their comments, which further affirm the value of our work.
>
> - GM A: **I'm quite particular about what makes a good chess puzzle—I look for natural-looking positions with reasonable play. Yet, I found myself enjoying many of the AI-generated examples in the booklet.**
> - IM for Chess Compositions B: **This booklet adds novel, AI-generated puzzles to the existing chess literature. I found several puzzles that were original, creative, and aesthetically pleasing, with surprising solutions that were satisfying to solve.**
> - GM C: **The puzzles in this booklet represent a pioneering step in human-AI collaboration for chess composition and hint at what might be achieved in the future for composing chess puzzles.**
>
> ## **Rebuttal**
>
> **Q1: The technical contributions of this work are somewhat limited... simply train typical generative models on an existing dataset and finetune the auto-regressive one using RL (mirroring standard practices in training LLMs e.g. for code generation).**
>
> We respectfully position our work as a Creative AI application, a contribution category (https://neurips.cc/Conferences/2025/CallForPapers) explicitly welcomed by the NeurIPS call for papers. We argue that for a topic as sensitive and nuanced as AI-driven creativity, a deep dive into a single, rich domain is not a limitation but a methodological necessity for drawing sharp, meaningful conclusions. As we cite in our introduction, the field has seen instances where shallow evaluations of creative systems yielded misleading results, a pitfall we were determined to avoid.
>
> Chess composition represents an ideal domain for such an in-depth study. It is a field with a profound history, recognized world titles, and established aesthetic principles, creating a perfect testbed for exploring the intersection of computation and creativity. Our focused investigation allowed for a rigorous analysis, culminating in the curated booklet of compositions we provided for expert review.
>
> Furthermore, our work makes a crucial contribution to the field demonstrating the ability of AI techniques to compose questions, and not only to solve them. It is often the case that even though solutions to specific problems appear more domain specific, they are important stepping stones to generalize later.
>
> We agree that building on prior work is standard in an application paper. However, we wish to emphasize the novel adaptations and contributions required to make these methods successful for our task, and we believe that the NeurIPS community can gain from reading our lessons. Specifically, our system is general and most of our contributions can be extended to other domains:
>
> - **Novel Counterintuitiveness Reward and generalizability:** Our primary technical contribution is a novel reward function that models counter-intuitiveness by analyzing the search statistics of a planning algorithm, effectively quantifying 'surprise.' This technique is directly transferable to other domains where search is applied, such as games (AlphaZero), mathematics (AlphaProof), and computer science. More broadly, we propose this concept generalizes beyond explicit search. By drawing an analogy between 'deep search' and the 'deep thinking' demonstrated by recent reasoning-focused models (e.g., O1, R1), our approach offers a pathway to measure the novelty or counter-intuitiveness of solutions in any domain that advanced language models can address.
>
> - **Diffusion Model & Auto-regressive model.** While the model architectures that we’ve trained are otherwise standard, the engineering setup enables this to be applied to discrete generative problems more broadly. In addition, our work provides compelling evidence that for this creative task, masked diffusion models can be superior to their AR counterparts, contributing a valuable data point to the field.
>
> - **Adapting RLHF for Novelty:** A naive implementation of RLHF-PPO proved insufficient, suffering from entropy collapse and failing to produce novel outputs (Section 2.1, with more details given in Appendix H). It is fairly new to point out this issue and only in very recent work [Wang 2025], [Cui 2025], researchers began to investigate the role of entropy and the potential for its collapse during the RL training of LLMs, as well as methods to mitigate it. In addition, our RL ablations detail the non-trivial modifications (specifically, measure novelty using various metrics and to either penalize or skip training on samples that are not novel) required to overcome this, offering generalizable lessons for any researcher seeking to optimize for novelty in conjunction with a reward signal. It is only in the latest LLM reasoning literature that techniques based on conceptually similar principles have started to be discussed. For instance, [Yu 2025] is using curriculum learning to actively filter out problems on which the model has already converged.
>
> Wang, Zihan, et al. "Ragen: Understanding self-evolution in llm agents via multi-turn reinforcement learning." preprint arXiv:2504.20073 (2025).
>
> Cui, Ganqu, et al. "The entropy mechanism of reinforcement learning for reasoning language models." preprint arXiv:2505.22617 (2025).
>
> Yu, Qiying, et al. "Dapo: An open-source llm reinforcement learning system at scale." preprint arXiv:2503.14476 (2025).
>
>
> **Q2: While the setup tackled in this work is unique, the final system that can generate puzzles is somewhat limited in scope. In particular, unlike chess engines which can start from custom positions, the puzzle generation system lacks the flexibility to help coach a player for specific scenarios......”.**
>
> Thank you for your feedback on potential future directions, such as conditional generation. It is greatly appreciated and aligns with our own plans for future work.
>
> While our current system doesn't yet have a dedicated “coaching mode”, it can produce counter-intuitive chess puzzles across a range of aesthetic themes. These themes can be used as further filtering mechanisms to identify puzzles specifically relevant for coaching a player. For example, we could filter for positions from a specific opening or with a particular aesthetic theme to create a customized puzzle set.
>
> **Q3: The performance of the final system, while comparable to puzzles in books, is not ‘superhuman’, and in fact requires manual effort in curation etc (for example the booklet required significant manual effort). Moreover, the uniqueness of the puzzles from the RL-based system is limited. As such, it is unclear of the desired quality.**
>
> - **Curation.** We need to clarify that humans also create puzzle books by curating positions. And our system has already saved a lot of manual efforts (we only have one chess expert curating the candidates).
>
> - **Quality.** Note that the uniqueness of our RL system is not low (the reason Lichess puzzle has such a high uniqueness ratio in Fig 4.c is because it is getting explicitly filtered for uniqueness in advance while our RL is not, mentioned in line 316). For the final quality, we would like to point out the feedback from the three chess experts who extensively reviewed our chess booklet. They agreed that some of the puzzles were creative, aesthetically pleasing, beautiful and counter-intuitive.
>
> - **Not superhuman.** As we mentioned in the final discussion section, unlike verifiable domains such as chess playing, human definition of counter-intuitiveness is subjective. Even for the same position, players may have varying opinions, even if they are of comparable playing strength, including grandmasters. Our estimate through chess engines and the amount of human data in our evaluation is therefore certainly approximate, and a proxy for a more general preference distribution. Such evaluations are common for example in the RLHF domain, where there would usually not be any claims of superhuman.
>
> **Q4: The results in Table 2 are rather surprising. One would expect a well-trained generative model to mirror the statistics of the training data, but the models here significantly underperform the original dataset in all metrics.....?**
>
> This is an excellent observation. The gap you've noted is not due to a poorly trained model, but rather a direct consequence of our model selection strategy, which deliberately prioritizes generalization over memorization.
>
> Here's the breakdown:
> - **Our Strategy Avoids Overfitting:** We strictly selected the model checkpoint with the lowest test loss. This standard practice ensures the model learns generalizable patterns from the data, rather than simply memorizing the training set.
> - **Overfitting Would Mirror the Training Data:** Crucially, we observed that if we continued training beyond this optimal point, the model does begin to overfit. As shown in Appendix Figure 7, this overfitting causes the puzzle score to rise. However, this comes at a steep price: the model's test loss increases, and its generation diversity significantly decreases. This also correlates with recent work on LLM RL research [Wu 2025].
> - **The Brittle Nature of Puzzles:** This effect is amplified because puzzles are "brittle"—a single incorrect piece can invalidate the entire puzzle. A model that generalizes will inevitably produce small, nuanced variations that may not pass the strict criteria of a puzzle.
>
> In essence, we made a conscious trade-off. We could have achieved higher puzzle scores by overfitting, but we chose to present a model that generalizes better and is more diverse, which we believe is a more meaningful benchmark.
>
> Wu, Mingqi, et al. "Reasoning or Memorization? Unreliable Results of Reinforcement Learning Due to Data Contamination." preprint arXiv:2507.10532 (2025).

---

> ### Author Response · Authors · 2025-08-05
>
> We thank you again for your constructive comments and detailed review, which have helped us improve the paper.
>
> As the discussion period is concluding in a few days, we wanted to briefly follow up and check if our rebuttal has sufficiently addressed your concerns. In summary, our main points were:
>
>   - **On Technical Contribution:** We positioned our work as a Creative AI application, highlighting our key technical contributions: (1) A novel, generalizable reward function for **counter-intuitiveness**; (2) Engineering design and rigorous comparison between masked diffusion models and AR models,  contributing a valuable data point to the model design field (e.g., recent trends on Diffusion Model vs AR); and (3) Non-trivial adaptations of RLHF to successfully **optimize for novelty**, a new challenge just gets discussed in very recent literature.
>
>   - **On Quality & "Superhuman" Performance:** Regarding the 'superhuman' bar, we clarified that this is difficult to address because, unlike game-playing, puzzle aesthetics and counter-intuitiveness are **inherently subjective**. In addition, the positive evaluations from Chess Grandmasters provide crucial evidence of our system's success. The fact that these experts, with their deep knowledge and critical standards, perceived our puzzles as novel and creative is a high bar for an AI system to meet.  This result is a key indicator of creative success, even if the puzzles are not labeled 'superhuman'. Finally, our goal is high-quality, human-AI collaborative composition, where manual curation by an expert is a standard and valuable part of the creative process, just as it is for human-authored puzzle books.
>
>   - **On Generative Model Performance (Table 2):** We explained that the performance gap you noted is a deliberate result of our model selection strategy, which **prioritizes generalization and diversity over overfitting** and simply memorizing the training set.
>
> We would be very grateful to know if our explanations and the new experiments have helped address your points. We are, of course, happy to answer any further questions you might have before the deadline.

---

> > ### Comment · Reviewer_MyRD · 2025-08-05
> >
> > I'd like to thank the authors for their detailed response. While the concerns about contribution still somewhat persist, I agree that this is indeed a novel and creative application of AI and would be interesting for the broader community to see. I also think the additional GM study further highlights the value of the system, and would like to update my rating to lean towards acceptance.

---

### Official Review · Reviewer_2cAd · 2025-07-01

**Clarity:** 4
**Significance:** 3
**Originality:** 3
**Rating:** 5
**Confidence:** 3

**Summary:**

The paper studies the problem of generating creative, aesthetic, and novel chess puzzles using generative models. The paper first defines numerical metrics for measuring these unique aspects of chess puzzles, then benchmarks multiple generative schemes, trained on a collection of chess game datasets. The author then adapts the autoregressive transformer with RLHF-PPO based fine-tuning using reward functions designed to maximize realism, novelty, and counter-intuitiveness in the output. From multiple numerical experiments, ablation studies, and human evaluation, the author showcases the effectiveness of GenAI and RL techniques for chess puzzle generation and provides a booklet of generated puzzles as an end product.

**Questions:**

- Graph on line 122 need minor fixing
- 3.3 and 3.4 might need to be introduced in chronological order in section 3 overview paragraph

**Ethical Concerns:**

["NO or VERY MINOR ethics concerns only"]

**Final Justification:**

The reviewer initial found this work to be good contribution and quite creative in the application aspect. The quality of the work is also really high, therefore will remain the recommendation of acceptance. However, given the work is application focused and optimized forward a really specific usage of chess puzzle generation, the reviewer does not believe this work qualifies for higher score than 5.

**Limitations:**

- Despite successfully generating puzzles that are evaluated favorably in both numerical experiment and human evaluation, the RL model still requires significant design effort to ensure good convergence.

**Paper Formatting Concerns:**

No concern raised

**Quality:**

4

**Strengths And Weaknesses:**

### Strength

- Well-written paper with tight structure and clear logic flow
- clear quantification of various measurement with articulated intuitions (such as quantification of puzzle uniqueness, novelty and counter intuitiveness) and solves unique challenge for chess puzzle generation
- meticulously designed reward with experiment demonstrate appropriate of design choices
- demonstration of interesting property such as emergent of aesthetics from AR and RL backbones

### Weakness

- the work mainly used well established generative technique (AR, diffusion, RLHF-PPO), therefore the contribution largely lies in engineering choices specifically for chess puzzle generation.

---

> ### Author Rebuttal · Authors · 2025-07-31
>
> We thank reviewer 2cAd for your constructive comments and feedback.
>
> ## **New human eval result with chess grandmasters (GM)**
>
> Before the rebuttal, we’d like to add a new human evaluation result with chess GM on our booklet. The following excerpts are their comments, which further affirm the value of our work.
>
> - GM A: **I'm quite particular about what makes a good chess puzzle—I look for natural-looking positions with reasonable play. Yet, I found myself enjoying many of the AI-generated examples in the booklet.**
> - IM for Chess Compositions B: **This booklet adds novel, AI-generated puzzles to the existing chess literature. I found several puzzles that were original, creative, and aesthetically pleasing, with surprising solutions that were satisfying to solve.**
> - GM C: **The puzzles in this booklet represent a pioneering step in human-AI collaboration for chess composition and hint at what might be achieved in the future for composing chess puzzles.**
>
> ## **Rebuttal**
>
> **Q1: The work mainly used well established generative techniques (AR, diffusion, RLHF-PPO), therefore the contribution largely lies in engineering choices specifically for chess puzzle generation.**
>
> We respectfully position our work as a Creative AI application, a contribution category (https://neurips.cc/Conferences/2025/CallForPapers) explicitly welcomed by the NeurIPS call for papers. We argue that for a topic as sensitive and nuanced as AI-driven creativity, a deep dive into a single, rich domain is not a limitation but a methodological necessity for drawing sharp, meaningful conclusions. As we cite in our introduction, the field has seen instances where shallow evaluations of creative systems yielded misleading results, a pitfall we were determined to avoid.
>
> Chess composition represents an ideal domain for such an in-depth study. It is a field with a profound history, recognized world titles, and established aesthetic principles, creating a perfect testbed for exploring the intersection of computation and creativity. Our focused investigation allowed for a rigorous analysis, culminating in the curated booklet of compositions we provided for expert (GrandMaster) review.
>
> We agree that building on prior work is standard in an application paper. However, we wish to emphasize the novel adaptations and contributions required to make these methods successful for our task, and we believe that the NeurIPS community can gain from reading our lessons. Specifically, our system is general and most of our contributions can be extended to other domains:
>
> - **Novel Counterintuitiveness Reward and generalizability**: Our primary technical contribution is a novel reward function that models counter-intuitiveness by analyzing the search statistics of an MCTS planning algorithm, effectively quantifying 'surprise.' This technique is directly transferable to other domains where MCTS is applied, such as games (AlphaZero), mathematics (AlphaProof), and computer science. More broadly, we propose this concept generalizes beyond explicit search. By drawing an analogy between 'deep search' and the 'deep thinking' demonstrated by recent reasoning-focused models (e.g., O1, R1), our approach offers a pathway to measure the novelty or counter-intuitiveness of solutions in any domain that advanced language models can address.
>
> - **Diffusion Model & Auto-regressive model.** While the model architectures that we’ve trained are otherwise standard, the engineering setup enables this to be applied to discrete generative problems more broadly, whether it is games or language. In addition, our work provides compelling evidence that for this creative task, masked diffusion models can be superior to their autoregressive counterparts, contributing a valuable data point to the field.
>
> - **Adapting RLHF for Novelty:** A naive implementation of RLHF-PPO proved insufficient, suffering from entropy collapse and failing to produce novel outputs (Section 2.1, with more details given in Appendix H). It is fairly new to point out this issue and only in very recent work [Wang 2025], [Cui 2025], researchers began to investigate the role of entropy and the potential for its collapse during the RL training of LLMs, as well as methods to mitigate it. In addition, our RL ablations detail the non-trivial modifications (specifically, measure novelty using various metrics and to either penalize or skip training on samples that are not novel) required to overcome this, offering generalizable lessons for any researcher seeking to optimize for novelty in conjunction with a reward signal. It is only in the latest LLM reasoning literature that techniques based on conceptually similar principles have started to be discussed. For instance, [Yu 2025] is using curriculum learning to actively filter out problems on which the model has already converged.
>
> Wang, Zihan, et al. "Ragen: Understanding self-evolution in llm agents via multi-turn reinforcement learning." arXiv preprint arXiv:2504.20073 (2025).
>
> Cui, Ganqu, et al. "The entropy mechanism of reinforcement learning for reasoning language models." arXiv preprint arXiv:2505.22617 (2025).
>
> Yu, Qiying, et al. "Dapo: An open-source llm reinforcement learning system at scale." arXiv preprint arXiv:2503.14476 (2025).
>
>
> **Q2: Graph on line 122 need minor fixing. 3.3 and 3.4 might need to be introduced in chronological order in section 3 overview paragraph**
>
> Thank you for your advice. We will update the graph and reorganize the order of section 3.3 and 3.4.
>
> **Q3: Despite successfully generating puzzles that are evaluated favorably in both numerical experiment and human evaluation, the RL model still requires significant design effort to ensure good convergence.**
>
> We argue that such design effort, instead, highlights a core contribution of our work. The challenge of stabilizing RL in a complex, sparse-reward generative domain while keeping generation diversity is precisely the problem we aimed to solve. Applying standard RL is notoriously difficult here. Therefore, our primary contribution is not just the final model, but the robust set of training techniques we developed to make it possible.
>
> As detailed in our response to Q1 (point 3), the value of this contribution is underscored by the fact that the challenges we overcame—such as entropy collapse—are now emerging as critical research problems in the RL training of Large Language Models. Our work provides a proven framework and generalizable lessons for tackling these very issues.

---

> > ### Comment · Reviewer_2cAd · 2025-08-06
> >
> > The reviewer appreciate the response and the questions and concerns have been addressed. The reviewer see this work as good contribution and will remain the current rating and recommendation.

---

### Official Review · Reviewer_MudV · 2025-07-02

**Clarity:** 3
**Significance:** 3
**Originality:** 4
**Rating:** 5
**Confidence:** 3

**Summary:**

This paper proposes a method for generating creative chess puzzles using a transformer model enhanced with reinforcement learning. By defining and quantifying creativity through uniqueness, novelty, counter-intuitiveness, and aesthetics, the authors design reward functions based on chess engine search statistics. Their RL approach significantly increases the generation of counter-intuitive puzzles while preserving aesthetic qualities. Human evaluations show that the generated puzzles rival or surpass human-composed ones in creativity and engagement.

**Questions:**

Q1) Typo in Section 2 "(latent diffusion [60, LDMs] )"?

Q2) In Table 1, the reported precision scores (e.g., 0.747504, 0.731198, 0.740115) have 6 decimal places, which is unnecessarily precise. Consider using only 4 dp?

Q3) A discussion on potential transfer to other creative domains or a proof of concept in a second domain would strengthen the paper's broader impact?

Q4) The Golden Set used to tune the counter-intuitiveness reward contains fewer than 100 samples. Would it  limit the robustness and generalizability of the learned scoring function? Will expanding this dataset or incorporating crowd-sourced annotations could improve the reliability of the reward design?

**Ethical Concerns:**

["NO or VERY MINOR ethics concerns only"]

**Final Justification:**

I think overall this work has a good originality, and W1 rebuttal is convincing. I also agree in Q4 that high-quality data collection for this task is uniquely challenging, its a good strategy to prioritize the data quality than quantity. I will maintain my score.

**Limitations:**

yes

**Paper Formatting Concerns:**

/

**Quality:**

3

**Strengths And Weaknesses:**

S1) This paper demonstrates high originality, introducing novel problem formulations and a metric that quantifies creativity in chess puzzles. It also has a good motivation as AI's role in creative domain is an important area.

W1) While the approach is compelling, the study is heavily domain specific, relying on chess engines, Lichess data, and hand crafted puzzle heuristics, which raises questions about the generalizability of the framework.

---

> ### Author Rebuttal · Authors · 2025-07-30
>
> We thank reviewer MudV for your constructive comments and feedback.
>
> ## **New human eval result with chess grandmasters (GM)**
>
> Before the rebuttal, we’d like to add a new human evaluation result with chess GM on our booklet. The following excerpts are their comments, which further affirm the value of our work.
>
> - GM A: **I'm quite particular about what makes a good chess puzzle—I look for natural-looking positions with reasonable play. Yet, I found myself enjoying many of the AI-generated examples in the booklet.**
> - IM for Chess Compositions B: **This booklet adds novel, AI-generated puzzles to the existing chess literature. I found several puzzles that were original, creative, and aesthetically pleasing, with surprising solutions that were satisfying to solve.**
> - GM C: **The puzzles in this booklet represent a pioneering step in human-AI collaboration for chess composition and hint at what might be achieved in the future for composing chess puzzles.**
>
> ## **Rebuttal**
>
> **W1: While the approach is compelling, the study is heavily domain specific, ..., which raises questions about the generalizability of the framework.**
>
> Thanks for your feedback. We respectfully position our work as a Creative AI application, a contribution category (https://neurips.cc/Conferences/2025/CallForPapers) explicitly welcomed by the NeurIPS call for papers. We argue that for a topic as sensitive and nuanced as AI-driven creativity, a deep dive into a single, rich domain is not a limitation but a methodological necessity for drawing sharp, meaningful conclusions. As we cite in our introduction, the field has seen instances where shallow evaluations of creative systems yielded misleading results, a pitfall we were determined to avoid.
>
> Chess composition represents an ideal domain for such an in-depth study. It is a field with a profound history, recognized world titles, and established aesthetic principles, creating a perfect testbed for exploring the intersection of computation and creativity. Our focused investigation allowed for a rigorous analysis, culminating in the curated booklet of compositions we provided for expert (GrandMaster) review. Furthermore, our work makes a crucial contribution to the field demonstrating the ability of AI techniques to compose questions, and not only to solve them. It is often the case that even though solutions to specific problems appear more domain specific, they are important stepping stones to generalize later.
>
> We also wish to emphasize the novel adaptations and contributions required to make these methods successful for our task, and we believe that the NeurIPS community can gain from reading our lessons. Specifically, our system is general and most of our contributions can be extended to other domains:
>
> - **Novel Counterintuitiveness Reward and generalizability**: Our primary technical contribution is a novel reward function that models counter-intuitiveness by analyzing the search statistics of an MCTS planning algorithm, effectively quantifying 'surprise.' This technique is directly transferable to other domains where MCTS is applied, such as games, mathematics, and computer science. More broadly, we propose this concept generalizes beyond explicit search. By drawing an analogy between 'deep search' and the 'deep thinking' demonstrated by recent reasoning-focused models (e.g., O1, R1), our approach offers a pathway to measure the novelty or counter-intuitiveness of solutions in any domain that advanced language models can address.
>
> - **Diffusion Model & Auto-regressive model.** While the model architectures that we’ve trained are otherwise standard, the engineering setup enables this to be applied to discrete generative problems more broadly, whether it is games or language.
>
> - **Adapting RLHF for Novelty**: A naive implementation of RLHF-PPO proved insufficient, suffering from entropy collapse and failing to produce novel outputs (Section 2.1, with more details given in Appendix H). It is fairly new to point out this entropy collapse issue and only in very recent work [Wang 2025], [Cui 2025], researchers began to investigate the role of entropy and the potential for its collapse during the RL training of LLMs, as well as methods to mitigate it. In addition, our RL ablations detail the non-trivial modifications (specifically, measure novelty using various metrics and to either penalize or skip training on samples that are not novel) required to overcome this, offering generalizable lessons for any researcher seeking to optimize for novelty in conjunction with a reward signal. It is only in the latest LLM reasoning literature that techniques based on conceptually similar principles have started to be discussed. For instance, [Yu 2025] is using curriculum learning to actively filter out problems on which the model has already converged.
>
> Wang, Zihan, et al. "Ragen: Understanding self-evolution in llm agents via multi-turn reinforcement learning." arXiv preprint arXiv:2504.20073 (2025).
>
> Cui, Ganqu, et al. "The entropy mechanism of reinforcement learning for reasoning language models." arXiv preprint arXiv:2505.22617 (2025).
>
> Yu, Qiying, et al. "Dapo: An open-source llm reinforcement learning system at scale." arXiv preprint arXiv:2503.14476 (2025).
>
> **Q1: Typo in Section 2 "(latent diffusion [60, LDMs] )"?  Q2: the reported precision scores in Table 1.**
>
> Thank you for pointing this out. LDM refers to the latent diffusion model. We will correct this and use 4 dp in the main paper.
>
> **Q3: A discussion on potential transfer to other creative domains .....?**
>
> Thank you for your constructive comments! We will add a broader impact discussion section in the main paper. Here is a draft:
>
> **Broader Impact and Generalizability**
> While this work is demonstrated within the specific domain of chess, its underlying principles and technical contributions possess significant potential for broader impact. The framework's most direct application lies in other domains that leverage guided search over a discrete state space, including not only complex board games like Go but also fields such as automated theorem proving. For instance, since systems like AlphaProof utilize AlphaZero, our counter-intuitiveness metric—which compares shallow and deep search evaluations—could be directly transferred to identify non-obvious or elegant problems, serving as a powerful reward signal for generating more creative mathematical problems. Conceptually, this work can also generalize to recent test-time scaling trends (OpenAI O1, DeepSeek R1). By taking analogy from deep search to deep thinking, we can also calculate the counter-intuitiveness for a given question of any domains that LLM can tackle.
>
> In addition, the diversity-filtering based RL framework is widely applicable in other domains. In developing our methods, we identified and provided a solution for the "entropy collapse" phenomenon, a critical issue that has only recently gained attention in the LLM community. Our solution—a dynamic diversity-filtering mechanism that promotes continuous exploration by penalizing non-novel samples—is domain-agnostic and can be applied to most tasks, from language to code generation, to prevent the policy from converging to a fixed solution. In summary, this research offers a transferable blueprint for quantifying and optimizing for non-obvious solutions and for building explorative RL systems capable of advancing the frontier of creative AI.
>
> **Q4: The Golden Set used to tune the ..... Would it limit the robustness and generalizability of the learned scoring function? Will expanding this dataset or incorporating crowd-sourced annotations could improve the reliability of the reward design?**
>
> While we acknowledge that a larger dataset would enhance robustness and performance, the effectiveness of our method with a small, curated set is also a key strength. This can highlight a few critical points:
>
> - **Effectiveness approximating crowdsource results.** Even though our curated set is small, in Appendix F, Figure 6 we have shown that there is a positive correlation between our counter intuitiveness reward and lichess puzzle’s rating score. This indicates that our method can indeed capture crowdsource human player’s difficulty preference.
>
> - **Efficiency of Engine-Based Feature Engineering.** Our approach does not learn a reward function from scratch. Instead, we are tuning a linear combination of a small set of powerful, semantically rich features (e.g., critical search depth) extracted from deep analysis by state-of-the-art chess engines. The purpose of the small Golden Set is simply to find the right weighting of these features that best aligns with human intuition. This stands in stark contrast to standard RLHF, which typically requires a dataset several orders of magnitude larger to learn a meaningful reward model. Our method's data efficiency is a significant practical advantage.
>
> - **The High Bar for Quality Data in this Domain:** We must also clarify that high-quality data collection for this task is uniquely challenging. Unlike tasks amenable to general crowdsourcing, reliably evaluating the "counter-intuitiveness" of a complex chess position requires substantial, specialized domain expertise. An annotation from a novice player would be of very low quality, as they may not be able to distinguish a brilliant, counter-intuitive move from a simple blunder. To ensure data quality, our human expert evaluators possessed Elo ratings between 2000 and 2400, placing them in the expert-to-master category. Sourcing a large number of annotations at this level of expertise is a significant undertaking.
>
> Therefore, while expanding the Golden Set with more expert-annotated examples is a valuable direction for future work, our current results demonstrate the power and efficiency of combining deep engine analysis with targeted, high-quality human guidance.

---

> > ### Comment · Reviewer_MudV · 2025-08-05
> >
> > Thank you for this detailed response. I think overall this work has a good originality, and W1 rebuttal is convincing. I also agree in Q4 that high-quality data collection for this task is uniquely challenging, its a good strategy to prioritize the data quality than quantity. I will maintain my score.

---

### Note · Authors · 2025-08-14

We sincerely thank all reviewers and AC for their time and insightful feedback. We hope our responses have managed to successfully address the points that were raised.

### **Significance of our work**
The new Grandmaster (GM)’s favorable results confirm the high quality and strategic depth of our AI-generated puzzles. We are confident this collection represents a landmark contribution, offering the chess community a powerful new training tool. **This work pioneers the use of AI for generating complex, creative chess problems and, to our knowledge, is the first significant advancement in this domain in over a decade.** Furthermore, it serves as a compelling demonstration of AI's capacity for genuine creativity.

### **Key theme in the discussion**
Our paper presents a complete framework to tackle the challenging problem of generating creative content—in this case, chess puzzles—by defining, quantifying, and optimizing for subjective qualities like uniqueness, novelty and counter-intuitiveness. We introduce a generative model refined by a novel diversity-filtered RL framework. The qualitative and human evaluation result affirm the creativity of our generation and the value of our work.

A key theme in the discussion was the generalizability and transferability of our work beyond chess. We believe our contributions are broadly applicable:
- Our primary technical innovation is a reward function that quantifies 'surprise' by comparing shallow vs. deep search. This principle is highly transferable to any domain using search or iterative reasoning (e.g., Go, theorem proving, LLM "deep thinking").

- We identified and solved the critical 'entropy collapse' problem when using RL for novelty. Our diversity-filtering framework offers a general blueprint for generating diverse outputs in creative AI.

- The success of our system, validated not just by metrics but by top human experts (GMs), establishes a strong benchmark for assessing AI-driven creativity and effective human-AI collaboration.

### **Revision**
Finally, we are fully committed to incorporating the suggestions for improving the paper's presentation. As discussed in our rebuttals, we will revise the manuscript to enhance clarity for non-domain experts, include discussion on our method's generalzability, correct all noted formatting issues, and ensure our technical contributions and metrics are presented as clearly as possible.

We thank the reviewers and the AC again for their valuable guidance.

---

### Decision · Program_Chairs · 2025-09-17

**Decision:**

Accept (poster)

**Comment:**

This paper proposes an RL framework for generating creative chess puzzles, introducing reward functions for novelty, counter-intuitiveness, and aesthetics, with strong quantitative and human expert validation.  Its strengths are originality, careful reward design, solid evaluation, and a valuable curated outcome.  Weaknesses lie in limited generalizability, modest technical novelty, and clarity issues.  Despite these, the work represents a distinctive and well-executed application of creative AI, justifying acceptance.  In discussion, reviewers’ concerns about scope and presentation were partly addressed, with clarifications on methodology and claims of broader applicability; most reviewers maintained or improved scores, leading to consensus that the contribution, while niche, is impactful enough for the conference.